# ON DOUBLE DESCENT IN REINFORCEMENT LEARNING WITH LSTD AND RANDOM FEATURES

**David Brellmann, Eloïse Berthier, David Filliat & Goran Frehse**
U2IS, ENSTA Paris, Institut Polytechnique de Paris, Palaiseau, FRANCE
{first_name.last_name}@ensta-paris.fr

## ABSTRACT

Temporal Difference (TD) algorithms are widely used in Deep Reinforcement Learning (RL). Their performance is heavily influenced by the size of the neural network. While in supervised learning, the regime of over-parameterization and its benefits are well understood, the situation in RL is much less clear. In this paper, we present a theoretical analysis of the influence of network size and $l_2$-regularization on performance. We identify the ratio between the number of parameters and the number of visited states as a crucial factor and define over-parameterization as the regime when it is larger than one. Furthermore, we observe a double descent phenomenon, i.e., a sudden drop in performance around the parameter/state ratio of one. Leveraging random features and the lazy training regime, we study the regularized Least-Squared Temporal Difference (LSTD) algorithm in an asymptotic regime, as both the number of parameters and states go to infinity, maintaining a constant ratio. We derive deterministic limits of both the empirical and the true Mean-Squared Bellman Error (MSBE) that feature correction terms responsible for the double descent. Correction terms vanish when the $l_2$-regularization is increased or the number of unvisited states goes to zero. Numerical experiments with synthetic and small real-world environments closely match the theoretical predictions.

## 1 INTRODUCTION

In recent years, neural networks have seen increased use in Reinforcement Learning (RL) (Mnih et al., 2015; Schulman et al., 2017; Haarnoja et al., 2018). While they can outperform traditional RL algorithms on challenging tasks, their theoretical understanding remains limited. Even for supervised learning, which can be considered a special case of RL with discount factor equal to zero, deep neural networks are still far from being fully understood despite significant research efforts (Arora et al., 2019; Mei et al., 2018; Rotskoff & Vanden-Eijnden, 2018; Lee et al., 2019; Bietti & Mairal, 2019; Cao et al., 2019). The difficulty is further exacerbated in RL by a myriad of new challenges that limit the scope of these works, such as the absence of true targets or the non-i.i.d nature of the collected samples (Kumar et al., 2020; Luo et al., 2020; Lyle et al., 2021; Dong et al., 2020). Temporal-Difference (TD) methods are widely used RL algorithms that frequently use neural networks, are simple, and efficient in practice. We use the regularized Least-squares Temporal Difference (LSTD) algorithm (Bradtke & Barto, 1996), which is easier to analyze since it doesn't use gradient descent, and because it converges to the same solution as other TD algorithms (Bradtke & Barto, 1996; Boyan, 1999; Berthier et al., 2022).

Theoretical studies of TD algorithms often explore asymptotic regimes where the number of samples $n \to \infty$ while the number of model parameters $N$ remains constant (Tsitsiklis & Van Roy, 1996; Sutton, 1988). When TD learning algorithms are applied to neural networks, it is commonly assumed that the number of parameters $N \to \infty$ with either a fixed or infinite number of samples without providing details on the relative magnitudes of these parameters (Cai et al., 2019; Agazzi & Lu, 2022; Berthier et al., 2022; Xiao et al., 2021). Inspired by advancements in supervised learning (Louart et al., 2018; Liao et al., 2020), we apply Random Matrix tools and propose a novel double asymptotic regime where the number of parameters $N$ and the number of distinct visited states $m$ go to infinity, maintaining a constant ratio, called model complexity. We use a linear model and nonlinear random features (RF) (Rahimi & Recht, 2007) to approximate an overparameterized

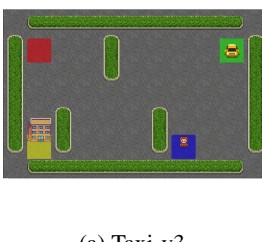

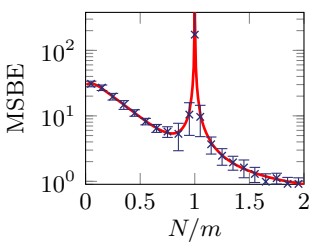

(a) Taxi-v3

(b) MSBE as a function of $N/m$

Figure 1: **As the model complexity $N/m$ (for $N$ parameters, $m$ distinct visited states) increases, the MSBE first shows a U-shaped curve, peaking around the interpolation threshold ($N = m$). Double descent refers to the phenomenon for $N/m > 1$ where the MSBE drops once again.** Continuous lines (red) indicate the theoretical values from Theorem 5.3, the crosses (blue) are numerical results averaged over 30 instances with their standard deviations after the learning with regularized LSTD on Taxi-v3 for $\gamma = 0.95, \lambda = 10^{-9}, n = 5000, m = 310$.

single-hidden-layer network in the lazy training regime (Chizat et al., 2019). The results of our theoretical and empirical analyses are outlined below.

**Contributions.** We make the following contributions, taking a step towards a better theoretical understanding of the influence of model complexity $N/m$ and $l_2$-regularization on the performance of Temporal Difference algorithms:

1. We propose a novel double asymptotic regime, where the number of parameters $N$ and distinct visited states $m$ go to infinity while maintaining a constant ratio. This leads to a precise assessment of the performance in both over-parameterized ($N/m > 1$) and under-parameterized regimes ($N/m < 1$). This is a nontrivial extension of existing work in supervised learning since several properties essential to proofs, such as the positive definiteness of key matrices, are voided by a discount factor in RL.

2. In the phase transition around $N/m = 1$, we observe a peak in the Mean-Squared Bellman Error (MSBE), as illustrated in figure 1, i.e., a double descent phenomenon similar to what has been reported in supervised learning (Mei & Montanari, 2022; Liao et al., 2020).

3. We identify the resolvent of a non-symmetric positive-definite matrix that emerges as a crucial factor in the performance analysis of TD learning algorithms in terms of the MSBE and we provide its deterministic limit form in the double asymptotic regime.

4. We derive analytical equations for both the asymptotic empirical MSBE on the collected transitions and the asymptotic true MSBE. The deterministic forms expose correction terms that we experimentally associate with the double descent phenomenon. We show that the correction terms vanish as the $l_2$-regularization is increased or $N/m$ goes to infinity. We also show that the influence of the $l_2$-regularization parameter decreases as $N/m$ increases.

5. Our theory closely matches empirical results on a range of both toy and small real-world Markov Reward Processes where $m$ and $N$ are fixed, but for which the asymptotic regime still gives accurate predictions. Notably, we observe a peak in the true MSBE around $N/m = 1$ that is not observed in the empirical MSBE. Correction terms, and therefore the difference between true and empirical MSBE, empirically vanish when the number of unvisited states goes to zero.

## 2 RELATED WORK

We review three related approaches to study neural networks in supervised learning or RL. Further technical results from the literature are cited where relevant throughout the paper.

**Neural Tangent Kernel (NTK) regime.** In the NTK regime, one considers that infinitely wide neural networks, with appropriate scaling and initial conditions, behave like the linearization of the

neural network around its initialization (Jacot et al., 2018). However, as highlighted by Chizat et al. (2019), this behavior is not specific to neural networks and is not so much due to over-parameterization than to an implicit choice of scaling. In such a scenario, neural networks can be modeled as a random feature model (Rahimi & Recht, 2007); we adopt this technique in order to abstract from the learning dynamics. The NTK regime was also considered in RL, in both finite and infinite state space, to prove the convergence of infinite-width neural TD learning algorithms towards the global optimum of the MSBE (Cai et al., 2019; Agazzi & Lu, 2022; Liu et al., 2019).

**Mean-Field regime.** Under appropriate initial conditions and scaling, the mean-field analysis models the neural network and its induced feature representation with an empirical distribution, which, at the infinite-width limit, corresponds to a population distribution. The evolution of such a population distribution is characterized by a partial differential equation (PDE) known as the continuity equation and captures Stochastic Gradient Descent (SGD) dynamics as a Wasserstein gradient flow of the objective function (Chizat & Bach, 2018; Rotskoff & Vanden-Eijnden, 2018; Mei et al., 2018). Although more challenging than the NTK regime, the mean-field regime is more realistic since the weights are not restricted to staying in their initial regions (Chizat et al., 2019). The mean-field regime was studied in RL to prove the convergence of infinite-width neural TD learning algorithms towards the global optimum of the MSBE (Zhang et al., 2021; Agazzi & Lu, 2022).

**Double Asymptotic regime.** In the above regimes, the number of data points $n$ is negligible compared to the number of parameters $N$ since $N$ grows to infinity. However, this is rarely the case in practice, which is why the double descent phenomenon can not be explained with the previous regimes (Zhang et al., 2020; Belkin et al., 2018). The double descent phenomenon is characterized by a peak near the interpolation threshold ($N = n$). For this reason, many studies in supervised learning consider a double asymptotic regime (Mei & Montanari, 2022; Louart et al., 2018; Liao et al., 2020; Belkin et al., 2020), where both $n, N$ go to infinity while maintaining their ratio constant. In the above work, techniques from Random Matrix theory are used to derive a precise description of the phase transition between under-($N < n$) and over-($N > n$) parameterization and the double descent phenomenon. Our work extends this approach from supervised learning to RL. Since several key properties from supervised learning are voided by the discount factor, the analysis is substantially more involved in the case of RL. Thomas (2022) investigated off-policy linear TD methods in the limit of large number of states and parameters on a transition matrix of rank 1, and observed a peaking behavior in the MSBE. We consider a more general setting on the on-policy setting and a different ratio in the double asymptotic regime without making assumptions about the rank of the transition matrix.

## 3 PRELIMINARIES

**Notations.** We define $[n] = \{1, 2, \ldots, n\}$. For a real matrix $\boldsymbol{A}$, $[\boldsymbol{A}]_{ij}$ denotes its $(i, j)^{th}$ entry. For $\boldsymbol{A}$ with real eigenvalues, we denote with $\nu_{\min}(\boldsymbol{A})$ the smallest and $\nu_{\max}(\boldsymbol{A})$ the largest eigenvalue. The symmetric part of $\boldsymbol{A}$ is $H(\boldsymbol{A}) = \frac{\boldsymbol{A} + \boldsymbol{A}^T}{2}$. The operator norm of a $\boldsymbol{A}$ is written $\|\boldsymbol{A}\|$. The norm induced by $\boldsymbol{A}$ on a vector $\boldsymbol{v}$ is $\|\boldsymbol{v}\|_{\boldsymbol{A}} = \boldsymbol{v}^T \boldsymbol{A} \boldsymbol{v}$ and $\|\boldsymbol{v}\|$ depicts the Euclidean norm of $\boldsymbol{v}$. $\mathcal{N}(0, 1)$ denotes the standard Gaussian distribution.

**Markov Reward Processes.** We consider a Markov Reward Process (MRP) $(\mathcal{S}, P, r, \gamma)$, where $\mathcal{S} \subseteq \mathbb{R}^d$ is the state space; $P : \mathcal{S} \times \mathcal{S} \to [0, 1]$ is the transition kernel (stochastic kernel) and $P(\boldsymbol{s}, \boldsymbol{s}')$ denotes the probability of transitioning to state $s'$ from state $\boldsymbol{s}$; $r : \mathcal{S} \times \mathcal{S} \to \mathbb{R}$ is the reward function; and $\gamma \in [0, 1)$ is the discount factor. For notational convenience, the state space $\mathcal{S}$ is described by the state matrix $\boldsymbol{S} \in \mathbb{R}^{d \times |\mathcal{S}|}$, where each column of $\boldsymbol{S}$, written $\boldsymbol{S}_i$, represents a state in $\mathcal{S}$. The transition probability matrix associated with the stochastic kernel $P$ is $\boldsymbol{P} \in \mathbb{R}^{|\mathcal{S}| \times |\mathcal{S}|}$. The goal is to learn the value function $V : \mathcal{S} \to \mathbb{R}$, which maps each state $\boldsymbol{s}$ to the expected discounted sum of rewards when starting from $\boldsymbol{s}$ and following the dynamics of the MRP defined by $\boldsymbol{P}$ as $V(\boldsymbol{s}) := \mathbb{E}_{\boldsymbol{P}}\left[\sum_{k=1}^{\infty} \gamma^{k-1} r(\boldsymbol{s}_k, \boldsymbol{s}_{k+1}) \mid \boldsymbol{s}_1 = \boldsymbol{s}\right]$. It is well-known that the value function is the unique fixed-point of Bellman's equation

$$\boldsymbol{V} = \bar{\boldsymbol{r}} + \gamma \boldsymbol{P} \boldsymbol{V}, \tag{1}$$

where $\boldsymbol{V} \in \mathbb{R}^{|\mathcal{S}|}$ is a vector whose $i$-th element is the value function of the $i$-th state $\boldsymbol{S}_i$; and $\bar{\boldsymbol{r}} \in \mathbb{R}^{|\mathcal{S}|}$ is the vector containing the expected rewards, for which $\bar{\boldsymbol{r}}_i = \mathbb{E}_{\boldsymbol{P}}[r|\boldsymbol{S}_i]$ for all $i \in [\,|\mathcal{S}|\,]$.

**Linear Function Approximation.** In practice, equation 1 cannot be solved since $\boldsymbol{P}$ is unknown and $|\mathcal{S}|$ is too large. One common solution is to use Linear Function Approximation (LFA). Using a parameter vector $\boldsymbol{\theta} \in \mathbb{R}^N$ and a feature matrix $\boldsymbol{\Sigma}_{\mathcal{S}} \in \mathbb{R}^{N \times |\mathcal{S}|}$, whose columns are the feature vectors for every state, $\boldsymbol{V}$ is approximated by $\boldsymbol{V} \approx \boldsymbol{\Sigma}_{\mathcal{S}}^T \boldsymbol{\theta}$. In Deep RL, the neural network learns both the feature vectors and the parameter vector. For a given feature matrix, the learning process based on equation 1 amounts to finding a parameter vector $\boldsymbol{\theta}$ that minimizes the Mean-Squared Bellman error (MSBE)

$$\text{MSBE}(\boldsymbol{\theta}) = \|\bar{\boldsymbol{r}} + \gamma \boldsymbol{P}\boldsymbol{\Sigma}_{\mathcal{S}}^T \boldsymbol{\theta} - \boldsymbol{\Sigma}_{\mathcal{S}}^T \boldsymbol{\theta}\|_{\boldsymbol{D}_{\boldsymbol{\pi}}}^2, \tag{2}$$

where $\boldsymbol{\pi} \in \mathbb{R}^{|\mathcal{S}|}$ is the stationary distribution induced by the MRP and $\boldsymbol{D}_{\boldsymbol{\pi}} \in \mathbb{R}^{|\mathcal{S}| \times |\mathcal{S}|}$ is its diagonal matrix. Since $\bar{\boldsymbol{r}} + \gamma \boldsymbol{P}\boldsymbol{\Sigma}_{\mathcal{S}}^T \boldsymbol{\theta}$ may not lie in the span of the bases $\boldsymbol{\Sigma}_{\mathcal{S}}$, there may not be a parameter vector $\boldsymbol{\theta}$ that brings the MSBE to zero.

**Linear Temporal-Difference Methods.** Linear Temporal-Difference (TD) methods are LFA methods that try to minimize the MSBE in equation 2 by replacing the second occurrence of $\boldsymbol{\theta}$ in equation 2 with an auxiliary vector $\boldsymbol{u}$, minimizing on $\boldsymbol{u}$ and then finding a $\boldsymbol{\theta}$ close to $\boldsymbol{u}$ (Dann et al., 2014):

$$\boldsymbol{u}^* = \underset{\boldsymbol{u} \in \mathbb{R}^N}{\arg\min} \ \|\bar{\boldsymbol{r}} + \gamma \boldsymbol{P}\boldsymbol{\Sigma}_{\mathcal{S}}^T \boldsymbol{\theta} - \boldsymbol{\Sigma}_{\mathcal{S}}^T \boldsymbol{u}\|_{\boldsymbol{D}_{\boldsymbol{\pi}}}^2 \qquad \text{(projection step)}, \tag{3}$$

$$\boldsymbol{\theta}^* = \underset{\boldsymbol{\theta} \in \mathbb{R}^N}{\arg\min} \ \|\boldsymbol{\Sigma}_{\mathcal{S}}^T \boldsymbol{u}^* - \boldsymbol{\Sigma}_{\mathcal{S}}^T \boldsymbol{\theta}\|_{\boldsymbol{D}_{\boldsymbol{\pi}}}^2 \qquad \text{(fixed-point step)}. \tag{4}$$

The projection step (equation 3) implies that TD methods actually minimize the Mean-Squared Projected Bellman error (MSPBE) rather than the MSBE (Dann et al., 2014). In our asymptotic regime, as the number of features $N \to \infty$, the class of representable value functions becomes richer, and the MSBPE converges to the MSBE (Cai et al., 2019; Agazzi & Lu, 2022).

## 4 SYSTEM MODEL

We now describe the key elements on which we base our asymptotic analysis of the MSBE in TD learning: Random features, the regularized LSTD algorithm, and the double asymptotic regime.

### 4.1 REGULARIZED LSTD WITH RANDOM FEATURES

**Random Features.** We consider value function approximation using the Random Feature (RF) mapping $\text{RF} : \mathcal{S} \to \mathbb{R}^N$ defined for all $\boldsymbol{s} \in \mathcal{S}$ as

$$\text{RF}(\boldsymbol{s}) = \sigma(\boldsymbol{W}\boldsymbol{s}), \tag{5}$$

where $\sigma : \mathbb{R} \to \mathbb{R}$ is $K_\sigma$-Lipschitz continuous and applied component-wise; $\boldsymbol{W} = \varphi(\tilde{\boldsymbol{W}}) \in \mathbb{R}^{N \times d}$ is a random weight matrix fixed throughout training for which $\tilde{\boldsymbol{W}} \in \mathbb{R}^{N \times d}$ has independent and identically distributed $\mathcal{N}(0, 1)$ entries, and $\varphi : \mathbb{R} \to \mathbb{R}$ is $K_\varphi$-Lipschitz continuous and applied component-wise. From the perspective of neural networks, the $N$ random features can be seen as $N$ outputs from a single-hidden-layer neural network. In our asymptotic regime, this simplification becomes even more accurate as the number of features $N$ of the single layer grows towards infinity and we enter into the lazy training regime, where weights barely deviate from their random initial values (Chizat et al., 2019). In the literature on Deep Learning and double descent, large-width neural networks are often modeled using asymptotic random features (Louart et al., 2018; Liao et al., 2020; Mei & Montanari, 2022), including in RL (Cai et al., 2019; Agazzi & Lu, 2022; Liu et al., 2019). In the following, we denote the random feature matrix of any state matrix $\boldsymbol{A} \in \mathbb{R}^{d \times p}$ as $\boldsymbol{\Sigma}_{\boldsymbol{A}}$ where RF is applied column-wise, i.e., $\boldsymbol{\Sigma}_{\boldsymbol{A}} = \sigma(\boldsymbol{W}\boldsymbol{A})$.

**Sample Matrices and Empirical MSBE.** We assume that the transition probability matrix $\boldsymbol{P}$ is unknown during the training phase. Instead, we have a dataset of $n$ transitions consisting of states, rewards, and next-states drawn from the MRP, i.e., $\mathcal{D}_{\text{train}} := \left\{ (\boldsymbol{s}_i, r_i, \boldsymbol{s}_i') \right\}_{i=1}^n$ where $\boldsymbol{s}_i' \sim P(\boldsymbol{s}_i, \cdot)$. We consider the on-policy setting, where $\mathcal{D}_{\text{train}}$ is derived from a sample path of the MRP or its stationary distribution $\boldsymbol{\pi}$. We collect the states and rewards in the sample matrices

$$\boldsymbol{X}_n = [\boldsymbol{s}_1, \dots, \boldsymbol{s}_n] \in \mathbb{R}^{d \times n}, \quad \boldsymbol{r} = [r_1, \dots, r_n]^T \in \mathbb{R}^n, \quad \boldsymbol{X}_n' = [\boldsymbol{s}_1', \dots, \boldsymbol{s}_n'] \in \mathbb{R}^{d \times n}. \tag{6}$$

Let $\hat{\mathcal{S}} \subseteq \mathcal{S}$ be the set of distinct states in $\mathcal{D}_{\text{train}}$, which we call underline{visited states}, and let $m = |\hat{\mathcal{S}}|$. Let $\hat{\boldsymbol{S}} \in \mathbb{R}^{d \times m}$ be the state matrix of $\hat{\mathcal{S}}$, i.e., each column $\hat{\boldsymbol{S}}_i$ of $\hat{\boldsymbol{S}}$ describes a state in $\hat{\mathcal{S}}$. We denote by $\boldsymbol{\Sigma}_{\hat{\mathcal{S}}} \in \mathbb{R}^{N \times m}$, $\boldsymbol{\Sigma}_{\boldsymbol{X}_n} \in \mathbb{R}^{N \times n}$, and $\boldsymbol{\Sigma}_{\boldsymbol{X}'_n} \in \mathbb{R}^{N \times n}$ the random feature matrices of $\hat{\mathcal{S}}$, $\boldsymbol{X}_n$, and $\boldsymbol{X}'_n$, respectively. For the proof of our results, it will be mathematically advantageous to express $\boldsymbol{\Sigma}_{\boldsymbol{X}_n}$ and $\boldsymbol{\Sigma}_{\boldsymbol{X}'_n}$ as the product of $\boldsymbol{\Sigma}_{\hat{\mathcal{S}}}$ with auxiliary matrices $\hat{U}_n \in \mathbb{R}^{m \times n}$ and $\hat{V}_n \in \mathbb{R}^{m \times n}$ as follows:

$$\boldsymbol{\Sigma}_{\boldsymbol{X}_n} = \sqrt{n} \boldsymbol{\Sigma}_{\hat{\mathcal{S}}} \hat{U}_n \quad \text{and} \quad \boldsymbol{\Sigma}_{\boldsymbol{X}'_n} = \sqrt{n} \boldsymbol{\Sigma}_{\hat{\mathcal{S}}} \hat{V}_n \,. \tag{7}$$

Each column $i$ of $\sqrt{n}\hat{U}_n$ is a one-hot vector, where the $j$-th element equals 1 if the $i$-th state $\boldsymbol{s}_i$ of $\boldsymbol{X}_n$ is $\hat{\boldsymbol{S}}_j$, and similarly for $\sqrt{n}\hat{V}_n$ and $\boldsymbol{X}'_n$. Since $\boldsymbol{P}$ is unknown, we aim to find $\boldsymbol{\theta}$ that minimizes the empirical version of the MSBE (equation 2) obtained with transitions collected in $\mathcal{D}_{\text{train}}$:

$$\widehat{\text{MSBE}}(\boldsymbol{\theta}) = \tfrac{1}{n} \| \boldsymbol{r} + \gamma \boldsymbol{\Sigma}_{\boldsymbol{X}'_n}^T \boldsymbol{\theta} - \boldsymbol{\Sigma}_{\boldsymbol{X}_n}^T \boldsymbol{\theta} \|^2, \tag{8}$$

which uses the Euclidean norm since the distribution is reflected by the samples. Assuming globally stable MRP, a fixed number of features, and all states being visited, $\widehat{\text{MSBE}}(\boldsymbol{\theta})$ converges to $\text{MSBE}(\boldsymbol{\theta})$ with probability 1, as the number of collected transitions $n \to \infty$. This follows from the law of large numbers (Stachurski, 2009). In our analysis, we will also consider the case where $n \to \infty$ without visiting all states, i.e., $m < |\mathcal{S}|$, such that there can be a significant difference between $\widehat{\text{MSBE}}(\boldsymbol{\theta})$ and $\text{MSBE}(\boldsymbol{\theta})$.

**Regularized Least-Square Temporal-Difference Methods.** Regularized Least-Square Temporal-Difference (LSTD) Methods (Bradtke & Barto, 1996) are linear TD methods that solve an empirical regularized version of equation 3 and 4 with transitions collected in $\mathcal{D}_{\text{train}}$:

$$\boldsymbol{u}^* = \underset{\boldsymbol{u} \in \mathbb{R}^N}{\arg\min} \ \| \boldsymbol{r} + \gamma \boldsymbol{\Sigma}_{\boldsymbol{X}'_n}^T \boldsymbol{\theta} - \boldsymbol{\Sigma}_{\boldsymbol{X}_n}^T \boldsymbol{u} \|^2 + \lambda_{m,n} \| \boldsymbol{u} \|^2, \tag{9}$$

$$\hat{\boldsymbol{\theta}} = \underset{\boldsymbol{\theta} \in \mathbb{R}^N}{\arg\min} \ \| \boldsymbol{\Sigma}_{\boldsymbol{X}_n}^T \boldsymbol{u}^* - \boldsymbol{\Sigma}_{\boldsymbol{X}_n}^T \boldsymbol{\theta} \|^2, \tag{10}$$

where $\lambda_{m,n} > 0$ is the underline{effective $l_2$-regularization parameter}, introduced to mitigate overfitting (Hoffman et al., 2011; Chen et al., 2013). It is well known that for $\lambda_{m,n} = 0$ and with the number of samples $n \to \infty$, the fixed point of the approximation equation 9 and 10 equals the fixed point of equation 3 and 4 with probability one. Solving the fixed-point of the linear system approximation given by equation 9 and 10 gives

$$\hat{\boldsymbol{\theta}} = \left[ \boldsymbol{\Sigma}_{\boldsymbol{X}_n} \left[ \boldsymbol{\Sigma}_{\boldsymbol{X}_n} - \gamma \boldsymbol{\Sigma}_{\boldsymbol{X}'_n} \right]^T + \lambda_{m,n} \boldsymbol{I}_N \right]^{-1} \boldsymbol{\Sigma}_{\boldsymbol{X}_n} \boldsymbol{r}. \tag{11}$$

Under appropriate learning rates, linear TD methods based on gradient-descent converge towards the same fixed-point $\hat{\boldsymbol{\theta}}$ (Robbins & Monro, 1951; Dann et al., 2014; Sutton & Barto, 2018). Besides reducing overfitting, an appropriate $\lambda_{m,n}$ ensures that $\boldsymbol{\Sigma}_{\boldsymbol{X}_n}[\boldsymbol{\Sigma}_{\boldsymbol{X}_n} - \gamma\boldsymbol{\Sigma}_{\boldsymbol{X}'_n}]^T + \lambda_{m,n}\boldsymbol{I}_N$ is invertible.

## 4.2 Double Asymptotic Regime and Resolvent in LSTD

We study the regularized LSTD in the following double asymptotic regime:

**Assumption 1** (Double Asymptotic Regime). *As $N, m, d \to \infty$, we have:*

1. $0 < \lim\min \left\{ \frac{N}{m}, \frac{d}{m}, \frac{m}{|\mathcal{S}|} \right\} < \lim\max \left\{ \frac{N}{m}, \frac{d}{m}, \frac{m}{|\mathcal{S}|} \right\} < \infty.$

2. *There exists $K_{\mathcal{S}}, K_r > 0$ such that $\lim\sup_{|\mathcal{S}|} \|\boldsymbol{S}\| < K_{\mathcal{S}}$ and $r(\cdot, \cdot)$ is bounded by $K_r$.*

In order to use Random Matrix tools, we rewrite equation 11 as (see proof in Lemma L.9)

$$\hat{\boldsymbol{\theta}} = \tfrac{1}{mn} \boldsymbol{\Sigma}_{\boldsymbol{X}_n} \left[ \tfrac{1}{mn} \left[ \boldsymbol{\Sigma}_{\boldsymbol{X}_n} - \gamma \boldsymbol{\Sigma}_{\boldsymbol{X}'_n} \right]^T \boldsymbol{\Sigma}_{\boldsymbol{X}_n} + \tfrac{\lambda_{m,n}}{mn} \boldsymbol{I}_n \right]^{-1} \boldsymbol{r}. \tag{12}$$

Instead of the effective $l_2$-regularization parameter $\lambda_{m,n}$, we will use its scaled version $\lambda = \frac{\lambda_{m,n}}{mn}$ in the remainder of this paper. We observe that $\hat{\boldsymbol{\theta}} = \frac{1}{mn} \boldsymbol{\Sigma}_{\boldsymbol{X}_n} \boldsymbol{Q}_m(\lambda) \boldsymbol{r}$ depends on the underline{resolvent}

$$\boldsymbol{Q}_m(\lambda) = \left[ \tfrac{1}{mn} \left[ \boldsymbol{\Sigma}_{\boldsymbol{X}_n} - \gamma \boldsymbol{\Sigma}_{\boldsymbol{X}'_n} \right]^T \boldsymbol{\Sigma}_{\boldsymbol{X}_n} + \lambda \boldsymbol{I}_n \right]^{-1} = \left[ \tfrac{1}{m} (\hat{U}_n - \gamma \hat{V}_n)^T \boldsymbol{\Sigma}_{\hat{\mathcal{S}}}^T \boldsymbol{\Sigma}_{\hat{\mathcal{S}}} \hat{U}_n + \lambda \boldsymbol{I}_n \right]^{-1} \tag{13}$$

when $\frac{1}{m}(\hat{U}_n - \gamma\hat{V}_n)^T\Sigma_{\hat{S}}^T\Sigma_{\hat{S}}\hat{U}_n + \lambda I_n$ is invertible, which in general may not be the case. We can guarantee invertibility if the underline{empirical transition model matrix} $\hat{A}_m \in \mathbb{R}^{m\times m}$

$$\hat{A}_m = \hat{U}_n(\hat{U}_n - \gamma\hat{V}_n)^T \tag{14}$$

has a positive-definite symmetric part (see Appendix H for a formal proof). For the remainder of the paper, we therefore make the following assumption on $\hat{A}_m$:

**Assumption 2** (Bounded Eigenspectrum). *There exist $0 < \xi_{\min} < \xi_{\max}$ such that for every $m$, all the eigenvalues of $H(\hat{A}_m)$ are in $[\xi_{\min}, \xi_{\max}]$.*

Note that the above assumption is satisfied for regularized pathwise LSTD (Lazaric et al., 2012), and also for sufficiently large $n$ (see Appendix H).

## 5 ASYMPTOTIC ANALYSIS OF REGULARIZED LSTD

In this section, we present our main theoretical results, which characterize the true and empirical MSBE under Assumptions 1 and 2.

### 5.1 AN EQUIVALENT DETERMINISTIC RESOLVENT

The resolvent $Q_m(\lambda)$ (in equation 13) plays a significant role in the performance of regularized LSTD since $\hat{\theta} = \frac{1}{\sqrt{n}}\frac{1}{m}\Sigma_{\hat{S}}\hat{U}_n Q_m(\lambda)r$. To assess the asymptotic $\widehat{\text{MSBE}}(\hat{\theta})$ and true $\text{MSBE}(\hat{\theta})$, we first find a deterministic equivalent for the resolvent $Q_m(\lambda)$. A natural deterministic equivalent would be $\mathbb{E}_W[Q_m(\lambda)]$, but it involves integration without having a closed form expression (due to the matrix inverse) and is inconvenient for practical computation. Leveraging Random Matrix tools, the following Theorem 5.1 proposes an asymptotic form that is $i$. close to $\mathbb{E}_W[Q_m(\lambda)]$ under Assumptions 1 and 2, and $ii$. numerically more accessible (for the proof, see Appendix E).

**Theorem 5.1** (Asymptotic Deterministic Resolvent). *Under Assumptions 1 (double asymptotic regime) and 2 (bounded spectrum), let $\lambda > 0$ and let the underline{deterministic resolvent} $\bar{Q}_m(\lambda) \in \mathbb{R}^{n\times n}$ be*

$$\bar{Q}_m(\lambda) = \left[\frac{N}{m}\frac{1}{1+\delta}(\hat{U}_n - \gamma\hat{V}_n)^T\Phi_{\hat{S}}\hat{U}_n + \lambda I_n\right]^{-1}, \tag{15}$$

*where the underline{deterministic Gram feature matrix} $\Phi_{\hat{S}} \in \mathbb{R}^{m\times m}$ is*

$$\Phi_{\hat{S}} = \mathbb{E}_{w\sim\mathcal{N}(0,I_d)}\left[\sigma(w^T\hat{S})^T\sigma(w^T\hat{S})\right], \tag{16}$$

*and the underline{correction factor} $\delta$ is the unique, positive, solution to*

$$\delta = \frac{1}{m}\text{Tr}\left((\hat{U}_n - \gamma\hat{V}_n)^T\Phi_{\hat{S}}\hat{U}_n\left[\frac{N}{m}\frac{1}{1+\delta}(\hat{U}_n - \gamma\hat{V}_n)^T\Phi_{\hat{S}}\hat{U}_n + \lambda I_n\right]^{-1}\right). \tag{17}$$

*Then $\lim_{m\to\infty}\left\|\mathbb{E}_W[Q_m(\lambda)] - \bar{Q}_m(\lambda)\right\| = 0$. The correction factor $\delta$ diminishes as $N$ or $\lambda$ grows (see Lemma J.3 and J.4).*

**Remark 1.** *Since $\delta \to 0$ when $N/m \to \infty$, the correction factor $\frac{1}{1+\delta}$ arises from our asymptotic regime, which keeps the ratio $N/m$ asymptotically constant (see Lemma J.1 for existence and uniqueness). Similar correction factors arise in related Random Matrix literature, which, however, mostly deals with positive semi-definite matrices (Couillet & Debbah, 2011; Liu et al., 2019; Liao et al., 2020). Our problem exceeds this frame, so we prove the result, including existence and uniqueness, with a somewhat more involved analysis based on the eigenspectrum of the products of matrices with positive-definite symmetric part and skew-symmetric matrices (see Appendix J).*

**Remark 2.** *For the case of supervised learning, a comparable proposition is presented by Louart et al. (2018, Theorem 1). It constitutes a special case of Theorem 5.1 with $\gamma = 0$, which corresponds to the case where we learn the reward function.*

**Remark 3.** *Note that the eigenvalues of $\bar{Q}_m(\lambda)$ are not necessarily real, which renders many tools from the related Random Matrix literature not applicable, e.g., Stieltjes transforms would provide information on the eigenspectrum density of matrices based on the trace of their resolvents.*

## 5.2 ASYMPTOTIC EMPIRICAL MEAN-SQUARED BELLMAN ERROR

TD methods learn by minimizing the empirical MSBE (equation 8) and, under appropriate learning rates, converge towards the empirical MSBE of LSTD, as mentioned in Section 4. It is straightforward to show that this leads to an optimal $\widehat{\mathrm{MSBE}}(\hat{\boldsymbol{\theta}}) = \frac{\lambda^2}{n}\|\boldsymbol{Q}_m(\lambda)\boldsymbol{r}\|^2$ (see Appendix F). Using concentration arguments for Gaussian distributions and Lipschitz applications, as well as Theorem 5.1, we derive the following deterministic form (see proof in Appendix F).

**Theorem 5.2** (Asymptotic Empirical MSBE). *Under the conditions of Theorem 5.1, the* *deterministic asymptotic empirical MSBE* *is* $\overline{\widehat{\mathrm{MSBE}}}(\hat{\boldsymbol{\theta}}) = \frac{\lambda^2}{n}\|\bar{\boldsymbol{Q}}_m(\lambda)\boldsymbol{r}\|^2 + \hat{\Delta}$, *with second order correction factor*

$$\hat{\Delta} = \frac{\lambda^2}{n}\frac{\frac{1}{N}\operatorname{Tr}\big(\bar{\boldsymbol{Q}}_m(\lambda)\boldsymbol{\Psi}_2\bar{\boldsymbol{Q}}_m(\lambda)^T\big)}{1-\frac{1}{N}\operatorname{Tr}\big(\boldsymbol{\Psi}_2\bar{\boldsymbol{Q}}_m(\lambda)^T\boldsymbol{\Psi}_1\bar{\boldsymbol{Q}}_m(\lambda)\big)}\|\bar{\boldsymbol{Q}}_m(\lambda)\boldsymbol{r}\|^2_{\boldsymbol{\Psi}_1}, \quad \text{where} \tag{18}$$

$$\boldsymbol{\Psi}_1 = \frac{N}{m}\frac{1}{1+\delta}\hat{\boldsymbol{U}}_n^T\boldsymbol{\Phi}_{\hat{\mathcal{S}}}\hat{\boldsymbol{U}}_n, \quad \text{and} \quad \boldsymbol{\Psi}_2 = \frac{N}{m}\frac{1}{1+\delta}(\hat{\boldsymbol{U}}_n - \gamma\hat{\boldsymbol{V}}_n)^T\boldsymbol{\Phi}_{\hat{\mathcal{S}}}(\hat{\boldsymbol{U}}_n - \gamma\hat{\boldsymbol{V}}_n). \tag{19}$$

*As* $N, m, d \to \infty$ *with asymptotic constant ratio* $N/m$, $\widehat{\mathrm{MSBE}}(\hat{\boldsymbol{\theta}}) - \overline{\widehat{\mathrm{MSBE}}}(\hat{\boldsymbol{\theta}}) \xrightarrow{a.s} 0$.

**Remark 4.** *As* $N/m \to \infty$, *we find* $\overline{\mathrm{MSBE}}(\hat{\boldsymbol{\theta}}) \to 0$.

**Remark 5.** *In supervised learning, a comparable proposition is presented by Louart et al. (2018, Theorem 3). It is a special case of Theorem 5.2 with* $\gamma = 0$, *where we learn the reward function* $r$.

## 5.3 ASYMPTOTIC MEAN-SQUARED BELLMAN ERROR

While the empirical MSBE only takes states from the data set into account, the true MSBE (equation 2) involves all states in $\mathcal{S}$. To extend the convergence results from the previous section to this case, we require some further notations. Using a decomposition similar to equation 7, we express $\boldsymbol{\Sigma}_{\boldsymbol{X}_n}$ and $\boldsymbol{\Sigma}_{\boldsymbol{X}'_n}$ as a product of the random feature matrix of the entire state space $\boldsymbol{\Sigma}_{\mathcal{S}} \in \mathbb{R}^{N \times |\mathcal{S}|}$ with $\boldsymbol{U}_n \in \mathbb{R}^{|\mathcal{S}| \times n}$ and $\boldsymbol{V}_n \in \mathbb{R}^{|\mathcal{S}| \times n}$ instead of $\boldsymbol{\Sigma}_{\hat{\mathcal{S}}}, \hat{\boldsymbol{U}}_n, \hat{\boldsymbol{V}}_n$. We obtain a decomposition of the transition model matrix $\boldsymbol{A}_n = \boldsymbol{U}_n(\boldsymbol{U}_n - \gamma\boldsymbol{V}_n)^T$. $\boldsymbol{A}_n$ was used by Boyan (1999) to interpret LSTD as model-based RL. Tsitsiklis & Van Roy (1996) and Nedić & Bertsekas (2003) showed that $\mathbb{E}[\boldsymbol{A}_n] \to \boldsymbol{D}_{\boldsymbol{\pi}}[\boldsymbol{I}_{|\mathcal{S}|} - \gamma\boldsymbol{P}]$ as $n \to \infty$. The bound on the difference $\|\boldsymbol{A}_n - \boldsymbol{D}_{\boldsymbol{\pi}}[\boldsymbol{I}_{|\mathcal{S}|} - \gamma\boldsymbol{P}]\|$ as a function of $n$ was studied by Tagorti & Scherrer (2015). We make the following assumption on this norm:

**Assumption 3.** *As* $n, m \to \infty$, *we have* $\|\boldsymbol{A}_n - \boldsymbol{D}_{\boldsymbol{\pi}}[\boldsymbol{I}_{|\mathcal{S}|} - \gamma\boldsymbol{P}]\| = \mathcal{O}\left(\frac{1}{\sqrt{m}}\right)$.

Using an approach similar to that of Theorem 5.2, plus a detailed analysis of operator norms, we obtain the following deterministic form of the asymptotic MSBE (proof in Appendix G):

**Theorem 5.3** (Asymptotic MSBE). *Under Assumptions 1, 2, and 3, the* *deterministic asymptotic MSBE* *is* $\overline{\mathrm{MSBE}}(\hat{\boldsymbol{\theta}}) = \left\|\bar{\boldsymbol{r}} + \gamma\frac{1}{\sqrt{n}}\frac{N}{m}\frac{1}{1+\delta}\boldsymbol{P}\boldsymbol{\Phi}_{\mathcal{S}}\boldsymbol{U}_n\bar{\boldsymbol{Q}}_m(\lambda)\boldsymbol{r} - \frac{1}{\sqrt{n}}\frac{N}{m}\frac{1}{1+\delta}\boldsymbol{\Phi}_{\mathcal{S}}\boldsymbol{U}_n\bar{\boldsymbol{Q}}_m(\lambda)\boldsymbol{r}\right\|^2_{\boldsymbol{D}_{\boldsymbol{\pi}}} + \Delta$, *with second-order correction factor*

$$\Delta = \frac{1}{n}\frac{\frac{1}{N}\operatorname{Tr}\big(\boldsymbol{\Lambda}_{\boldsymbol{P}}\big[\boldsymbol{\Theta}_{\mathcal{S}}\boldsymbol{\Psi}_2\boldsymbol{\Theta}_{\mathcal{S}}^T - 2\boldsymbol{\Theta}_{\mathcal{S}}(\boldsymbol{U}_n - \gamma\boldsymbol{V}_n)^T\boldsymbol{\Psi}_{\mathcal{S}} + \boldsymbol{\Psi}_{\mathcal{S}}\big]\big)}{1-\frac{1}{N}\operatorname{Tr}\big(\boldsymbol{\Psi}_2\bar{\boldsymbol{Q}}_m(\lambda)^T\boldsymbol{\Psi}_1\bar{\boldsymbol{Q}}_m(\lambda)\big)}\|\bar{\boldsymbol{Q}}_m(\lambda)\boldsymbol{r}\|^2_{\boldsymbol{\Psi}_1}, \quad \text{where} \tag{20}$$

$$\boldsymbol{\Psi}_{\mathcal{S}} = \frac{N}{m}\frac{1}{1+\delta}\boldsymbol{\Phi}_{\mathcal{S}}, \quad \boldsymbol{\Lambda}_{\boldsymbol{P}} = [\boldsymbol{I}_{|\mathcal{S}|} - \gamma\boldsymbol{P}]^T\boldsymbol{D}_{\boldsymbol{\pi}}[\boldsymbol{I}_{|\mathcal{S}|} - \gamma\boldsymbol{P}], \quad \text{and} \quad \boldsymbol{\Theta}_{\mathcal{S}} = \boldsymbol{\Psi}_{\mathcal{S}}\boldsymbol{U}_n\bar{\boldsymbol{Q}}_m(\lambda). \tag{21}$$

*As* $N, m, d \to \infty$ *with asymptotic constant ratio* $N/m$, $\mathrm{MSBE}(\hat{\boldsymbol{\theta}}) - \overline{\mathrm{MSBE}}(\hat{\boldsymbol{\theta}}) \xrightarrow{a.s} 0$.

**Remark 6.** *Like the empirical* $\overline{\widehat{\mathrm{MSBE}}}(\hat{\boldsymbol{\theta}})$ *in Theorem 5.2, the true* $\overline{\mathrm{MSBE}}(\hat{\boldsymbol{\theta}})$ *is also influenced by the correction terms* $\delta$ *and* $\Delta$. *Note that in asymptotic regimes where* $N/m \to \infty$ *or* $\lambda \to \infty$, *the correction terms vanish. When* $N/m \to \infty$, $\overline{\mathrm{MSBE}}(\hat{\boldsymbol{\theta}})$ *is independent of* $\lambda$ *as shown in details in Appendix C.*

**Remark 7.** *When all states have been visited, the common subexpressions in the second-order correction factors* $\hat{\Delta}$ *and* $\Delta$ *dominate so that* $\hat{\Delta}$, $\Delta$ *become similar (for a proof, see Lemma G.8).*

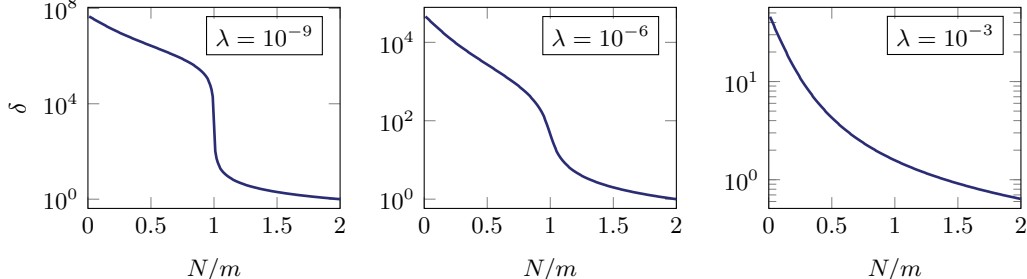

Figure 2: **The correction factor $\delta$ is a decreasing function of the number of parameters $N$. For a small $l_2$-regularization parameter $\lambda$, we observe a sharp decrease near the interpolation threshold ($N = m$ for $m$ distinct visited states). As $\lambda$ increases, the function becomes smoother and smaller (note the different scales of the y-axis).** $\delta$ is computed with equation 17 on Taxi-v3 with $\gamma = 0.95, m = 310, n = 5000$.

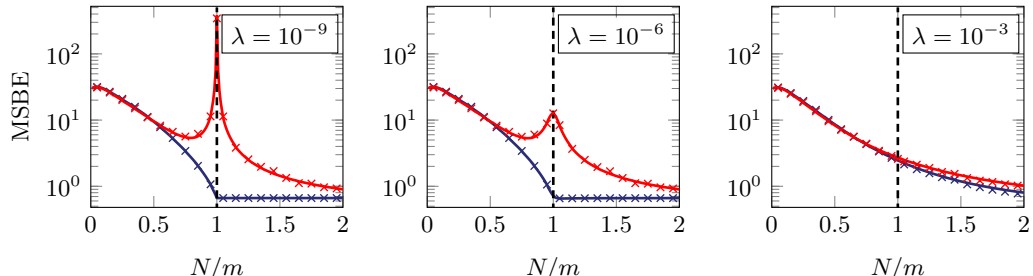

Figure 3: **The double descent phenomenon occurs in the true MSBE (red) of regularized LSTD, peaking around the interpolation threshold ($N = m$ for $N$ parameters, $m$ distinct visited states) when the empirical $\widehat{\text{MSBE}}$ (blue) vanishes. It diminishes as the $l_2$-regularization parameter $\lambda$ increases.** Continuous lines indicate the theoretical values from Theorem 5.2 and Theorem 5.3, the crosses are numerical results averaged over 30 instances after the learning with regularized LSTD in Taxi-v3 with $\gamma = 0.95, m = 310, n = 5000$.

## 6 NUMERICAL EXPERIMENTS

In this section, we provide an empirical evaluation, including a discussion of the behavior of the correction factor $\delta$ from Theorem 5.1, and its impact on the empirical and true MSBE from Theorem 5.2 and Theorem 5.3. Additional experiments can be found in Appendix A.

**Experimental Setup.** We use the recursive regularized LSTD implementation of Dann et al. (2014) on three MRPs: a synthetic ergodic MRP (500 states); a gridworld MRP (400 states) obtained from a random policy in a $20 \times 20$ gridworld (Ahmed, 2018); and a Taxi-v3 MRP (356 states) obtained from a learned policy in OpenAI gym Taxi-v3 (Towers et al., 2023) (Figure 1a). In all cases, states are described by $d$-Gaussian vectors where $d = 50$. For the random features, $\boldsymbol{W}$ is drawn from a Gaussian distribution and $\sigma(\cdot) = \max(0, \cdot)$ is the ReLU function. For all experiments, $\mathcal{D}_{\text{train}} := \{(\boldsymbol{s}_i, r_i, \boldsymbol{s}'_i)\}_{i=1}^n$ is derived from a sample path of $n$ transitions with the same seed (42). For each instance $i$, we sample random features using the seed $i$. The following graphs show averages over 30 instances.

**Correction Factor $\delta$ vs. Model Complexity.** The correction factor $\delta$ (equation 17) plays a key role in the asymptotic $\widehat{\text{MSBE}}$ and MSBE. Figure 2 shows $\delta$ as a function of the model complexity $N/m$ and for different values of the regularization parameter $\lambda$. It confirms that, as stated in Theorem 5.1, $\delta$ is a decreasing function of $N/m$. For a small $\lambda$, we observe a sharp decrease at the interpolation threshold ($N = m$). E.g., for $\lambda = 10^{-9}$, $\delta$ falls from an order of $10^7$ in under-parameterized models ($N < m$) to an order of $10^1$ in over-parameterized models ($N > m$). For larger values of $\lambda$, $\delta$ decreases more smoothly and has smaller values. Further experiments on the behavior of $\delta$ and experiments for other environments are provided in Appendix A.

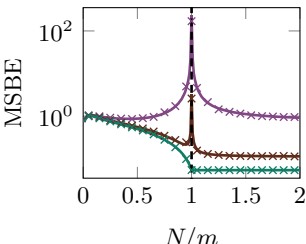
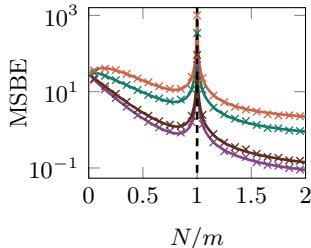

Figure 4: **With more distinct states $m$ visited, the double descent in the MSBE diminishes, disappearing for $m = |\mathcal{S}|$.** MSBE from Theorem 5.3 (lines) and avg. numerical results over 30 instances (crosses) in a synthetic ergodic MRP for $m = 0.86|\mathcal{S}|$ (purple), $m = 0.998|\mathcal{S}|$ (maroon), and $m = |\mathcal{S}|$ (green) with $\gamma = 0.95, s = |\mathcal{S}|, n = 3000$.

Figure 5: **The discount factor $\gamma$ has little effect on the double descent in the MSBE.** Results in the Gridworld MRP for $\gamma = 0$ (purple), $\gamma = 0.5$ (maroon), $\gamma = 0.95$ (green), and $\gamma = 0.99$ (orange) with $m = 386, n = 5000$. MSBE from Theorem 5.3 (lines) and avg. numerical results over 30 instances (crosses).

**Double Descent Behavior.** As a consequence of the sharp transition of the correction factor $\delta$ for small $l_2$-regularization parameters, as discussed above, Theorem 5.2 and Theorem 5.3 predict a change in behavior of the empirical $\widehat{\text{MSBE}}$ and true MSBE between the under- and overparameterized regimes. Figure 3 shows both $\widehat{\text{MSBE}}$ and MSBE as a function of the model complexity $N/m$ with different $l_2$-regularization penalties $\lambda$ in Taxi-v3. Despite the fact that the equations for $\widehat{\text{MSBE}}$ and MSBE were derived for the asymptotic case $N \to \infty$, we observe an almost perfect match with the numerically evaluated original definitions in equation 8 and equation 2. For small $\lambda$, the true MSBE exhibits a peak around the interpolation threshold $N = m$, leading to a double descent phenomenon. In contrast, the empirical $\widehat{\text{MSBE}}$ is close to its minimum at $N = m$ and almost constant for $N \geq m$, so no double descent is observed. While for the Taxi-v3, the empirical $\widehat{\text{MSBE}}$ is smaller than the true MSBE, this is not necessarily the case in other environments, where the empirical $\widehat{\text{MSBE}}$ can be larger overall than the true MSBE (see further experiments in Appendix A). For larger $\lambda$, the double descent in the true MSBE disappears and the difference between the true MSBE and the empirical $\widehat{\text{MSBE}}$ is less pronounced, although it may not vanish. Appendix B shows a similar double descent phenomenon for the Mean-Squared Value Error. All the above observations are in accordance with established results in supervised learning (Liao et al., 2020).

**Impact of the Number of Unvisited States and the Discount Factor $\gamma$.** Once all states have been visited, MSBE and $\widehat{\text{MSBE}}$ have similar behavior, with no peak at the interpolation threshold ($N = m$), see also Remark 7. The experiments in Figure 4 depict this behavior. They also illustrate that the double descent phenomenon diminishes as the number of distinct unvisited states goes to zero. The experiments in Figure 5 illustrate that the discount factor $\gamma$ has little impact on the double descent phenomenon.

## 7 CONCLUSION

In this work, we have analyzed the performance of regularized LSTD with random features in a novel double asymptotic regime, where the number of parameters $N$ and distinct visited states $m$ go to infinity with a constant ratio. We have established deterministic limit forms for both the empirical MSBE and true MSBE that feature correction terms. We have observed that these correction terms are responsible for a double descent phenomenon in the true MSBE, similar to supervised learning, resulting in a sudden drop in performance for $N = m$. The correction terms vanish, and so does the double descent phenomenon when the $l_2$-regularization is increased or the number of unvisited states goes to zero. Directions for future work include a study of the off-policy setting, extending our results beyond one hidden layer to deep neural networks, and going beyond policy evaluation in order to investigate other RL algorithms, such as Q-Learning.

# 8 ACKNOWLEDGEMENTS

This work was supported by Ecole Polytechnique. We also thank Zhenyu Liao for his helpful dis-´
cussions on this work.

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

# A ADDITIONAL EXPERIMENTS

This appendix shows additional empirical results which cannot be put in the main body due to space limitations.

## A.1 $\delta$ IN THE DOUBLE ASYMPTOTIC REGIME

Like Figure 2 in Section 6, Figure 6 depicts the correction factor $\delta$ (equation 17) as a function of the complexity model $N/m$ in synthetic ergodic and Gridworld MRPs. $\delta$ shows a similar behavior than for the one observed in Taxi-v3 in Figure 2.

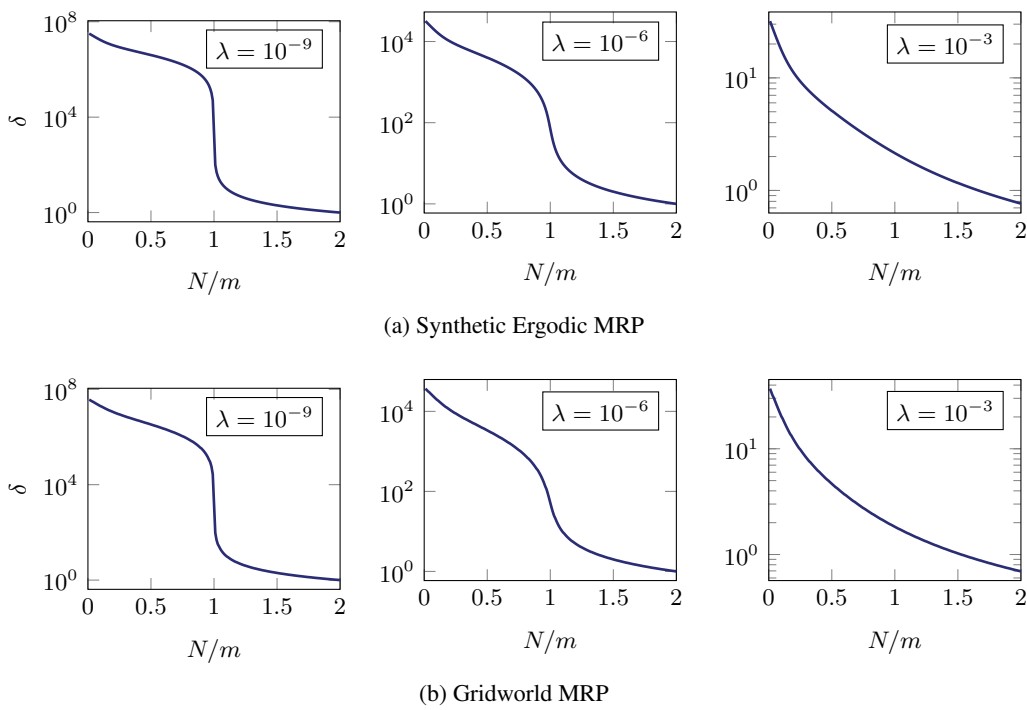

Figure 6: **The correction factor $\delta$ is a decreasing function of the number of parameters $N$. For small $l_2$-regularization parameter $\lambda$, we observe a sharp decrease near the interpolation threshold ($N = m$, for $m$ distinct visited states). As $\lambda$ increases, the function becomes smoother and smaller (note the different scales of the y-axis).** $\delta$ is computed with equation 17 in synthetic ergodic and Girdworld MRPs with $\gamma = 0.95, m = 499, n = 3000$ and $\gamma = 0.95, m = 386, n = 5000$, respectively.

Figure 7 depicts $\delta$ as a function of the $l_2$-regularization parameter for different ratio $N/m$. It confirms $\delta$ decreases monotonically as the $l_2$-regularization parameter $\lambda$ increases, as stated by Lemma J.4. Furthermore, we observe the impact of regularization parameter $\lambda$ becomes less significant as the model complexity $N/m$ increases. Indeed, as $N/m$ increases, we observe a larger initially flat region and smaller values of $\delta$.

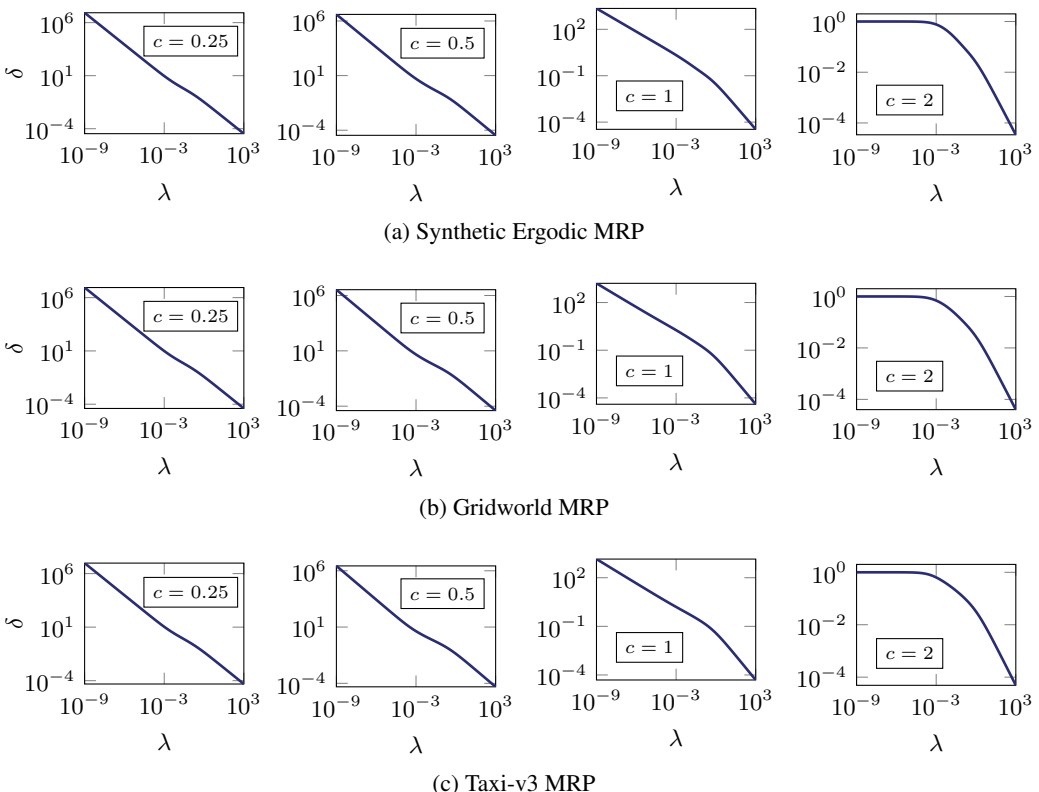

(a) Synthetic Ergodic MRP

(b) Gridworld MRP

(c) Taxi-v3 MRP

Figure 7: **The correction factor $\delta$ is a decreasing function of the $l_2$-regularization parameter $\lambda$. As the model complexity $c = N/m$ increases, the impact of regularization parameter $\lambda$ becomes less significant (note the different scales of the y-axis).** $\delta$ is computed with equation 17 in synthetic ergodic, Girdworld and Taxi-v3 MRPs with $\gamma = 0.95, m = 499, n = 3000, \gamma = 0.95, m = 386, n = 5000$ and $\gamma = 0.95, m = 310, n = 5000$, respectively.

## A.2 DOUBLE DESCENT BEHAVIOR

Figure 8 shows both $\widehat{\text{MSBE}}$ and MSBE as a function of the model complexity $N/m$ with different $l_2$-regularization penalties $\lambda$, in synthetic ergodic and Gridworld MRPs. Both MSBE and $\widehat{\text{MSBE}}$ depict a similar double descent behavior for small $\lambda$ than in Figure 3 in Section 6. We observe the empirical $\widehat{\text{MSBE}}$ is not necessarily lower for over-parameterized ($N > m$) models.

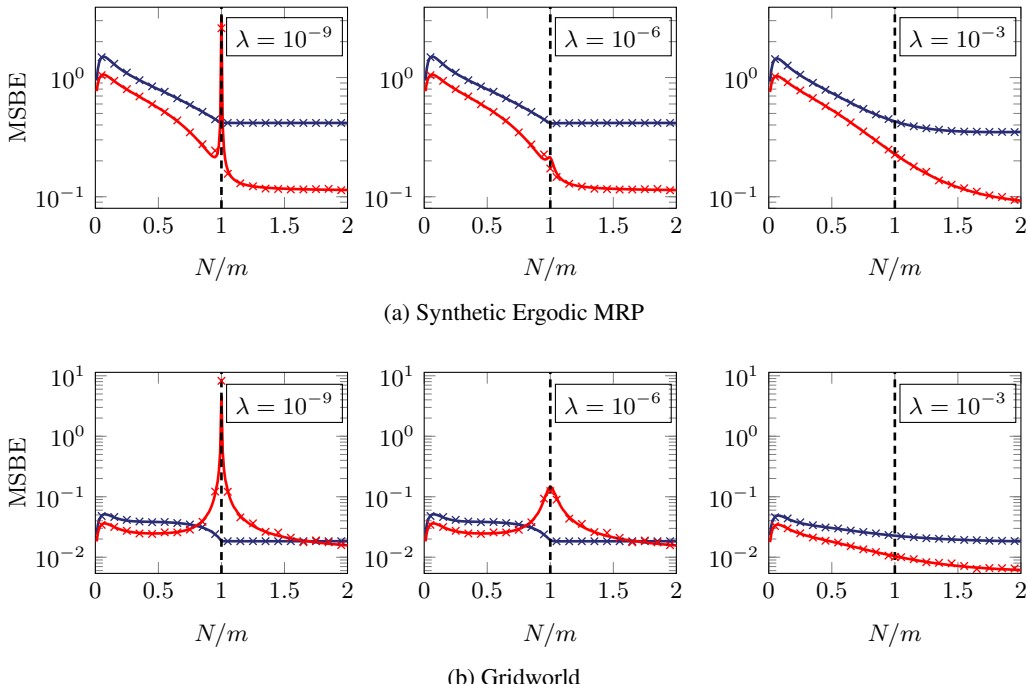

(a) Synthetic Ergodic MRP

(b) Gridworld

Figure 8: **The double descent phenomenon occurs in the true** MSBE **(red) of regularized LSTD, peaking around the interpolation threshold ($N = m$ for $N$ parameters, $m$ distinct visited states) when the empirical** $\widehat{\text{MSBE}}$ **(blue) vanishes. It diminishes as the $l_2$-regularization parameter $\lambda$ increases.** Continuous lines indicate the theoretical values from Theorem 5.2 and Theorem 5.3, the crosses are numerical results averaged over 30 instances after the learning with regularized LSTD in synthetic ergodic and Gridworld MRPs with $\gamma = 0.95, d = 50, n = 3000$ and $n = 5000$, respectively.

### A.3 IMPACT OF THE NUMBER OF UNVISITED STATES AND OF THE DISCOUNT FACTOR $\gamma$

Like Figure 4, Figure 9 depicts the behavior of the true MSBE for different numbers of distinct visited states $m$ and shows that as the number of distinct unvisited states goes to zero, the double descent phenomenon diminishes.

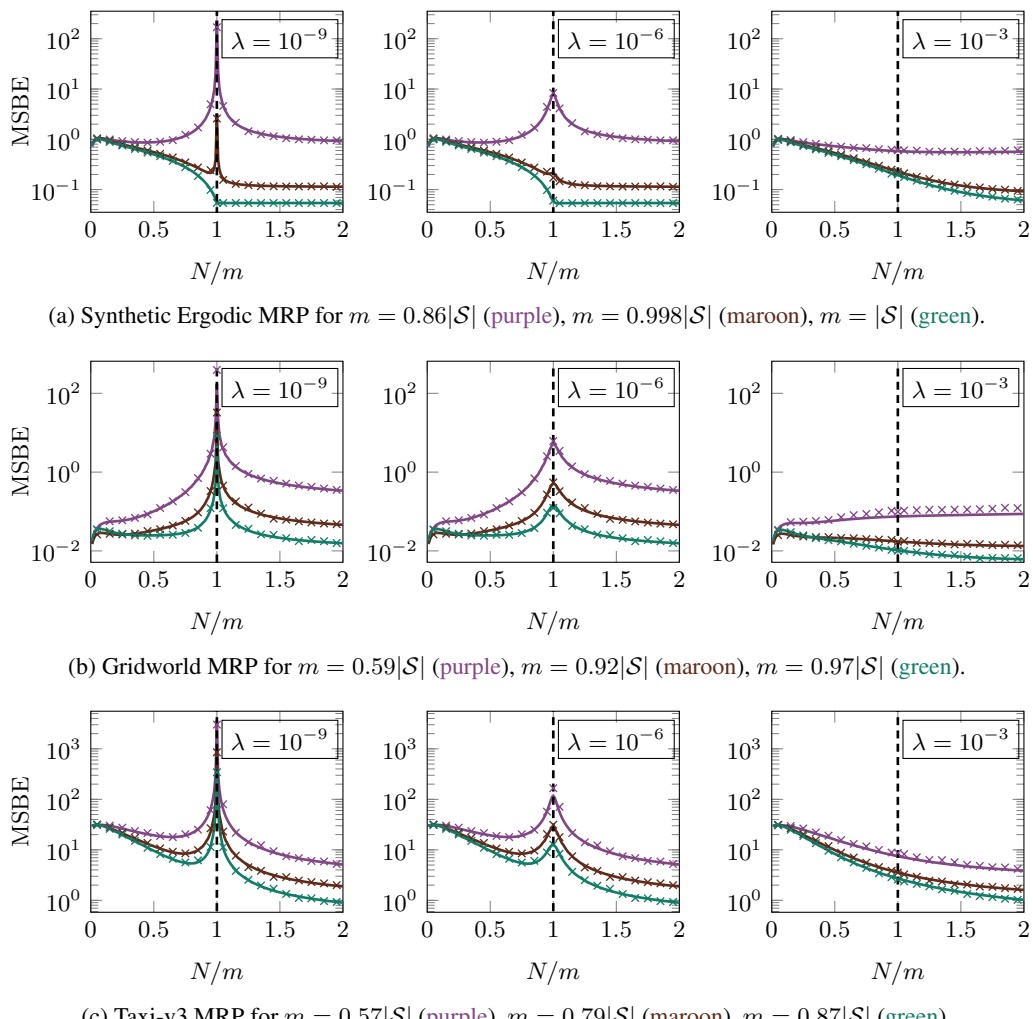

(a) Synthetic Ergodic MRP for $m = 0.86|\mathcal{S}|$ (purple), $m = 0.998|\mathcal{S}|$ (maroon), $m = |\mathcal{S}|$ (green).

(b) Gridworld MRP for $m = 0.59|\mathcal{S}|$ (purple), $m = 0.92|\mathcal{S}|$ (maroon), $m = 0.97|\mathcal{S}|$ (green).

(c) Taxi-v3 MRP for $m = 0.57|\mathcal{S}|$ (purple), $m = 0.79|\mathcal{S}|$ (maroon), $m = 0.87|\mathcal{S}|$ (green).

Figure 9: **With more distinct states m visited, the double descent in the MSBE diminishes, disappearing for $m = |\mathcal{S}|$.** Continuous lines indicate the theoretical values of MSBE from Theorem 5.3 for different numbers of distinct visited states $m$; the crosses are numerical results averaged over 30 instances after the learning with regularized LSTD in synthetic ergodic, Gridworld and Taxi-v3 MRPs with $\gamma = 0.95, d = 50$.

Figure 10 describes the impact of the discount factor $\gamma$ on the double descent phenomenon, and shows it remains true for all $\gamma$.

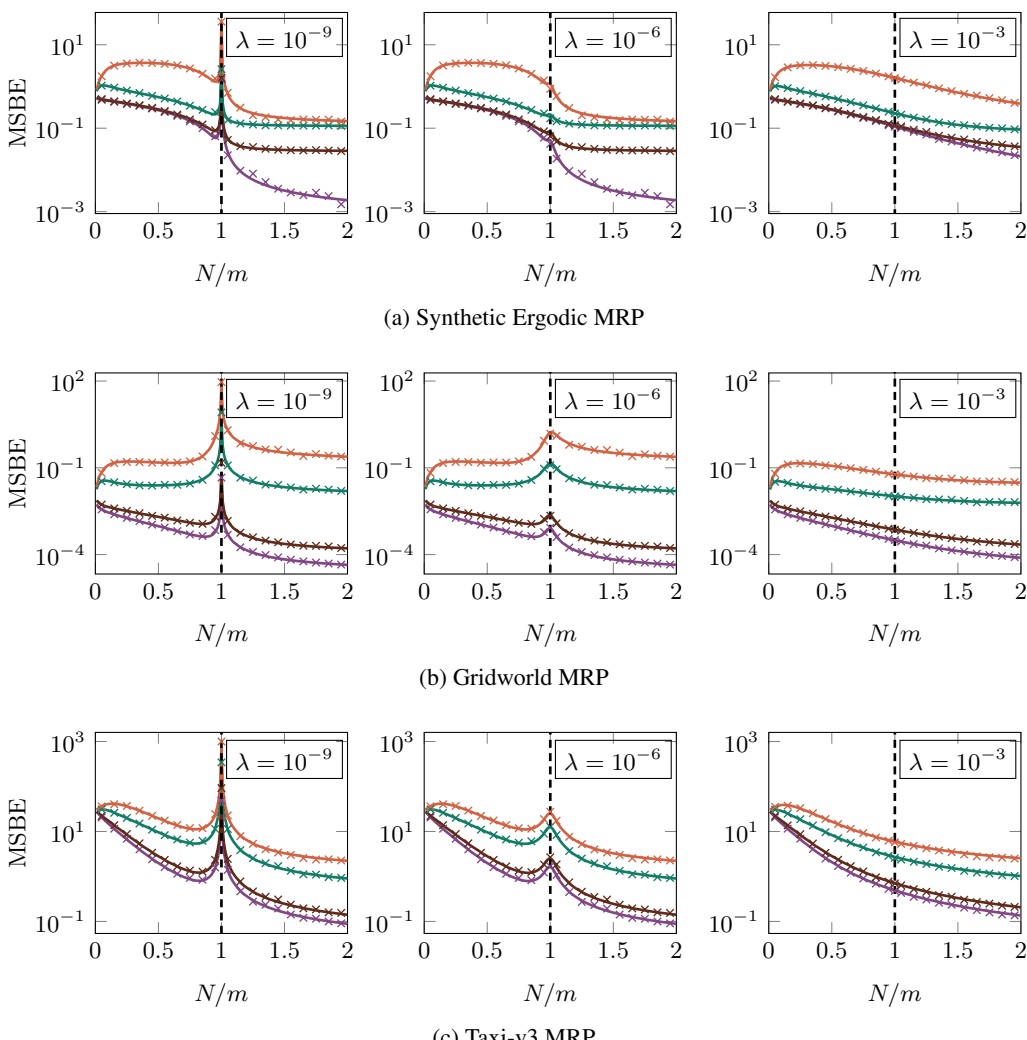

(a) Synthetic Ergodic MRP

(b) Gridworld MRP

(c) Taxi-v3 MRP

Figure 10: **The discount factor $\gamma$ has little effect on the double descent in the MSBE.** Continuous lines indicate the theoretical values of MSBE from Theorem 5.3 for $\gamma = 0$ (purple), $\gamma = 0.5$ (maroon), $\gamma = 0.95$ (green), and $\gamma = 0.99$ (orange); the crosses are numerical results averaged over 30 instances after the learning with regularized LSTD in synthetic ergodic, Gridworld and Taxi-v3 MRPs for $d = 50, n = 3000$, $d = 50, n = 5000$ and $d = 50, n = 5000$, respectively.

## A.4 IMPACT OF THE $l_2$-REGULARIZATION PARAMETER ON THE MSBE

Figure 11 and Figure 12 depict the empirical $\widehat{\mathrm{MSBE}}$ and the true MSBE as a function of the $l_2$-regularization parameter. In supervised learning, the training error is an increasing function of the $l_2$-regularization parameter $\lambda$ (Liao et al., 2020), whereas the discount factor induces a more intricated behavior in RL as described by Figure 11.

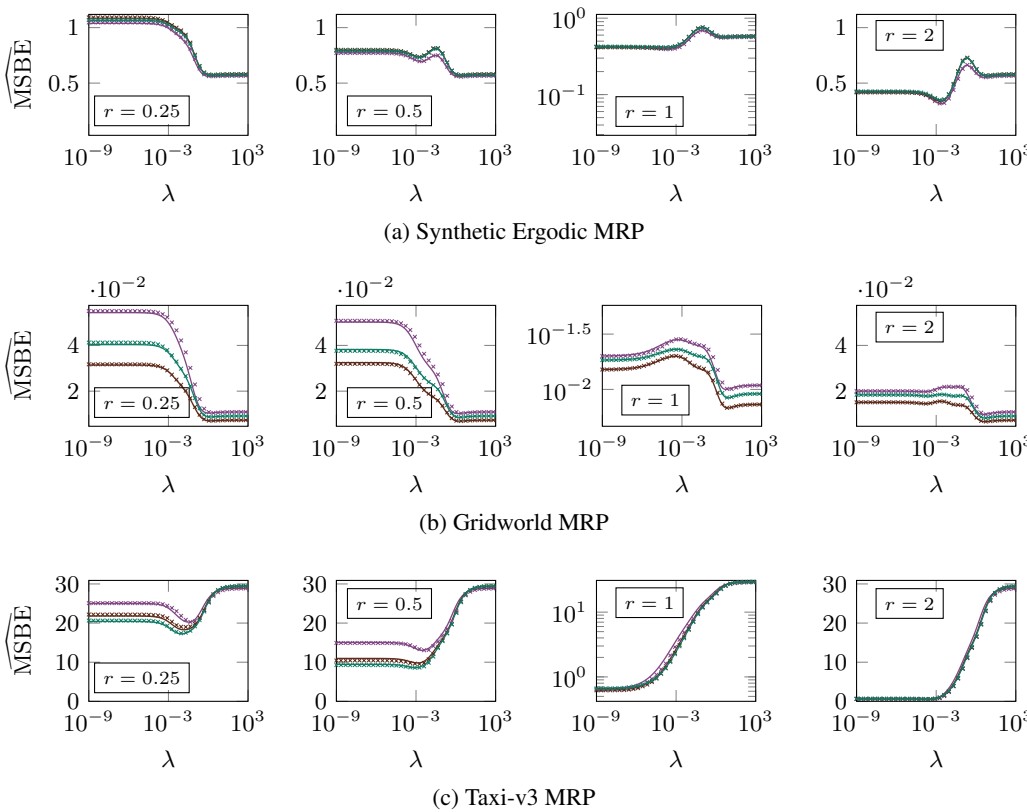

(a) Synthetic Ergodic MRP

(b) Gridworld MRP

(c) Taxi-v3 MRP

Figure 11: $\widehat{\mathrm{MSBE}}$ **is not an increasing function of the $l_2$-regularization parameter $\lambda$.** Continuous lines indicate the theoretical values of $\widehat{\mathrm{MSBE}}$ from Theorem 5.2 for $n = 1000$ (purple), $n = 3000$ (maroon), and $n = 5000$ (green); the crosses are numerical results averaged over 30 instances after the learning with regularized LSTD in synthetic ergodic, Gridworld and Taxi-v3 MRPs for $d = 50, n = 3000$, $d = 50, n = 5000$ and $d = 50, n = 5000$, respectively. Note that the y-axis has a logarithmic scale for $r = N/m = 1$.

In Figure 12 at the interpolation threshold ($N = m$), we observe that as $\lambda$ increases, MSBE decreases and so does the peak observed in the previous experiments. For other ratios, the true MSBE depicts complex behaviors that may differ between under- and over-parameterized models and depends on the environment. Yet, both empirical and true MSBE show similar and opposite trends for the same $\lambda = \frac{\lambda_{m,n}}{mn}$, regardless of the number of samples $n$ and distinct visited states $m$. In practice, the effective $l_2$-regularization parameter $\lambda_{m,n}$ is tuned. This suggests that it depends on both the number of transitions collected $n$ and the number of distinct visited states $m$, and not just on the number of samples $n$ as it is commonly suggested (Hoffman et al., 2011; Chen et al., 2013).

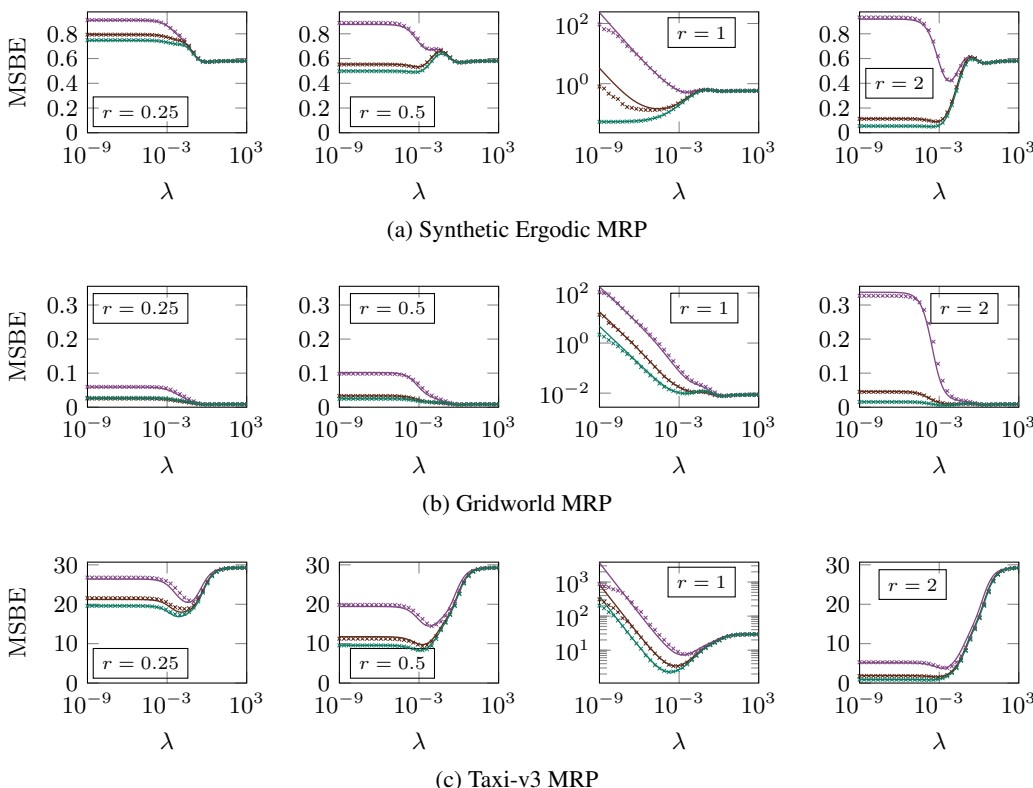

(a) Synthetic Ergodic MRP

(b) Gridworld MRP

(c) Taxi-v3 MRP

Figure 12: **At the interpolation threshold ($r = N/m = 1$), as $\lambda$ increases,** MSBE **decreases and so does the double descent phenomenon.** Continuous lines indicate the theoretical values of MSBE from Theorem 5.3 for $n = 1000$ (purple), $n = 3000$ (maroon), and $n = 5000$ (green); the crosses are numerical results averaged over 30 instances after the learning with regularized LSTD in synthetic ergodic, Gridworld and Taxi-v3 MRPs for $d = 50, n = 3000$, $d = 50, n = 5000$ and $d = 50, n = 5000$, respectively. Note that the y-axis has a logarithmic scale for $r = N/m = 1$.

A.5   IMPACT OF THE SECOND-ORDER CORRECTION FACTOR $\Delta$ IN THE TRUE MSBE

Figure 13 depicts $\Delta$ in Theorem 5.3 as a function of the model complexity $N/m$ for small $l_2$-regularization parameter ($\lambda = 10^{-9}$). It shows the double descent phenomenon in the true MSBE is mainly due to the second-order correction term $\Delta$.

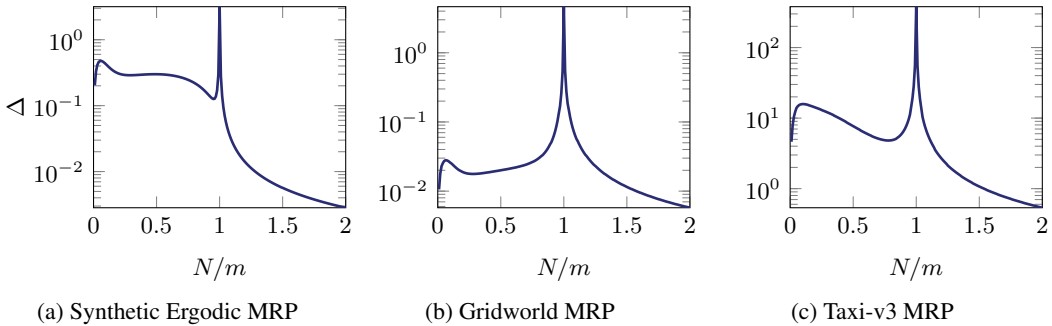

| (a) Synthetic Ergodic MRP | (b) Gridworld MRP | (c) Taxi-v3 MRP |

Figure 13: **The double descent phenomenon in the true** MSBE **is mainly due to the second-order correction term** $\Delta$. $\Delta$ is computed with equation 20 in synthetic ergodic, Girdworld, and Taxi-v3 MRPs with $\lambda = 10^{-9}, \gamma = 0.95, m = 499, n = 3000, \lambda = 10^{-9}, \gamma = 0.95, m = 386, n = 5000$ and $\lambda = 10^{-9}, \gamma = 0.95, m = 310, n = 5000$, respectively.

## B   MEAN-SQUARED VALUE ERROR

In this section, we study the Mean-Squared Value Error (MSVE) and observes a similar double descent behavior than the one observed for the true MSBE. The MSVE is defined as

$$\text{MSVE}(\hat{\boldsymbol{\theta}}) = \left\| \boldsymbol{V} - \boldsymbol{\Sigma}_{\mathcal{S}}^T \hat{\boldsymbol{\theta}} \right\|_{\boldsymbol{D}_{\boldsymbol{\pi}}}^2. \tag{22}$$

Using a similar approach than for Theorem 5.3, we obtain the following deterministic form of the asymptotic MSVE:

**Corollary B.0.1** (Asymptotic MSVE). *Under Assumptions 1, 2, and 3, the deterministic asymptotic MSVE is* $\overline{\text{MSVE}}(\hat{\boldsymbol{\theta}}) = \left\| \boldsymbol{V} - \frac{1}{\sqrt{n}} \frac{N}{m} \frac{1}{1+\delta} \boldsymbol{\Phi}_{\mathcal{S}} \boldsymbol{U}_n \bar{\boldsymbol{Q}}_m(\lambda) \boldsymbol{r} \right\|_{\boldsymbol{D}_{\boldsymbol{\pi}}}^2 + \Delta'$, *with second-order correction factor*

$$\Delta' = \frac{1}{n} \frac{\frac{1}{N} \text{Tr}\left( \boldsymbol{D}_{\boldsymbol{\pi}} \left[ \boldsymbol{\Theta}_{\mathcal{S}} \boldsymbol{\Psi}_2 \boldsymbol{\Theta}_{\mathcal{S}}^T - 2\boldsymbol{\Theta}_{\mathcal{S}} (\boldsymbol{U}_n - \gamma \boldsymbol{V}_n)^T \boldsymbol{\Psi}_{\mathcal{S}} + \boldsymbol{\Psi}_{\mathcal{S}} \right] \right)}{1 - \frac{1}{N} \text{Tr}\left( \boldsymbol{\Psi}_2 \bar{\boldsymbol{Q}}_m(\lambda)^T \boldsymbol{\Psi}_1 \bar{\boldsymbol{Q}}_m(\lambda) \right)} \| \bar{\boldsymbol{Q}}_m(\lambda) \boldsymbol{r} \|_{\boldsymbol{\Psi}_1}^2. \tag{23}$$

*As $N, m, d \to \infty$ with asymptotic constant ratio $N/m$, $\text{MSVE}(\hat{\boldsymbol{\theta}}) - \overline{\text{MSVE}}(\hat{\boldsymbol{\theta}}) \xrightarrow{a.s} 0$.*

*Proof.* Using $\boldsymbol{D}_{\boldsymbol{\pi}} = [\boldsymbol{I}_m - \gamma \boldsymbol{P}]^T \boldsymbol{D}_{\boldsymbol{\pi}} \boldsymbol{D}_{\boldsymbol{\pi}}^{-1} [\boldsymbol{I}_m - \gamma \boldsymbol{P}]^{-1T} \boldsymbol{D}_{\boldsymbol{\pi}} [\boldsymbol{I}_m - \gamma \boldsymbol{P}]^{-1} \boldsymbol{D}_{\boldsymbol{\pi}}^{-1} \boldsymbol{D}_{\boldsymbol{\pi}} [\boldsymbol{I}_m - \gamma \boldsymbol{P}]$ and a with similar proof than for Theorem 5.3, we find Corollary B.0.1.                              □

**Remark 8.** *Like $\overline{\text{MSBE}}(\hat{\boldsymbol{\theta}})$ in Theorem 5.3, the $\overline{\text{MSVE}}(\hat{\boldsymbol{\theta}})$ is also influenced by the correction terms $\delta$ and $\Delta'$. Note that in asymptotic regimes where $N/m \to \infty$ or $\lambda \to \infty$, the correction terms vanish. When $N/m \to \infty$, $\overline{\text{MSVE}}(\hat{\boldsymbol{\theta}})$ is independent of $\lambda$ as shown in details in Appendix C.*

Figure 14 shows both the empirical $\widehat{\text{MSVE}}$ and the true MSVE as a function of the model complexity $N/m$ with different $l_2$-regularization penalties $\lambda$ in the synthetic ergodic, Girdworld and Taxi MRPs. Like the true MSBE, we observe an almost perfect match with the numerically evaluated original definition in equation 22.

For small $\lambda$, like for the true MSBE, the true MSVE exhibits a peak around the interpolation threshold $N = m$, leading to a double descent phenomenon. In contrast, the empirical $\widehat{\text{MSVE}}$ is close to its minimum at $N = m$ and almost constant for $N \geq m$, so no double descent is observed. Unlike than for the true MSBE, we observe that the empirical $\widehat{\text{MSVE}}$ is always smaller than the true

MSVE. For larger $\lambda$, the double descent in the true MSVE disappears and the difference between the true MSVE and the empirical $\widehat{\text{MSVE}}$ is less pronounced, although it may not vanish.

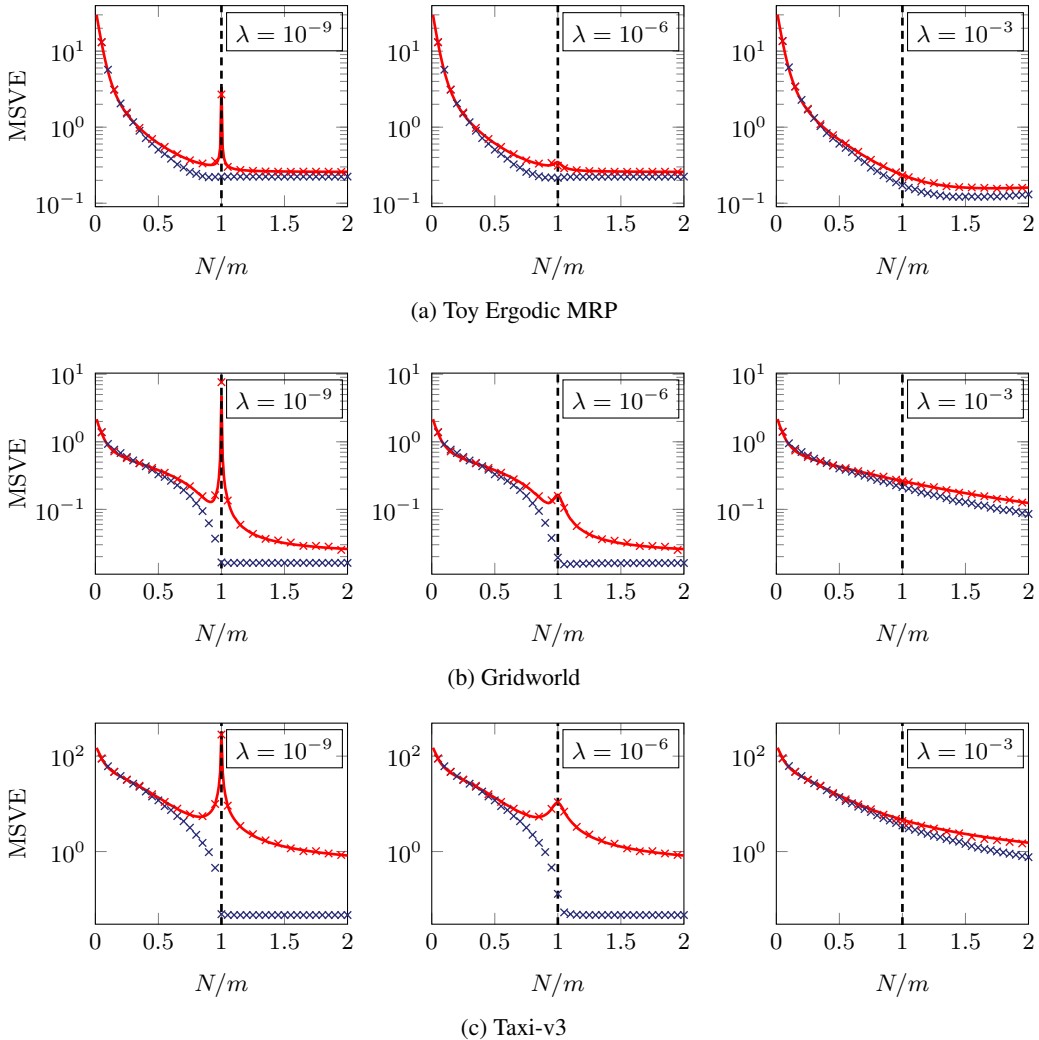

(a) Toy Ergodic MRP

(b) Gridworld

(c) Taxi-v3

Figure 14: **The double descent phenomenon occurs in the true** MSVE **(red) of regularized LSTD, peaking around the interpolation threshold ($N = m$ for $N$ parameters, $m$ distinct visited states) when the empirical $\widehat{\text{MSVE}}$ (blue) vanishes. It diminishes as the $l_2$-regularization parameter $\lambda$ increases.** Continuous lines indicate the theoretical values from Corollary B.0.1, the crosses are numerical results averaged over 30 instances after the learning with regularized LSTD in synthetic ergodic, Gridworld and Taxi MRPs with $\gamma = 0.95, d = 50, n = 3000, n = 5000$ and $n = 5000$, respectively.

## C  REFORMULATION OF THE MAIN RESULTS

Let $\mathcal{C}$ be a compact set such that $\mathcal{S} \subset \mathcal{C}$. We extend the stationary distribution $\pi$ on $\mathcal{C}$ by setting $\pi(s) = 0$ for $s \in \mathcal{S}\backslash\mathcal{C}$. We denote by $L^2(\mathcal{C}, \pi)$ the set of squared integrable functions $f : \mathcal{C} \to \mathbb{R}$ with respect to the distribution $\pi$ on $\mathcal{C}$, and the norm on $L^2(\mathcal{C}, \pi)$ defined as $\|f\|^2_{L^2(\mathcal{C}, \pi)} = \langle f, f \rangle = \int f(x)^2 \pi(dx)$. The Frobenius norm of a matrix $\boldsymbol{A}$ is denoted as $\|\boldsymbol{A}\|_F = \sqrt{\langle \boldsymbol{A}, \boldsymbol{A} \rangle_F} = \sqrt{\text{Tr}(\boldsymbol{A}^T\boldsymbol{A})}$.

In this section, we aim to reformulate the results from Section 5 in a feature space derived from the eigendecomposition of $\boldsymbol{\Phi}_{\mathcal{S}}$. The section is organized into three subsections: in Section C.1 we introduce the new asymptotic feature space, we reformulate the results of Section 5 in the new feature space in Section C.2 and we provide the proofs for these results in Section C.3.

## C.1 ASYMPTOTIC FEATURE SPACE

**Asymptotic Feature Space.** Both Theorem 5.2 and Theorem 5.3 use either the deterministic Gram feature matrix $\boldsymbol{\Phi}_{\hat{\mathcal{S}}}$ (defined in equation 16) or the deterministic Gram feature matrix $\boldsymbol{\Phi}_{\mathcal{S}}$ that are given by the continuous kernel function $\Phi : \mathcal{C} \times \mathcal{C} \to \mathbb{R}$ defined as

$$\Phi(\boldsymbol{s}, \boldsymbol{s}') = \mathbb{E}_{\boldsymbol{w} \sim \mathcal{N}(\boldsymbol{0}, \boldsymbol{I}_d)} \left[ \sigma(\boldsymbol{w}^T \boldsymbol{s})^T \sigma(\boldsymbol{w}^T \boldsymbol{s}') \right].$$

Since $\Phi$ is continuous and $\mathcal{C}$ is compact, the Mercer's theorem states (Schölkopf & Smola, 2002) that

$$\Phi(\boldsymbol{s}, \boldsymbol{s}') = \sum_{i=1}^{M} \nu_i \varphi_i(\boldsymbol{s}) \varphi_i(\boldsymbol{s}') = \sum_{i=1}^{M} \omega_i(\boldsymbol{s}) \omega_i(\boldsymbol{s}'); \tag{24}$$

where $\{\nu_i\}_{i=1}^{M}$ and $\{\varphi_i\}_{i=1}^{M}$ are the eigenvalues and eigenfunctions of the Hilbert-Schmidt integral operators $T_\Phi : L^2(\mathcal{C}, \pi) \to L^2(\mathcal{C}, \pi), f \mapsto T_\Phi(f)(\boldsymbol{s}') = \int_{\mathbb{R}^d} \Phi(\boldsymbol{s}, \boldsymbol{s}') f(\boldsymbol{s}) \pi(\boldsymbol{s}) d\boldsymbol{s}$ and $\{\omega_i(\cdot) = \sqrt{\nu_i} \varphi_i(\cdot)\}_{i=1}^{M}$ are the rescaled eigenfunction. $\{\omega_i(\cdot)\}_{i=1}^{M}$ forms an orthogonal basis in $L^2(\mathcal{C}, \pi)$. Usually, $M$ is infinite. In the following of this section, we will find convenient to define a vector representation of functions in the asymptotic feature space defined by the feature map $\{\omega_i\}_{i=1}^{M}$. For any state matrix $\boldsymbol{A} \in \mathbb{R}^{d \times p}$, we denote by $\boldsymbol{\Omega}_{\boldsymbol{A}} \in \mathbb{R}^{M \times p}$ the feature matrices of $\boldsymbol{A}$ so that $[\boldsymbol{\Omega}_{\boldsymbol{A}}]_{ij} = \omega_i(\boldsymbol{A}_j)$ for $\boldsymbol{A}_j$ the $j^{th}$ column of $\boldsymbol{A}$. With those new notations, we can decompose $\boldsymbol{\Phi}_{\hat{\mathcal{S}}}$ and $\boldsymbol{\Phi}_{\mathcal{S}}$ as

$$\boldsymbol{\Phi}_{\hat{\mathcal{S}}} = \boldsymbol{\Omega}_{\hat{\mathcal{S}}}^T \boldsymbol{\Omega}_{\hat{\mathcal{S}}} \qquad \text{and} \qquad \boldsymbol{\Phi}_{\mathcal{S}} = \boldsymbol{\Omega}_{\mathcal{S}}^T \boldsymbol{\Omega}_{\mathcal{S}}.$$

**Regularized LSTD in the Asymptotic Feature Space.** Let $\boldsymbol{v}_{\bar{\theta}} \in \mathbb{R}^M$ the weight vector returned by the regularized LSTD with the rescaled $l_2$-regularization parameter $\lambda \frac{m(1+\delta)}{N}$ for the asymptotic features $\{\omega_i\}_{i=1}^{M}$ on transitions collected in $\mathcal{D}_{\text{train}}$:

$$\boldsymbol{v}_{\bar{\theta}} = \frac{1}{\sqrt{n}} \left[ \bar{\boldsymbol{A}} + \lambda \frac{m(1+\delta)}{N} \boldsymbol{I}_M \right]^{-1} \boldsymbol{\Omega}_{\hat{\mathcal{S}}}^T \hat{\boldsymbol{U}}_n \boldsymbol{r},$$

where

$$\bar{\boldsymbol{A}} = \boldsymbol{\Omega}_{\hat{\mathcal{S}}} \hat{\boldsymbol{U}}_n (\hat{\boldsymbol{U}}_n - \gamma \hat{\boldsymbol{V}}_n)^T \boldsymbol{\Omega}_{\hat{\mathcal{S}}}^T. \tag{25}$$

**Second-Order Correction Factor in the Asymptotic Feature Space.** Let $f_\Delta(\cdot)$ be the second-order correction function defined as

$$f_\Delta(\boldsymbol{B}) = \frac{\lambda^2}{n} \frac{m^2(1+\delta)^2}{N^2} \frac{\frac{1}{N} \|\boldsymbol{v}_{\bar{\theta}}\|^2}{1 - \frac{1}{N} \left\| \left[ \bar{\boldsymbol{A}} + \lambda \frac{m(1+\delta)}{N} \boldsymbol{I}_M \right]^{-1} \bar{\boldsymbol{A}} \right\|_F^2} \left\| \boldsymbol{B} \left[ \bar{\boldsymbol{A}} + \lambda \frac{m(1+\delta)}{N} \boldsymbol{I}_M \right]^{-1} \right\|_F^2.$$

From the definition of $f_\Delta(\boldsymbol{B})$, it directly follows that $f_\Delta(\boldsymbol{B}) \to 0$ as $N/m \to \infty$ or as $\lambda \to \infty$.

## C.2 REFORMULATION OF THE MAIN RESULTS

### C.2.1 EMPIRICAL MEAN-SQUARED BELLMAN ERROR

**Theorem C.1** (Asymptotic Empirical MSBE). *Under the conditions of Theorem 5.1, the deterministic asymptotic empirical MSBE is*

$$\widehat{\overline{\text{MSBE}}}(\hat{\boldsymbol{\theta}}) = \frac{1}{n} \left\| \boldsymbol{r} + \gamma \boldsymbol{\Omega}_{\boldsymbol{X}'}^T \boldsymbol{v}_{\bar{\theta}} - \boldsymbol{\Omega}_{\boldsymbol{X}}^T \boldsymbol{v}_{\bar{\theta}} \right\|^2 + f_\Delta \left( (\hat{\boldsymbol{U}}_n - \gamma \hat{\boldsymbol{V}}_n)^T \boldsymbol{\Omega}_{\hat{\mathcal{S}}}^T \right).$$

*As $N, m, d \to \infty$ with asymptotic constant ratio $N/m$,*

$$\widehat{\text{MSBE}}(\hat{\boldsymbol{\theta}}) - \widehat{\overline{\text{MSBE}}}(\hat{\boldsymbol{\theta}}) \xrightarrow{a.s} 0.$$

**Remark 9.** *The left-hand term $\frac{1}{n}\left\|r + \gamma\boldsymbol{\Omega}_{X'}^T \boldsymbol{v}_{\bar{\theta}} - \boldsymbol{\Omega}_X^T \boldsymbol{v}_{\bar{\theta}}\right\|^2$ in $\widehat{\overline{\text{MSBE}}}(\hat{\boldsymbol{\theta}})$ depicts the empirical MSBE for the regularized LSTD in the asymptotic features $\{\omega_i\}_{i=1}^M$ with the rescaled $l_2$-regularization parameter $\lambda\frac{m(1+\delta)}{N}$ on transitions collected in $\mathcal{D}_{train}$.*

**Remark 10.** *As $N/m$ increases, the influence of the $l_2$-regularization parameter $\lambda$ decreases.*

**Remark 11.** *As $N/m \to \infty$, $\widehat{\overline{\text{MSBE}}}(\hat{\boldsymbol{\theta}}) \to 0$.*

### C.2.2 MEAN-SQUARED BELLMAN ERROR

**Theorem C.2** (Asymptotic MSBE). *Under Assumptions 1, 2, and 3, the deterministic asymptotic MSBE is*

$$\overline{\text{MSBE}}(\hat{\boldsymbol{\theta}}) = \left\|r + \gamma\boldsymbol{P}\boldsymbol{\Omega}_{\mathcal{S}}^T \boldsymbol{v}_{\bar{\theta}} - \boldsymbol{\Omega}_{\mathcal{S}}^T \boldsymbol{v}_{\bar{\theta}}\right\|_{\boldsymbol{D}_{\boldsymbol{\pi}}}^2 + f_\Delta\left(\boldsymbol{D}_{\boldsymbol{\pi}}^{\frac{1}{2}}(\boldsymbol{I}_{|\mathcal{S}|} - \gamma\boldsymbol{P})\boldsymbol{\Omega}_{\mathcal{S}}^T\right).$$

*As $N, m, d \to \infty$ with asymptotic constant ratio $N/m$,*

$$\text{MSBE}(\hat{\boldsymbol{\theta}}) - \overline{\text{MSBE}}(\hat{\boldsymbol{\theta}}) \xrightarrow{a.s} 0.$$

**Remark 12.** *The left-hand term $\left\|r + \gamma\boldsymbol{P}\boldsymbol{\Omega}_{\mathcal{S}}^T \boldsymbol{v}_{\bar{\theta}} - \boldsymbol{\Omega}_{\mathcal{S}}^T \boldsymbol{v}_{\bar{\theta}}\right\|_{\boldsymbol{D}_{\boldsymbol{\pi}}}^2$ in $\overline{\text{MSBE}}(\hat{\boldsymbol{\theta}})$ depicts the true MSBE for the regularized LSTD in the asymptotic features $\{\omega_i\}_{i=1}^M$ with the rescaled $l_2$-regularization parameter $\lambda\frac{m(1+\delta)}{N}$ on transitions collected in $\mathcal{D}_{train}$.*

**Remark 13.** *As $N/m$ increases, the influence of the $l_2$-regularization parameter $\lambda$ decreases.*

**Remark 14.** *As $N/m \to \infty$, $\overline{\text{MSBE}}(\hat{\boldsymbol{\theta}})$ converges to the MSBE of $\boldsymbol{v}_{\bar{\theta}}$ without the $l_2$ regularization parameter.*

### C.2.3 MEAN-SQUARED VALUE ERROR

**Corollary C.2.1** (Asymptotic MSVE). *Under Assumptions 1, 2, and 3, the deterministic asymptotic MSVE is*

$$\overline{\text{MSVE}}(\hat{\boldsymbol{\theta}}) = \left\|\boldsymbol{V} - \boldsymbol{\Omega}_{\mathcal{S}}^T \boldsymbol{v}_{\bar{\theta}}\right\|_{\boldsymbol{D}_{\boldsymbol{\pi}}}^2 + f_\Delta\left(\boldsymbol{D}_{\boldsymbol{\pi}}^{\frac{1}{2}}\boldsymbol{\Omega}_{\mathcal{S}}^T\right).$$

*As $N, m, d \to \infty$ with asymptotic constant ratio $N/m$,*

$$\text{MSVE}(\hat{\boldsymbol{\theta}}) - \overline{\text{MSVE}}(\hat{\boldsymbol{\theta}}) \xrightarrow{a.s} 0.$$

**Remark 15.** *The left-hand term $\left\|\boldsymbol{V} - \boldsymbol{\Omega}_{\mathcal{S}}^T \boldsymbol{v}_{\bar{\theta}}\right\|_{\boldsymbol{D}_{\boldsymbol{\pi}}}^2$ in $\overline{\text{MSVE}}(\hat{\boldsymbol{\theta}})$ depicts the MSVE for the regularized LSTD in the asymptotic features $\{\omega_i\}_{i=1}^M$ with the rescaled $l_2$-regularization parameter $\lambda\frac{m(1+\delta)}{N}$ on transitions collected in $\mathcal{D}_{train}$.*

**Remark 16.** *As $N/m$ increases, the influence of the $l_2$-regularization parameter $\lambda$ decreases.*

**Remark 17.** *As $N/m \to \infty$, $\overline{\text{MSVE}}(\hat{\boldsymbol{\theta}})$ converges to the MSVE of $\boldsymbol{v}_{\bar{\theta}}$ without the $l_2$ regularization parameter.*

### C.3 PROOFS OF RESULTS FOUND IN SECTION C.2

**Theorem C.3** (Asymptotic Empirical MSBE). *Under the conditions of Theorem 5.1, the deterministic asymptotic empirical MSBE is*

$$\widehat{\overline{\text{MSBE}}}(\hat{\boldsymbol{\theta}}) = \frac{1}{n}\left\|r + \gamma\boldsymbol{\Omega}_{X'}^T \boldsymbol{v}_{\bar{\theta}} - \boldsymbol{\Omega}_X^T \boldsymbol{v}_{\bar{\theta}}\right\|^2 + f_\Delta\left((\hat{\boldsymbol{U}}_n - \gamma\hat{\boldsymbol{V}}_n)^T\boldsymbol{\Omega}_{\hat{\mathcal{S}}}^T\right).$$

*As $N, m, d \to \infty$ with asymptotic constant ratio $N/m$,*

$$\widehat{\text{MSBE}}(\hat{\boldsymbol{\theta}}) - \widehat{\overline{\text{MSBE}}}(\hat{\boldsymbol{\theta}}) \xrightarrow{a.s} 0.$$

*Proof.* We have

$$\frac{\lambda^2}{n}\|\bar{\boldsymbol{Q}}_m(\lambda)r\|^2 = \frac{1}{n}\left\|r + \gamma\frac{N}{m}\frac{1}{1+\delta}\hat{\boldsymbol{V}}_n^T\boldsymbol{\Omega}_{\hat{\mathcal{S}}}^T\boldsymbol{\Omega}_{\hat{\mathcal{S}}}\hat{\boldsymbol{U}}_n\bar{\boldsymbol{Q}}_m(\lambda)r - \frac{N}{m}\frac{1}{1+\delta}\hat{\boldsymbol{U}}_n\boldsymbol{\Omega}_{\mathcal{S}}^T\boldsymbol{\Omega}_{\hat{\mathcal{S}}}\hat{\boldsymbol{U}}_n\bar{\boldsymbol{Q}}_m(\lambda)r\right\|^2$$
$$= \frac{1}{n}\left\|r + \gamma\boldsymbol{\Omega}_{X'}^T \boldsymbol{v}_{\bar{\theta}} - \boldsymbol{\Omega}_X^T \boldsymbol{v}_{\bar{\theta}}\right\|^2.$$

For $\hat{\Delta}$, we have

$$\text{Tr}\left(\bar{\boldsymbol{Q}}_m(\lambda)\boldsymbol{\Psi}_2\bar{\boldsymbol{Q}}_m(\lambda)^T\right)$$

$$= \frac{N}{m}\frac{1}{1+\delta}\text{Tr}\left(\bar{\boldsymbol{Q}}_m(\lambda)(\hat{\boldsymbol{U}}_n - \gamma\hat{\boldsymbol{V}}_n)^T\boldsymbol{\Phi}_{\hat{\mathcal{S}}}(\hat{\boldsymbol{U}}_n - \gamma\hat{\boldsymbol{V}}_n)\bar{\boldsymbol{Q}}_m(\lambda)^T\right)$$

$$= \frac{m(1+\delta)}{N}\text{Tr}\left((\hat{\boldsymbol{U}}_n - \gamma\hat{\boldsymbol{V}}_n)^T\boldsymbol{\Omega}_{\hat{\mathcal{S}}}^T\left[\bar{\boldsymbol{A}} + \lambda\frac{m(1+\delta)}{N}\boldsymbol{I}_M\right]^{-1}\left[\bar{\boldsymbol{A}} + \lambda\frac{m(1+\delta)}{N}\boldsymbol{I}_M\right]^{-1T}\boldsymbol{\Omega}_{\hat{\mathcal{S}}}(\hat{\boldsymbol{U}}_n - \gamma\hat{\boldsymbol{V}}_n)\right)$$

$$= \frac{m(1+\delta)}{N}\left\|(\hat{\boldsymbol{U}}_n - \gamma\hat{\boldsymbol{V}}_n)^T\boldsymbol{\Omega}_{\hat{\mathcal{S}}}^T\left[\bar{\boldsymbol{A}} + \lambda\frac{m(1+\delta)}{N}\boldsymbol{I}_M\right]^{-1}\right\|_F^2$$

$$\text{Tr}\left(\boldsymbol{\Psi}_2\bar{\boldsymbol{Q}}_m(\lambda)^T\boldsymbol{\Psi}_1\bar{\boldsymbol{Q}}_m(\lambda)\right)$$

$$= \frac{N^2}{m^2}\frac{1}{(1+\delta)^2}\text{Tr}\left((\hat{\boldsymbol{U}}_n - \gamma\hat{\boldsymbol{V}}_n)^T\boldsymbol{\Omega}_{\hat{\mathcal{S}}}^T\boldsymbol{\Omega}_{\hat{\mathcal{S}}}(\hat{\boldsymbol{U}}_n - \gamma\hat{\boldsymbol{V}}_n)\bar{\boldsymbol{Q}}_m(\lambda)^T\hat{\boldsymbol{U}}_n^T\boldsymbol{\Omega}_{\hat{\mathcal{S}}}^T\boldsymbol{\Omega}_{\hat{\mathcal{S}}}\hat{\boldsymbol{U}}_n\bar{\boldsymbol{Q}}_m(\lambda)\right)$$

$$= \text{Tr}\left(\bar{\boldsymbol{A}}^T\left[\bar{\boldsymbol{A}} + \lambda\frac{m(1+\delta)}{N}\boldsymbol{I}_M\right]^{-1T}\left[\bar{\boldsymbol{A}} + \lambda\frac{m(1+\delta)}{N}\boldsymbol{I}_M\right]^{-1}\bar{\boldsymbol{A}}\right)$$

$$= \left\|\left[\bar{\boldsymbol{A}} + \lambda\frac{m(1+\delta)}{N}\boldsymbol{I}_M\right]^{-1}\bar{\boldsymbol{A}}\right\|_F^2,$$

and

$$\|\bar{\boldsymbol{Q}}_m(\lambda)\boldsymbol{r}\|_{\boldsymbol{\Psi}_1}^2 = \frac{m(1+\delta)}{n}\|\boldsymbol{v}_{\boldsymbol{\theta}}\|^2$$

$$\square$$

**Theorem C.4** (Asymptotic MSBE). *Under Assumptions 1, 2, and 3, the deterministic asymptotic MSBE is*

$$\overline{\text{MSBE}}(\hat{\boldsymbol{\theta}}) = \left\|\boldsymbol{r} + \gamma\boldsymbol{P}\boldsymbol{\Omega}_{\mathcal{S}}^T\boldsymbol{v}_{\boldsymbol{\theta}} - \boldsymbol{\Omega}_{\mathcal{S}}^T\boldsymbol{v}_{\boldsymbol{\theta}}\right\|_{\boldsymbol{D}_{\boldsymbol{\pi}}}^2 + f_{\Delta}\left(\boldsymbol{D}_{\boldsymbol{\pi}}^{\frac{1}{2}}(\boldsymbol{I}_{|\mathcal{S}|} - \gamma\boldsymbol{P})\boldsymbol{\Omega}_{\mathcal{S}}^T\right).$$

*As $N, m, d \to \infty$ with asymptotic constant ratio $N/m$,*

$$\text{MSBE}(\hat{\boldsymbol{\theta}}) - \overline{\text{MSBE}}(\hat{\boldsymbol{\theta}}) \xrightarrow{a.s} 0.$$

*Proof.* We can rewrite

$$\text{Tr}\left(\boldsymbol{\Lambda}_{\boldsymbol{P}}\boldsymbol{\Theta}_{\mathcal{S}}\boldsymbol{\Psi}_2\boldsymbol{\Theta}_{\mathcal{S}}^T\right)$$

$$= \frac{N}{m}\frac{1}{1+\delta}\text{Tr}\left(\left[\bar{\boldsymbol{A}} + \lambda\frac{m(1+\delta)}{N}\boldsymbol{I}_M\right]^{-1T}\bar{\boldsymbol{A}}^T\boldsymbol{\Omega}_{\mathcal{S}}[\boldsymbol{I}_{|\mathcal{S}|} - \gamma\boldsymbol{P}]^T\boldsymbol{D}_{\boldsymbol{\pi}}[\boldsymbol{I}_{|\mathcal{S}|} - \gamma\boldsymbol{P}]\boldsymbol{\Omega}_{\mathcal{S}}^T\bar{\boldsymbol{A}}\left[\bar{\boldsymbol{A}} + \lambda\frac{m(1+\delta)}{N}\boldsymbol{I}_M\right]^{-1}\right)$$

$$= \frac{N}{m}\frac{1}{1+\delta}\left\|[\boldsymbol{I}_{|\mathcal{S}|} - \gamma\boldsymbol{P}]\boldsymbol{\Omega}_{\mathcal{S}}^T\bar{\boldsymbol{A}}\left[\bar{\boldsymbol{A}} + \lambda\frac{m(1+\delta)}{N}\boldsymbol{I}_M\right]^{-1}\right\|_{F,\boldsymbol{D}_{\boldsymbol{\pi}}}^2,$$

$$\frac{N}{m}\frac{1}{1+\delta}\text{Tr}\left(\boldsymbol{\Lambda}_{\boldsymbol{P}}\boldsymbol{\Phi}_{\mathcal{S}}\right) = \frac{N}{m}\frac{1}{1+\delta}\left\|[\boldsymbol{I}_{|\mathcal{S}|} - \gamma\boldsymbol{P}]\boldsymbol{\Omega}_{\mathcal{S}}^T\right\|_{F,\boldsymbol{D}_{\boldsymbol{\pi}}}^2,$$

and

$$\text{Tr}\left(\boldsymbol{\Lambda}_{\boldsymbol{P}}\boldsymbol{\Theta}_{\mathcal{S}}(\boldsymbol{U}_n - \gamma\boldsymbol{V}_n)^T\boldsymbol{\Psi}_{\mathcal{S}}\right) = \frac{N}{m}\frac{1}{1+\delta}\text{Tr}\left(\boldsymbol{\Lambda}_{\boldsymbol{P}}\boldsymbol{\Omega}_{\mathcal{S}}^T\bar{\boldsymbol{A}}\left[\bar{\boldsymbol{A}} + \lambda\frac{m(1+\delta)}{N}\boldsymbol{I}_M\right]^{-1}\boldsymbol{\Omega}_{\mathcal{S}}\right)$$

$$= \frac{N}{m}\frac{1}{1+\delta}\text{Tr}\left(\boldsymbol{\Omega}_{\mathcal{S}}[\boldsymbol{I}_{|\mathcal{S}|} - \gamma\boldsymbol{P}]^T\boldsymbol{D}_{\boldsymbol{\pi}}[\boldsymbol{I}_{|\mathcal{S}|} - \gamma\boldsymbol{P}]\boldsymbol{\Omega}_{\mathcal{S}}^T\bar{\boldsymbol{A}}\left[\bar{\boldsymbol{A}} + \lambda\frac{m(1+\delta)}{N}\boldsymbol{I}_M\right]^{-1}\right)$$

$$= \frac{N}{m}\frac{1}{1+\delta}\left\langle[\boldsymbol{I}_{|\mathcal{S}|} - \gamma\boldsymbol{P}]\boldsymbol{\Omega}_{\mathcal{S}}^T, [\boldsymbol{I}_{|\mathcal{S}|} - \gamma\boldsymbol{P}]\boldsymbol{\Omega}_{\mathcal{S}}^T\bar{\boldsymbol{A}}\left[\bar{\boldsymbol{A}} + \lambda\frac{m(1+\delta)}{N}\boldsymbol{I}_M\right]^{-1}\right\rangle_{F,\boldsymbol{D}_{\boldsymbol{\pi}}}.$$

Therefore,

$$\text{Tr}\left(\boldsymbol{\Lambda}_{\boldsymbol{P}}\left[\boldsymbol{\Theta}_{\mathcal{S}}\boldsymbol{\Psi}_2\boldsymbol{\Theta}_{\mathcal{S}}^T - 2\boldsymbol{\Theta}_{\mathcal{S}}(\boldsymbol{U}_n - \gamma\boldsymbol{V}_n)^T\boldsymbol{\Psi}_{\mathcal{S}} + \boldsymbol{\Psi}_{\mathcal{S}}\right]\right)$$

$$= \frac{N}{m}\frac{1}{1+\delta}\left\|[\boldsymbol{I}_{|\mathcal{S}|} - \gamma\boldsymbol{P}]\boldsymbol{\Omega}_{\mathcal{S}}^T - [\boldsymbol{I}_{|\mathcal{S}|} - \gamma\boldsymbol{P}]\boldsymbol{\Omega}_{\mathcal{S}}^T\bar{\boldsymbol{A}}\left[\bar{\boldsymbol{A}} + \lambda\frac{m(1+\delta)}{N}\boldsymbol{I}_M\right]^{-1}\right\|_{F,\boldsymbol{D}_{\boldsymbol{\pi}}}^2$$

$$= \lambda^2\left\|[\boldsymbol{I}_{|\mathcal{S}|} - \gamma\boldsymbol{P}]\boldsymbol{\Omega}_{\mathcal{S}}^T\left[\bar{\boldsymbol{A}} + \lambda\frac{m(1+\delta)}{N}\boldsymbol{I}_M\right]^{-1}\right\|_{F,\boldsymbol{D}_{\boldsymbol{\pi}}}^2.$$

$$\square$$

# D   EVALUATION OF $\boldsymbol{\Phi}_{\mathcal{S}}$ OR $\boldsymbol{\Phi}_{\hat{\mathcal{S}}}$

The evaluation of $\boldsymbol{\Phi}_{\hat{\mathcal{S}}} = \mathbb{E}_{\boldsymbol{w}}\big[\sigma(\boldsymbol{w}^T\hat{\boldsymbol{S}})^T\sigma(\boldsymbol{w}^T\hat{\boldsymbol{S}})\big]$ or $\boldsymbol{\Phi}_{\mathcal{S}} = \mathbb{E}_{\boldsymbol{w}}\big[\sigma(\boldsymbol{w}^T\boldsymbol{S})^T\sigma(\boldsymbol{w}^T\boldsymbol{S})\big]$ naturally boils down to the evaluation of its individual entries and thus to the calculus, for arbitrary vectors $\boldsymbol{a}, \boldsymbol{b} \in \mathbb{R}^d$ of

$$\boldsymbol{\Phi}_{ab} = \mathbb{E}\big[\sigma(\boldsymbol{w}^T\boldsymbol{a})\sigma(\boldsymbol{w}^T\boldsymbol{b})\big] = (2\pi)^{-\frac{p}{2}} \int \sigma(\varphi(\tilde{\boldsymbol{w}})^T\boldsymbol{a})\sigma(\varphi(\tilde{\boldsymbol{w}})^T\boldsymbol{b})e^{-\frac{1}{2}\|\tilde{\boldsymbol{w}}\|^2}d\tilde{\boldsymbol{w}}. \qquad (26)$$

The evaluation of equation 26 can be obtained through various integration tricks for a wide family of mappings $\varphi(\cdot)$ and activation functions $\sigma(\cdot)$. We provide in Table 1 (found in Louart et al. (2018)) the values of $\boldsymbol{\Phi}_{ab}$ for $\boldsymbol{w} \sim \mathcal{N}(0, \boldsymbol{I}_d)$ (i.e., for $\varphi(t) = t$) and for a set of activation functions $\sigma(\cdot)$ not necessarily satisfying the Lipschitz continuity. In experiments in Section 6, we focus only on the ReLU function, i.e., $\sigma(t) = \max(t, 0)$.

Table 1: Values of $\boldsymbol{\Phi}_{ab}$ for $\boldsymbol{w} \sim \mathcal{N}(0, \boldsymbol{I}_d)$, $\angle(\boldsymbol{a}, \boldsymbol{b}) \equiv \frac{\boldsymbol{a}^T\boldsymbol{b}}{\|\boldsymbol{a}\|\|\boldsymbol{b}\|}$ (Louart et al., 2018).

| $\sigma(t)$ | $\boldsymbol{\Phi}_{ab}$ |
|---|---|
| $t$ | $\boldsymbol{a}^T\boldsymbol{b}$ |
| $\max(t, 0)$ | $\frac{1}{2\pi}\|\boldsymbol{a}\|\|\boldsymbol{b}\| \left(\angle(\boldsymbol{a}, \boldsymbol{b})\arccos(-\angle(\boldsymbol{a}, \boldsymbol{b})) + \sqrt{1 - \angle(\boldsymbol{a}, \boldsymbol{b})^2}\right)$ |
| $|t|$ | $\frac{2}{\pi}\|\boldsymbol{a}\|\|\boldsymbol{b}\| \left(\angle(\boldsymbol{a}, \boldsymbol{b})\arcsin(\angle(\boldsymbol{a}, \boldsymbol{b})) + \sqrt{1 - \angle(\boldsymbol{a}, \boldsymbol{b})^2}\right)$ |
| $\mathrm{erf}(t)$ | $\frac{2}{\pi}\arcsin\left(\frac{2\boldsymbol{a}^T\boldsymbol{b}}{\sqrt{(1+2\|\boldsymbol{a}\|^2)(1+2\|\boldsymbol{b}\|^2)}}\right)$ |
| $1_{\{t>0\}}$ | $\frac{1}{2} - \frac{1}{2\pi}\arccos(\angle(\boldsymbol{a}, \boldsymbol{b}))$ |
| $\mathrm{sign}(t)$ | $\frac{2}{\pi}\arcsin(\angle(\boldsymbol{a}, \boldsymbol{b}))$ |
| $\cos(t)$ | $\exp(-\frac{1}{2}(\|\boldsymbol{a}\|^2 + \|\boldsymbol{b}\|^2))\cosh(\boldsymbol{a}^T\boldsymbol{b})$ |
| $\sin(t)$ | $\exp(-\frac{1}{2}(\|\boldsymbol{a}\|^2 + \|\boldsymbol{b}\|^2))\sinh(\boldsymbol{a}^T\boldsymbol{b})$. |

# E  PROOF OF THEOREM 5.1

Under Assumptions 1 and 2, this section is dedicated to prove the asymptotic equivalence between $\mathbb{E}[\boldsymbol{Q}_m(\lambda)]$ and

$$\bar{\boldsymbol{Q}}_m(\lambda) = \left[ \frac{N}{m} \frac{1}{1+\delta} (\hat{\boldsymbol{U}}_n - \gamma \hat{\boldsymbol{V}}_n)^T \boldsymbol{\Phi}_{\hat{\mathcal{S}}} \hat{\boldsymbol{U}}_n + \lambda \boldsymbol{I}_n \right]^{-1}$$

defined in Theorem 5.1, when $N, m \to \infty$. In order to prove Theorem 5.1, we shall proceed by introducing an intermediary resolvent $\tilde{\boldsymbol{Q}}_m(\lambda)$ (defined in equation 27), and show subsequently under Assumptions 1 and 2 that

$$\|\mathbb{E}[\boldsymbol{Q}_m(\lambda)] - \tilde{\boldsymbol{Q}}_m(\lambda)\| \to 0 \quad \text{and} \quad \|\tilde{\boldsymbol{Q}}_m(\lambda) - \bar{\boldsymbol{Q}}_m(\lambda)\| \to 0,$$

as $N, m \to \infty$.

In order to simplify the notations, we denote by $\boldsymbol{Q}_m$ the resolvent $\boldsymbol{Q}_m(\lambda)$. The first half of the proof is dedicated to Lemma E.1, which proposes a first characterization of $\mathbb{E}[\boldsymbol{Q}_m]$ by $\tilde{\boldsymbol{Q}}_m$ as $N, m \to \infty$ under Assumptions 1 and 2. This preliminary step is classical in studying resolvents in Random Matrix literature (Louart et al., 2018; Liao et al., 2020) as the direct comparison of $\mathbb{E}[\boldsymbol{Q}_m]$ to $\bar{\boldsymbol{Q}}_m$ with the implicit $\delta$ (equation 17) may be cumbersome.

**Lemma E.1.** *Under Assumptions 1 and 2, let $\lambda > 0$ and let $\tilde{\boldsymbol{Q}}_m(\lambda) \in \mathbb{R}^{n \times n}$ be the resolvent defined as*

$$\tilde{\boldsymbol{Q}}_m(\lambda) = \left[ \frac{N}{m} \frac{1}{1+\alpha} (\hat{\boldsymbol{U}}_n - \gamma \hat{\boldsymbol{V}}_n)^T \boldsymbol{\Phi}_{\hat{\mathcal{S}}} \hat{\boldsymbol{U}}_n + \lambda \boldsymbol{I}_n \right]^{-1} ; \tag{27}$$

*for the deterministic Gram feature matrix*

$$\boldsymbol{\Phi}_{\hat{\mathcal{S}}} = \mathbb{E}_{\boldsymbol{w} \sim \mathcal{N}(\boldsymbol{0}, \boldsymbol{I}_d)} \left[ \sigma(\boldsymbol{w}^T \hat{\boldsymbol{S}})^T \sigma(\boldsymbol{w}^T \hat{\boldsymbol{S}}) \right],$$

*and*

$$\alpha = \frac{1}{m} \operatorname{Tr} \left( (\hat{\boldsymbol{U}}_n - \gamma \hat{\boldsymbol{V}}_n)^T \boldsymbol{\Phi}_{\hat{\mathcal{S}}} \hat{\boldsymbol{U}}_n \mathbb{E}[\boldsymbol{Q}_-(\lambda)] \right), \tag{28}$$

*where*

$$\boldsymbol{Q}_-(\lambda) = \left[ \frac{1}{m} (\hat{\boldsymbol{U}}_n - \gamma \hat{\boldsymbol{V}}_n)^T \sigma(\boldsymbol{W}_- \hat{\boldsymbol{S}})^T \sigma(\boldsymbol{W}_- \hat{\boldsymbol{S}}) \hat{\boldsymbol{U}}_n + \lambda \boldsymbol{I}_n \right]^{-1}, \tag{29}$$

*for which $\boldsymbol{W}_- \in \mathbb{R}^{(N-1) \times d}$ depicts the submatrix of the weight matrix $\boldsymbol{W}$ (defined in equation 5) without the first row. Then,*

$$\lim_{m \to \infty} \|\mathbb{E}_{\boldsymbol{W}}[\boldsymbol{Q}_m(\lambda)] - \tilde{\boldsymbol{Q}}_m(\lambda)\| = 0.$$

**Remark 18.** *Firstly, we can note that $\alpha$ is uniformly bounded. Since $\frac{1}{m} \operatorname{Tr}(\boldsymbol{\Phi}_{\hat{\mathcal{S}}}) = \mathbb{E}\left[ \frac{1}{m} \|\sigma(\boldsymbol{w}^T \hat{\boldsymbol{S}})\|^2 \right]$ and from Lemma K.2, we have*

$$\frac{1}{m} \operatorname{Tr}(\boldsymbol{\Phi}_{\hat{\mathcal{S}}}) = \int_0^\infty \Pr\left( \frac{1}{m} \|\sigma(\boldsymbol{w}^T \hat{\boldsymbol{S}})\|^2 > t \right) dt = \int_0^\infty 2t \Pr\left( \frac{1}{m} \|\sigma(\boldsymbol{w}^T \hat{\boldsymbol{S}})\| > t \right) dt = \mathcal{O}(1). \tag{30}$$

*We deduce that*

$$\alpha = \frac{1}{m} \operatorname{Tr} \left( (\hat{\boldsymbol{U}}_n - \gamma \hat{\boldsymbol{V}}_n)^T \boldsymbol{\Phi}_{\hat{\mathcal{S}}} \hat{\boldsymbol{U}}_n \mathbb{E}[\boldsymbol{Q}_-] \right) \leq \|\hat{\boldsymbol{U}}_n \mathbb{E}[\boldsymbol{Q}_-] (\hat{\boldsymbol{U}}_n - \gamma \hat{\boldsymbol{V}}_n)^T \| \frac{1}{m} \operatorname{Tr}(\boldsymbol{\Phi}_{\hat{\mathcal{S}}}) = \mathcal{O}(1), \tag{31}$$

*where we used $|\operatorname{Tr}(\boldsymbol{AB})| \leq \|\boldsymbol{A}\| \operatorname{Tr}(\boldsymbol{B})$ for non-negative definite matrix $\boldsymbol{B}$ together with Lemma I.1 which asserts the operator norm of the resolvent $\boldsymbol{Q}_-$ is uniformly bounded. Furthermore, both $\|\hat{\boldsymbol{U}}_n\|$ and $\|\hat{\boldsymbol{V}}_n\|$ are upper bounded by $1$.*

*Proof.* We decompose the matrix $\boldsymbol{\Sigma}_{\hat{\mathcal{S}}}^T \boldsymbol{\Sigma}_{\hat{\mathcal{S}}}$ as

$$\boldsymbol{\Sigma}_{\hat{\mathcal{S}}}^T \boldsymbol{\Sigma}_{\hat{\mathcal{S}}} = \sum_{i=1}^N \boldsymbol{\sigma}_i \boldsymbol{\sigma}_i^T, \tag{32}$$

where $\boldsymbol{\sigma}_i = \sigma(\hat{\boldsymbol{S}}^T \boldsymbol{w}_i) \in \mathbb{R}^m$ for which $\boldsymbol{w}_i \in \mathbb{R}^d$ denotes the i-th row of $\boldsymbol{W}$ (defined in equation 5). Using the resolvent identity (Lemma L.2), we write

$$\mathbb{E}[\boldsymbol{Q}_m] - \tilde{\boldsymbol{Q}}_m$$

$$= \mathbb{E}\left[\boldsymbol{Q}_m \left[\tilde{\boldsymbol{Q}}_m^{-1} - \lambda \boldsymbol{I}_n - \frac{1}{m}(\hat{\boldsymbol{U}}_n - \gamma \hat{\boldsymbol{V}}_n)^T \boldsymbol{\Sigma}_{\hat{\mathcal{S}}}^T \boldsymbol{\Sigma}_{\hat{\mathcal{S}}} \hat{\boldsymbol{U}}_n \right] \tilde{\boldsymbol{Q}}_m \right]$$

$$= \frac{N}{m} \frac{1}{1+\alpha} \mathbb{E}[\boldsymbol{Q}_m](\hat{\boldsymbol{U}}_n - \gamma \hat{\boldsymbol{V}}_n)^T \boldsymbol{\Phi}_{\hat{\mathcal{S}}} \hat{\boldsymbol{U}}_n \tilde{\boldsymbol{Q}}_m - \frac{1}{m} \sum_{i=1}^{N} \mathbb{E}\left[\boldsymbol{Q}_m(\hat{\boldsymbol{U}}_n - \gamma \hat{\boldsymbol{V}}_n)^T \boldsymbol{\sigma}_i \boldsymbol{\sigma}_i^T \hat{\boldsymbol{U}}_n \right] \tilde{\boldsymbol{Q}}_m$$

$$= \frac{N}{m} \frac{1}{1+\alpha} \mathbb{E}[\boldsymbol{Q}_m](\hat{\boldsymbol{U}}_n - \gamma \hat{\boldsymbol{V}}_n)^T \boldsymbol{\Phi}_{\hat{\mathcal{S}}} \hat{\boldsymbol{U}}_n \tilde{\boldsymbol{Q}}_m - \frac{1}{m} \sum_{i=1}^{N} \mathbb{E}\left[\boldsymbol{Q}_{-i} \frac{(\hat{\boldsymbol{U}}_n - \gamma \hat{\boldsymbol{V}}_n)^T \boldsymbol{\sigma}_i \boldsymbol{\sigma}_i^T \hat{\boldsymbol{U}}_n}{1 + \frac{1}{m} \boldsymbol{\sigma}_i^T \hat{\boldsymbol{U}}_n \boldsymbol{Q}_{-i}(\hat{\boldsymbol{U}}_n - \gamma \hat{\boldsymbol{V}}_n)^T \boldsymbol{\sigma}_i} \right] \tilde{\boldsymbol{Q}}_m,$$

where the last equality is obtained with the Sherman identity (Lemma L.4) for

$$\boldsymbol{Q}_{-i} = \left[ \frac{1}{m}(\hat{\boldsymbol{U}}_n - \gamma \hat{\boldsymbol{V}}_n)^T \boldsymbol{\Sigma}_{\hat{\mathcal{S}}}^T \boldsymbol{\Sigma}_{\hat{\mathcal{S}}} \hat{\boldsymbol{U}}_n - \frac{1}{m}(\hat{\boldsymbol{U}}_n - \gamma \hat{\boldsymbol{V}}_n)^T \boldsymbol{\sigma}_i \boldsymbol{\sigma}_i^T \hat{\boldsymbol{U}}_n + \lambda \boldsymbol{I}_n \right]^{-1} \tag{33}$$

independent of $\boldsymbol{\sigma}_i$ and thus $\boldsymbol{w}_i$. Exploiting this independence, we decompose

$$\mathbb{E}[\boldsymbol{Q}_m] - \tilde{\boldsymbol{Q}}_m \tag{34}$$

$$= \frac{N}{m} \frac{1}{1+\alpha} \mathbb{E}[\boldsymbol{Q}_m](\hat{\boldsymbol{U}}_n - \gamma \hat{\boldsymbol{V}}_n)^T \boldsymbol{\Phi}_{\hat{\mathcal{S}}} \hat{\boldsymbol{U}}_n \tilde{\boldsymbol{Q}}_m - \frac{1}{1+\alpha} \frac{1}{m} \sum_{i=1}^{N} \mathbb{E}\left[\boldsymbol{Q}_{-i}(\hat{\boldsymbol{U}}_n - \gamma \hat{\boldsymbol{V}}_n)^T \boldsymbol{\sigma}_i \boldsymbol{\sigma}_i^T \hat{\boldsymbol{U}}_n \right] \tilde{\boldsymbol{Q}}_m$$

$$+ \frac{1}{m} \frac{1}{1+\alpha} \sum_{i=1}^{N} \mathbb{E}\left[\boldsymbol{Q}_{-i} \frac{(\hat{\boldsymbol{U}}_n - \gamma \hat{\boldsymbol{V}}_n)^T \boldsymbol{\sigma}_i \boldsymbol{\sigma}_i^T \hat{\boldsymbol{U}}_n (\frac{1}{m} \boldsymbol{\sigma}_i^T \hat{\boldsymbol{U}}_n \boldsymbol{Q}_{-i}(\hat{\boldsymbol{U}}_n - \gamma \hat{\boldsymbol{V}}_n)^T \boldsymbol{\sigma}_i - \alpha)}{1 + \frac{1}{m} \boldsymbol{\sigma}_i^T \hat{\boldsymbol{U}}_n \boldsymbol{Q}_{-i}(\hat{\boldsymbol{U}}_n - \gamma \hat{\boldsymbol{V}}_n)^T \boldsymbol{\sigma}_i} \right] \tilde{\boldsymbol{Q}}_m \tag{35}$$

$$= \underbrace{\frac{1}{m} \frac{1}{1+\alpha} \sum_{i=1}^{N} \mathbb{E}[\boldsymbol{Q}_m - \boldsymbol{Q}_{-i}](\hat{\boldsymbol{U}}_n - \gamma \hat{\boldsymbol{V}}_n)^T \boldsymbol{\Phi}_{\hat{\mathcal{S}}} \hat{\boldsymbol{U}}_n \tilde{\boldsymbol{Q}}_m}_{=\boldsymbol{Z}_1}$$

$$+ \underbrace{\frac{1}{m} \frac{1}{1+\alpha} \sum_{i=1}^{N} \mathbb{E}\left[\boldsymbol{Q}_m \left(\hat{\boldsymbol{U}}_n - \gamma \hat{\boldsymbol{V}}_n\right)^T \boldsymbol{\sigma}_i \boldsymbol{\sigma}_i^T \hat{\boldsymbol{U}}_n \tilde{\boldsymbol{Q}}_m \left(\frac{1}{m} \boldsymbol{\sigma}_i^T \hat{\boldsymbol{U}}_n \boldsymbol{Q}_{-i}(\hat{\boldsymbol{U}}_n - \gamma \hat{\boldsymbol{V}}_n)^T \boldsymbol{\sigma}_i - \alpha\right) \right]}_{=\boldsymbol{Z}_2}.$$

$$\tag{36}$$

The last equality is obtained by exploiting the Sherman identity (Lemma L.4) in reverse on the rightmost term, and from the independence of $\boldsymbol{Q}_{-i}$ and $\boldsymbol{\sigma}_i \boldsymbol{\sigma}_i^T$ for the second right-hand term. We want to prove that both $\boldsymbol{Z}_1$ and $\boldsymbol{Z}_2$ have a vanishing spectral norm under Assumptions 1 and 2. With both the resolvent identity (Lemma L.2) and the Sherman identity (Lemma L.4), we rewrite $\boldsymbol{Z}_1$ as

$$\boldsymbol{Z}_1 = \frac{1}{m} \frac{1}{1+\alpha} \sum_{i=1}^{N} \mathbb{E}[\boldsymbol{Q}_m - \boldsymbol{Q}_{-i}](\hat{\boldsymbol{U}}_n - \gamma \hat{\boldsymbol{V}}_n)^T \boldsymbol{\Phi}_{\hat{\mathcal{S}}} \hat{\boldsymbol{U}}_n \tilde{\boldsymbol{Q}}_m$$

$$= -\frac{1}{m^2} \frac{1}{1+\alpha} \sum_{i=1}^{N} \mathbb{E}\left[\boldsymbol{Q}_m(\hat{\boldsymbol{U}}_n - \gamma \hat{\boldsymbol{V}}_n)^T \boldsymbol{\sigma}_i \boldsymbol{\sigma}_i^T \hat{\boldsymbol{U}}_n \boldsymbol{Q}_{-i} \right](\hat{\boldsymbol{U}}_n - \gamma \hat{\boldsymbol{V}}_n)^T \boldsymbol{\Phi}_{\hat{\mathcal{S}}} \hat{\boldsymbol{U}}_n \tilde{\boldsymbol{Q}}_m$$

$$= -\frac{1}{m^2} \frac{1}{1+\alpha} \sum_{i=1}^{N} \mathbb{E}\left[\boldsymbol{Q}_m(\hat{\boldsymbol{U}}_n - \gamma \hat{\boldsymbol{V}}_n)^T \boldsymbol{\sigma}_i \boldsymbol{D}_i \boldsymbol{\sigma}_i^T \hat{\boldsymbol{U}}_n \boldsymbol{Q}_m \right](\hat{\boldsymbol{U}}_n - \gamma \hat{\boldsymbol{V}}_n)^T \boldsymbol{\Phi}_{\hat{\mathcal{S}}} \hat{\boldsymbol{U}}_n \tilde{\boldsymbol{Q}}_m$$

$$= -\frac{1}{m^2} \frac{1}{1+\alpha} \mathbb{E}\left[\boldsymbol{Q}_m(\hat{\boldsymbol{U}}_n - \gamma \hat{\boldsymbol{V}}_n)^T \boldsymbol{\Sigma}_{\hat{\mathcal{S}}}^T \boldsymbol{D} \boldsymbol{\Sigma}_{\hat{\mathcal{S}}} \hat{\boldsymbol{U}}_n \boldsymbol{Q}_m \right](\hat{\boldsymbol{U}}_n - \gamma \hat{\boldsymbol{V}}_n)^T \boldsymbol{\Phi}_{\hat{\mathcal{S}}} \hat{\boldsymbol{U}}_n \tilde{\boldsymbol{Q}}_m,$$

where $\boldsymbol{D} \in \mathbb{R}^{N \times N}$ is a diagonal matrix for which, for all $i \in [N]$, we have

$$\boldsymbol{D}_i = \left( 1 + \frac{1}{m} \boldsymbol{\sigma}_i^T \hat{\boldsymbol{U}}_n \boldsymbol{Q}_{-i}(\hat{\boldsymbol{U}}_n - \gamma \hat{\boldsymbol{V}}_n)^T \boldsymbol{\sigma}_i \right). \tag{37}$$

With a similar proof than for Lemma I.1, we can show there exists a $K_{\tilde{\boldsymbol{Q}}_m}$ such that, for all $m$, we have $\|\tilde{\boldsymbol{Q}}_m\| \leq K_{\tilde{\boldsymbol{Q}}_m}$ and then

$$\left\| \frac{1}{1+\alpha} (\hat{\boldsymbol{U}}_n - \gamma \hat{\boldsymbol{V}}_n)^T \boldsymbol{\Phi}_{\hat{\mathcal{S}}} \hat{\boldsymbol{U}}_n \tilde{\boldsymbol{Q}}_m \right\| = \left\| \frac{m}{N} (\boldsymbol{I}_n - \lambda \tilde{\boldsymbol{Q}}_m) \right\| \leq \frac{m}{N} (1 + \lambda K_{\tilde{\boldsymbol{Q}}_m}). \tag{38}$$

Furthermore, from Lemma E.4, we have

$$\left\| \frac{1}{m^2} \mathbb{E}\left[ \boldsymbol{Q}_m (\hat{\boldsymbol{U}}_n - \gamma \hat{\boldsymbol{V}}_n)^T \boldsymbol{\Sigma}_{\hat{\mathcal{S}}}^T \boldsymbol{D} \boldsymbol{\Sigma}_{\hat{\mathcal{S}}} \hat{\boldsymbol{U}}_n \boldsymbol{Q}_m \right] \right\| = \mathcal{O}\left( \frac{1}{m} \right). \tag{39}$$

Therefore, by combining both equation 38 and equation 39, we conclude that $\boldsymbol{Z}_1$ has a vanishing spectral norm, i.e.,

$$\|\boldsymbol{Z}_1\| = \left\| \frac{1}{m} \frac{1}{1+\alpha} \sum_{i=1}^N \mathbb{E}[\boldsymbol{Q}_m - \boldsymbol{Q}_{-i}](\hat{\boldsymbol{U}}_n - \gamma \hat{\boldsymbol{V}}_n)^T \boldsymbol{\Phi}_{\hat{\mathcal{S}}} \hat{\boldsymbol{U}}_n \tilde{\boldsymbol{Q}}_m \right\| = \mathcal{O}\left( \frac{1}{m} \right). \tag{40}$$

We want to show now that $\boldsymbol{Z}_2$ has also a vanishing operator norm. For $i \in [N]$, by setting

$$\boldsymbol{B}_i = m^{\frac{1}{4}} \boldsymbol{Q}_m \left( \hat{\boldsymbol{U}}_n - \gamma \hat{\boldsymbol{V}}_n \right)^T \boldsymbol{\sigma}_i \left( \frac{1}{m} \boldsymbol{\sigma}_i^T \hat{\boldsymbol{U}}_n \boldsymbol{Q}_{-i} (\hat{\boldsymbol{U}}_n - \gamma \hat{\boldsymbol{V}}_n)^T \boldsymbol{\sigma}_i - \alpha \right)$$

and

$$\boldsymbol{C}_i = m^{-\frac{1}{4}} \tilde{\boldsymbol{Q}}_m^T \hat{\boldsymbol{U}}_n^T \boldsymbol{\sigma}_i,$$

we decompose $\boldsymbol{Z}_2$ with its symmetric and its skew-symmetric part as

$$\boldsymbol{Z}_2 = \frac{1}{1+\alpha} \frac{1}{m} \sum_{i=1}^N \mathbb{E}\left[ \boldsymbol{Q}_m \left( \hat{\boldsymbol{U}}_n - \gamma \hat{\boldsymbol{V}}_n \right)^T \boldsymbol{\sigma}_i \boldsymbol{\sigma}_i^T \hat{\boldsymbol{U}}_n \tilde{\boldsymbol{Q}}_m \left( \frac{1}{m} \boldsymbol{\sigma}_i^T \hat{\boldsymbol{U}}_n \boldsymbol{Q}_{-i} (\hat{\boldsymbol{U}}_n - \gamma \hat{\boldsymbol{V}}_n)^T \boldsymbol{\sigma}_i - \alpha \right) \right]$$

$$= \frac{1}{1+\alpha} \frac{1}{m} \sum_{i=1}^N \mathbb{E}\left[ \boldsymbol{B}_i \boldsymbol{C}_i^T \right]$$

$$= \frac{1}{1+\alpha} \frac{1}{m} \sum_{i=1}^N \mathbb{E}\left[ \frac{\boldsymbol{B}_i \boldsymbol{C}_i^T + \boldsymbol{C}_i \boldsymbol{B}_i^T}{2} \right] + \frac{1}{1+\alpha} \frac{1}{m} \sum_{i=1}^N \mathbb{E}\left[ \frac{\boldsymbol{B}_i \boldsymbol{C}_i^T - \boldsymbol{C}_i \boldsymbol{B}_i^T}{2} \right].$$

For the symmetric part, we use the relations $(\boldsymbol{B}_i - \boldsymbol{C}_i)(\boldsymbol{B}_i - \boldsymbol{C}_i)^T \succeq 0$ and $(\boldsymbol{B}_i + \boldsymbol{C}_i)(\boldsymbol{B}_i + \boldsymbol{C}_i)^T \succeq 0$ to deduce that

$$-\boldsymbol{B}_i \boldsymbol{B}_i^T - \boldsymbol{C}_i \boldsymbol{C}_i^T \preceq \boldsymbol{B}_i \boldsymbol{C}_i^T + \boldsymbol{C}_i \boldsymbol{B}_i^T \preceq \boldsymbol{B}_i \boldsymbol{B}_i^T + \boldsymbol{C}_i \boldsymbol{C}_i^T,$$

where $\preceq$ is the Loewner order for semi-positive-definite matrices. For the skew-symmetric part, we observe that $\|\mathbb{E}\left[\boldsymbol{B}_i \boldsymbol{C}_i^T - \boldsymbol{C}_i \boldsymbol{B}_i^T\right]\| = \|i \mathbb{E}\left[\boldsymbol{B}_i \boldsymbol{C}_i^T - \boldsymbol{C}_i \boldsymbol{B}_i^T\right]\|$ for $i^2 = -1$. With a similar reasoning than above, using the relations $(\boldsymbol{B}_i + i\boldsymbol{C}_i)(\boldsymbol{B}_i + i\boldsymbol{C}_i)^* \succeq 0$ and $-(\boldsymbol{B}_i - i\boldsymbol{C}_i)(\boldsymbol{B}_i - i\boldsymbol{C}_i)^* \preceq 0$, we deduce the relation

$$-\boldsymbol{B}_i \boldsymbol{B}_i^T - \boldsymbol{C}_i \boldsymbol{C}_i^T \preceq i(\boldsymbol{B}_i \boldsymbol{C}_i^T - \boldsymbol{C}_i \boldsymbol{B}_i^T) \preceq \boldsymbol{B}_i \boldsymbol{B}_i^T + \boldsymbol{C}_i \boldsymbol{C}_i^T.$$

From those relations, for both the symmetric and skew-symmetric parts, we have

$$\|\boldsymbol{Z}_2\| = \left\| \frac{1}{1+\alpha} \frac{1}{m} \sum_{i=1}^N \mathbb{E}\left[ \boldsymbol{Q} \left( \hat{\boldsymbol{U}}_n - \gamma \hat{\boldsymbol{V}}_n \right)^T \boldsymbol{\sigma}_i \boldsymbol{\sigma}_i^T \hat{\boldsymbol{U}}_n \tilde{\boldsymbol{Q}}_m \left( \frac{1}{m} \boldsymbol{\sigma}_i^T \hat{\boldsymbol{U}}_n \boldsymbol{Q}_{-i} (\hat{\boldsymbol{U}}_n - \gamma \hat{\boldsymbol{V}}_n)^T \boldsymbol{\sigma}_i - \alpha \right) \right] \right\|$$

$$\leq \frac{1}{1+\alpha} \left( \left\| \sum_{i=1}^N \mathbb{E}\left[ \frac{1}{m} \boldsymbol{B}_i \boldsymbol{B}_i^T \right] \right\| + \left\| \sum_{i=1}^N \mathbb{E}\left[ \frac{1}{m} \boldsymbol{C}_i \boldsymbol{C}_i^T \right] \right\| \right). \tag{41}$$

From Lemma I.4, we know there exists a real $K'_{\boldsymbol{Q}_m} > 0$ such that, for all $m$, we have

$$\left\| \frac{1}{\sqrt{m}} \boldsymbol{Q}_m (\hat{\boldsymbol{U}}_n - \gamma \hat{\boldsymbol{V}}_n)^T \boldsymbol{\Sigma}_{\hat{\mathcal{S}}}^T \right\| \leq K'_{\boldsymbol{Q}_m}.$$

At this point,

$$\left\| \sum_{i=1}^{N} \mathbb{E}\left[ \frac{1}{m} \boldsymbol{B}_i \boldsymbol{B}_i^T \right] \right\|$$

$$= \left\| \sum_{i=1}^{N} \mathbb{E}\left[ \frac{1}{\sqrt{m}} \boldsymbol{Q}_m \left( \hat{\boldsymbol{U}}_n - \gamma \hat{\boldsymbol{V}}_n \right)^T \boldsymbol{\sigma}_i \boldsymbol{\sigma}_i^T \left( \hat{\boldsymbol{U}}_n - \gamma \hat{\boldsymbol{V}}_n \right)^T \boldsymbol{Q}_m^T \left( \frac{1}{m} \boldsymbol{\sigma}_i^T \hat{\boldsymbol{U}}_n \boldsymbol{Q}_{-i} (\hat{\boldsymbol{U}}_n - \gamma \hat{\boldsymbol{V}}_n)^T \boldsymbol{\sigma}_i - \alpha \right)^2 \right] \right\|$$

$$= \left\| \sqrt{m}\, \mathbb{E}\left[ \frac{1}{m} \boldsymbol{Q}_m \left( \hat{\boldsymbol{U}}_n - \gamma \hat{\boldsymbol{V}}_n \right)^T \boldsymbol{\Sigma}_{\hat{\mathcal{S}}}^T \boldsymbol{D}_2^2 \boldsymbol{\Sigma}_{\hat{\mathcal{S}}} \left( \hat{\boldsymbol{U}}_n - \gamma \hat{\boldsymbol{V}}_n \right)^T \boldsymbol{Q}_m^T \right] \right\|$$

$$\leq \sqrt{m} K_{\boldsymbol{Q}_m}'^2 \, \mathbb{E}[\|\boldsymbol{D}_2^2\|],$$

where $\boldsymbol{D}_2 \in \mathbb{R}^{N \times N}$ is a diagonal matrix for which, for all $i \in [N]$, we have

$$[\boldsymbol{D}_2]_i = \left( \frac{1}{m} \boldsymbol{\sigma}_i^T \hat{\boldsymbol{U}}_n \boldsymbol{Q}_{-i} (\hat{\boldsymbol{U}}_n - \gamma \hat{\boldsymbol{V}}_n)^T \boldsymbol{\sigma}_i - \alpha \right).$$

From both Lemma K.5 and the union bound, we have

$$\Pr\left( \|\boldsymbol{D}_2\| > t \right) = \Pr\left( \max_{1 \leq i \leq N} [\boldsymbol{D}_2]_i > t \right) \leq CN e^{-cm \min(t, t^2)}$$

for some $c, C > 0$ independent of $m$ and $N$. We have thus

$$\mathbb{E}\left( \|\boldsymbol{D}_2\|^2 \right) = \mathbb{E}\left( \max_{1 \leq i \leq N} [\boldsymbol{D}_2^2]_i \right) = \int_0^\infty \Pr\left( \max_{1 \leq i \leq N} [\boldsymbol{D}_2^2]_i > t \right) dt$$

$$= \int_0^\infty 2t \Pr\left( \max_{1 \leq i \leq N} [\boldsymbol{D}_2]_i > t \right) dt$$

$$\leq \int_0^\infty 2t C N e^{-cm \min(t, t^2)} dt$$

$$= \int_0^1 2t C N e^{-cmt^2} dt + \int_1^\infty 2t C N e^{-cmt} dt$$

$$\leq \int_0^\infty 2t C N e^{-cmt^2} dt + \int_0^\infty 2t C N e^{-cmt} dt$$

$$= \frac{1}{m} \frac{2C}{c} \int_0^\infty t N e^{-t^2} dt + \frac{1}{m^2} \frac{2C}{c^2} \int_0^\infty t N e^{-t} dt$$

$$= \mathcal{O}\left( \frac{1}{m} \right).$$

We deduce that

$$\left\| \sum_{i=1}^{N} \mathbb{E}\left[ \frac{1}{m} \boldsymbol{B}_i \boldsymbol{B}_i^T \right] \right\| = \mathcal{O}\left( \frac{1}{\sqrt{m}} \right).$$

In addition, with a similar proof than for Lemma I.4, we can show there exists a real $K_{\tilde{\boldsymbol{Q}}_m}' > 0$ such that, for all $m$, we have

$$\left\| \sqrt{\frac{N}{m}} \sqrt{\frac{1}{1+\alpha}} \bar{\boldsymbol{Z}}^T \hat{\boldsymbol{U}}_n \tilde{\boldsymbol{Q}}_m \right\| \leq K_{\tilde{\boldsymbol{Q}}_m}',$$

where $\bar{\boldsymbol{Z}} \bar{\boldsymbol{Z}}^T$ is the Cholesky decomposition of $\boldsymbol{\Phi}_{\hat{\mathcal{S}}}$. Therefore,

$$\left\| \sum_{i=1}^{N} \mathbb{E}\left[ \frac{1}{m} \boldsymbol{C}_i \boldsymbol{C}_i^T \right] \right\| = \left\| \sum_{i=1}^{N} \mathbb{E}\left[ \frac{1}{m\sqrt{m}} \tilde{\boldsymbol{Q}}_m^T \hat{\boldsymbol{U}}_n^T \boldsymbol{\sigma}_i \boldsymbol{\sigma}_i^T \hat{\boldsymbol{U}}_n \tilde{\boldsymbol{Q}}_m \right] \right\|$$

$$= \left\| \frac{1}{\sqrt{m}} \frac{N}{m} \tilde{\boldsymbol{Q}}_m^T \hat{\boldsymbol{U}}_n^T \boldsymbol{\Phi}_{\hat{\mathcal{S}}} \hat{\boldsymbol{U}}_n \tilde{\boldsymbol{Q}}_m \right\|$$

$$= \mathcal{O}\left( \frac{1}{\sqrt{m}} \right).$$

From equation 41 and above, we deduce that $\boldsymbol{Z}_2$ vanishes under the operator nom, i.e.,

$$\|\boldsymbol{Z}_2\| = \mathcal{O}\left(\frac{1}{\sqrt{m}}\right). \tag{42}$$

Using both equation 40 and equation 42 into equation 36, we conclude that

$$\|\mathbb{E}[\boldsymbol{Q}_m] - \tilde{\boldsymbol{Q}}_m\| = \mathcal{O}\left(\frac{1}{\sqrt{m}}\right). \tag{43}$$

$\square$

To get Theorem 5.1, we start from Lemma E.1 and we show that

$$\|\bar{\boldsymbol{Q}}_m(\lambda) - \tilde{\boldsymbol{Q}}_m(\lambda)\| \to 0,$$

as $N, m \to \infty$.

**Theorem E.2** (Asymptotic Deterministic Resolvent). *Under Assumptions 1 and 2, let $\lambda > 0$ and let $\bar{\boldsymbol{Q}}_m(\lambda) \in \mathbb{R}^{n \times n}$ be the resolvent defined as*

$$\bar{\boldsymbol{Q}}_m(\lambda) = \left[\frac{N}{m}\frac{1}{1+\delta}(\hat{\boldsymbol{U}}_n - \gamma\hat{\boldsymbol{V}}_n)^T \boldsymbol{\Phi}_{\hat{\mathcal{S}}}\hat{\boldsymbol{U}}_n + \lambda\boldsymbol{I}_n\right]^{-1},$$

*where $\delta$ is the correction factor defined as the unique positive solution to*

$$\delta = \frac{1}{m}\mathrm{Tr}\left((\hat{\boldsymbol{U}}_n - \gamma\hat{\boldsymbol{V}}_n)^T \boldsymbol{\Phi}_{\hat{\mathcal{S}}}\hat{\boldsymbol{U}}_n\left[\frac{N}{m}\frac{1}{1+\delta}(\hat{\boldsymbol{U}}_n - \gamma\hat{\boldsymbol{V}}_n)^T \boldsymbol{\Phi}_{\hat{\mathcal{S}}}\hat{\boldsymbol{U}}_n + \lambda\boldsymbol{I}_n\right]^{-1}\right).$$

*Then,*

$$\lim_{m\to\infty}\left\|\mathbb{E}_{\boldsymbol{W}}\left[\boldsymbol{Q}_m(\lambda)\right] - \bar{\boldsymbol{Q}}_m(\lambda)\right\| = 0.$$

*Proof.* From Lemma J.1 in Appendix J, we know that $\delta$ exists and is the unique positive solution of equation 17 under Assumptions 1 and 2. From Lemma E.1 we have a first asymptotic equivalent of $\mathbb{E}_{\boldsymbol{W}}[\boldsymbol{Q}_m]$ given by

$$\tilde{\boldsymbol{Q}}_m = \left[\frac{N}{m}\frac{1}{1+\alpha}\left(\hat{\boldsymbol{U}}_n - \gamma\hat{\boldsymbol{V}}_n\right)^T \boldsymbol{\Phi}_{\hat{\mathcal{S}}}\hat{\boldsymbol{U}}_n + \lambda\boldsymbol{I}_n\right]^{-1},$$

where

$$\alpha = \frac{1}{m}\mathrm{Tr}\left((\hat{\boldsymbol{U}}_n - \gamma\hat{\boldsymbol{V}}_n)^T \boldsymbol{\Phi}_{\hat{\mathcal{S}}}\hat{\boldsymbol{U}}_n\mathbb{E}[\boldsymbol{Q}_-]\right),$$

since

$$\lim_{m\to\infty}\|\mathbb{E}_{\boldsymbol{W}}[\boldsymbol{Q}_m] - \tilde{\boldsymbol{Q}}_m\| = 0.$$

In order to finish the proof of the Theorem, we want to show that

$$\lim_{m\to\infty}\|\tilde{\boldsymbol{Q}}_m - \bar{\boldsymbol{Q}}_m\| = 0. \tag{44}$$

From the resolvent identity (Lemma L.2), we have

$$\|\tilde{\boldsymbol{Q}}_m - \bar{\boldsymbol{Q}}_m\| = \frac{N}{m}\frac{|\alpha - \delta|}{(1+\delta)(1+\alpha)}\left\|\tilde{\boldsymbol{Q}}_m\left(\hat{\boldsymbol{U}}_n - \gamma\hat{\boldsymbol{V}}_n\right)^T \boldsymbol{\Phi}_{\hat{\mathcal{S}}}\hat{\boldsymbol{U}}_n\bar{\boldsymbol{Q}}_m\right\|. \tag{45}$$

Let $\bar{\boldsymbol{Z}}\bar{\boldsymbol{Z}}^T$ be the Cholesky decomposition of $\boldsymbol{\Phi}_{\hat{\mathcal{S}}}$. With a similar proof than for Lemma I.4, we can show there exists a real $K'_{\tilde{\boldsymbol{Q}}} > 0$ such that, for all $m$, we have

$$\left\|\sqrt{\frac{1}{1+\alpha}}\sqrt{\frac{N}{m}}\tilde{\boldsymbol{Q}}_m\left(\hat{\boldsymbol{U}}_n - \gamma\hat{\boldsymbol{V}}_n\right)^T \bar{\boldsymbol{Z}}\right\| \le K'_{\tilde{\boldsymbol{Q}}}.$$

Similarly, we can show there exists a real $K'_{\bar{\boldsymbol{Q}}} > 0$ such that, for all $m$, we have

$$\left\|\sqrt{\frac{1}{1+\delta}}\sqrt{\frac{N}{m}}\bar{\boldsymbol{Z}}^T\hat{\boldsymbol{U}}_n\bar{\boldsymbol{Q}}_m\right\| \le K'_{\bar{\boldsymbol{Q}}}.$$

Therefore,

$$\left\| \tilde{\boldsymbol{Q}}_m \left( \hat{\boldsymbol{U}}_n - \gamma \hat{\boldsymbol{V}}_n \right)^T \boldsymbol{\Phi}_{\hat{\mathcal{S}}} \hat{\boldsymbol{U}}_n \bar{\boldsymbol{Q}}_m \right\| \leq \sqrt{(1+\delta)(1+\alpha)} \frac{m}{N} K'_{\bar{\boldsymbol{Q}}} K'_{\tilde{\boldsymbol{Q}}}.$$

As a consequence, in order to prove equation 44, it remains to prove that

$$\lim_{m \to \infty} |\alpha - \delta| = 0.$$

We decompose $|\alpha - \delta|$ as

$$|\alpha - \delta| = \left| \frac{1}{m} \operatorname{Tr}\left( (\hat{\boldsymbol{U}}_n - \gamma \hat{\boldsymbol{V}}_n)^T \boldsymbol{\Phi}_{\hat{\mathcal{S}}} \hat{\boldsymbol{U}}_n \big[ \mathbb{E}[\boldsymbol{Q}_-] - \bar{\boldsymbol{Q}}_m \big] \right) \right| \tag{46}$$

$$\leq \underbrace{\left| \frac{1}{m} \operatorname{Tr}\left( (\hat{\boldsymbol{U}}_n - \gamma \hat{\boldsymbol{V}}_n)^T \boldsymbol{\Phi}_{\hat{\mathcal{S}}} \hat{\boldsymbol{U}}_n \big[ \mathbb{E}[\boldsymbol{Q}_-] - \tilde{\boldsymbol{Q}}_m \big] \right) \right|}_{= Z_1} + \underbrace{\left| \frac{1}{m} \operatorname{Tr}\left( (\hat{\boldsymbol{U}}_n - \gamma \hat{\boldsymbol{V}}_n)^T \boldsymbol{\Phi}_{\hat{\mathcal{S}}} \hat{\boldsymbol{U}}_n \big[ \tilde{\boldsymbol{Q}}_m - \bar{\boldsymbol{Q}}_m \big] \right) \right|}_{= Z_2}.$$

$$\tag{47}$$

To show $Z_1$ vanishes, we write $\alpha$ as

$$\alpha = \frac{1}{m} \operatorname{Tr}\left( (\hat{\boldsymbol{U}}_n - \gamma \hat{\boldsymbol{V}}_n)^T \boldsymbol{\Phi}_{\hat{\mathcal{S}}} \hat{\boldsymbol{U}}_n \mathbb{E}[\boldsymbol{Q}_-] \right)$$

$$= \frac{1}{m} \operatorname{Tr}\left( (\hat{\boldsymbol{U}}_n - \gamma \hat{\boldsymbol{V}}_n)^T \boldsymbol{\Phi}_{\hat{\mathcal{S}}} \hat{\boldsymbol{U}}_n \mathbb{E}[\boldsymbol{Q}_m] \right) + \frac{1}{m} \operatorname{Tr}\left( (\hat{\boldsymbol{U}}_n - \gamma \hat{\boldsymbol{V}}_n)^T \boldsymbol{\Phi}_{\hat{\mathcal{S}}} \hat{\boldsymbol{U}}_n \big[ \mathbb{E}[\boldsymbol{Q}_-] - \mathbb{E}[\boldsymbol{Q}_m] \big] \right).$$

There exists a real $K > 0$ such that

$$\frac{1}{m} \operatorname{Tr}\left( (\hat{\boldsymbol{U}}_n - \gamma \hat{\boldsymbol{V}}_n)^T \boldsymbol{\Phi}_{\hat{\mathcal{S}}} \hat{\boldsymbol{U}}_n \big[ \mathbb{E}[\boldsymbol{Q}_-] - \mathbb{E}[\boldsymbol{Q}_m] \big] \right) \leq K \left\| \big[ \mathbb{E}[\boldsymbol{Q}_-] - \mathbb{E}[\boldsymbol{Q}_m] \big] \right\|;$$

since both $\|\hat{\boldsymbol{U}}_n\|$ and $\|\hat{\boldsymbol{V}}_n\|$ are upper bounded by 1, $|\operatorname{Tr}(\boldsymbol{AB})| \leq \|\boldsymbol{A}\| \operatorname{Tr}(\boldsymbol{B})$ for non-negative definite matrix $\boldsymbol{B}$, and from equation 30 that uniformly bounds $\frac{1}{m} \operatorname{Tr}(\boldsymbol{\Phi}_{\hat{\mathcal{S}}})$. From Lemma E.4, we have

$$\|\mathbb{E}[\boldsymbol{Q}_m - \boldsymbol{Q}_-]\| = \left\| \frac{1}{m} \frac{m}{N} \sum_{i=1}^N \mathbb{E}[\boldsymbol{Q}_m - \boldsymbol{Q}_{-i}] \right\|$$

$$= \left\| \frac{1}{m^2} \frac{m}{N} \mathbb{E} \left[ \boldsymbol{Q}_m (\hat{\boldsymbol{U}}_n - \gamma \hat{\boldsymbol{V}}_n)^T \boldsymbol{\Sigma}_{\hat{\mathcal{S}}}^T \boldsymbol{D} \boldsymbol{\Sigma}_{\hat{\mathcal{S}}} \hat{\boldsymbol{U}}_n \boldsymbol{Q}_m \right] \right\|$$

$$= \mathcal{O}\left( \frac{1}{m} \right),$$

where $\boldsymbol{D} \in \mathbb{R}^{N \times N}$ is a diagonal matrix for which, for all $i \in [N]$, we have

$$\boldsymbol{D}_i = \left( 1 + \frac{1}{m} \boldsymbol{\sigma}_i^T \hat{\boldsymbol{U}}_n \boldsymbol{Q}_{-i} (\hat{\boldsymbol{U}}_n - \gamma \hat{\boldsymbol{V}}_n)^T \boldsymbol{\sigma}_i \right).$$

As a consequence, by combining the results above and from Lemma E.1, we conclude for $Z_1$ that

$$|Z_1| = \left| \alpha - \frac{1}{m} \operatorname{Tr}\left( (\hat{\boldsymbol{U}}_n - \gamma \hat{\boldsymbol{V}}_n)^T \boldsymbol{\Phi}_{\hat{\mathcal{S}}} \hat{\boldsymbol{U}}_n \tilde{\boldsymbol{Q}}_m \right) \right| = \mathcal{O}\left( \frac{1}{\sqrt{m}} \right).$$

Using the vanishing result of $Z_1$ into equation 47 and applying the resolvent identity (Lemma L.2) on $Z_2$, we get

$$|\alpha - \delta| \leq \frac{N}{m} \frac{|\alpha - \delta|}{(1+\delta)(1+\alpha)} \left| \frac{1}{m} \operatorname{Tr}\left( (\hat{\boldsymbol{U}}_n - \gamma \hat{\boldsymbol{V}}_n)^T \boldsymbol{\Phi}_{\hat{\mathcal{S}}} \hat{\boldsymbol{U}}_n \tilde{\boldsymbol{Q}}_m (\hat{\boldsymbol{U}}_n - \gamma \hat{\boldsymbol{V}}_n)^T \boldsymbol{\Phi}_{\hat{\mathcal{S}}} \hat{\boldsymbol{U}}_n \bar{\boldsymbol{Q}}_m \right) \right| + \mathcal{O}\left( \frac{1}{\sqrt{m}} \right),$$

which implies that

$$|\alpha - \delta| \left( 1 - \frac{N}{m} \frac{1}{(1+\delta)(1+\alpha)} \frac{1}{m} \operatorname{Tr}\left( (\hat{\boldsymbol{U}}_n - \gamma \hat{\boldsymbol{V}}_n)^T \boldsymbol{\Phi}_{\hat{\mathcal{S}}} \hat{\boldsymbol{U}}_n \tilde{\boldsymbol{Q}}_m (\hat{\boldsymbol{U}}_n - \gamma \hat{\boldsymbol{V}}_n)^T \boldsymbol{\Phi}_{\hat{\mathcal{S}}} \hat{\boldsymbol{U}}_n \bar{\boldsymbol{Q}}_m \right) \right) = \mathcal{O}\left( \frac{1}{\sqrt{m}} \right).$$

It remains to show

$$\lim_{m \to \infty} \sup \frac{1}{m} \frac{N}{m} \frac{1}{m} \frac{1}{(1+\delta)(1+\alpha)} \operatorname{Tr}\left( (\hat{\boldsymbol{U}}_n - \gamma \hat{\boldsymbol{V}}_n)^T \boldsymbol{\Phi}_{\hat{\mathcal{S}}} \hat{\boldsymbol{U}}_n \tilde{\boldsymbol{Q}}_m (\hat{\boldsymbol{U}}_n - \gamma \hat{\boldsymbol{V}}_n)^T \boldsymbol{\Phi}_{\hat{\mathcal{S}}} \hat{\boldsymbol{U}}_n \bar{\boldsymbol{Q}}_m \right) < 1.$$

Let the matrices $\boldsymbol{B}_n = (\hat{\boldsymbol{U}}_n - \gamma\hat{\boldsymbol{V}}_n)^T \boldsymbol{\Phi}_{\hat{\mathcal{S}}} \hat{\boldsymbol{U}}_n$, $\boldsymbol{B}'_n = \bar{\boldsymbol{Z}}^T \hat{\boldsymbol{A}}_m \boldsymbol{Z}$, $\bar{\boldsymbol{Q}}'_m = \left[\frac{N}{m}\frac{1}{1+\delta}\boldsymbol{B}'_n + \lambda\boldsymbol{I}_m\right]^{-1}$, and

$\tilde{\boldsymbol{Q}}'_m = \left[\frac{N}{m}\frac{1}{1+\alpha}\boldsymbol{B}'_n + \lambda\boldsymbol{I}_m\right]^{-1}$; where $\hat{\boldsymbol{A}}_m = \hat{\boldsymbol{U}}_n(\hat{\boldsymbol{U}}_n - \gamma\hat{\boldsymbol{V}}_n)^T$ is the empirical transition model matrix defined in equation 14. Using the Cauchy–Schwarz inequality, we write

$$
\frac{1}{m}\frac{N}{m}\frac{1}{(1+\delta)(1+\alpha)}\operatorname{Tr}\big(\boldsymbol{B}_n\tilde{\boldsymbol{Q}}_m\boldsymbol{B}_n\bar{\boldsymbol{Q}}_m\big)
$$

$$
= \frac{1}{m}\frac{N}{m}\frac{1}{(1+\delta)(1+\alpha)}\operatorname{Tr}\big(\boldsymbol{B}'_n\tilde{\boldsymbol{Q}}'_m\boldsymbol{B}'_n\bar{\boldsymbol{Q}}'_m\big)
$$

$$
\leq \sqrt{\underbrace{\frac{N}{m}\frac{1}{m}\frac{1}{(1+\delta)^2}\operatorname{Tr}\big(\boldsymbol{B}'_n\bar{\boldsymbol{Q}}'_m\bar{\boldsymbol{Q}}'^T_m\boldsymbol{B}'^T_n\big)}_{Z'_1}\underbrace{\frac{N}{m}\frac{1}{m}\frac{1}{(1+\alpha)^2}\operatorname{Tr}\big(\boldsymbol{B}'_n\tilde{\boldsymbol{Q}}'_m\tilde{\boldsymbol{Q}}'^T_m\boldsymbol{B}'^T_n\big)}_{Z'_2}}.
$$

We observe that

$$
\delta = \frac{1}{m}\operatorname{Tr}\big(\boldsymbol{B}_n\bar{\boldsymbol{Q}}_m\big) = \frac{1}{m}\operatorname{Tr}\big(\boldsymbol{B}'_n\bar{\boldsymbol{Q}}'_m\big) = \frac{1}{m}\operatorname{Tr}\big(\boldsymbol{B}'_n\bar{\boldsymbol{Q}}'_m\boldsymbol{Q}'^T_m\boldsymbol{Q}'^{-1T}_m\big)
$$

$$
= \frac{1}{m}\frac{N}{m}\frac{1+\delta}{(1+\delta)^2}\operatorname{Tr}\big(\boldsymbol{B}'_n\bar{\boldsymbol{Q}}'_m\bar{\boldsymbol{Q}}'^T_m\boldsymbol{B}'^T_n\big) + \frac{\lambda}{m}\operatorname{Tr}\big(\boldsymbol{B}'_n\bar{\boldsymbol{Q}}'_m\bar{\boldsymbol{Q}}'^T_m\big).
$$

Since $H(\boldsymbol{B}'_n)$ is at least semi-positive-definite under Assumption 2, we have

$$
\operatorname{Tr}\big(\boldsymbol{B}'_n\bar{\boldsymbol{Q}}'_m\bar{\boldsymbol{Q}}'^T_m\big) = \operatorname{Tr}\big(\bar{\boldsymbol{Q}}'^T_m\boldsymbol{B}'_n\bar{\boldsymbol{Q}}'_m\big) = \operatorname{Tr}\big(\bar{\boldsymbol{Q}}'^T_m H(\boldsymbol{B}'_n)\bar{\boldsymbol{Q}}'_m\big) \geq 0.
$$

As a consequence we have

$$
\frac{1}{m}\frac{N}{m}\frac{1}{(1+\delta)^2}\operatorname{Tr}\big(\boldsymbol{B}'_n\bar{\boldsymbol{Q}}'_m\bar{\boldsymbol{Q}}'^T_m\boldsymbol{B}'^T_n\big) \leq \frac{\delta - \frac{\lambda}{m}\operatorname{Tr}\big(\boldsymbol{B}'_n\bar{\boldsymbol{Q}}'_m\bar{\boldsymbol{Q}}'^T_m\big)}{1+\delta} \leq \frac{\delta}{1+\delta}.
$$

To prove $\frac{\delta}{1+\delta} < 1$, it remains to show that $\delta < \infty$. With a similar proof than for Lemma I.1, we can show there exists a real $K_{\bar{Q}} > 0$ such that, for all $m$, we have $\|\bar{\boldsymbol{Q}}_m\| \leq K_{\bar{Q}}$, and thus

$$
\delta = \frac{1}{m}\operatorname{Tr}\big(\boldsymbol{B}_n\bar{\boldsymbol{Q}}_m\big) = \frac{1}{m}\operatorname{Tr}\big((\hat{\boldsymbol{U}}_n - \gamma\hat{\boldsymbol{V}}_n)^T\boldsymbol{\Phi}_{\hat{\mathcal{S}}}\hat{\boldsymbol{U}}_n\bar{\boldsymbol{Q}}_m\big) \leq \frac{2}{m}\operatorname{Tr}(\boldsymbol{\Phi}_{\hat{\mathcal{S}}})\|\bar{\boldsymbol{Q}}_m(\delta)\| \leq \frac{2}{m}\operatorname{Tr}(\boldsymbol{\Phi}_{\hat{\mathcal{S}}})K_{\bar{Q}} < \infty
$$

where we used for the first inequality the relation $|\operatorname{Tr}(\boldsymbol{AB})| \leq \|\boldsymbol{A}\|\operatorname{Tr}(\boldsymbol{B})$ for non-negative definite matrix $\boldsymbol{B}$. Furthermore, from equation 30, $\frac{1}{m}\operatorname{Tr}(\boldsymbol{\Phi}_{\hat{\mathcal{S}}})$ is bounded under Assumptions 1 and 2, and both $\|\hat{\boldsymbol{U}}_n\|$ and $\|\hat{\boldsymbol{V}}_n\|$ are upper bounded by 1. We thus conclude for $Z'_1$ that

$$
\limsup_m \frac{1}{m}\frac{N}{m}\frac{1}{(1+\delta)^2}\operatorname{Tr}\big(\boldsymbol{B}'_n\bar{\boldsymbol{Q}}'_m\bar{\boldsymbol{Q}}'^T_m\boldsymbol{B}'^T_n\big) < 1. \tag{48}
$$

With similar arguments, we can show for $Z'_2$ that

$$
\limsup_m \frac{1}{m}\frac{N}{m}\frac{1}{(1+\alpha)^2}\operatorname{Tr}\big(\boldsymbol{B}'_n\tilde{\boldsymbol{Q}}'_m\tilde{\boldsymbol{Q}}'^T_m\boldsymbol{B}'^T_n\big) < 1,
$$

which concludes the proof that

$$
|\alpha - \delta| = \mathcal{O}\left(\frac{1}{\sqrt{m}}\right). \tag{49}
$$

Using the result above with equation 45, we get

$$
\|\tilde{\boldsymbol{Q}}_m - \bar{\boldsymbol{Q}}_m\| = |\alpha - \delta|\left\|\frac{N}{m}\frac{1}{(1+\delta)(1+\alpha)}\tilde{\boldsymbol{Q}}_m\left(\hat{\boldsymbol{U}}_n - \gamma\hat{\boldsymbol{V}}_n\right)^T\boldsymbol{\Phi}_{\hat{\mathcal{S}}}\hat{\boldsymbol{U}}_n\bar{\boldsymbol{Q}}_m\right\|
$$

$$
= \mathcal{O}\left(\frac{1}{\sqrt{m}}\right),
$$

which concludes the proof. $\qquad\square$

**Lemma E.3.** *Under Assumptions 1 and 2, let $\boldsymbol{D} \in \mathbb{R}^{N \times N}$ be the diagonal matrix defined in equation 37 for which, for all $i \in [N]$, we have*

$$\boldsymbol{D}_i = 1 + \frac{1}{m} \boldsymbol{\sigma}_i^T \hat{\boldsymbol{U}}_n \boldsymbol{Q}_{-i} (\hat{\boldsymbol{U}}_n - \gamma \hat{\boldsymbol{V}}_n)^T \boldsymbol{\sigma}_i. \tag{50}$$

*Then*

$$\mathbb{E} \left[ \|\boldsymbol{D}\| \right] = \mathcal{O}(1).$$

*Proof.* Let $\alpha = \frac{1}{m} \text{Tr} \left( (\hat{\boldsymbol{U}}_n - \gamma \hat{\boldsymbol{V}}_n)^T \boldsymbol{\Phi}_{\hat{\mathcal{S}}} \hat{\boldsymbol{U}}_n \mathbb{E}[\boldsymbol{Q}_-(\lambda)] \right)$ defined in equation 28. From equation 31, $\alpha$ is uniformly bounded, i.e., there exists a real $K_\alpha > 0$ such that $\alpha \leq K_\alpha$. From both Lemma K.5 and the union bound, we have

$$\Pr \left( \|\boldsymbol{D}\| > 1 + \alpha + t \right) = \Pr \left( \max_{1 \leq i \leq N} \boldsymbol{D}_i > 1 + \alpha + t \right) \leq CN e^{-cm \min(t, t^2)},$$

for some $c, C > 0$ independent of $m$ and $N$. Therefore,

$$
\begin{aligned}
\mathbb{E} \left[ \|\boldsymbol{D}\| \right] = \mathbb{E} \left[ \max_{1 \leq i \leq N} \boldsymbol{D}_i \right] &= \int_0^\infty \Pr \left( \max_{1 \leq i \leq N} \boldsymbol{D}_i > t \right) dt \\
&= \int_0^{2(1+K_\alpha)} \Pr \left( \max_{1 \leq i \leq N} \boldsymbol{D}_i > t \right) dt + \int_{2(1+K_\alpha)}^\infty \Pr \left( \max_{1 \leq i \leq N} \boldsymbol{D}_i > t \right) dt \\
&\leq 2(1 + K_\alpha) + \int_{2(1+K_\alpha)}^\infty CN e^{-cm \min \left( (t - (1+K_\alpha))^2, t - (1+K_\alpha) \right)} dt \\
&= 2(1 + K_\alpha) + \int_{1+K_\alpha}^\infty CN e^{-cmt} dt \\
&= 2(1 + K_\alpha) + \frac{CN}{cm} e^{-Cm(1+K_\alpha)} \\
&= \mathcal{O}(1).
\end{aligned}
$$

$\square$

**Lemma E.4.** *Under Assumptions 1 and 2, let $\boldsymbol{D} \in \mathbb{R}^{N \times N}$ be the diagonal matrix defined in equation 37 for which, for all $i \in [N]$, we have*

$$\boldsymbol{D}_i = 1 + \frac{1}{m} \boldsymbol{\sigma}_i^T \hat{\boldsymbol{U}}_n \boldsymbol{Q}_{-i} (\hat{\boldsymbol{U}}_n - \gamma \hat{\boldsymbol{V}}_n)^T \boldsymbol{\sigma}_i.$$

*Then*

$$\left\| \mathbb{E} \left[ \frac{1}{m} \boldsymbol{Q}_m (\hat{\boldsymbol{U}}_n - \gamma \hat{\boldsymbol{V}}_n)^T \boldsymbol{\Sigma}_{\hat{\mathcal{S}}}^T \boldsymbol{D} \boldsymbol{\Sigma}_{\hat{\mathcal{S}}} \hat{\boldsymbol{U}}_n \boldsymbol{Q}_m \right] \right\| = \mathcal{O}(1).$$

*Proof.* From Lemma I.4, there exists $K'_{\boldsymbol{Q}_m} > 0$ such that, for all $m$, we have $\left\| \frac{1}{\sqrt{m}} \boldsymbol{Q}_m (\hat{\boldsymbol{U}}_n - \gamma \hat{\boldsymbol{V}}_n)^T \boldsymbol{\Sigma}_{\hat{\mathcal{S}}} \right\| \leq 2K'_{\boldsymbol{Q}_m}$ and $\left\| \frac{1}{\sqrt{m}} \boldsymbol{\Sigma}_{\hat{\mathcal{S}}} \hat{\boldsymbol{U}}_n \boldsymbol{Q}_m \right\| \leq K'_{\boldsymbol{Q}_m}$. Therefore,

$$\left\| \mathbb{E} \left[ \frac{1}{m} \boldsymbol{Q}_m (\hat{\boldsymbol{U}}_n - \gamma \hat{\boldsymbol{V}}_n)^T \boldsymbol{\Sigma}_{\hat{\mathcal{S}}}^T \boldsymbol{D} \boldsymbol{\Sigma}_{\hat{\mathcal{S}}} \hat{\boldsymbol{U}}_n \boldsymbol{Q}_m \right] \right\| \leq 2K'^2_{\boldsymbol{Q}_m} \mathbb{E} \left[ \|\boldsymbol{D}\| \right].$$

From Lemma E.3, we have

$$\mathbb{E} \left[ \|\boldsymbol{D}\| \right] = \mathcal{O}(1).$$

As a consequence, we deduce that

$$\left\| \mathbb{E} \left[ \frac{1}{m} \boldsymbol{Q}_m (\hat{\boldsymbol{U}}_n - \gamma \hat{\boldsymbol{V}}_n)^T \boldsymbol{\Sigma}_{\hat{\mathcal{S}}}^T \boldsymbol{D} \boldsymbol{\Sigma}_{\hat{\mathcal{S}}} \hat{\boldsymbol{U}}_n \boldsymbol{Q}_m \right] \right\| = \mathcal{O}(1). \tag{51}$$

$\square$

# F  PROOF OF THEOREM 5.2

This section is dedicated to find an asymptotic deterministic limit of the empirical $\widehat{\text{MSBE}}(\hat{\boldsymbol{\theta}})$ (equation 8) under Assumptions 1 and 2. First, we observe that the empirical MSBE depends on the quadratic form $\boldsymbol{r}^T \boldsymbol{Q}_m(\lambda)^T \boldsymbol{Q}_m(\lambda) \boldsymbol{r}$ since

$$\widehat{\text{MSBE}}(\hat{\boldsymbol{\theta}}) = \frac{1}{n} \| \boldsymbol{r} + \gamma \boldsymbol{\Sigma}_{\boldsymbol{X}'_n}^T \hat{\boldsymbol{\theta}} - \boldsymbol{\Sigma}_{\boldsymbol{X}_n}^T \hat{\boldsymbol{\theta}} \|^2 \tag{52}$$

$$= \frac{1}{n} \left\| \boldsymbol{r} - \frac{1}{mn} \left( \boldsymbol{\Sigma}_{\boldsymbol{X}_n} - \gamma \boldsymbol{\Sigma}_{\boldsymbol{X}'_n} \right)^T \boldsymbol{\Sigma}_{\boldsymbol{X}_n} \boldsymbol{Q}_m(\lambda) \boldsymbol{r} \right\|^2 \tag{53}$$

$$= \frac{1}{n} \left\| \left[ \frac{1}{mn} \left( \boldsymbol{\Sigma}_{\boldsymbol{X}_n} - \gamma \boldsymbol{\Sigma}_{\boldsymbol{X}'_n} \right)^T \boldsymbol{\Sigma}_{\boldsymbol{X}_n} + \lambda \boldsymbol{I}_n - \frac{1}{mn} \left( \boldsymbol{\Sigma}_{\boldsymbol{X}_n} - \gamma \boldsymbol{\Sigma}_{\boldsymbol{X}'_n} \right)^T \boldsymbol{\Sigma}_{\boldsymbol{X}_n} \right] \boldsymbol{Q}_m(\lambda) \boldsymbol{r} \right\|^2 \tag{54}$$

$$= \frac{\lambda^2}{n} \| \boldsymbol{Q}_m(\lambda) \boldsymbol{r} \|^2 . \tag{55}$$

We determine in Theorem 5.2 a deterministic limit of $\widehat{\text{MSBE}}(\hat{\boldsymbol{\theta}})$ by combining Theorem 5.1, which provides an asymptotically more tractable approximation of $\mathbb{E}_{\boldsymbol{W}} \left[ \boldsymbol{Q}_m(\lambda) \right]$ under the form of a fixed-point equation, with concentration arguments. Theorem 5.2 is corollary of Lemma F.2 and of the concentration result of Lemma K.2 in Section K. Both Lemma F.4 and Lemma F.5 are key Lemma used in the proof of Theorem 5.2 and Theorem 5.3.

To simplify the notations, we denote the matrix $\boldsymbol{Q}_m$ as the resolvent $\boldsymbol{Q}_m(\lambda)$ (defined in equation 13). We define the matrix $\boldsymbol{\Psi}_{\hat{\mathcal{S}}} \in \mathbb{R}^{m \times m}$ as

$$\boldsymbol{\Psi}_{\hat{\mathcal{S}}} = \frac{N}{m} \frac{1}{1+\delta} \boldsymbol{\Phi}_{\hat{\mathcal{S}}}.$$

Furthermore, the notation $\boldsymbol{A} = \boldsymbol{B} + \mathcal{O}_{\|\cdot\|} \left( \frac{1}{\sqrt{m}} \right)$ means that $\|\boldsymbol{A} - \boldsymbol{B}\| = \mathcal{O} \left( \frac{1}{\sqrt{m}} \right)$.

**Theorem F.1** (Asymptotic Empirical MSBE). *Under the conditions of Theorem 5.1, the deterministic asymptotic empirical MSBE is*

$$\widehat{\widehat{\text{MSBE}}}(\hat{\boldsymbol{\theta}}) = \tfrac{\lambda^2}{n} \| \bar{\boldsymbol{Q}}_m(\lambda) \boldsymbol{r} \|^2 + \hat{\Delta},$$

*with second-order correction factor*

$$\hat{\Delta} = \tfrac{\lambda^2}{n} \frac{\frac{1}{N} \text{Tr} \left( \bar{\boldsymbol{Q}}_m(\lambda) \boldsymbol{\Psi}_2 \bar{\boldsymbol{Q}}_m(\lambda)^T \right)}{1 - \frac{1}{N} \text{Tr} \left( \bar{\boldsymbol{Q}}_m(\lambda) \boldsymbol{\Psi}_2 \bar{\boldsymbol{Q}}_m(\lambda)^T \boldsymbol{\Psi}_1 \right)} \| \bar{\boldsymbol{Q}}_m(\lambda) \boldsymbol{r} \|_{\boldsymbol{\Psi}_1}^2,$$

*where*

$$\boldsymbol{\Psi}_1 = \hat{\boldsymbol{U}}_n^T \boldsymbol{\Psi}_{\hat{\mathcal{S}}} \hat{\boldsymbol{U}}_n, \quad \text{and} \quad \boldsymbol{\Psi}_2 = (\hat{\boldsymbol{U}}_n - \gamma \hat{\boldsymbol{V}}_n)^T \boldsymbol{\Psi}_{\hat{\mathcal{S}}} (\hat{\boldsymbol{U}}_n - \gamma \hat{\boldsymbol{V}}_n).$$

*As $N, m, d \to \infty$ with asymptotic constant ratio $N/m$,*

$$\widehat{\text{MSBE}}(\hat{\boldsymbol{\theta}}) - \widehat{\widehat{\text{MSBE}}}(\hat{\boldsymbol{\theta}}) \xrightarrow{a.s} 0.$$

*Proof.* From equation 55, we have

$$\widehat{\text{MSBE}}(\hat{\boldsymbol{\theta}}) = \frac{\lambda^2}{n} \| \boldsymbol{Q}_m \, \boldsymbol{r} \|^2 = \frac{\lambda^2}{n} \boldsymbol{r}^T \boldsymbol{Q}_m^T \boldsymbol{Q}_m \boldsymbol{r}.$$

From Lemma K.6, we have

$$\Pr \left( \left| \frac{\lambda^2}{n} \boldsymbol{r}^T \boldsymbol{Q}_m^T \boldsymbol{Q}_m \boldsymbol{r} - \frac{\lambda^2}{n} \boldsymbol{r}^T \mathbb{E}[\boldsymbol{Q}_m^T \boldsymbol{Q}_m] \boldsymbol{r} \right| > t \right) \leq C e^{-c n^2 m t^2},$$

for some $C, c > 0$ independent of $m, n$ and $N$. Furthermore, from Lemma F.2, we have

$$\left\| \mathbb{E}[\boldsymbol{Q}_m^T \boldsymbol{Q}_m] - \bar{\boldsymbol{Q}}_m^T \bar{\boldsymbol{Q}}_m - \frac{\frac{1}{N} \text{Tr} \left( \boldsymbol{\Psi}_2 \bar{\boldsymbol{Q}}_m^T \bar{\boldsymbol{Q}}_m \right)}{1 - \frac{1}{N} \text{Tr} \left( \boldsymbol{\Psi}_2 \bar{\boldsymbol{Q}}_m^T \boldsymbol{\Psi}_1 \bar{\boldsymbol{Q}}_m \right)} \bar{\boldsymbol{Q}}_m^T \boldsymbol{\Psi}_1 \bar{\boldsymbol{Q}}_m \right\| = \mathcal{O} \left( \frac{1}{\sqrt{m}} \right).$$

As a consequence, we have

$$\widehat{\text{MSBE}}(\hat{\boldsymbol{\theta}}) - \widehat{\widehat{\text{MSBE}}}(\hat{\boldsymbol{\theta}}) \xrightarrow{a.s} 0,$$

as $m \to \infty$. $\qquad\square$

**Lemma F.2.** *Under Assumptions 1 and 2, let $\boldsymbol{Q}_m \in \mathbb{R}^{n \times n}$ be the resolvent defined in equation 13, let $\bar{\boldsymbol{Q}}_m \in \mathbb{R}^{n \times n}$ be the deterministic resolvent defined in equation 15, and let $\boldsymbol{M} \in \mathbb{R}^{n \times n}$ be any matrix with a bounded operator norm. Then,*

$$\left\| \mathbb{E}\left[\boldsymbol{Q}_m^T \boldsymbol{M} \boldsymbol{Q}_m\right] - \bar{\boldsymbol{Q}}_m^T \boldsymbol{M} \bar{\boldsymbol{Q}}_m - \frac{\frac{1}{N} \operatorname{Tr}\left(\boldsymbol{\Psi}_2 \bar{\boldsymbol{Q}}_m^T \boldsymbol{M} \bar{\boldsymbol{Q}}_m\right)}{1 - \frac{1}{N} \operatorname{Tr}\left(\boldsymbol{\Psi}_2 \bar{\boldsymbol{Q}}_m^T \boldsymbol{\Psi}_1 \bar{\boldsymbol{Q}}_m\right)} \bar{\boldsymbol{Q}}_m^T \boldsymbol{\Psi}_1 \bar{\boldsymbol{Q}}_m \right\| = \mathcal{O}\left(\frac{1}{\sqrt{m}}\right),$$

*for $\boldsymbol{\Psi}_1, \boldsymbol{\Psi}_2 \in \mathbb{R}^{n \times n}$ defined in equation 19.*

*Proof.* From Lemma F.6, we have

$$\mathbb{E}\left[\boldsymbol{Q}_m^T \boldsymbol{M} \boldsymbol{Q}_m\right] = \bar{\boldsymbol{Q}}_m^T \boldsymbol{M} \bar{\boldsymbol{Q}}_m + \mathbb{E}\left[\boldsymbol{Q}_m^T \boldsymbol{M} \bar{\boldsymbol{Q}}_m (\hat{\boldsymbol{U}}_n - \gamma \hat{\boldsymbol{V}}_n)^T \boldsymbol{\Psi}_{\hat{\mathcal{S}}} \hat{\boldsymbol{U}}_n \boldsymbol{Q}_m\right]$$
$$- \mathbb{E}\left[\boldsymbol{Q}_-^T \boldsymbol{M} \bar{\boldsymbol{Q}}_m (\hat{\boldsymbol{U}}_n - \gamma \hat{\boldsymbol{V}}_n)^T \boldsymbol{\Psi}_{\hat{\mathcal{S}}} \hat{\boldsymbol{U}}_n \boldsymbol{Q}_-\right] + \frac{1}{N} \operatorname{Tr}\left(\boldsymbol{\Psi}_2 \bar{\boldsymbol{Q}}_m^T \boldsymbol{M} \bar{\boldsymbol{Q}}_m\right) \mathbb{E}\left[\boldsymbol{Q}_-^T \boldsymbol{\Psi}_1 \boldsymbol{Q}_-\right]$$
$$+ \mathcal{O}_{\|\cdot\|}\left(\frac{1}{\sqrt{m}}\right).$$

Let

$$\boldsymbol{M}' = \boldsymbol{M} \bar{\boldsymbol{Q}}_m (\hat{\boldsymbol{U}}_n - \gamma \hat{\boldsymbol{V}}_n)^T \boldsymbol{\Psi}_{\hat{\mathcal{S}}} \hat{\boldsymbol{U}}_n = \boldsymbol{M}\left[\boldsymbol{I}_n - \lambda \bar{\boldsymbol{Q}}_m\right]$$

With a similar proof than for Lemma I.1, we can show that there exists a real $K_{\bar{\boldsymbol{Q}}}$ such that, for all $m$, we have $\|\bar{\boldsymbol{Q}}_m\| \leq K_{\bar{\boldsymbol{Q}}}$. We deduce thus that $\boldsymbol{M}'$ is a matrix with a bounded operator norm since $\|\boldsymbol{M}'\| \leq (1 + \lambda K_{\bar{\boldsymbol{Q}}}) \|\boldsymbol{M}\|$. From Lemma F.3, we have

$$\left\| \mathbb{E}\left[\boldsymbol{Q}_m^T \boldsymbol{M} \bar{\boldsymbol{Q}}_m (\hat{\boldsymbol{U}}_n - \gamma \hat{\boldsymbol{V}}_n)^T \boldsymbol{\Psi}_{\hat{\mathcal{S}}} \hat{\boldsymbol{U}}_n \boldsymbol{Q}_m\right] - \mathbb{E}\left[\boldsymbol{Q}_-^T \boldsymbol{M} \bar{\boldsymbol{Q}}_m (\hat{\boldsymbol{U}}_n - \gamma \hat{\boldsymbol{V}}_n)^T \boldsymbol{\Psi}_{\hat{\mathcal{S}}} \hat{\boldsymbol{U}}_n \boldsymbol{Q}_-\right]\right\| = \mathcal{O}\left(\frac{1}{m}\right).$$

Therefore,

$$\mathbb{E}\left[\boldsymbol{Q}_m^T \boldsymbol{M} \boldsymbol{Q}_m\right] = \bar{\boldsymbol{Q}}_m^T \boldsymbol{M} \bar{\boldsymbol{Q}}_m + \frac{1}{N} \operatorname{Tr}\left(\boldsymbol{\Psi}_2 \bar{\boldsymbol{Q}}_m^T \boldsymbol{M} \bar{\boldsymbol{Q}}_m\right) \mathbb{E}\left[\boldsymbol{Q}_-^T \boldsymbol{\Psi}_1 \boldsymbol{Q}_-\right] + \mathcal{O}_{\|\cdot\|}\left(\frac{1}{\sqrt{m}}\right).$$

Furthermore, from Lemma F.4, we have

$$\left\| \mathbb{E}\left[\boldsymbol{Q}_m^T \boldsymbol{\Psi}_1 \boldsymbol{Q}_m\right] - \mathbb{E}\left[\boldsymbol{Q}_-^T \boldsymbol{\Psi}_1 \boldsymbol{Q}_-\right]\right\| = \mathcal{O}\left(\frac{1}{\sqrt{m}}\right),$$

and from Lemma F.5 we have

$$\mathbb{E}\left[\boldsymbol{Q}_m^T \boldsymbol{\Psi}_1 \boldsymbol{Q}_m\right] = \bar{\boldsymbol{Q}}_m^T \boldsymbol{\Psi}_1 \bar{\boldsymbol{Q}}_m + \frac{1}{1 - \frac{1}{N} \operatorname{Tr}\left(\boldsymbol{\Psi}_2 \bar{\boldsymbol{Q}}_m^T \boldsymbol{\Psi}_1 \bar{\boldsymbol{Q}}_m\right)} \bar{\boldsymbol{Q}}_m^T \boldsymbol{\Psi}_1 \bar{\boldsymbol{Q}}_m + \mathcal{O}_{\|\cdot\|}\left(\frac{1}{\sqrt{m}}\right).$$

We conclude thus

$$\mathbb{E}\left[\boldsymbol{Q}_m^T \boldsymbol{M} \boldsymbol{Q}_m\right] = \bar{\boldsymbol{Q}}_m^T \boldsymbol{M} \bar{\boldsymbol{Q}}_m + \frac{\frac{1}{N} \operatorname{Tr}\left(\boldsymbol{\Psi}_2 \bar{\boldsymbol{Q}}_m^T \boldsymbol{M} \bar{\boldsymbol{Q}}_m\right)}{1 - \frac{1}{N} \operatorname{Tr}\left(\boldsymbol{\Psi}_2 \bar{\boldsymbol{Q}}_m^T \boldsymbol{\Psi}_1 \bar{\boldsymbol{Q}}_m\right)} \bar{\boldsymbol{Q}}_m^T \boldsymbol{\Psi}_1 \bar{\boldsymbol{Q}}_m + \mathcal{O}_{\|\cdot\|}\left(\frac{1}{\sqrt{m}}\right).$$

$\square$

**Lemma F.3.** *Under Assumptions 1 and 2, let $\boldsymbol{M} \in \mathbb{R}^{n \times n}$ be any matrix with a bounded operator norm, let $\boldsymbol{Q}_m \in \mathbb{R}^{n \times n}$ be the resolvent defined in equation 13, and let $\boldsymbol{Q}_- \in \mathbb{R}^{n \times n}$ be the resolvent defined in equation 29. Then,*

$$\left\| \mathbb{E}\left[\boldsymbol{Q}_m^T \boldsymbol{M} \boldsymbol{Q}_m\right] - \mathbb{E}\left[\boldsymbol{Q}_-^T \boldsymbol{M} \boldsymbol{Q}_-\right]\right\| = \mathcal{O}\left(\frac{1}{m}\right).$$

*Proof.* We observe that

$$\left\| \mathbb{E}\left[\boldsymbol{Q}_m^T \boldsymbol{M} \boldsymbol{Q}_m\right] - \mathbb{E}\left[\boldsymbol{Q}_-^T \boldsymbol{M} \boldsymbol{Q}_-\right]\right\| \leq \left\| \mathbb{E}\left[\boldsymbol{Q}_m^T \boldsymbol{M} \boldsymbol{Q}_m\right] - \mathbb{E}\left[\boldsymbol{Q}_-^T \boldsymbol{M} \boldsymbol{Q}_m\right]\right\|$$
$$+ \left\| \mathbb{E}\left[\boldsymbol{Q}_-^T \boldsymbol{M} \boldsymbol{Q}_m\right] - \mathbb{E}\left[\boldsymbol{Q}_-^T \boldsymbol{M} \boldsymbol{Q}_-\right]\right\|.$$

The objective is to show that both terms vanish. By exchangeability arguments, we have

$$\left\| \mathbb{E}\left[\boldsymbol{Q}_m^T \boldsymbol{M} \boldsymbol{Q}_m\right] - \mathbb{E}\left[\boldsymbol{Q}_-^T \boldsymbol{M} \boldsymbol{Q}_m\right] \right\|$$

$$= \left\| \frac{1}{N} \mathbb{E}\left[ \sum_{i=1}^{N} [\boldsymbol{Q}_m - \boldsymbol{Q}_{-i}]^T \boldsymbol{M} \boldsymbol{Q}_m \right] \right\|$$

$$= \left\| \frac{1}{N} \mathbb{E}\left[ \sum_{i=1}^{N} \frac{1}{m} \boldsymbol{Q}_m^T \hat{\boldsymbol{U}}_n^T \boldsymbol{\sigma}_i \boldsymbol{\sigma}_i^T (\hat{\boldsymbol{U}}_n - \gamma \hat{\boldsymbol{V}}_n) \boldsymbol{Q}_{-i}^T \boldsymbol{M} \boldsymbol{Q}_m \right] \right\| \quad \text{(Lemma L.2)}$$

$$= \left\| \frac{1}{N} \mathbb{E}\left[ \sum_{i=1}^{N} \frac{1}{m} \boldsymbol{Q}_m^T \hat{\boldsymbol{U}}_n^T \boldsymbol{\sigma}_i \boldsymbol{\sigma}_i^T (\hat{\boldsymbol{U}}_n - \gamma \hat{\boldsymbol{V}}_n) \boldsymbol{Q}_m^T \boldsymbol{M} \boldsymbol{Q}_m \left( 1 + \frac{1}{m} \boldsymbol{\sigma}_i^T \hat{\boldsymbol{U}}_n \boldsymbol{Q}_{-i} (\hat{\boldsymbol{U}}_n - \gamma \hat{\boldsymbol{V}}_n)^T \boldsymbol{\sigma}_i \right) \right] \right\|$$

$$= \left\| \frac{1}{N} \mathbb{E}\left[ \frac{1}{m} \boldsymbol{Q}_m^T \hat{\boldsymbol{U}}_n^T \boldsymbol{\Sigma}_{\hat{\mathcal{S}}}^T \boldsymbol{D} \boldsymbol{\Sigma}_{\hat{\mathcal{S}}} (\hat{\boldsymbol{U}}_n - \gamma \hat{\boldsymbol{V}}_n) \boldsymbol{Q}_m^T \boldsymbol{M} \boldsymbol{Q}_m \right] \right\|,$$

where $\boldsymbol{D} \in \mathbb{R}^{N \times N}$ is a diagonal matrix for which, for all $i \in [N]$, we have

$$\boldsymbol{D}_i = 1 + \frac{1}{m} \boldsymbol{\sigma}_i^T \hat{\boldsymbol{U}}_n \boldsymbol{Q}_{-i} (\hat{\boldsymbol{U}}_n - \gamma \hat{\boldsymbol{V}}_n)^T \boldsymbol{\sigma}_i.$$

From Lemma E.3, we know

$$\mathbb{E}\left[\|\boldsymbol{D}\|\right] = \mathcal{O}(1).$$

Furthermore, from Lemma I.4, we know there exists a real $K_{\boldsymbol{Q}}' > 0$ such that, for all $m$, we have

$$\left\| \frac{1}{\sqrt{m}} \boldsymbol{\Sigma}_{\hat{\mathcal{S}}} \hat{\boldsymbol{U}}_n \boldsymbol{Q}_m \right\| \leq K_{\boldsymbol{Q}}'$$

and

$$\left\| \frac{1}{\sqrt{m}} \boldsymbol{Q}_m (\hat{\boldsymbol{U}}_n - \gamma \hat{\boldsymbol{V}}_n)^T \boldsymbol{\Sigma}_{\hat{\mathcal{S}}}^T \right\| \leq 2 K_{\boldsymbol{Q}}'.$$

We deduce thus

$$\left\| \mathbb{E}\left[\boldsymbol{Q}_m^T \boldsymbol{M} \boldsymbol{Q}_m\right] - \mathbb{E}\left[\boldsymbol{Q}_-^T \boldsymbol{M} \boldsymbol{Q}_m\right] \right\| = \left\| \frac{1}{m} \frac{m}{N} \mathbb{E}\left[ \frac{1}{m} \boldsymbol{Q}_m^T \hat{\boldsymbol{U}}_n^T \boldsymbol{\Sigma}_{\hat{\mathcal{S}}}^T \boldsymbol{D} \boldsymbol{\Sigma}_{\hat{\mathcal{S}}} (\hat{\boldsymbol{U}}_n - \gamma \hat{\boldsymbol{V}}_n) \boldsymbol{Q}_m^T \boldsymbol{M} \boldsymbol{Q}_m \right] \right\|$$

$$= \mathcal{O}\left( \frac{1}{m} \right).$$

With a similar reasoning, we can show that

$$\left\| \mathbb{E}\left[\boldsymbol{Q}_-^T \boldsymbol{M} \boldsymbol{Q}_m\right] - \mathbb{E}\left[\boldsymbol{Q}_-^T \boldsymbol{M} \boldsymbol{Q}_-\right] \right\| = \mathcal{O}\left( \frac{1}{m} \right),$$

and we conclude thus

$$\left\| \mathbb{E}\left[\boldsymbol{Q}_m^T \boldsymbol{M} \boldsymbol{Q}_m\right] - \mathbb{E}\left[\boldsymbol{Q}_-^T \boldsymbol{M} \boldsymbol{Q}_-\right] \right\| = \mathcal{O}\left( \frac{1}{m} \right).$$

$\square$

**Lemma F.4.** *Under Assumptions 1 and 2, let $\boldsymbol{Q}_m \in \mathbb{R}^{n \times n}$ be the resolvent defined in equation 13, let $\boldsymbol{Q}_- \in \mathbb{R}^{n \times n}$ be the resolvent defined in equation 29, and let $\boldsymbol{\Psi}_1 \in \mathbb{R}^{n \times n}$ be the matrix defined in equation 19. Then,*

$$\left\| \mathbb{E}\left[\boldsymbol{Q}_m^T \boldsymbol{\Psi}_1 \boldsymbol{Q}_m\right] - \mathbb{E}\left[\boldsymbol{Q}_-^T \boldsymbol{\Psi}_1 \boldsymbol{Q}_-\right] \right\| = \mathcal{O}\left( \frac{1}{\sqrt{m}} \right).$$

*Proof.* We observe that

$$\left\| \mathbb{E}\left[\boldsymbol{Q}_m^T \boldsymbol{\Psi}_1 \boldsymbol{Q}_m\right] - \mathbb{E}\left[\boldsymbol{Q}_-^T \boldsymbol{\Psi}_1 \boldsymbol{Q}_-\right] \right\| \leq \left\| \mathbb{E}\left[\boldsymbol{Q}_m^T \boldsymbol{\Psi}_1 \boldsymbol{Q}_m\right] - \mathbb{E}\left[\boldsymbol{Q}_-^T \boldsymbol{\Psi}_1 \boldsymbol{Q}_m\right] \right\|$$

$$+ \left\| \mathbb{E}\left[\boldsymbol{Q}_-^T \boldsymbol{\Psi}_1 \boldsymbol{Q}_m\right] - \mathbb{E}\left[\boldsymbol{Q}_-^T \boldsymbol{\Psi}_1 \boldsymbol{Q}_-\right] \right\|.$$

The objective is to show that both terms vanish. By exchangeability arguments, we have

$$\left\| \mathbb{E}\left[ \boldsymbol{Q}_m^T \boldsymbol{\Psi}_1 \boldsymbol{Q}_m \right] - \mathbb{E}\left[ \boldsymbol{Q}_-^T \boldsymbol{\Psi}_1 \boldsymbol{Q}_m \right] \right\|$$

$$= \left\| \frac{1}{N} \sum_{i=1}^{N} \mathbb{E}\left[ \left[ \boldsymbol{Q}_m - \boldsymbol{Q}_{-i} \right]^T \boldsymbol{\Psi}_1 \boldsymbol{Q}_m \right] \right\|$$

$$= \left\| \frac{1}{N} \sum_{i=1}^{N} \mathbb{E}\left[ \frac{1}{m} \boldsymbol{Q}_m^T \hat{\boldsymbol{U}}_n^T \boldsymbol{\sigma}_i \boldsymbol{\sigma}_i^T (\hat{\boldsymbol{U}}_n - \gamma \hat{\boldsymbol{V}}_n) \boldsymbol{Q}_{-i}^T \boldsymbol{\Psi}_1 \boldsymbol{Q}_m \right] \right\| \qquad \text{(Lemma L.2)}$$

$$= \left\| \frac{1}{N} \sum_{i=1}^{N} \mathbb{E}\left[ \frac{1}{m} \boldsymbol{Q}_m^T \hat{\boldsymbol{U}}_n^T \boldsymbol{\sigma}_i \boldsymbol{\sigma}_i^T (\hat{\boldsymbol{U}}_n - \gamma \hat{\boldsymbol{V}}_n) \boldsymbol{Q}_m^T \boldsymbol{\Psi}_1 \boldsymbol{Q}_m \left( 1 + \frac{1}{m} \boldsymbol{\sigma}_i^T \hat{\boldsymbol{U}}_n \boldsymbol{Q}_{-i} (\hat{\boldsymbol{U}}_n - \gamma \hat{\boldsymbol{V}}_n)^T \boldsymbol{\sigma}_i \right) \right] \right\|$$

$$= \left\| \underbrace{\frac{1}{N} \mathbb{E}\left[ \frac{1}{m} \boldsymbol{Q}_m^T \hat{\boldsymbol{U}}_n^T \boldsymbol{\Sigma}_{\hat{\mathcal{S}}}^T \boldsymbol{D} \boldsymbol{\Sigma}_{\hat{\mathcal{S}}} (\hat{\boldsymbol{U}}_n - \gamma \hat{\boldsymbol{V}}_n) \boldsymbol{Q}_m^T \boldsymbol{\Psi}_1 \boldsymbol{Q}_m \right]}_{=\boldsymbol{Z}} \right\|,$$

where $\boldsymbol{D} \in \mathbb{R}^{N \times N}$ is a diagonal matrix for which, for all $i \in [N]$, we have

$$\boldsymbol{D}_i = 1 + \frac{1}{m} \boldsymbol{\sigma}_i^T \hat{\boldsymbol{U}}_n \boldsymbol{Q}_{-i} (\hat{\boldsymbol{U}}_n - \gamma \hat{\boldsymbol{V}}_n)^T \boldsymbol{\sigma}_i.$$

Let the matrices

$$\boldsymbol{B} = \frac{1}{N} \frac{1}{m^{\frac{1}{4}}} \boldsymbol{Q}_m^T \hat{\boldsymbol{U}}_n^T \boldsymbol{\Sigma}_{\hat{\mathcal{S}}}^T \boldsymbol{D} \boldsymbol{\Sigma}_{\hat{\mathcal{S}}} (\hat{\boldsymbol{U}}_n - \gamma \hat{\boldsymbol{V}}_n) \boldsymbol{Q}_m^T$$

and

$$\boldsymbol{C}^T = \frac{1}{m^{\frac{3}{4}}} \boldsymbol{\Psi}_1 \boldsymbol{Q}_m.$$

We decompose $\boldsymbol{Z}$ with its symmetric and its skew-symmetric parts as

$$\boldsymbol{Z} = \mathbb{E}\left[ \boldsymbol{B} \boldsymbol{C}^T \right] = \mathbb{E}\left[ \frac{\boldsymbol{B} \boldsymbol{C}^T + \boldsymbol{C} \boldsymbol{B}^T}{2} \right] + \mathbb{E}\left[ \frac{\boldsymbol{B} \boldsymbol{C}^T - \boldsymbol{C} \boldsymbol{B}^T}{2} \right].$$

With the same reasoning on the symmetric part and the skew-symmetric part than for equation 41, we get for the operator norm

$$\left\| \boldsymbol{Z} \right\| \leq \left\| \mathbb{E}\left[ \boldsymbol{B} \boldsymbol{B}^T \right] \right\| + \left\| \mathbb{E}\left[ \boldsymbol{C} \boldsymbol{C}^T \right] \right\|.$$

We want to show that both $\left\| \mathbb{E}\left[ \boldsymbol{B} \boldsymbol{B}^T \right] \right\|$ and $\left\| \mathbb{E}\left[ \boldsymbol{C} \boldsymbol{C}^T \right] \right\|$ vanish. We have

$$\mathbb{E}\left[ \boldsymbol{B} \boldsymbol{B}^T \right] = \mathbb{E}\left[ \frac{m^2}{N^2} \frac{1}{m^2 \sqrt{m}} \boldsymbol{Q}_m^T \hat{\boldsymbol{U}}_n^T \boldsymbol{\Sigma}_{\hat{\mathcal{S}}}^T \boldsymbol{D} \boldsymbol{\Sigma}_{\hat{\mathcal{S}}} (\hat{\boldsymbol{U}}_n - \gamma \hat{\boldsymbol{V}}_n) \boldsymbol{Q}_m^T \boldsymbol{Q}_m (\hat{\boldsymbol{U}}_n - \gamma \hat{\boldsymbol{V}}_n)^T \boldsymbol{\Sigma}_{\hat{\mathcal{S}}}^T \boldsymbol{D} \boldsymbol{\Sigma}_{\hat{\mathcal{S}}} \hat{\boldsymbol{U}}_n \boldsymbol{Q}_m \right].$$

From Lemma E.3, we know

$$\mathbb{E}\left[ \left\| \boldsymbol{D} \right\| \right] = \mathcal{O}(1).$$

Furthermore, from Lemma I.4, we know there exists a real $K_{\boldsymbol{Q}}' > 0$ such that, for all $m$, we have

$$\left\| \frac{1}{\sqrt{m}} \boldsymbol{\Sigma}_{\hat{\mathcal{S}}} \hat{\boldsymbol{U}}_n \boldsymbol{Q}_m \right\| \leq K_{\boldsymbol{Q}}'$$

and

$$\left\| \frac{1}{\sqrt{m}} \boldsymbol{Q}_m (\hat{\boldsymbol{U}}_n - \gamma \hat{\boldsymbol{V}}_n)^T \boldsymbol{\Sigma}_{\hat{\mathcal{S}}}^T \right\| \leq 2 K_{\boldsymbol{Q}}'.$$

We have therefore

$$\left\| \mathbb{E}\left[ \boldsymbol{B} \boldsymbol{B}^T \right] \right\| = \mathcal{O}\left( \frac{1}{\sqrt{m}} \right).$$

For $\mathbb{E}\left[ \boldsymbol{C} \boldsymbol{C}^T \right]$, we have

$$\mathbb{E}\left[ \boldsymbol{C} \boldsymbol{C}^T \right] = \mathbb{E}\left[ \frac{1}{m \sqrt{m}} \boldsymbol{Q}_m^T \boldsymbol{\Psi}_1^2 \boldsymbol{Q}_m \right].$$

Let $\boldsymbol{\sigma}_{N+1}$ and $\boldsymbol{\sigma}_{N+2}$ be independent vectors with the same law as $\boldsymbol{\sigma}_i$, we have

$$\mathbb{E}\left[\frac{1}{m\sqrt{m}}\boldsymbol{Q}_m^T\boldsymbol{\Psi}_1\boldsymbol{\Psi}_1\boldsymbol{Q}_m\right] = \mathbb{E}\left[\frac{1}{m\sqrt{m}}\frac{N^2}{m^2}\frac{1}{(1+\delta)^2}\boldsymbol{Q}_m^T\hat{\boldsymbol{U}}_n^T\boldsymbol{\sigma}_{N+1}\boldsymbol{\sigma}_{N+1}^T\hat{\boldsymbol{U}}_n\hat{\boldsymbol{U}}_n^T\boldsymbol{\sigma}_{N+2}\boldsymbol{\sigma}_{N+2}^T\hat{\boldsymbol{U}}_n\boldsymbol{Q}_m\right].$$

Let

$$\boldsymbol{B}' = \frac{1}{m^{\frac{3}{4}}}\frac{N}{m}\frac{1}{1+\delta}\boldsymbol{Q}_m^T\hat{\boldsymbol{U}}_n^T\boldsymbol{\sigma}_{N+1}\boldsymbol{\sigma}_{N+1}^T\hat{\boldsymbol{U}}_n$$

and

$$\boldsymbol{C}'^T = \frac{1}{m^{\frac{3}{4}}}\frac{N}{m}\frac{1}{1+\delta}\hat{\boldsymbol{U}}_n^T\boldsymbol{\sigma}_{N+2}\boldsymbol{\sigma}_{N+2}^T\hat{\boldsymbol{U}}_n\boldsymbol{Q}_m.$$

We decompose $\mathbb{E}\left[\boldsymbol{C}\boldsymbol{C}^T\right]$ with its symmetric and its skew-symmetric parts as

$$\mathbb{E}\left[\boldsymbol{C}\boldsymbol{C}^T\right] = \mathbb{E}\left[\boldsymbol{B}'\boldsymbol{C}'^T\right] = \mathbb{E}\left[\frac{\boldsymbol{B}'\boldsymbol{C}'^T + \boldsymbol{C}'\boldsymbol{B}'^T}{2}\right] + \mathbb{E}\left[\frac{\boldsymbol{B}'\boldsymbol{C}'^T - \boldsymbol{C}'\boldsymbol{B}'^T}{2}\right],$$

and we get for the operator norm

$$\left\|\mathbb{E}\left[\boldsymbol{C}\boldsymbol{C}^T\right]\right\| \leq \left\|\mathbb{E}\left[\boldsymbol{B}'\boldsymbol{B}'^T\right]\right\| + \left\|\mathbb{E}\left[\boldsymbol{C}'\boldsymbol{C}'^T\right]\right\|.$$

To prove $\left\|\mathbb{E}\left[\boldsymbol{C}\boldsymbol{C}^T\right]\right\|$ vanish, we prove both $\left\|\mathbb{E}\left[\boldsymbol{B}'\boldsymbol{B}'^T\right]\right\|$ and $\left\|\mathbb{E}\left[\boldsymbol{C}'\boldsymbol{C}'^T\right]\right\|$ vanish. Let $K = \frac{1}{(1+\delta)^2}\frac{N}{N+1}\frac{N}{m}$, we write $\mathbb{E}\left[\boldsymbol{B}'\boldsymbol{B}'^T\right]$ as

$$\begin{aligned}
\mathbb{E}\left[\boldsymbol{B}'\boldsymbol{B}'^T\right] &= \mathbb{E}\left[\frac{1}{m\sqrt{m}}\frac{N^2}{m^2}\frac{1}{(1+\delta)^2}\boldsymbol{Q}_m^T\hat{\boldsymbol{U}}_n^T\boldsymbol{\sigma}_{N+1}\boldsymbol{\sigma}_{N+1}^T\hat{\boldsymbol{U}}_n\hat{\boldsymbol{U}}_n^T\boldsymbol{\sigma}_{N+1}\boldsymbol{\sigma}_{N+1}^T\hat{\boldsymbol{U}}_n\boldsymbol{Q}_m\right] \\
&= \mathbb{E}\left[\frac{1}{m\sqrt{m}}\frac{N^2}{m^2}\frac{1}{(1+\delta)^2}\boldsymbol{Q}_{-N-1}^T\hat{\boldsymbol{U}}_n^T\boldsymbol{\sigma}_{N+1}\boldsymbol{\sigma}_{N+1}^T\hat{\boldsymbol{U}}_n\hat{\boldsymbol{U}}_n^T\boldsymbol{\sigma}_{N+1}\boldsymbol{\sigma}_{N+1}^T\hat{\boldsymbol{U}}_n\boldsymbol{Q}_{-N-1}\right] \\
&= \mathbb{E}\left[K\frac{1}{m\sqrt{m}}\sum_{i=1}^{N+1}\boldsymbol{Q}_{-i}^T\hat{\boldsymbol{U}}_n^T\boldsymbol{\sigma}_i\boldsymbol{\sigma}_i^T\hat{\boldsymbol{U}}_n\boldsymbol{Q}_{-i}\left(\frac{1}{m}\boldsymbol{\sigma}_i^T\hat{\boldsymbol{U}}_n\hat{\boldsymbol{U}}_n^T\boldsymbol{\sigma}_i\right)\right] \\
&= \mathbb{E}\left[K\frac{1}{m\sqrt{m}}\sum_{i=1}^{N+1}\boldsymbol{Q}_{-i}^T\hat{\boldsymbol{U}}_n^T\boldsymbol{\sigma}_i\boldsymbol{\sigma}_i^T\hat{\boldsymbol{U}}_n\boldsymbol{Q}_{-i}\frac{1}{m}\operatorname{Tr}\left(\hat{\boldsymbol{U}}_n\hat{\boldsymbol{U}}_n^T\boldsymbol{\Phi}_{\hat{\mathcal{S}}}\right)\right] \\
&\quad + \mathbb{E}\left[K\frac{1}{m\sqrt{m}}\sum_{i=1}^{N+1}\boldsymbol{Q}_{-i}^T\hat{\boldsymbol{U}}_n^T\boldsymbol{\sigma}_i\boldsymbol{\sigma}_i^T\hat{\boldsymbol{U}}_n\boldsymbol{Q}_{-i}\left(\frac{1}{m}\boldsymbol{\sigma}_i^T\hat{\boldsymbol{U}}_n\hat{\boldsymbol{U}}_n^T\boldsymbol{\sigma}_i - \frac{1}{m}\operatorname{Tr}\left(\hat{\boldsymbol{U}}_n\hat{\boldsymbol{U}}_n^T\boldsymbol{\Phi}_{\hat{\mathcal{S}}}\right)\right)\right] \\
&= \underbrace{\mathbb{E}\left[K\frac{1}{m}\operatorname{Tr}\left(\hat{\boldsymbol{U}}_n\hat{\boldsymbol{U}}_n^T\boldsymbol{\Phi}_{\hat{\mathcal{S}}}\right)\frac{1}{m\sqrt{m}}\boldsymbol{Q}_m^T\hat{\boldsymbol{U}}_n^T\boldsymbol{\Sigma}_{\hat{\mathcal{S}}}^T\boldsymbol{D}^2\boldsymbol{\Sigma}_{\hat{\mathcal{S}}}\hat{\boldsymbol{U}}_n\boldsymbol{Q}_m\right]}_{=\boldsymbol{Z}_1} \\
&\quad + \underbrace{\mathbb{E}\left[K\frac{1}{m}\operatorname{Tr}\left(\hat{\boldsymbol{U}}_n\hat{\boldsymbol{U}}_n^T\boldsymbol{\Phi}_{\hat{\mathcal{S}}}\right)\frac{1}{m\sqrt{m}}\boldsymbol{Q}_m^T\hat{\boldsymbol{U}}_n^T\boldsymbol{\Sigma}_{\hat{\mathcal{S}}}^T\boldsymbol{D}^2\boldsymbol{D}'\boldsymbol{\Sigma}_{\hat{\mathcal{S}}}\hat{\boldsymbol{U}}_n\boldsymbol{Q}_m\right]}_{=\boldsymbol{Z}_2},
\end{aligned}$$

where $\boldsymbol{D}' \in \mathbb{R}^{N \times N}$ is a diagonal matrices for which, for all $i \in [N]$, we have

$$\boldsymbol{D}_i' = \frac{1}{m}\boldsymbol{\sigma}_i^T\hat{\boldsymbol{U}}_n\hat{\boldsymbol{U}}_n^T\boldsymbol{\sigma}_i - \frac{1}{m}\operatorname{Tr}\left(\hat{\boldsymbol{U}}_n\hat{\boldsymbol{U}}_n^T\boldsymbol{\Phi}_{\hat{\mathcal{S}}}\right).$$

From Lemma E.3, from Lemma I.4, and from equation 30, we have

$$\|\boldsymbol{Z}_1\| = \mathcal{O}\left(\frac{1}{\sqrt{m}}\right).$$

From Lemma K.2, we have

$$\mathbb{E}\left[\|\boldsymbol{D}'\|\right] = \mathcal{O}\left(\frac{1}{\sqrt{m}}\right).$$

and thus

$$\|\boldsymbol{Z}_2\| = \mathcal{O}\left(\frac{1}{m}\right).$$

We conclude that

$$\left\| \mathbb{E}\left[\boldsymbol{B}'\boldsymbol{B}'^T\right]\right\| = \mathcal{O}\left(\frac{1}{\sqrt{m}}\right)$$

and

$$\left\| \mathbb{E}\left[\boldsymbol{C}'\boldsymbol{C}'^T\right]\right\| = \mathcal{O}\left(\frac{1}{\sqrt{m}}\right).$$

Therefore,

$$\left\| \mathbb{E}\left[\boldsymbol{C}\boldsymbol{C}^T\right]\right\| = \mathcal{O}\left(\frac{1}{\sqrt{m}}\right)$$

and

$$\left\| \mathbb{E}\left[\boldsymbol{Q}_m^T\hat{\boldsymbol{U}}_n^T\boldsymbol{\Phi}_{\hat{\mathcal{S}}}\hat{\boldsymbol{U}}_n\boldsymbol{Q}_m\right] - \mathbb{E}\left[\boldsymbol{Q}_-^T\hat{\boldsymbol{U}}_n^T\boldsymbol{\Phi}_{\hat{\mathcal{S}}}\hat{\boldsymbol{U}}_n\boldsymbol{Q}_m\right]\right\| = \mathcal{O}\left(\frac{1}{\sqrt{m}}\right).$$

With a similar reasoning, we can show

$$\left\| \mathbb{E}\left[\boldsymbol{Q}_-^T\boldsymbol{\Psi}_1\boldsymbol{Q}_m\right] - \mathbb{E}\left[\boldsymbol{Q}_-^T\boldsymbol{\Psi}_1\boldsymbol{Q}_-\right]\right\| = \mathcal{O}\left(\frac{1}{\sqrt{m}}\right).$$

We conclude thus

$$\left\| \mathbb{E}\left[\boldsymbol{Q}_m^T\boldsymbol{\Psi}_1\boldsymbol{Q}_m\right] - \mathbb{E}\left[\boldsymbol{Q}_-^T\boldsymbol{\Psi}_1\boldsymbol{Q}_-\right]\right\| = \mathcal{O}\left(\frac{1}{\sqrt{m}}\right).$$

$\square$

**Lemma F.5.** *Under Assumptions 1 and 2, let $\boldsymbol{Q}_m \in \mathbb{R}^{n\times n}$ be the resolvent defined in equation 13, let $\bar{\boldsymbol{Q}}_m \in \mathbb{R}^{n\times n}$ be the deterministic resolvent defined in equation 15, let $\boldsymbol{\Psi}_1, \boldsymbol{\Psi}_2 \in \mathbb{R}^{n\times n}$ be the matrices defined in equation 19. Then,*

$$\left\| \mathbb{E}\left[\boldsymbol{Q}_m^T\boldsymbol{\Psi}_1\boldsymbol{Q}_m\right] - \frac{1}{1 - \frac{1}{N}\operatorname{Tr}\left(\boldsymbol{\Psi}_2\bar{\boldsymbol{Q}}_m^T\boldsymbol{\Psi}_1\bar{\boldsymbol{Q}}_m\right)}\bar{\boldsymbol{Q}}_m^T\boldsymbol{\Psi}_1\bar{\boldsymbol{Q}}_m\right\| = \mathcal{O}\left(\frac{1}{\sqrt{m}}\right).$$

*Proof.* From Lemma F.6, we know that

$$\mathbb{E}\left[\boldsymbol{Q}_m^T\boldsymbol{\Psi}_1\boldsymbol{Q}_m\right] = \bar{\boldsymbol{Q}}_m^T\boldsymbol{\Psi}_1\bar{\boldsymbol{Q}}_m + \mathbb{E}\left[\boldsymbol{Q}_m^T\boldsymbol{\Psi}_1\bar{\boldsymbol{Q}}_m(\hat{\boldsymbol{U}}_n - \gamma\hat{\boldsymbol{V}}_n)^T\boldsymbol{\Psi}_{\hat{\mathcal{S}}}\hat{\boldsymbol{U}}_n\boldsymbol{Q}_m\right]$$
$$- \mathbb{E}\left[\boldsymbol{Q}_-^T\boldsymbol{\Psi}_1\bar{\boldsymbol{Q}}_m(\hat{\boldsymbol{U}}_n - \gamma\hat{\boldsymbol{V}}_n)^T\boldsymbol{\Psi}_{\hat{\mathcal{S}}}\hat{\boldsymbol{U}}_n\boldsymbol{Q}_-\right] + \frac{1}{N}\operatorname{Tr}\left(\boldsymbol{\Psi}_2\bar{\boldsymbol{Q}}_m^T\boldsymbol{\Psi}_1\bar{\boldsymbol{Q}}_m\right)\mathbb{E}\left[\boldsymbol{Q}_-^T\boldsymbol{\Psi}_1\boldsymbol{Q}_-\right]$$
$$+ \mathcal{O}_{\|\cdot\|}\left(\frac{1}{\sqrt{m}}\right).$$

Exploiting $\bar{\boldsymbol{Q}}_m(\hat{\boldsymbol{U}}_n - \gamma\hat{\boldsymbol{V}}_n)^T\boldsymbol{\Psi}_{\hat{\mathcal{S}}}\hat{\boldsymbol{U}}_n = \boldsymbol{I}_n - \lambda\bar{\boldsymbol{Q}}_n$ in the above equation, and from Lemma F.4, we obtain the simplification

$$\mathbb{E}\left[\boldsymbol{Q}_m^T\boldsymbol{\Psi}_1\boldsymbol{Q}_m\right] = \bar{\boldsymbol{Q}}_m^T\boldsymbol{\Psi}_1\bar{\boldsymbol{Q}}_m + \frac{1}{N}\operatorname{Tr}\left(\boldsymbol{\Psi}_2\bar{\boldsymbol{Q}}_m^T\boldsymbol{\Psi}_1\bar{\boldsymbol{Q}}_m\right)\mathbb{E}\left[\boldsymbol{Q}_m^T\boldsymbol{\Psi}_1\boldsymbol{Q}_m\right] + \mathcal{O}_{\|\cdot\|}\left(\frac{1}{\sqrt{m}}\right).$$

or equivalently

$$\mathbb{E}\left[\boldsymbol{Q}_m^T\boldsymbol{\Psi}_1\boldsymbol{Q}_m\right]\left(1 - \frac{1}{N}\operatorname{Tr}\left(\boldsymbol{\Psi}_2\bar{\boldsymbol{Q}}_m^T\boldsymbol{\Psi}_1\bar{\boldsymbol{Q}}_m\right)\mathbb{E}\left[\boldsymbol{Q}_m^T\boldsymbol{\Psi}_1\boldsymbol{Q}_m\right]\right) = \bar{\boldsymbol{Q}}_m^T\boldsymbol{\Psi}_1\bar{\boldsymbol{Q}}_m + \mathcal{O}_{\|\cdot\|}\left(\frac{1}{\sqrt{m}}\right).$$

Let $\boldsymbol{B}'_n = \bar{\boldsymbol{Z}}^T\hat{\boldsymbol{A}}_m\boldsymbol{Z}$ and $\bar{\boldsymbol{Q}}'_m = \left[\frac{N}{m}\frac{1}{1+\delta}\boldsymbol{B}'_n + \lambda\boldsymbol{I}_m\right]^{-1}$, for which $\hat{\boldsymbol{A}}_m = \hat{\boldsymbol{U}}_n(\hat{\boldsymbol{U}}_n - \gamma\hat{\boldsymbol{V}}_n)^T$ is the empirical transition model matrix (equation 14) and $\bar{\boldsymbol{Z}}\bar{\boldsymbol{Z}}^T = \boldsymbol{\Phi}_{\hat{\mathcal{S}}}$ is the Cholesky decompositon of $\boldsymbol{\Phi}_{\hat{\mathcal{S}}}$. We have from the cyclic properties of the trace

$$\frac{1}{N}\operatorname{Tr}\left(\boldsymbol{\Psi}_2\bar{\boldsymbol{Q}}_m^T\boldsymbol{\Psi}_1\bar{\boldsymbol{Q}}_m\right)$$
$$= \frac{1}{m}\frac{N}{m}\frac{1}{(1+\delta)^2}\operatorname{Tr}\left((\hat{\boldsymbol{U}}_n - \gamma\hat{\boldsymbol{V}}_n)^T\boldsymbol{\Phi}_{\hat{\mathcal{S}}}(\hat{\boldsymbol{U}}_n - \gamma\hat{\boldsymbol{V}}_n)\bar{\boldsymbol{Q}}_m^T\hat{\boldsymbol{U}}_n^T\boldsymbol{\Phi}_{\hat{\mathcal{S}}}\hat{\boldsymbol{U}}_n\bar{\boldsymbol{Q}}_m\right)$$
$$= \frac{1}{m}\frac{N}{m}\frac{1}{(1+\delta)^2}\operatorname{Tr}\left(\boldsymbol{B}'_n\boldsymbol{Q}'_m\boldsymbol{Q}'^T_m\boldsymbol{B}'^T_n\right).$$

From equation 48, we have

$$\limsup_m \frac{1}{m} \frac{N}{m} \frac{1}{(1+\delta)^2} \operatorname{Tr}\!\big(\boldsymbol{B}_n' \bar{\boldsymbol{Q}}_m' \bar{\boldsymbol{Q}}_m'^T \boldsymbol{B}_n'^T\big) < 1.$$

Therefore,

$$\mathbb{E}\big[\boldsymbol{Q}_m^T \boldsymbol{\Psi}_1 \boldsymbol{Q}_m\big] = \bar{\boldsymbol{Q}}_m^T \boldsymbol{\Psi}_1 \bar{\boldsymbol{Q}}_m + \frac{1}{1 - \frac{1}{N} \operatorname{Tr}\big(\boldsymbol{\Psi}_2 \bar{\boldsymbol{Q}}_m^T \boldsymbol{\Psi}_1 \bar{\boldsymbol{Q}}_m\big)} \bar{\boldsymbol{Q}}_m^T \boldsymbol{\Psi}_1 \bar{\boldsymbol{Q}}_m + \mathcal{O}_{\|\cdot\|}\left(\frac{1}{\sqrt{m}}\right).$$

□

**Lemma F.6.** *Under Assumptions 1 and 2, let $\boldsymbol{Q}_m \in \mathbb{R}^{n \times n}$ be the resolvent defined in equation 13, let $\boldsymbol{Q}_- \in \mathbb{R}^{n \times n}$ be the resolvent defined in equation 29, let $\bar{\boldsymbol{Q}}_m \in \mathbb{R}^{n \times n}$ be the deterministic resolvent defined in equation 15, let $\hat{\boldsymbol{U}}_n, \hat{\boldsymbol{V}}_n \in \mathbb{R}^{m \times n}$ be the shift matrices defined in equation 7, and let $\boldsymbol{M}$ be either any matrix with a bounded operator norm or $\boldsymbol{M} = \boldsymbol{\Psi}_1$. Then,*

$$\left\| \mathbb{E}\big[\boldsymbol{Q}_m^T \boldsymbol{M} \boldsymbol{Q}_m\big] - \bar{\boldsymbol{Q}}_m^T \boldsymbol{M} \bar{\boldsymbol{Q}}_m - \mathbb{E}\left[\boldsymbol{Q}_m^T \boldsymbol{M} \bar{\boldsymbol{Q}}_m (\hat{\boldsymbol{U}}_n - \gamma \hat{\boldsymbol{V}}_n)^T \boldsymbol{\Psi}_{\hat{\mathcal{S}}} \hat{\boldsymbol{U}}_n \boldsymbol{Q}_m\right] \right.$$

$$\left. + \mathbb{E}\left[\boldsymbol{Q}_-^T \boldsymbol{M} \bar{\boldsymbol{Q}}_m (\hat{\boldsymbol{U}}_n - \gamma \hat{\boldsymbol{V}}_n)^T \boldsymbol{\Psi}_{\hat{\mathcal{S}}} \hat{\boldsymbol{U}}_n \boldsymbol{Q}_-\right] - \frac{1}{N} \operatorname{Tr}\big(\boldsymbol{\Psi}_2 \bar{\boldsymbol{Q}}_m^T \boldsymbol{M} \bar{\boldsymbol{Q}}_m\big) \mathbb{E}\big[\boldsymbol{Q}_-^T \boldsymbol{\Psi}_1 \boldsymbol{Q}_-\big] \right\|$$

$$= \mathcal{O}\left(\frac{1}{\sqrt{m}}\right),$$

*for $\boldsymbol{\Psi}_1, \boldsymbol{\Psi}_2 \in \mathbb{R}^{n \times n}$ defined in equation 19.*

*Proof.* With the resolvent identity (Lemma L.2), we decompose $\mathbb{E}\big[\boldsymbol{Q}_m^T \boldsymbol{M} \boldsymbol{Q}_m\big]$ as

$$\mathbb{E}\big[\boldsymbol{Q}_m^T \boldsymbol{M} \boldsymbol{Q}_m\big] = \mathbb{E}\big[\boldsymbol{Q}_m^T \boldsymbol{M} \bar{\boldsymbol{Q}}_m\big] - \mathbb{E}\big[\boldsymbol{Q}_m^T \boldsymbol{M} [\bar{\boldsymbol{Q}}_m - \boldsymbol{Q}_m]\big] \qquad (56)$$

$$= \mathbb{E}\big[\boldsymbol{Q}_m^T \boldsymbol{M} \bar{\boldsymbol{Q}}_m\big]$$

$$- \mathbb{E}\left[\boldsymbol{Q}_m^T \boldsymbol{M} \bar{\boldsymbol{Q}}_m \left[\frac{1}{m}(\hat{\boldsymbol{U}}_n - \gamma \hat{\boldsymbol{V}}_n)^T \boldsymbol{\Sigma}_{\hat{\mathcal{S}}}^T \boldsymbol{\Sigma}_{\hat{\mathcal{S}}} \hat{\boldsymbol{U}}_n - (\hat{\boldsymbol{U}}_n - \gamma \hat{\boldsymbol{V}}_n)^T \boldsymbol{\Psi}_{\hat{\mathcal{S}}} \hat{\boldsymbol{U}}_n\right] \boldsymbol{Q}_m\right]$$

$$= \underbrace{\mathbb{E}\big[\boldsymbol{Q}_m^T \boldsymbol{M} \bar{\boldsymbol{Q}}_m\big]}_{=\boldsymbol{Z}_1} + \mathbb{E}\left[\boldsymbol{Q}_m^T \boldsymbol{M} \bar{\boldsymbol{Q}}_m (\hat{\boldsymbol{U}}_n - \gamma \hat{\boldsymbol{V}}_n)^T \boldsymbol{\Psi}_{\hat{\mathcal{S}}} \hat{\boldsymbol{U}}_n \boldsymbol{Q}_m\right]$$

$$- \underbrace{\frac{1}{m} \sum_{i=1}^N \mathbb{E}\left[\boldsymbol{Q}_m^T \boldsymbol{M} \bar{\boldsymbol{Q}}_m (\hat{\boldsymbol{U}}_n - \gamma \hat{\boldsymbol{V}}_n)^T \boldsymbol{\sigma}_i \boldsymbol{\sigma}_i^T \hat{\boldsymbol{U}}_n \boldsymbol{Q}_m\right]}_{=\boldsymbol{Z}_2},$$

$$(57)$$

where $\boldsymbol{\Sigma}_{\hat{\mathcal{S}}}^T \boldsymbol{\Sigma}_{\hat{\mathcal{S}}} = \sum_{i=1}^N \boldsymbol{\sigma}_i \boldsymbol{\sigma}_i^T$ is the same decompositon of $\boldsymbol{\Sigma}_{\hat{\mathcal{S}}}^T \boldsymbol{\Sigma}_{\hat{\mathcal{S}}}$ than the one used in equation 32. From Theorem 5.1, we have

$$\big\| \mathbb{E}[\boldsymbol{Q}_m] - \bar{\boldsymbol{Q}}_m \big\| = \mathcal{O}\left(\frac{1}{\sqrt{m}}\right).$$

Therefore, from above and from Lemma F.9 which upper bounds $\|\boldsymbol{M} \bar{\boldsymbol{Q}}_m\|$, we deduce for $\boldsymbol{Z}_1$ that

$$\|\boldsymbol{Z}_1\| - \big\| \bar{\boldsymbol{Q}}_m^T \boldsymbol{M} \bar{\boldsymbol{Q}}_m \big\| = \big\| \mathbb{E}\big[\boldsymbol{Q}_m^T \boldsymbol{M} \bar{\boldsymbol{Q}}_m\big] \big\| - \big\| \bar{\boldsymbol{Q}}_m^T \boldsymbol{M} \bar{\boldsymbol{Q}}_m \big\|$$

$$\leq \big\| \mathbb{E}[\boldsymbol{Q}_m] - \bar{\boldsymbol{Q}}_m \big\| \|\boldsymbol{M} \bar{\boldsymbol{Q}}_m\|$$

$$= \mathcal{O}\left(\frac{1}{\sqrt{m}}\right).$$

We want to find now a deterministic approximation for $\boldsymbol{Z}_2$ in equation 57. From the Sherman identity (Lemma L.4) and with the resolvent $\boldsymbol{Q}_{-i}$ defined in equation 33 as

$$\boldsymbol{Q}_{-i} = \left[\frac{1}{m}(\hat{\boldsymbol{U}}_n - \gamma \hat{\boldsymbol{V}}_n)^T \boldsymbol{\Sigma}_{\hat{\mathcal{S}}}^T \boldsymbol{\Sigma}_{\hat{\mathcal{S}}} \hat{\boldsymbol{U}}_n - \frac{1}{m}(\hat{\boldsymbol{U}}_n - \gamma \hat{\boldsymbol{V}}_n)^T \boldsymbol{\sigma}_i \boldsymbol{\sigma}_i^T \hat{\boldsymbol{U}}_n + \lambda \boldsymbol{I}_n\right]^{-1},$$

we obtain the following relation

$$\boldsymbol{Q}_m = \boldsymbol{Q}_{-i} - \frac{\frac{1}{m}\boldsymbol{Q}_{-i}(\hat{\boldsymbol{U}}_n - \gamma\hat{\boldsymbol{V}}_n)^T\boldsymbol{\sigma}_i\boldsymbol{\sigma}_i^T\hat{\boldsymbol{U}}_n\boldsymbol{Q}_{-i}}{1 + \frac{1}{m}\boldsymbol{\sigma}_i^T\hat{\boldsymbol{U}}_n\boldsymbol{Q}_{-i}(\hat{\boldsymbol{U}}_n - \gamma\hat{\boldsymbol{V}}_n)^T\boldsymbol{\sigma}_i}.$$

By remarking that for all $i \in [N]$, we have

$$\boldsymbol{Q}_m^T\boldsymbol{M}\bar{\boldsymbol{Q}}_m(\hat{\boldsymbol{U}}_n - \gamma\hat{\boldsymbol{V}}_n)^T\boldsymbol{\sigma}_i\boldsymbol{\sigma}_i^T\hat{\boldsymbol{U}}_n\boldsymbol{Q}_m$$

$$= \boldsymbol{Q}_m^T\boldsymbol{M}\bar{\boldsymbol{Q}}_m(\hat{\boldsymbol{U}}_n - \gamma\hat{\boldsymbol{V}}_n)^T\boldsymbol{\sigma}_i\boldsymbol{\sigma}_i^T\hat{\boldsymbol{U}}_n\boldsymbol{Q}_{-i}\frac{1}{1 + \frac{1}{m}\boldsymbol{\sigma}_i^T\hat{\boldsymbol{U}}_n\boldsymbol{Q}_{-i}(\hat{\boldsymbol{U}}_n - \gamma\hat{\boldsymbol{V}}_n)^T\boldsymbol{\sigma}_i}$$

$$= \frac{1}{1 + \delta}\boldsymbol{Q}_m^T\boldsymbol{M}\bar{\boldsymbol{Q}}_m(\hat{\boldsymbol{U}}_n - \gamma\hat{\boldsymbol{V}}_n)^T\boldsymbol{\sigma}_i\boldsymbol{\sigma}_i^T\hat{\boldsymbol{U}}_n\boldsymbol{Q}_{-i}$$

$$+ \frac{1}{1 + \delta}\boldsymbol{Q}_m^T\boldsymbol{M}\bar{\boldsymbol{Q}}_m(\hat{\boldsymbol{U}}_n - \gamma\hat{\boldsymbol{V}}_n)^T\boldsymbol{\sigma}_i\boldsymbol{\sigma}_i^T\hat{\boldsymbol{U}}_n\boldsymbol{Q}_{-i}\frac{\delta - \frac{1}{m}\boldsymbol{\sigma}_i^T\hat{\boldsymbol{U}}_n\boldsymbol{Q}_{-i}(\hat{\boldsymbol{U}}_n - \gamma\hat{\boldsymbol{V}}_n)^T\boldsymbol{\sigma}_i}{1 + \frac{1}{m}\boldsymbol{\sigma}_i^T\hat{\boldsymbol{U}}_n\boldsymbol{Q}_{-i}(\hat{\boldsymbol{U}}_n - \gamma\hat{\boldsymbol{V}}_n)^T\boldsymbol{\sigma}_i},$$

we decompose $\boldsymbol{Z}_2$ as

$$\boldsymbol{Z}_2 = \frac{1}{m}\mathbb{E}\left[\sum_{i=1}^N \boldsymbol{Q}_m^T\boldsymbol{M}\bar{\boldsymbol{Q}}_m(\hat{\boldsymbol{U}}_n - \gamma\hat{\boldsymbol{V}}_n)^T\boldsymbol{\sigma}_i\boldsymbol{\sigma}_i^T\hat{\boldsymbol{U}}_n\boldsymbol{Q}_m\right] \tag{58}$$

$$= \underbrace{\mathbb{E}\left[\boldsymbol{Q}_-^T\boldsymbol{M}\bar{\boldsymbol{Q}}_m(\hat{\boldsymbol{U}}_n - \gamma\hat{\boldsymbol{V}}_n)^T\boldsymbol{\Psi}_{\hat{\mathcal{S}}}\hat{\boldsymbol{U}}_n\boldsymbol{Q}_-\right]}_{=\boldsymbol{Z}_{21}} \tag{59}$$

$$-\underbrace{\frac{1}{m}\frac{1}{1 + \delta}\sum_{i=1}^N\mathbb{E}\left[\boldsymbol{Q}_{-i}^T\hat{\boldsymbol{U}}_n^T\boldsymbol{\sigma}_i\boldsymbol{\sigma}_i^T\hat{\boldsymbol{U}}_n\boldsymbol{Q}_{-i}\frac{\frac{1}{m}\boldsymbol{\sigma}_i^T(\hat{\boldsymbol{U}}_n - \gamma\hat{\boldsymbol{V}}_n)\boldsymbol{Q}_{-i}^T\boldsymbol{M}\bar{\boldsymbol{Q}}_m(\hat{\boldsymbol{U}}_n - \gamma\hat{\boldsymbol{V}}_n)^T\boldsymbol{\sigma}_i}{1 + \frac{1}{m}\boldsymbol{\sigma}_i^T\hat{\boldsymbol{U}}_n\boldsymbol{Q}_{-i}(\hat{\boldsymbol{U}}_n - \gamma\hat{\boldsymbol{V}}_n)^T\boldsymbol{\sigma}_i}\right]}_{=\boldsymbol{Z}_{22}}$$

$$\tag{60}$$

$$+\underbrace{\frac{1}{m}\frac{1}{1 + \delta}\sum_{i=1}^N\mathbb{E}\left[\boldsymbol{Q}_{-i}^T\boldsymbol{M}\bar{\boldsymbol{Q}}_m(\hat{\boldsymbol{U}}_n - \gamma\hat{\boldsymbol{V}}_n)^T\boldsymbol{\sigma}_i\boldsymbol{\sigma}_i^T\hat{\boldsymbol{U}}_n\boldsymbol{Q}_{-i}\frac{\delta - \frac{1}{m}\boldsymbol{\sigma}_i^T\hat{\boldsymbol{U}}_n\boldsymbol{Q}_{-i}(\hat{\boldsymbol{U}}_n - \gamma\hat{\boldsymbol{V}}_n)^T\boldsymbol{\sigma}_i}{1 + \frac{1}{m}\boldsymbol{\sigma}_i^T\hat{\boldsymbol{U}}_n\boldsymbol{Q}_{-i}(\hat{\boldsymbol{U}}_n - \gamma\hat{\boldsymbol{V}}_n)^T\boldsymbol{\sigma}_i}\right]}_{=\boldsymbol{Z}_{23}}$$

$$\tag{61}$$

$$-\underbrace{\frac{1}{m}\frac{1}{1 + \delta}\sum_{i=1}^N\mathbb{E}\Bigg[\boldsymbol{Q}_{-i}^T\hat{\boldsymbol{U}}_n^T\boldsymbol{\sigma}_i\boldsymbol{\sigma}_i^T\hat{\boldsymbol{U}}_n\boldsymbol{Q}_{-i}}$$

$$\underbrace{\frac{\left(\frac{1}{m}\boldsymbol{\sigma}_i^T(\hat{\boldsymbol{U}}_n - \gamma\hat{\boldsymbol{V}}_n)\boldsymbol{Q}_{-i}^T\boldsymbol{M}\bar{\boldsymbol{Q}}_m(\hat{\boldsymbol{U}}_n - \gamma\hat{\boldsymbol{V}}_n)^T\boldsymbol{\sigma}_i\right)\left(\delta - \frac{1}{m}\boldsymbol{\sigma}_i^T\hat{\boldsymbol{U}}_n\boldsymbol{Q}_{-i}(\hat{\boldsymbol{U}}_n - \gamma\hat{\boldsymbol{V}}_n)^T\boldsymbol{\sigma}_i\right)}{\left(1 + \frac{1}{m}\boldsymbol{\sigma}_i^T\hat{\boldsymbol{U}}_n\boldsymbol{Q}_{-i}(\hat{\boldsymbol{U}}_n - \gamma\hat{\boldsymbol{V}}_n)^T\boldsymbol{\sigma}_i\right)^2}\Bigg]}_{=\boldsymbol{Z}_{24}}$$

$$\tag{62}$$

$$= \boldsymbol{Z}_{21} - \boldsymbol{Z}_{22} + \boldsymbol{Z}_{23} - \boldsymbol{Z}_{24}. \tag{63}$$

From Lemma F.7, we have

$$\left\|\boldsymbol{Z}_{22} - \frac{1}{N}\text{Tr}(\boldsymbol{\Psi}_2\bar{\boldsymbol{Q}}_m^T\boldsymbol{M}\bar{\boldsymbol{Q}}_m)\mathbb{E}\left[\boldsymbol{Q}_-^T\boldsymbol{\Psi}_1\boldsymbol{Q}_-\right]\right\| = \mathcal{O}\left(\frac{1}{\sqrt{m}}\right).$$

With a similar proof than for $\boldsymbol{Z}_{22}$, we can show for $\boldsymbol{Z}_{24}$ that

$$\|\boldsymbol{Z}_{24}\| = \mathcal{O}\left(\frac{1}{\sqrt{m}}\right).$$

From Lemma F.8, we have

$$\|\boldsymbol{Z}_{23}\| = \mathcal{O}\left(\frac{1}{\sqrt{m}}\right).$$

As a consequence, we conclude that

$$\mathbb{E}\big[\boldsymbol{Q}_m^T \boldsymbol{M} \boldsymbol{Q}_m\big] = \bar{\boldsymbol{Q}}_m^T \boldsymbol{M} \bar{\boldsymbol{Q}}_m + \mathbb{E}\left[\boldsymbol{Q}_m^T \boldsymbol{M} \bar{\boldsymbol{Q}}_m (\hat{\boldsymbol{U}}_n - \gamma \hat{\boldsymbol{V}}_n)^T \boldsymbol{\Psi}_{\hat{\mathcal{S}}} \hat{\boldsymbol{U}}_n \boldsymbol{Q}_m\right]$$

$$- \mathbb{E}\left[\boldsymbol{Q}_-^T \boldsymbol{M} \bar{\boldsymbol{Q}}_m (\hat{\boldsymbol{U}}_n - \gamma \hat{\boldsymbol{V}}_n)^T \boldsymbol{\Psi}_{\hat{\mathcal{S}}} \hat{\boldsymbol{U}}_n \boldsymbol{Q}_-\right] + \frac{1}{N} \operatorname{Tr}\big(\boldsymbol{\Psi}_2 \bar{\boldsymbol{Q}}_m^T \boldsymbol{M} \bar{\boldsymbol{Q}}_m\big) \mathbb{E}\big[\boldsymbol{Q}_-^T \boldsymbol{\Psi}_1 \boldsymbol{Q}_-\big]$$

$$+ \mathcal{O}_{\|\cdot\|}\left(\frac{1}{\sqrt{m}}\right).$$

$\square$

**Lemma F.7.** *Under Assumptions 1 and 2, let $\boldsymbol{Z}_{22} \in \mathbb{R}^{n \times n}$ be the matrix defined in equation 60 as*

$$\boldsymbol{Z}_{22} = \frac{1}{m} \frac{1}{1+\delta} \sum_{i=1}^N \mathbb{E}\left[\boldsymbol{Q}_{-i}^T \hat{\boldsymbol{U}}_n^T \boldsymbol{\sigma}_i \boldsymbol{\sigma}_i^T \hat{\boldsymbol{U}}_n \boldsymbol{Q}_{-i} \frac{\frac{1}{m}\boldsymbol{\sigma}_i^T (\hat{\boldsymbol{U}}_n - \gamma \hat{\boldsymbol{V}}_n) \boldsymbol{Q}_{-i}^T \boldsymbol{M} \bar{\boldsymbol{Q}}_m (\hat{\boldsymbol{U}}_n - \gamma \hat{\boldsymbol{V}}_n)^T \boldsymbol{\sigma}_i}{1 + \frac{1}{m}\boldsymbol{\sigma}_i^T \hat{\boldsymbol{U}}_n \boldsymbol{Q}_{-i} (\hat{\boldsymbol{U}}_n - \gamma \hat{\boldsymbol{V}}_n)^T \boldsymbol{\sigma}_i}\right].$$

*Then,*

$$\left\|\boldsymbol{Z}_{22} - \frac{1}{N} \operatorname{Tr}\big(\boldsymbol{\Psi}_2 \bar{\boldsymbol{Q}}_m^T \boldsymbol{M} \bar{\boldsymbol{Q}}_m\big) \mathbb{E}\big[\boldsymbol{Q}_-^T \boldsymbol{\Psi}_1 \boldsymbol{Q}_-\big]\right\| = \mathcal{O}\left(\frac{1}{\sqrt{m}}\right),$$

*where $\bar{\boldsymbol{Q}}_m \in \mathbb{R}^{n \times n}$ is the deterministic resolvent defined in equation 15, $\boldsymbol{Q}_- \in \mathbb{R}^{n \times n}$ is the resolvent defined in equation 29, and $\boldsymbol{\Psi}_1, \boldsymbol{\Psi}_2 \in \mathbb{R}^{n \times n}$ defined in equation 19.*

*Proof.* Let $\boldsymbol{D} \in \mathbb{R}^{N \times N}$ be a diagonal matrix for which, for all $i \in [N]$, we have

$$\boldsymbol{D}_i = 1 + \frac{1}{m} \boldsymbol{\sigma}_i^T \hat{\boldsymbol{U}}_n \boldsymbol{Q}_{-i} (\hat{\boldsymbol{U}}_n - \gamma \hat{\boldsymbol{V}}_n)^T \boldsymbol{\sigma}_i,$$

and $\boldsymbol{D}_2 \in \mathbb{R}^{N \times N}$ be another diagonal matrix for which, for all $i \in [N]$, we have

$$[\boldsymbol{D}_2]_i = \frac{\frac{1}{m}\boldsymbol{\sigma}_i^T (\hat{\boldsymbol{U}}_n - \gamma \hat{\boldsymbol{V}}_n) \boldsymbol{Q}_{-i}^T \boldsymbol{M} \bar{\boldsymbol{Q}}_m (\hat{\boldsymbol{U}}_n - \gamma \hat{\boldsymbol{V}}_n)^T \boldsymbol{\sigma}_i}{1 + \frac{1}{m}\boldsymbol{\sigma}_i^T \hat{\boldsymbol{U}}_n \boldsymbol{Q}_{-i} (\hat{\boldsymbol{U}}_n - \gamma \hat{\boldsymbol{V}}_n)^T \boldsymbol{\sigma}_i} - \frac{1}{N} \operatorname{Tr}\big(\boldsymbol{\Psi}_2 \bar{\boldsymbol{Q}}_m^T \boldsymbol{M} \bar{\boldsymbol{Q}}_m\big)$$

$$= \frac{\frac{1}{m}\boldsymbol{\sigma}_i^T (\hat{\boldsymbol{U}}_n - \gamma \hat{\boldsymbol{V}}_n) \boldsymbol{Q}_{-i}^T \boldsymbol{M} \bar{\boldsymbol{Q}}_m (\hat{\boldsymbol{U}}_n - \gamma \hat{\boldsymbol{V}}_n)^T \boldsymbol{\sigma}_i}{1 + \frac{1}{m}\boldsymbol{\sigma}_i^T \hat{\boldsymbol{U}}_n \boldsymbol{Q}_{-i} (\hat{\boldsymbol{U}}_n - \gamma \hat{\boldsymbol{V}}_n)^T \boldsymbol{\sigma}_i}$$

$$- \frac{\frac{1}{m} \operatorname{Tr}\left((\hat{\boldsymbol{U}}_n - \gamma \hat{\boldsymbol{V}}_n)^T \boldsymbol{\Phi}_{\hat{\mathcal{S}}} (\hat{\boldsymbol{U}}_n - \gamma \hat{\boldsymbol{V}}_n) \bar{\boldsymbol{Q}}_m^T \boldsymbol{M} \bar{\boldsymbol{Q}}_m\right)}{1+\delta}$$

$$= \frac{\frac{1}{m}\boldsymbol{\sigma}_i^T (\hat{\boldsymbol{U}}_n - \gamma \hat{\boldsymbol{V}}_n) \boldsymbol{Q}_{-i}^T \boldsymbol{M} \bar{\boldsymbol{Q}}_m (\hat{\boldsymbol{U}}_n - \gamma \hat{\boldsymbol{V}}_n)^T \boldsymbol{\sigma}_i}{1 + \frac{1}{m}\boldsymbol{\sigma}_i^T \hat{\boldsymbol{U}}_n \boldsymbol{Q}_{-i} (\hat{\boldsymbol{U}}_n - \gamma \hat{\boldsymbol{V}}_n)^T \boldsymbol{\sigma}_i}$$

$$- \frac{\frac{1}{m} \operatorname{Tr}\left((\hat{\boldsymbol{U}}_n - \gamma \hat{\boldsymbol{V}}_n) \bar{\boldsymbol{Q}}_m^T \boldsymbol{M} \bar{\boldsymbol{Q}}_m (\hat{\boldsymbol{U}}_n - \gamma \hat{\boldsymbol{V}}_n)^T \boldsymbol{\Phi}_{\hat{\mathcal{S}}}\right)}{1+\delta}.$$

We have

$$\left\|\boldsymbol{Z}_{22} - \frac{1}{N}\mathbb{E}\big[\boldsymbol{Q}_-^T \boldsymbol{\Psi}_1 \boldsymbol{Q}_-\big] \operatorname{Tr}\big(\boldsymbol{\Psi}_2 \bar{\boldsymbol{Q}}_m^T \boldsymbol{M} \bar{\boldsymbol{Q}}_m\big)\right\|$$

$$= \left\|\boldsymbol{Z}_{22} - \frac{1}{m} \frac{1}{1+\delta}\mathbb{E}\left[\sum_{i=1}^N \boldsymbol{Q}_{-i}^T \hat{\boldsymbol{U}}_n^T \boldsymbol{\sigma}_i \boldsymbol{\sigma}_i^T \hat{\boldsymbol{U}}_n \boldsymbol{Q}_{-i}\right] \frac{1}{N} \operatorname{Tr}\big(\boldsymbol{\Psi}_2 \bar{\boldsymbol{Q}}_m^T \boldsymbol{M} \bar{\boldsymbol{Q}}_m\big)\right\|$$

$$= \left\|\frac{1}{m} \frac{1}{1+\delta}\mathbb{E}\left[\sum_{i=1}^N \boldsymbol{Q}_{-i}^T \hat{\boldsymbol{U}}_n^T \boldsymbol{\sigma}_i \boldsymbol{\sigma}_i^T \hat{\boldsymbol{U}}_n \boldsymbol{Q}_{-i} \left(\frac{\frac{1}{m}\boldsymbol{\sigma}_i^T (\hat{\boldsymbol{U}}_n - \gamma \hat{\boldsymbol{V}}_n) \boldsymbol{Q}_{-i}^T \boldsymbol{M} \bar{\boldsymbol{Q}}_m (\hat{\boldsymbol{U}}_n - \gamma \hat{\boldsymbol{V}}_n)^T \boldsymbol{\sigma}_i}{1 + \frac{1}{m}\boldsymbol{\sigma}_i^T \hat{\boldsymbol{U}}_n \boldsymbol{Q}_{-i} (\hat{\boldsymbol{U}}_n - \gamma \hat{\boldsymbol{V}}_n)^T \boldsymbol{\sigma}_i}\right.\right.\right.$$

$$\left.\left.\left. - \frac{1}{N} \operatorname{Tr}\big(\boldsymbol{\Psi}_2 \bar{\boldsymbol{Q}}_m^T \boldsymbol{M} \bar{\boldsymbol{Q}}_m\big)\right)\right]\right\|$$

$$= \left\|\underbrace{\frac{1}{m} \frac{1}{1+\delta}\mathbb{E}\big[\boldsymbol{Q}_m^T \hat{\boldsymbol{U}}_n^T \boldsymbol{\Sigma}_{\hat{\mathcal{S}}}^T \boldsymbol{D}^2 \boldsymbol{D}_2 \boldsymbol{\Sigma}_{\hat{\mathcal{S}}} \hat{\boldsymbol{U}}_n \boldsymbol{Q}_m\big]}_{=\boldsymbol{Z}_{221}}\right\|.$$

Let the matrices

$$\boldsymbol{B} = m^{-\frac{1}{4}} \frac{1}{\sqrt{m}} \boldsymbol{Q}_m^T \hat{\boldsymbol{U}}_n^T \boldsymbol{\Sigma}_{\hat{\mathcal{S}}}^T \boldsymbol{D}^2,$$

and

$$\boldsymbol{C}^T = m^{\frac{1}{4}} \frac{1}{\sqrt{m}} \boldsymbol{D}_2 \boldsymbol{\Sigma}_{\hat{\mathcal{S}}} \hat{\boldsymbol{U}}_n \boldsymbol{Q}_m.$$

Using the matrices above, we have

$$\boldsymbol{Z}_{221} = \frac{1}{m} \frac{1}{1+\delta} \mathbb{E}\big[\boldsymbol{Q}_m^T \hat{\boldsymbol{U}}_n^T \boldsymbol{\Sigma}_{\hat{\mathcal{S}}}^T \boldsymbol{D}^2 \boldsymbol{D}_2 \boldsymbol{\Sigma}_{\hat{\mathcal{S}}} \hat{\boldsymbol{U}}_n \boldsymbol{Q}_m\big] = \frac{1}{1+\delta} \mathbb{E}\big[\boldsymbol{BC}^T\big] = \frac{1}{1+\delta} \mathbb{E}\left[\frac{\boldsymbol{BC}^T + \boldsymbol{CB}^T}{2}\right],$$

since $\boldsymbol{Z}_{221}$ is symmetric. We use the relations $(\boldsymbol{B} - \boldsymbol{C})(\boldsymbol{B} - \boldsymbol{C})^T \succeq 0$ and $(\boldsymbol{B} + \boldsymbol{C})(\boldsymbol{B} + \boldsymbol{C})^T \succeq 0$ to deduce the following relation

$$-\boldsymbol{BB}^T - \boldsymbol{CC}^T \preceq \boldsymbol{BC}^T + \boldsymbol{CB}^T \preceq \boldsymbol{BB}^T + \boldsymbol{CC}^T.$$

From this relation, we obtain

$$\|\boldsymbol{Z}_{221}\| \leq \frac{1}{2(1+\delta)}\Big(\mathbb{E}\big[\|\boldsymbol{BB}^T\|\big] + \mathbb{E}\big[\|\boldsymbol{CC}^T\|\big]\Big),$$

where

$$\boldsymbol{BB}^T = \frac{1}{m\sqrt{m}} \boldsymbol{Q}_m^T \hat{\boldsymbol{U}}_n^T \boldsymbol{\Sigma}_{\hat{\mathcal{S}}}^T \boldsymbol{D}^4 \boldsymbol{\Sigma}_{\hat{\mathcal{S}}} \hat{\boldsymbol{U}}_n \boldsymbol{Q}_m$$

and

$$\boldsymbol{CC}^T = \frac{1}{\sqrt{m}} \boldsymbol{Q}_m^T \hat{\boldsymbol{U}}_n^T \boldsymbol{\Sigma}_{\hat{\mathcal{S}}}^T \boldsymbol{D}_2^2 \boldsymbol{\Sigma}_{\hat{\mathcal{S}}} \hat{\boldsymbol{U}}_n \boldsymbol{Q}_m.$$

To get the Lemma, we prove that both $\mathbb{E}\big[\|\boldsymbol{BB}^T\|\big]$ and $\mathbb{E}\big[\|\boldsymbol{CC}^T\|\big]$ vanish. From Lemma I.4, we know there exists a real $K'_{\boldsymbol{Q}} > 0$ such that, for all $m$, we have

$$\left\|\frac{1}{\sqrt{m}} \boldsymbol{\Sigma}_{\hat{\mathcal{S}}} \hat{\boldsymbol{U}}_n \boldsymbol{Q}_m\right\| \leq K'_{\boldsymbol{Q}}$$

and

$$\left\|\frac{1}{\sqrt{m}} \boldsymbol{Q}_m (\hat{\boldsymbol{U}}_n - \gamma \hat{\boldsymbol{V}}_n)^T \boldsymbol{\Sigma}_{\hat{\mathcal{S}}}^T\right\| \leq 2K'_{\boldsymbol{Q}}.$$

Furthermore, from Lemma E.3, we know

$$\mathbb{E}\left[\|\boldsymbol{D}^4\|\right] = \mathcal{O}(1).$$

We conclude that

$$\mathbb{E}\big[\|\boldsymbol{BB}^T\|\big] = \mathcal{O}\left(\frac{1}{\sqrt{m}}\right).$$

For $\mathbb{E}\big[\|\boldsymbol{CC}^T\|\big]$, we remark that

$$\Pr\big([\boldsymbol{D}_2]_i \geq t\big) \leq \Pr\bigg(\frac{\frac{1}{m}\boldsymbol{\sigma}_i^T(\hat{\boldsymbol{U}}_n - \gamma\hat{\boldsymbol{V}}_n)\boldsymbol{Q}_{-i}^T \boldsymbol{M}\bar{\boldsymbol{Q}}_m(\hat{\boldsymbol{U}}_n - \gamma\hat{\boldsymbol{V}}_n)^T\boldsymbol{\sigma}_i}{1 + \frac{1}{m}\boldsymbol{\sigma}_i^T\hat{\boldsymbol{U}}_n\boldsymbol{Q}_{-i}(\hat{\boldsymbol{U}}_n - \gamma\hat{\boldsymbol{V}}_n)^T\boldsymbol{\sigma}_i}$$
$$- \frac{\frac{1}{m}\operatorname{Tr}\big(\boldsymbol{\Phi}_{\hat{\mathcal{S}}}(\hat{\boldsymbol{U}}_n - \gamma\hat{\boldsymbol{V}}_n)\bar{\boldsymbol{Q}}_m^T\boldsymbol{M}\bar{\boldsymbol{Q}}_m(\hat{\boldsymbol{U}}_n - \gamma\hat{\boldsymbol{V}}_n)^T\big)}{1 + \frac{1}{m}\boldsymbol{\sigma}_i^T\hat{\boldsymbol{U}}_n\boldsymbol{Q}_{-i}(\hat{\boldsymbol{U}}_n - \gamma\hat{\boldsymbol{V}}_n)^T\boldsymbol{\sigma}_i} \geq \frac{t}{2}\bigg)$$
$$+ \Pr\bigg(\frac{\frac{1}{m}\operatorname{Tr}\big(\boldsymbol{\Phi}_{\hat{\mathcal{S}}}(\hat{\boldsymbol{U}}_n - \gamma\hat{\boldsymbol{V}}_n)\bar{\boldsymbol{Q}}_m^T\boldsymbol{M}\bar{\boldsymbol{Q}}_m(\hat{\boldsymbol{U}}_n - \gamma\hat{\boldsymbol{V}}_n)^T\big)}{1 + \frac{1}{m}\boldsymbol{\sigma}_i^T\hat{\boldsymbol{U}}_n\boldsymbol{Q}_{-i}(\hat{\boldsymbol{U}}_n - \gamma\hat{\boldsymbol{V}}_n)^T\boldsymbol{\sigma}_i}$$
$$- \frac{\frac{1}{m}\operatorname{Tr}\big(\boldsymbol{\Phi}_{\hat{\mathcal{S}}}(\hat{\boldsymbol{U}}_n - \gamma\hat{\boldsymbol{V}}_n)\bar{\boldsymbol{Q}}_m^T\boldsymbol{M}\bar{\boldsymbol{Q}}_m(\hat{\boldsymbol{U}}_n - \gamma\hat{\boldsymbol{V}}_n)^T\big)}{1+\delta} \geq \frac{t}{2}\bigg).$$

Since $\|\boldsymbol{M}\bar{\boldsymbol{Q}}_m\|$ is bounded from Lemma F.9, with a similar proof than for Lemma K.5, we can prove that

$$\Pr\Bigg(\ \bigg|\frac{1}{m}\boldsymbol{\sigma}^T(\hat{\boldsymbol{U}}_n - \gamma\hat{\boldsymbol{V}}_n)\boldsymbol{Q}_{-i}^T\boldsymbol{M}\bar{\boldsymbol{Q}}_m(\hat{\boldsymbol{U}}_n - \gamma\hat{\boldsymbol{V}}_n)^T\boldsymbol{\sigma}$$

$$-\frac{1}{m}\operatorname{Tr}\big((\hat{\boldsymbol{U}}_n - \gamma\hat{\boldsymbol{V}}_n)\mathbb{E}[\boldsymbol{Q}_{-i}^T]\boldsymbol{M}\bar{\boldsymbol{Q}}_m(\hat{\boldsymbol{U}}_n - \gamma\hat{\boldsymbol{V}}_n)^T\boldsymbol{\Phi}_{\hat{\mathcal{S}}}\big)\bigg| > t\Bigg) \leq Ce^{-cm\max(t,t^2)},$$

for some $C, c$ independent of $N, m$. Besides, from the proof of Theorem 5.1, we also have

$$\bigg|\frac{1}{m}\operatorname{Tr}\big((\hat{\boldsymbol{U}}_n - \gamma\hat{\boldsymbol{V}}_n)\mathbb{E}[\boldsymbol{Q}_{-i}^T]\boldsymbol{M}\bar{\boldsymbol{Q}}_m(\hat{\boldsymbol{U}}_n - \gamma\hat{\boldsymbol{V}}_n)^T\boldsymbol{\Phi}_{\hat{\mathcal{S}}})$$

$$-\frac{1}{m}\operatorname{Tr}\big((\hat{\boldsymbol{U}}_n - \gamma\hat{\boldsymbol{V}}_n)\bar{\boldsymbol{Q}}_m^T\boldsymbol{M}\bar{\boldsymbol{Q}}_m(\hat{\boldsymbol{U}}_n - \gamma\hat{\boldsymbol{V}}_n)^T\boldsymbol{\Phi}_{\hat{\mathcal{S}}})\bigg| = \mathcal{O}\left(\frac{1}{\sqrt{m}}\right),$$

as both $\|\hat{\boldsymbol{U}}_n\|$ and $\|\hat{\boldsymbol{V}}_n\|$ are upper bounded by 1, $|\operatorname{Tr}(\boldsymbol{A}\boldsymbol{B})| \leq \|\boldsymbol{A}\|\operatorname{Tr}(\boldsymbol{B})$ for non-negative definite matrix $\boldsymbol{B}$, and from equation 30 that bounds $\frac{1}{m}\operatorname{Tr}(\boldsymbol{\Phi}_{\hat{\mathcal{S}}})$. From Lemma K.5, we have

$$\Pr\left(\frac{1}{m}\boldsymbol{\sigma}_i^T\hat{\boldsymbol{U}}_n\boldsymbol{Q}_{-i}(\hat{\boldsymbol{U}}_n - \gamma\hat{\boldsymbol{V}}_n)^T\boldsymbol{\sigma}_i - \alpha > t\right) \leq C'e^{-mc'\max(t,t^2)},$$

for some $C', c'$ independent of $N, m$. From equation 49 in the proof of Theorem 5.1, we have

$$|\alpha - \delta| = \mathcal{O}\left(\frac{1}{\sqrt{m}}\right).$$

Combining all results above, we deduce that

$$\mathbb{E}\left(\|\boldsymbol{D}_2\|^2\right) = \mathbb{E}\left(\max_{1\leq i\leq N}[\boldsymbol{D}_2^2]_i\right) = \int_0^\infty \Pr\left(\max_{1\leq i\leq N}[\boldsymbol{D}_2^2]_i > t\right)dt = \mathcal{O}\left(\frac{1}{m}\right),$$

and therefore

$$\mathbb{E}[\|\boldsymbol{C}\boldsymbol{C}^T\|] = \mathcal{O}\left(\frac{1}{\sqrt{m}}\right).$$

$\square$

**Lemma F.8.** *Under Assumptions 1 and 2, let $\boldsymbol{Z}_{23} \in \mathbb{R}^{n\times n}$ be the matrix defined in equation 61 as*

$$\boldsymbol{Z}_{23} = \frac{1}{m}\frac{1}{1+\delta}\sum_{i=1}^N\mathbb{E}\left[\boldsymbol{Q}_{-i}^T\boldsymbol{M}\bar{\boldsymbol{Q}}_m(\hat{\boldsymbol{U}}_n - \gamma\hat{\boldsymbol{V}}_n)^T\boldsymbol{\sigma}_i\boldsymbol{\sigma}_i^T\hat{\boldsymbol{U}}_n\boldsymbol{Q}_{-i}\frac{\delta - \frac{1}{m}\boldsymbol{\sigma}_i^T\hat{\boldsymbol{U}}_n\boldsymbol{Q}_{-i}(\hat{\boldsymbol{U}}_n - \gamma\hat{\boldsymbol{V}}_n)^T\boldsymbol{\sigma}_i}{1+\frac{1}{m}\boldsymbol{\sigma}_i^T\hat{\boldsymbol{U}}_n\boldsymbol{Q}_{-i}(\hat{\boldsymbol{U}}_n - \gamma\hat{\boldsymbol{V}}_n)^T\boldsymbol{\sigma}_i}\right].$$

*Then,*

$$\|\boldsymbol{Z}_{23}\| = \mathcal{O}\left(\frac{1}{\sqrt{m}}\right).$$

*Proof.* Let the matrices

$$\boldsymbol{B}_i = m^{-\frac{1}{4}}\frac{1}{\sqrt{m}}\boldsymbol{Q}_{-i}^T\boldsymbol{M}\bar{\boldsymbol{Q}}_m(\hat{\boldsymbol{U}}_n - \gamma\hat{\boldsymbol{V}}_n)^T\boldsymbol{\sigma}_i$$

and

$$\boldsymbol{C}_i^T = m^{\frac{1}{4}}\frac{1}{\sqrt{m}}\boldsymbol{\sigma}_i^T\hat{\boldsymbol{U}}_n\boldsymbol{Q}_{-i}\frac{\delta - \frac{1}{m}\boldsymbol{\sigma}_i^T\hat{\boldsymbol{U}}_n\boldsymbol{Q}_{-i}(\hat{\boldsymbol{U}}_n - \gamma\hat{\boldsymbol{V}}_n)^T\boldsymbol{\sigma}_i}{1+\frac{1}{m}\boldsymbol{\sigma}_i^T\hat{\boldsymbol{U}}_n\boldsymbol{Q}_{-i}(\hat{\boldsymbol{U}}_n - \gamma\hat{\boldsymbol{V}}_n)^T\boldsymbol{\sigma}_i}.$$

We decompose $\boldsymbol{Z}_{23}$ with its symmetric and skew-symmetric parts as

$$\boldsymbol{Z}_{23} = \frac{1}{1+\delta}\sum_{i=1}^N\mathbb{E}[\boldsymbol{B}_i\boldsymbol{C}_i^T]$$

$$= \frac{1}{1+\delta}\sum_{i=1}^N\mathbb{E}\left[\frac{\boldsymbol{B}_i\boldsymbol{C}_i^T + \boldsymbol{C}_i\boldsymbol{B}_i^T}{2}\right] + \frac{1}{1+\delta}\mathbb{E}\left[\sum_{i=1}^N\frac{\boldsymbol{B}_i\boldsymbol{C}_i^T - \boldsymbol{C}_i\boldsymbol{B}_i^T}{2}\right].$$

With the same reasoning on the symmetric part and on the skew-symmetric part than for equation 41, we get for the operator norm

$$\|\boldsymbol{Z}_{23}\| \leq \frac{1}{1+\delta} \left\| \mathbb{E}\left[ \sum_{i=1}^{N} \boldsymbol{B}_i \boldsymbol{B}_i^T \right] \right\| + \frac{1}{1+\delta} \left\| \mathbb{E}\left[ \sum_{i=1}^{N} \boldsymbol{C}_i \boldsymbol{C}_i^T \right] \right\|.$$

We want to show that both $\left\| \mathbb{E}\left[ \sum_{i=1}^{N} \boldsymbol{B}_i \boldsymbol{B}_i^T \right] \right\|$ and $\left\| \mathbb{E}\left[ \sum_{i=1}^{N} \boldsymbol{C}_i \boldsymbol{C}_i^T \right] \right\|$ vanish. We write $\mathbb{E}\left[ \sum_{i=1}^{N} \boldsymbol{C}_i \boldsymbol{C}_i^T \right]$ as

$$\mathbb{E}\left[ \sum_{i=1}^{N} \boldsymbol{C}_i \boldsymbol{C}_i^T \right] = \mathbb{E}\left[ \sum_{i=1}^{N} \frac{1}{\sqrt{m}} \boldsymbol{Q}_{-i}^T \hat{\boldsymbol{U}}_n^T \boldsymbol{\sigma}_i \boldsymbol{\sigma}_i^T \hat{\boldsymbol{U}}_n \boldsymbol{Q}_{-i} \frac{\left(\delta - \frac{1}{m} \boldsymbol{\sigma}_i^T \hat{\boldsymbol{U}}_n \boldsymbol{Q}_{-i}(\hat{\boldsymbol{U}}_n - \gamma \hat{\boldsymbol{V}}_n)^T \boldsymbol{\sigma}_i\right)^2}{\left(1 + \frac{1}{m} \boldsymbol{\sigma}_i^T \hat{\boldsymbol{U}}_n \boldsymbol{Q}_{-i}(\hat{\boldsymbol{U}}_n - \gamma \hat{\boldsymbol{V}}_n)^T \boldsymbol{\sigma}_i\right)^2} \right]$$

$$= \mathbb{E}\left[ \sum_{i=1}^{N} \frac{1}{\sqrt{m}} \boldsymbol{Q}_m^T \hat{\boldsymbol{U}}_n^T \boldsymbol{\sigma}_i \boldsymbol{\sigma}_i^T \hat{\boldsymbol{U}}_n \boldsymbol{Q}_m \left(\delta - \frac{1}{m} \boldsymbol{\sigma}_i^T \hat{\boldsymbol{U}}_n \boldsymbol{Q}_{-i}(\hat{\boldsymbol{U}}_n - \gamma \hat{\boldsymbol{V}}_n)^T \boldsymbol{\sigma}_i\right)^2 \right]$$

$$= \mathbb{E}\left[ \frac{1}{\sqrt{m}} \boldsymbol{Q}_m^T \hat{\boldsymbol{U}}_n^T \boldsymbol{\Sigma}_{\hat{\mathcal{S}}}^T \boldsymbol{D}_3^2 \boldsymbol{\Sigma}_{\hat{\mathcal{S}}} \hat{\boldsymbol{U}}_n \boldsymbol{Q}_m \right],$$

where $\boldsymbol{D}_3 \in \mathbb{R}^{N \times N}$ is a diagonal matrix for which, for all $i \in [N]$, we have

$$[\boldsymbol{D}_3]_i = \delta - \frac{1}{m} \boldsymbol{\sigma}_i^T \hat{\boldsymbol{U}}_n \boldsymbol{Q}_{-i}(\hat{\boldsymbol{U}}_n - \gamma \hat{\boldsymbol{V}}_n)^T \boldsymbol{\sigma}_i.$$

With a similar proof than for Lemma E.3, and from Lemma E.1 and Theorem 5.1, we find that

$$\mathbb{E}\left( \|\boldsymbol{D}_3\|^2 \right) = \mathcal{O}\left( \frac{1}{m} \right).$$

From Lemma I.4, we know there exists a real $K'_{\boldsymbol{Q}} > 0$ such that, for all $m$, we have

$$\left\| \frac{1}{\sqrt{m}} \boldsymbol{\Sigma}_{\hat{\mathcal{S}}} \hat{\boldsymbol{U}}_n \boldsymbol{Q}_m \right\| \leq K'_{\boldsymbol{Q}}$$

and

$$\left\| \frac{1}{\sqrt{m}} \boldsymbol{Q}_m (\hat{\boldsymbol{U}}_n - \gamma \hat{\boldsymbol{V}}_n)^T \boldsymbol{\Sigma}_{\hat{\mathcal{S}}}^T \right\| \leq 2 K'_{\boldsymbol{Q}}.$$

We deduce thus

$$\left\| \mathbb{E}\left[ \sum_{i=1}^{N} \boldsymbol{C}_i \boldsymbol{C}_i^T \right] \right\| = \mathcal{O}\left( \frac{1}{\sqrt{m}} \right).$$

We write $\mathbb{E}\left[ \sum_{i=1}^{N} \boldsymbol{B}_i \boldsymbol{B}_i^T \right]$ as

$$\mathbb{E}\left[ \sum_{i=1}^{N} \boldsymbol{B}_i \boldsymbol{B}_i^T \right] = \mathbb{E}\left[ \sum_{i=1}^{N} \frac{1}{m\sqrt{m}} \boldsymbol{Q}_{-i}^T \boldsymbol{M} \bar{\boldsymbol{Q}}_m (\hat{\boldsymbol{U}}_n - \gamma \hat{\boldsymbol{V}}_n)^T \boldsymbol{\sigma}_i \boldsymbol{\sigma}_i^T (\hat{\boldsymbol{U}}_n - \gamma \hat{\boldsymbol{V}}_n) \bar{\boldsymbol{Q}}_m^T \boldsymbol{M}^T \boldsymbol{Q}_{-i} \right]$$

$$= \frac{1}{\sqrt{m}} \frac{N}{m} \mathbb{E}\left[ \boldsymbol{Q}_-^T \boldsymbol{M} \bar{\boldsymbol{Q}}_m (\hat{\boldsymbol{U}}_n - \gamma \hat{\boldsymbol{V}}_n)^T \boldsymbol{\Phi}_{\hat{\mathcal{S}}} (\hat{\boldsymbol{U}}_n - \gamma \hat{\boldsymbol{V}}_n) \bar{\boldsymbol{Q}}_m^T \boldsymbol{M}^T \boldsymbol{Q}_- \right],$$

With a similar proof than for Lemma I.1, we can show there exists a real $K_{\bar{\boldsymbol{Q}}} > 0$ such that, for all $m$, we have

$$\|\bar{\boldsymbol{Q}}_m\| \leq K_{\bar{\boldsymbol{Q}}}.$$

Let $\hat{\boldsymbol{A}}_m = \hat{\boldsymbol{U}}_n (\hat{\boldsymbol{U}}_n - \gamma \hat{\boldsymbol{V}}_n)^T$ be the empirical transition model matrix defined in equation 14. Under Assumption 2, $\hat{\boldsymbol{A}}_m$ is invertible. From Lemma I.3, we have

$$\left\| \bar{\boldsymbol{Q}}_m (\hat{\boldsymbol{U}}_n - \gamma \hat{\boldsymbol{V}}_n)^T \boldsymbol{\Phi}_{\hat{\mathcal{S}}} \right\| = \left\| \bar{\boldsymbol{Q}}_m (\hat{\boldsymbol{U}}_n - \gamma \hat{\boldsymbol{V}}_n)^T \boldsymbol{\Phi}_{\hat{\mathcal{S}}} \hat{\boldsymbol{U}}_n (\hat{\boldsymbol{U}}_n - \gamma \hat{\boldsymbol{V}}_n)^T \hat{\boldsymbol{A}}_m^{-1} \right\|$$

$$= \left\| \frac{m}{N} (1+\delta) \left[ \boldsymbol{I}_n - \lambda \bar{\boldsymbol{Q}}_m \right] (\hat{\boldsymbol{U}}_n - \gamma \hat{\boldsymbol{V}}_n)^T \hat{\boldsymbol{A}}_m^{-1} \right\|$$

$$\leq 2 \frac{m}{N} \frac{1+\delta}{\xi_{\min}} (1 + K_{\bar{\boldsymbol{Q}}}).$$

From above and from Lemma F.9 that upper bounds $\|M\bar{Q}_m\|$, we conclude that

$$\left\| \mathbb{E}\left[\sum_{i=1}^{N} B_i B_i^T\right] \right\| = \mathcal{O}\left(\frac{1}{\sqrt{m}}\right).$$

$\square$

**Lemma F.9.** *Under Assumptions 1 and 2, let $\bar{Q}_m \in \mathbb{R}^{n \times n}$ be the deterministic resolvent defined in equation 15, and let $M$ be either any matrix with a bounded operator norm or $M = \hat{U}_n^T \Psi_{\hat{S}} \hat{U}_n$. Then there exists a real $K > 0$ such that, for all $m$, we have*

$$\|M\bar{Q}_m\| \le K.$$

*Proof.* With a similar proof than for Lemma I.1, we can show there exists a real $K_{\bar{Q}} > 0$ such that, for all $m$, we have

$$\|\bar{Q}_m\| \le K_{\bar{Q}}.$$

In the case where $M$ is a matrix with a bounded operator norm, i.e., $\|M\| \le K_M$ we have

$$\|M\bar{Q}_m\| \le K_M K_{\bar{Q}}.$$

Otherwise, when $M = \hat{U}_n^T \Psi_{\hat{S}} \hat{U}_n$, we consider $\hat{A}_m = \hat{U}_n(\hat{U}_n - \gamma\hat{V}_n)^T$ the empirical transition model matrix defined in equation 14. Under Assumption 2, $\hat{A}_m$ is invertible. From Lemma I.3, we have

$$
\begin{aligned}
\|M\bar{Q}_m\| &= \left\| \frac{N}{m}\frac{1}{1+\delta}\hat{U}_n^T \Phi_{\hat{S}} \hat{U}_n \bar{Q}_m \right\| \\
&= \left\| \frac{N}{m}\frac{1}{1+\delta}\hat{U}_n^T \hat{A}_m^{-1}\hat{U}_n(\hat{U}_n - \gamma\hat{V}_n)^T \Phi_{\hat{S}} \hat{U}_n \bar{Q}_m \right\| \\
&= \left\| \hat{U}_n^T \hat{A}_m^{-1}\hat{U}_n \left[ I_n - \lambda\bar{Q}_m \right] \right\| \\
&\le \frac{1}{\xi_{\min}}(1 + \lambda K_{\bar{Q}}).
\end{aligned}
$$

$\square$

## G   PROOF OF THEOREM 5.3

To simplify the notations, we denote the matrix $Q_m$ as the resolvent $Q_m(\lambda)$ (defined in equation 13), and we set $p = |\mathcal{S}|$. We define the matrices $\Psi_{\hat{S}} \in \mathbb{R}^{m \times m}$ and $\Psi_{\mathcal{S}} \in \mathbb{R}^{p \times p}$ as

$$\Psi_{\hat{S}} = \frac{N}{m}\frac{1}{1+\delta}\Phi_{\hat{S}} \quad \text{and} \quad \Psi_{\mathcal{S}} = \frac{N}{m}\frac{1}{1+\delta}\Phi_{\mathcal{S}}.$$

We also add the notation $A = B + \mathcal{O}_{\|\cdot\|}\left(\frac{1}{\sqrt{m}}\right)$ which means that $\|A - B\| = \mathcal{O}\left(\frac{1}{\sqrt{m}}\right)$.

Under Assumptions 1, 2 and 3, this section is dedicated to find an asymptotic deterministic version of the true $\mathrm{MSBE}(\hat{\theta})$ defined in equation 2 with a similar approach than the one used in Appendix F for $\widehat{\mathrm{MSBE}}(\hat{\theta})$. In proofs of both Theorem 5.1 and Theorem 5.2, we constantly use the fact that $\|Q_m\|$ is uniformly bounded by a constant $K_Q > 0$. We can also easily bound the operator norm of $\frac{1}{m}\Sigma_{\hat{S}}^T \Sigma_{\hat{S}} \hat{U}_n Q_m$ since the empirical transition model matrix $\hat{A}_m = \hat{U}_n(\hat{U}_n - \gamma\hat{V}_n)^T$ (equation 14) is invertible under Assumption 2. Indeed,

$$\left\| \frac{1}{m}\Sigma_{\hat{S}}^T \Sigma_{\hat{S}} \hat{U}_n Q_m \right\| = \left\| \frac{1}{m}\hat{A}_m^{-1}\hat{U}_n(\hat{U}_n - \gamma\hat{V}_n)^T \Sigma_{\hat{S}}^T \Sigma_{\hat{S}} \hat{U}_n Q_m \right\| = \left\| \hat{A}_m^{-1}\hat{U}_n \left[ I_n - \lambda Q_m \right] \right\|$$

$$\le \frac{1}{\xi_{\min}}\left(1 + \lambda K_Q\right)$$

However, we do not have such simple control for $\frac{1}{m}\boldsymbol{\Sigma}_{\mathcal{S}}^T\boldsymbol{\Sigma}_{\mathcal{S}}\boldsymbol{U}_n\boldsymbol{Q}_m$ since $\boldsymbol{U}_n(\boldsymbol{U}_n - \gamma\boldsymbol{V}_n)^T$ is not invertible until all states are visited. Furthermore, from Corollary K.1.1, only a $\mathcal{O}(\sqrt{m})$ upper bound can be derived for $\left\|\frac{1}{\sqrt{m}}\boldsymbol{\Sigma}_{\hat{\mathcal{S}}}\right\|$ or $\left\|\frac{1}{\sqrt{p}}\boldsymbol{\Sigma}_{\mathcal{S}}\right\|$. Nonetheless, with the additional Assumption 3, we can bound $\frac{1}{m}\boldsymbol{D}_{\boldsymbol{\pi}}(\boldsymbol{I}_p - \gamma\boldsymbol{P})\boldsymbol{\Sigma}_{\mathcal{S}}^T\boldsymbol{\Sigma}_{\mathcal{S}}\boldsymbol{U}_n\boldsymbol{Q}_m$ as stated by Lemma G.1. Fortunately for us, the proof of Theorem 5.3 indicates that controlling the operator norm of $\frac{1}{m}\boldsymbol{D}_{\boldsymbol{\pi}}(\boldsymbol{I}_p - \gamma\boldsymbol{P})\boldsymbol{\Sigma}_{\mathcal{S}}^T\boldsymbol{\Sigma}_{\mathcal{S}}\boldsymbol{U}_n\boldsymbol{Q}_m$ is sufficient.

**Lemma G.1.** *Under Assumptions 1, 2 and 3, there exists $K > 0$ such that, for all $m$, we have*

$$\left\|\frac{1}{m}\boldsymbol{D}_{\boldsymbol{\pi}}(\boldsymbol{I}_p - \gamma\boldsymbol{P})\boldsymbol{\Sigma}_{\mathcal{S}}^T\boldsymbol{\Sigma}_{\mathcal{S}}\boldsymbol{U}_n\boldsymbol{Q}_m\right\| \le K.$$

*Proof.* The main idea of the proof is to use a similar reasoning than for $\left\|\frac{1}{m}\hat{\boldsymbol{A}}_m\boldsymbol{\Sigma}_{\hat{\mathcal{S}}}^T\boldsymbol{\Sigma}_{\hat{\mathcal{S}}}\boldsymbol{Q}_m\right\|$. To this end, we use the triangular inequality, the decomposition $\boldsymbol{D}_{\boldsymbol{\pi}}(\boldsymbol{I}_p - \gamma\boldsymbol{P}) = \boldsymbol{U}_n(\boldsymbol{U}_n - \gamma\boldsymbol{V}_n)^T + \boldsymbol{D}_{\boldsymbol{\pi}}(\boldsymbol{I}_p - \gamma\boldsymbol{P}) - \boldsymbol{U}_n(\boldsymbol{U}_n - \gamma\boldsymbol{V}_n)^T$, and Assumption 3 as follows:

$$\left\|\frac{1}{m}\boldsymbol{D}_{\boldsymbol{\pi}}(\boldsymbol{I}_p - \gamma\boldsymbol{P})\boldsymbol{\Sigma}_{\mathcal{S}}^T\boldsymbol{\Sigma}_{\mathcal{S}}\boldsymbol{U}_n\boldsymbol{Q}_m\right\| \le \left\|\frac{1}{m}\boldsymbol{U}_n(\boldsymbol{U}_n - \gamma\boldsymbol{V}_n)^T\boldsymbol{\Sigma}_{\mathcal{S}}^T\boldsymbol{\Sigma}_{\mathcal{S}}\boldsymbol{U}_n\boldsymbol{Q}_m\right\|$$

$$+ \left\|\frac{1}{m}\left[\boldsymbol{D}_{\boldsymbol{\pi}}(\boldsymbol{I}_p - \gamma\boldsymbol{P}) - \boldsymbol{U}_n(\boldsymbol{U}_n - \gamma\boldsymbol{V}_n)^T\right]\boldsymbol{\Sigma}_{\mathcal{S}}^T\boldsymbol{\Sigma}_{\mathcal{S}}\boldsymbol{U}_n\boldsymbol{Q}_m\right\|$$

$$\le \underbrace{\left\|\boldsymbol{U}_n\left[\boldsymbol{I}_n - \lambda\boldsymbol{Q}_m\right]\right\|}_{=Z_1}$$

$$+ \underbrace{\left\|\frac{1}{\sqrt{m}}\boldsymbol{U}_n(\boldsymbol{U}_n - \gamma\boldsymbol{V}_n)^T\boldsymbol{\Sigma}_{\mathcal{S}}^T - \frac{1}{\sqrt{m}}\boldsymbol{D}_{\boldsymbol{\pi}}\left[\boldsymbol{I}_{|\mathcal{S}|} - \gamma\boldsymbol{P}\right]\boldsymbol{\Sigma}_{\mathcal{S}}^T\right\|\left\|\frac{1}{\sqrt{m}}\boldsymbol{\Sigma}_{\mathcal{S}}\boldsymbol{U}_n\boldsymbol{Q}_m\right\|}_{=Z_2}.$$

From Lemma I.1, we know there exists $K_{\boldsymbol{Q}} > 0$ such that, for all $m$, we have $\|\boldsymbol{Q}_m\| \le K_{\boldsymbol{Q}}$. Therefore, for the left-hand part $Z_1$, we have

$$\|Z_1\| = \left\|\boldsymbol{U}_n\left[\boldsymbol{I}_n - \lambda\boldsymbol{Q}_m\right]\right\| \le 1 + \lambda K_{\boldsymbol{Q}}.$$

From Assumption 3, for the right-hand part $Z_2$, we have

$$\left\|\frac{1}{\sqrt{m}}\boldsymbol{U}_n(\boldsymbol{U}_n - \gamma\boldsymbol{V}_n)^T\boldsymbol{\Sigma}_{\mathcal{S}}^T - \frac{1}{\sqrt{m}}\boldsymbol{D}_{\boldsymbol{\pi}}\left[\boldsymbol{I}_{|\mathcal{S}|} - \gamma\boldsymbol{P}\right]\boldsymbol{\Sigma}_{\mathcal{S}}^T\right\|$$

$$\le \left\|\boldsymbol{U}_n(\boldsymbol{U}_n - \gamma\boldsymbol{V}_n)^T - \boldsymbol{D}_{\boldsymbol{\pi}}\left[\boldsymbol{I}_{|\mathcal{S}|} - \gamma\boldsymbol{P}\right]\right\|\left\|\frac{1}{\sqrt{m}}\boldsymbol{\Sigma}_{\mathcal{S}}\right\|$$

$$= \mathcal{O}(1),$$

since $\|\boldsymbol{\Sigma}_{\mathcal{S}}\| = \mathcal{O}(m)$ from Corollary K.1.1. Furthermore, from Lemma I.4, we know there exists a real $K'_{\boldsymbol{Q}} > 0$ such that, for all $m$, we have

$$\left\|\frac{1}{\sqrt{m}}\boldsymbol{\Sigma}_{\mathcal{S}}\boldsymbol{U}_n\boldsymbol{Q}_m\right\| = \left\|\frac{1}{\sqrt{m}}\boldsymbol{\Sigma}_{\hat{\mathcal{S}}}\hat{\boldsymbol{U}}_n\boldsymbol{Q}_m\right\| \le K'_{\boldsymbol{Q}}.$$

We have $\|Z_2\| = \mathcal{O}(1)$ which concludes the proof. $\square$

Assumption 3 may hold for sufficiently large $n$ since $\mathbb{E}\left[\boldsymbol{U}_n(\boldsymbol{U}_n - \gamma\boldsymbol{V}_n)^T\right] \to \boldsymbol{D}_{\boldsymbol{\pi}}\left[\boldsymbol{I}_p - \gamma\boldsymbol{P}\right]$ as $n \to \infty$ (Tsitsiklis & Van Roy, 1996; Nedić & Bertsekas, 2003). Indeed, Tagorti & Scherrer (2015) has established that we can control $\|\boldsymbol{U}_n(\boldsymbol{U}_n - \gamma\boldsymbol{V}_n)^T - \boldsymbol{D}_{\boldsymbol{\pi}}\left[\boldsymbol{I}_{|\mathcal{S}|} - \gamma\boldsymbol{P}\right]\|$ with a sufficiently large $n$.

With the additional Assumption 3, we can now present the following result on the asymptotic Mean-Squared Bellman error.

**Theorem G.2** (Asymptotic MSBE). *Under Assumptions 1, 2, and 3, the determinsitic asymptotic MSBE is*

$$\overline{\mathrm{MSBE}}(\hat{\boldsymbol{\theta}}) = \left\| \bar{\boldsymbol{r}} + \gamma \frac{1}{\sqrt{n}} \boldsymbol{P} \boldsymbol{\Psi}_{\mathcal{S}} \boldsymbol{U}_n \bar{\boldsymbol{Q}}_m(\lambda) \boldsymbol{r} - \frac{1}{\sqrt{n}} \boldsymbol{\Psi}_{\mathcal{S}} \boldsymbol{U}_n \bar{\boldsymbol{Q}}_m(\lambda) \boldsymbol{r} \right\|_{\boldsymbol{D}_{\boldsymbol{\pi}}}^2 + \Delta,$$

*with second-order correction factor*

$$\Delta = \frac{1}{n} \frac{\frac{1}{N} \mathrm{Tr}\big( \boldsymbol{\Lambda}_{\boldsymbol{P}} \big[ \boldsymbol{\Theta}_{\mathcal{S}} \boldsymbol{\Psi}_2 \boldsymbol{\Theta}_{\mathcal{S}}^T - 2\boldsymbol{\Theta}_{\mathcal{S}} (\boldsymbol{U}_n - \gamma \boldsymbol{V}_n)^T \boldsymbol{\Psi}_{\mathcal{S}} + \boldsymbol{\Psi}_{\mathcal{S}} \big] \big)}{1 - \frac{1}{N} \mathrm{Tr}\big( \boldsymbol{\Psi}_2 \bar{\boldsymbol{Q}}_m(\lambda)^T \boldsymbol{\Psi}_1 \bar{\boldsymbol{Q}}_m(\lambda) \big)} \| \bar{\boldsymbol{Q}}_m(\lambda) \boldsymbol{r} \|_{\boldsymbol{\Psi}_1}^2,$$

*where*

$$\boldsymbol{\Lambda}_{\boldsymbol{P}} = [\boldsymbol{I}_{|\mathcal{S}|} - \gamma \boldsymbol{P}]^T \boldsymbol{D}_{\boldsymbol{\pi}} [\boldsymbol{I}_{|\mathcal{S}|} - \gamma \boldsymbol{P}],$$
$$\boldsymbol{\Theta}_{\mathcal{S}} = \boldsymbol{\Psi}_{\mathcal{S}} \boldsymbol{U}_n \bar{\boldsymbol{Q}}_m(\lambda).$$

*As $N, m, d \to \infty$ with asymptotic constant ratio $N/m$, $\mathrm{MSBE}(\hat{\boldsymbol{\theta}}) - \overline{\mathrm{MSBE}}(\hat{\boldsymbol{\theta}}) \xrightarrow{a.s} 0$.*

*Proof.* We decompose $\mathrm{MSBE}(\hat{\boldsymbol{\theta}})$ as

$$\mathrm{MSBE}(\hat{\boldsymbol{\theta}}) = \| \bar{\boldsymbol{r}} + \gamma \boldsymbol{P} \boldsymbol{\Sigma}_{\mathcal{S}}^T \hat{\boldsymbol{\theta}} - \boldsymbol{\Sigma}_{\mathcal{S}}^T \hat{\boldsymbol{\theta}} \|_{\boldsymbol{D}_{\boldsymbol{\pi}}}^2 = \| \bar{\boldsymbol{r}} + \big[ \gamma \boldsymbol{P} - \boldsymbol{I}_p \big] \boldsymbol{\Sigma}_{\mathcal{S}}^T \boldsymbol{\theta} \|_{\boldsymbol{D}_{\boldsymbol{\pi}}}^2 \tag{64}$$

$$= \left\| \bar{\boldsymbol{r}} - \frac{1}{m\sqrt{n}} \big[ \boldsymbol{I}_p - \gamma \boldsymbol{P} \big] \boldsymbol{\Sigma}_{\mathcal{S}}^T \boldsymbol{\Sigma}_{\mathcal{S}} \boldsymbol{U}_n \boldsymbol{Q}_m \boldsymbol{r} \right\|_{\boldsymbol{D}_{\boldsymbol{\pi}}}^2 \tag{65}$$

$$= \| \bar{\boldsymbol{r}} \|_{\boldsymbol{D}_{\boldsymbol{\pi}}}^2 \tag{66}$$

$$\underbrace{- \frac{2}{m\sqrt{n}} \bar{\boldsymbol{r}}^T \boldsymbol{D}_{\boldsymbol{\pi}} \big[ \boldsymbol{I}_p - \gamma \boldsymbol{P} \big] \boldsymbol{\Sigma}_{\mathcal{S}}^T \boldsymbol{\Sigma}_{\mathcal{S}} \boldsymbol{U}_n \boldsymbol{Q}_m \boldsymbol{r}}_{=Z_2} \tag{67}$$

$$\underbrace{+ \left\| \frac{1}{m\sqrt{n}} \big[ \boldsymbol{I}_p - \gamma \boldsymbol{P} \big] \boldsymbol{\Sigma}_{\mathcal{S}}^T \boldsymbol{\Sigma}_{\mathcal{S}} \boldsymbol{U}_n \boldsymbol{Q}_m \boldsymbol{r} \right\|_{\boldsymbol{D}_{\boldsymbol{\pi}}}^2}_{=Z_3}. \tag{68}$$

We want to find an asymptotic equivalent for both $Z_2$ and $Z_3$. From Lemma G.3, we have

$$\mathbb{E}\big[ Z_2 \big] = \frac{2}{\sqrt{n}} \bar{\boldsymbol{r}}^T \boldsymbol{D}_{\boldsymbol{\pi}} \big[ \boldsymbol{I}_p - \gamma \boldsymbol{P} \big] \boldsymbol{\Psi}_{\mathcal{S}} \boldsymbol{U}_n \bar{\boldsymbol{Q}}_m \boldsymbol{r} + \mathcal{O}\left( \frac{1}{\sqrt{m}} \right).$$

For $Z_3$, we have

$$Z_3 = \left\| \frac{1}{m\sqrt{n}} \big[ \boldsymbol{I}_p - \gamma \boldsymbol{P} \big] \boldsymbol{\Sigma}_{\mathcal{S}}^T \boldsymbol{\Sigma}_{\mathcal{S}} \boldsymbol{U}_n \boldsymbol{Q}_m \boldsymbol{r} \right\|_{\boldsymbol{D}_{\boldsymbol{\pi}}}^2 = \frac{1}{nm^2} \boldsymbol{r}^T \boldsymbol{Q}_m^T \boldsymbol{U}_n^T \boldsymbol{\Sigma}_{\mathcal{S}}^T \boldsymbol{\Sigma}_{\mathcal{S}} \boldsymbol{\Lambda}_{\boldsymbol{P}} \boldsymbol{\Sigma}_{\mathcal{S}}^T \boldsymbol{\Sigma}_{\mathcal{S}} \boldsymbol{U}_n \boldsymbol{Q}_m \boldsymbol{r}.$$

From Lemma G.4, we have

$$\mathbb{E}\big[ Z_3 \big] = \frac{1}{n} \boldsymbol{r}^T \bar{\boldsymbol{Q}}_m^T \boldsymbol{U}_n^T \boldsymbol{\Psi}_{\mathcal{S}} \boldsymbol{\Lambda}_{\boldsymbol{P}} \boldsymbol{\Psi}_{\mathcal{S}} \boldsymbol{U}_n \bar{\boldsymbol{Q}}_m \boldsymbol{r}$$

$$+ \frac{1}{n} \frac{\frac{1}{N} \mathrm{Tr}\big( \boldsymbol{\Lambda}_{\boldsymbol{P}} \big[ \boldsymbol{\Theta}_{\mathcal{S}} \boldsymbol{\Psi}_2 \boldsymbol{\Theta}_{\mathcal{S}}^T - 2\boldsymbol{\Theta}_{\mathcal{S}} (\boldsymbol{U}_n - \gamma \boldsymbol{V}_n)^T \boldsymbol{\Psi}_{\mathcal{S}} + \boldsymbol{\Psi}_{\mathcal{S}} \big] \big)}{1 - \frac{1}{N} \mathrm{Tr}\big( \boldsymbol{\Psi}_2 \bar{\boldsymbol{Q}}_m^T \boldsymbol{\Psi}_1 \bar{\boldsymbol{Q}}_m \big)} \| \bar{\boldsymbol{Q}}_m \boldsymbol{r} \|_{\boldsymbol{\Psi}_1}^2$$

$$+ \mathcal{O}\left( \frac{1}{\sqrt{m}} \right).$$

With a similar proof than for Lemma K.6 we can deduce that

$$\mathrm{MSBE}(\hat{\boldsymbol{\theta}}) - \overline{\mathrm{MSBE}}(\hat{\boldsymbol{\theta}}) \xrightarrow{a.s} 0,$$

as $m \to \infty$. $\qquad \square$

**Lemma G.3.** *Under Assumptions 1, 2 and 3, let $Z_2 \in \mathbb{R}$ defined in equation 67 as*

$$Z_2 = \frac{1}{m\sqrt{n}} \mathbb{E}\big[ \bar{\boldsymbol{r}}^T \boldsymbol{D}_{\boldsymbol{\pi}} \big[ \boldsymbol{I}_p - \gamma \boldsymbol{P} \big] \boldsymbol{\Sigma}_{\mathcal{S}}^T \boldsymbol{\Sigma}_{\mathcal{S}} \boldsymbol{U}_n \boldsymbol{Q}_m \boldsymbol{r} \big].$$

*Then*

$$\left| Z_2 - \frac{1}{\sqrt{n}} \bar{\boldsymbol{r}}^T \boldsymbol{D}_{\boldsymbol{\pi}} \big[ \boldsymbol{I}_p - \gamma \boldsymbol{P} \big] \boldsymbol{\Psi}_{\mathcal{S}} \boldsymbol{U}_n \bar{\boldsymbol{Q}}_m \boldsymbol{r} \right| = \mathcal{O}\left( \frac{1}{\sqrt{m}} \right),$$

*for $\bar{\boldsymbol{Q}}_m$ the deterministic resolvent defined in equation 15, and $\boldsymbol{\Psi}_{\mathcal{S}} \in \mathbb{R}^{p \times p}$ defined in equation 21.*

*Proof.* As in equation 32, we decompose the matrix $\boldsymbol{\Sigma}_{\mathcal{S}}^T \boldsymbol{\Sigma}_{\mathcal{S}}$ as

$$\boldsymbol{\Sigma}_{\mathcal{S}}^T \boldsymbol{\Sigma}_{\mathcal{S}} = \sum_{i=1}^N \boldsymbol{\sigma}_i \boldsymbol{\sigma}_i^T,$$

where $\boldsymbol{\sigma}_i = \sigma(\boldsymbol{S}^T \boldsymbol{w}_i) \in \mathbb{R}^m$ for which $\boldsymbol{w}_i \in \mathbb{R}^d$ denotes the i-th row of $\boldsymbol{W}$ defined in equation 5. Let $\boldsymbol{Q}_{-i} \in \mathbb{R}^{n \times n}$ be the following resolvent

$$\boldsymbol{Q}_{-i} = \left[ \frac{1}{m}(\boldsymbol{U}_n - \gamma \boldsymbol{V}_n)^T \boldsymbol{\Sigma}_{\mathcal{S}}^T \boldsymbol{\Sigma}_{\mathcal{S}} \boldsymbol{U}_n - \frac{1}{m}(\boldsymbol{U}_n - \gamma \boldsymbol{V}_n)^T \boldsymbol{\sigma}_i \boldsymbol{\sigma}_i^T \boldsymbol{U}_n + \lambda \boldsymbol{I}_n \right]^{-1},$$

independent of $\boldsymbol{\sigma}_i$ and thus $\boldsymbol{w}_i$. From the Sherman identity (Lemma L.4), we have

$$Z_2 = \frac{1}{m\sqrt{n}} \mathbb{E}\left[ \bar{\boldsymbol{r}}^T \boldsymbol{D}_{\boldsymbol{\pi}} [\boldsymbol{I}_p - \gamma \boldsymbol{P}] \boldsymbol{\Sigma}_{\mathcal{S}}^T \boldsymbol{\Sigma}_{\mathcal{S}} \boldsymbol{U}_n \boldsymbol{Q}_m \boldsymbol{r} \right]$$

$$= \frac{1}{m\sqrt{n}} \mathbb{E}\left[ \sum_{i=1}^N \bar{\boldsymbol{r}}^T \boldsymbol{D}_{\boldsymbol{\pi}} [\boldsymbol{I}_p - \gamma \boldsymbol{P}] \boldsymbol{\sigma}_i \boldsymbol{\sigma}_i^T \boldsymbol{U}_n \boldsymbol{Q}_m \boldsymbol{r} \right]$$

$$= \frac{1}{m\sqrt{n}} \mathbb{E}\left[ \sum_{i=1}^N \frac{\bar{\boldsymbol{r}}^T \boldsymbol{D}_{\boldsymbol{\pi}} [\boldsymbol{I}_p - \gamma \boldsymbol{P}] \boldsymbol{\sigma}_i \boldsymbol{\sigma}_i^T \boldsymbol{U}_n \boldsymbol{Q}_{-i} \boldsymbol{r}}{1 + \frac{1}{m}\boldsymbol{\sigma}_i^T \boldsymbol{U}_n \boldsymbol{Q}_{-i}(\boldsymbol{U}_n - \gamma \boldsymbol{V}_n)^T \boldsymbol{\sigma}_i} \right].$$

Let $\boldsymbol{D} \in \mathbb{R}^{N \times N}$ be a diagonal matrix for which, for all $i \in [N]$, we have

$$\boldsymbol{D}_i = \delta - \frac{1}{m}\boldsymbol{\sigma}_i^T \boldsymbol{U}_n \boldsymbol{Q}_{-i}(\boldsymbol{U}_n - \gamma \boldsymbol{V}_n)^T \boldsymbol{\sigma}_i.$$

We replace $1 + \frac{1}{m}\boldsymbol{\sigma}_j^T \boldsymbol{U}_n \boldsymbol{Q}_{-j}(\boldsymbol{U}_n - \gamma \boldsymbol{V}_n)^T \boldsymbol{\sigma}_j$ by $1 + \delta$ as following

$$Z_2 = \underbrace{\frac{1}{m\sqrt{n}} \frac{1}{1+\delta} \mathbb{E}\left[ \sum_{i=1}^N \bar{\boldsymbol{r}}^T \boldsymbol{D}_{\boldsymbol{\pi}} [\boldsymbol{I}_p - \gamma \boldsymbol{P}] \boldsymbol{\sigma}_i \boldsymbol{\sigma}_i^T \boldsymbol{U}_n \boldsymbol{Q}_{-i} \boldsymbol{r} \right]}_{Z_{21}}$$

$$+ \underbrace{\frac{1}{m\sqrt{n}} \frac{1}{1+\delta} \mathbb{E}\left[ \sum_{i=1}^N \frac{\bar{\boldsymbol{r}}^T \boldsymbol{D}_{\boldsymbol{\pi}} [\boldsymbol{I}_p - \gamma \boldsymbol{P}] \boldsymbol{\sigma}_i \boldsymbol{\sigma}_i^T \boldsymbol{U}_n \boldsymbol{Q}_{-i} \boldsymbol{D}_i \boldsymbol{r}}{1 + \frac{1}{m}\boldsymbol{\sigma}_i^T \boldsymbol{U}_n \boldsymbol{Q}_{-i}(\boldsymbol{U}_n - \gamma \boldsymbol{V}_n)^T \boldsymbol{\sigma}_i} \right]}_{Z_{22}}.$$

We have $Z_{22}$ vanishing since $\mathbb{E}\big[\|\boldsymbol{D}\|\big] = \mathcal{O}\left(\frac{1}{\sqrt{m}}\right)$ and from Lemma G.1. From Theorem 5.1, we have thus

$$Z_2 = \frac{1}{m\sqrt{n}} \frac{1}{1+\delta} \mathbb{E}\left[ \sum_{i=1}^N \bar{\boldsymbol{r}}^T \boldsymbol{D}_{\boldsymbol{\pi}} [\boldsymbol{I}_p - \gamma \boldsymbol{P}] \boldsymbol{\sigma}_i \boldsymbol{\sigma}_i^T \boldsymbol{U}_n \boldsymbol{Q}_{-i} \boldsymbol{r} \right] + \mathcal{O}\left(\frac{1}{\sqrt{m}}\right)$$

$$= \frac{1}{\sqrt{n}} \frac{N}{m} \frac{1}{1+\delta} \bar{\boldsymbol{r}}^T \boldsymbol{D}_{\boldsymbol{\pi}} [\boldsymbol{I}_p - \gamma \boldsymbol{P}] \boldsymbol{\Phi}_{\mathcal{S}} \boldsymbol{U}_n \mathbb{E}[\boldsymbol{Q}_-] \boldsymbol{r} + \mathcal{O}\left(\frac{1}{\sqrt{m}}\right)$$

$$= \frac{1}{\sqrt{n}} \bar{\boldsymbol{r}}^T \boldsymbol{D}_{\boldsymbol{\pi}} [\boldsymbol{I}_p - \gamma \boldsymbol{P}] \boldsymbol{\Psi}_{\mathcal{S}} \boldsymbol{U}_n \bar{\boldsymbol{Q}}_m \boldsymbol{r} + \mathcal{O}\left(\frac{1}{\sqrt{m}}\right).$$

$\square$

**Lemma G.4.** *Under Assumptions 1, 2 and 3, let $\boldsymbol{\Lambda}_{\boldsymbol{P}} \in \mathbb{R}^{p \times p}$ be the matrix defined in equation 21, and let $Z_3 \in \mathbb{R}$ be defined in equation 68 as*

$$Z_3 = \mathbb{E}\left[ \frac{1}{nm^2} \boldsymbol{r}^T \boldsymbol{Q}_m^T \boldsymbol{U}_n^T \boldsymbol{\Sigma}_{\mathcal{S}}^T \boldsymbol{\Sigma}_{\mathcal{S}} \boldsymbol{\Lambda}_{\boldsymbol{P}} \boldsymbol{\Sigma}_{\mathcal{S}}^T \boldsymbol{\Sigma}_{\mathcal{S}} \boldsymbol{U}_n \boldsymbol{Q}_m \boldsymbol{r} \right].$$

*Then*

$$\left| Z_3 - \frac{1}{n} \boldsymbol{r}^T \bar{\boldsymbol{Q}}_m^T \boldsymbol{U}_n^T \boldsymbol{\Psi}_{\mathcal{S}} \boldsymbol{\Lambda}_{\boldsymbol{P}} \boldsymbol{\Psi}_{\mathcal{S}} \boldsymbol{U}_n \bar{\boldsymbol{Q}}_m \boldsymbol{r} \right.$$

$$\left. - \frac{1}{n} \frac{\frac{1}{N} \operatorname{Tr}\left( \boldsymbol{\Lambda}_{\boldsymbol{P}} \big[ \boldsymbol{\Theta}_{\mathcal{S}} \boldsymbol{\Psi}_2 \boldsymbol{\Theta}_{\mathcal{S}}^T - 2\boldsymbol{\Theta}_{\mathcal{S}}(\boldsymbol{U}_n - \gamma \boldsymbol{V}_n)^T \boldsymbol{\Psi}_{\mathcal{S}} + \boldsymbol{\Psi}_{\mathcal{S}} \big] \right)}{1 - \frac{1}{N} \operatorname{Tr}\left( \boldsymbol{\Psi}_2 \bar{\boldsymbol{Q}}_m^T \boldsymbol{\Psi}_1 \bar{\boldsymbol{Q}}_m \right)} \|\bar{\boldsymbol{Q}}_m \boldsymbol{r}\|_{\boldsymbol{\Psi}_1}^2 \right| = \mathcal{O}\left(\frac{1}{\sqrt{m}}\right),$$

where $\bar{Q}_m$ is the deterministic resolvent defined in equation 15, $\Psi_1, \Psi_2 \in \mathbb{R}^{n \times n}$ are defined in equation 19, $\Psi_{\mathcal{S}} \in \mathbb{R}^{p \times p}$ and $\Theta_{\mathcal{S}} \in \mathbb{R}^{p \times n}$ are defined in equation 21.

*Proof.* As in equation 32, we decompose the matrix $\Sigma_{\mathcal{S}}^T \Sigma_{\mathcal{S}}$ as

$$\Sigma_{\mathcal{S}}^T \Sigma_{\mathcal{S}} = \sum_{i=1}^{N} \sigma_i \sigma_i^T,$$

where $\sigma_i = \sigma(S^T w_i) \in \mathbb{R}^m$ for which $w_i \in \mathbb{R}^d$ denotes the i-th row of $W$ defined in equation 5. Let $Q_{-i} \in \mathbb{R}^{n \times n}$ be the following resolvent

$$Q_{-i} = \left[ \frac{1}{m}(U_n - \gamma V_n)^T \Sigma_{\mathcal{S}}^T \Sigma_{\mathcal{S}} U_n - \frac{1}{m}(U_n - \gamma V_n)^T \sigma_i \sigma_i^T U_n + \lambda I_n \right]^{-1}$$

independent of $\sigma_i$ and thus $w_i$. From the Sherman identity (Lemma L.4), we decompose $Z_3$ as

$$Z_3 = \mathbb{E}\left[ \frac{1}{nm^2} r^T Q_m^T U_n^T \Sigma_{\mathcal{S}}^T \Sigma_{\mathcal{S}} \Lambda_P \Sigma_{\mathcal{S}}^T \Sigma_{\mathcal{S}} U_n Q_m r \right] \tag{69}$$

$$= \sum_{i,j=1}^{N} \mathbb{E}\left[ \frac{1}{nm^2} r^T Q_m^T U_n^T \sigma_i \sigma_i^T \Lambda_P \sigma_j \sigma_j^T U_n Q_m r \right] \tag{70}$$

$$= \sum_{i,j=1}^{N} \mathbb{E}\left[ \frac{1}{nm^2} r^T \frac{Q_{-i}^T U_n^T \sigma_i \sigma_i^T}{1 + \frac{1}{m}\sigma_i^T U_n Q_{-i}(U_n - \gamma V_n)^T \sigma_i} \Lambda_P \frac{\sigma_j \sigma_j^T U_n Q_{-j}}{1 + \frac{1}{m}\sigma_j^T U_n Q_{-j}(U_n - \gamma V_n)^T \sigma_j} r \right] \tag{71}$$

$$= \underbrace{\sum_{i=1}^{N} \sum_{j \neq i} \mathbb{E}\left[ \frac{1}{nm^2} r^T \frac{Q_{-i}^T U_n^T \sigma_i \sigma_i^T}{1 + \frac{1}{m}\sigma_i^T U_n Q_{-i}(U_n - \gamma V_n)^T \sigma_i} \Lambda_P \frac{\sigma_j \sigma_j^T U_n Q_{-j}}{1 + \frac{1}{m}\sigma_j^T U_n Q_{-j}(U_n - \gamma V_n)^T \sigma_j} r \right]}_{=Z_{31}}$$

$$+ \underbrace{\sum_{i=1}^{N} \mathbb{E}\left[ \frac{1}{nm^2} r^T \frac{Q_{-i}^T U_n^T \sigma_i \sigma_i^T \Lambda_P \sigma_i \sigma_i^T U_n Q_{-i}}{\left(1 + \frac{1}{m}\sigma_i^T U_n Q_{-i}(U_n - \gamma V_n)^T \sigma_i\right)^2} r \right]}_{=Z_{32}}. \tag{72}$$

From Lemma G.5, we have

$$Z_{31} = \frac{1}{n} r^T \bar{Q}_m^T U_n^T \Psi_{\mathcal{S}} \Lambda_P \Psi_{\mathcal{S}} U_n \bar{Q}_m r$$

$$+ \frac{1}{n} \frac{\frac{1}{N} \operatorname{Tr}\left( \Lambda_P \left[ \Theta_{\mathcal{S}} \Psi_2 \Theta_{\mathcal{S}}^T - 2\Theta_{\mathcal{S}}(U_n - \gamma V_n)^T \Psi_{\mathcal{S}} \right] \right)}{1 - \frac{1}{N} \operatorname{Tr}\left( \Psi_2 \bar{Q}_m^T \Psi_1 \bar{Q}_m \right)} \|\bar{Q}_m r\|_{\Psi_1}^2$$

$$+ \mathcal{O}\left( \frac{1}{\sqrt{m}} \right).$$

For the second term $Z_{32}$, we have from Lemma G.6,

$$Z_{32} = \frac{1}{n} \frac{\frac{1}{N} \operatorname{Tr}(\Lambda_P \Psi_{\mathcal{S}})}{1 - \frac{1}{N} \operatorname{Tr}\left( \Psi_2 \bar{Q}_m^T \Psi_1 \bar{Q}_m \right)} \|\bar{Q}_m r\|_{\Psi_1}^2 + \mathcal{O}\left( \frac{1}{\sqrt{m}} \right).$$

We conclude that

$$Z_3 = Z_{32} + Z_{32}$$

$$= \frac{1}{n} r^T \bar{Q}_m^T U_n^T \Psi_{\mathcal{S}} \Lambda_P \Psi_{\mathcal{S}} U_n \bar{Q}_m r$$

$$+ \frac{1}{n} \frac{\frac{1}{N} \operatorname{Tr}\left( \Lambda_P \left[ \Theta_{\mathcal{S}} \Psi_2 \Theta_{\mathcal{S}}^T - 2\Theta_{\mathcal{S}}(U_n - \gamma V_n)^T \Psi_{\mathcal{S}} + \Psi_{\mathcal{S}} \right] \right)}{1 - \frac{1}{N} \operatorname{Tr}\left( \Psi_2 \bar{Q}_m^T \Psi_1 \bar{Q}_m \right)} \|\bar{Q}_m r\|_{\Psi_1}^2$$

$$+ \mathcal{O}\left( \frac{1}{\sqrt{m}} \right).$$

□

**Lemma G.5.** *Under Assumptions 1, 2 and 3, let $Z_{31} \in \mathbb{R}$ defined in equation 72 as*

$$Z_{31} = \frac{1}{nm^2} \sum_{i=1}^{N} \sum_{j \neq i} \mathbb{E}\left[ \frac{\boldsymbol{r}^T \boldsymbol{Q}_{-i}^T \boldsymbol{U}_n^T \boldsymbol{\sigma}_i \boldsymbol{\sigma}_i^T \boldsymbol{\Lambda}_{\boldsymbol{P}} \boldsymbol{\sigma}_j \boldsymbol{\sigma}_j^T \boldsymbol{U}_n \boldsymbol{Q}_{-j} \boldsymbol{r}}{\left(1 + \frac{1}{m} \boldsymbol{\sigma}_i^T \boldsymbol{U}_n \boldsymbol{Q}_{-i} (\boldsymbol{U}_n - \gamma \boldsymbol{V}_n)^T \boldsymbol{\sigma}_i\right) \left(1 + \frac{1}{m} \boldsymbol{\sigma}_j^T \boldsymbol{U}_n \boldsymbol{Q}_{-j} (\boldsymbol{U}_n - \gamma \boldsymbol{V}_n)^T \boldsymbol{\sigma}_j\right)} \right],$$

*Then*

$$\left| Z_{31} - \frac{1}{n} \boldsymbol{r}^T \bar{\boldsymbol{Q}}_m^T \boldsymbol{U}_n^T \boldsymbol{\Psi}_{\mathcal{S}} \boldsymbol{\Lambda}_{\boldsymbol{P}} \boldsymbol{\Psi}_{\mathcal{S}} \boldsymbol{U}_n \bar{\boldsymbol{Q}}_m \boldsymbol{r} \right.$$

$$\left. - \frac{1}{n} \frac{\frac{1}{N} \operatorname{Tr} \left(\boldsymbol{\Lambda}_{\boldsymbol{P}} \left[\boldsymbol{\Theta}_{\mathcal{S}} \boldsymbol{\Psi}_2 \boldsymbol{\Theta}_{\mathcal{S}}^T - 2\boldsymbol{\Theta}_{\mathcal{S}} (\boldsymbol{U}_n - \gamma \boldsymbol{V}_n)^T \boldsymbol{\Psi}_{\mathcal{S}} \right]\right)}{1 - \frac{1}{N} \operatorname{Tr} \left(\boldsymbol{\Psi}_2 \bar{\boldsymbol{Q}}_m^T \boldsymbol{\Psi}_1 \bar{\boldsymbol{Q}}_m\right)} \|\bar{\boldsymbol{Q}}_m \boldsymbol{r}\|_{\boldsymbol{\Psi}_1}^2 \right| = \mathcal{O}\left(\frac{1}{\sqrt{m}}\right),$$

*where $\bar{\boldsymbol{Q}}_m$ is the deterministic resolvent defined in equation 15, $\boldsymbol{\Psi}_1, \boldsymbol{\Psi}_2 \in \mathbb{R}^{n \times n}$ are defined in equation 19, $\boldsymbol{\Psi}_{\mathcal{S}} \in \mathbb{R}^{p \times p}$ and $\boldsymbol{\Theta}_{\mathcal{S}} \in \mathbb{R}^{p \times n}$ are defined in equation 21.*

*Proof.* Using the Sherman identity (Lemma L.4), we decompose $Z_{31}$ as

$$Z_{31} = \frac{1}{nm^2} \sum_{i=1}^{N} \sum_{j \neq i} \mathbb{E}\left[ \frac{\boldsymbol{r}^T \boldsymbol{Q}_{-i}^T \boldsymbol{U}_n^T \boldsymbol{\sigma}_i \boldsymbol{\sigma}_i^T \boldsymbol{\Lambda}_{\boldsymbol{P}} \boldsymbol{\sigma}_j \boldsymbol{\sigma}_j^T \boldsymbol{U}_n \boldsymbol{Q}_{-j} \boldsymbol{r}}{\left(1 + \frac{1}{m} \boldsymbol{\sigma}_i^T \boldsymbol{U}_n \boldsymbol{Q}_{-i} (\boldsymbol{U}_n - \gamma \boldsymbol{V}_n)^T \boldsymbol{\sigma}_i\right) \left(1 + \frac{1}{m} \boldsymbol{\sigma}_j^T \boldsymbol{U}_n \boldsymbol{Q}_{-j} (\boldsymbol{U}_n - \gamma \boldsymbol{V}_n)^T \boldsymbol{\sigma}_j\right)} \right]$$

$$= \frac{1}{nm^2} \sum_{i=1}^{N} \sum_{j \neq i} \mathbb{E}\left[ \boldsymbol{r}^T \frac{\boldsymbol{Q}_m^T \boldsymbol{U}_n^T \boldsymbol{\sigma}_i \boldsymbol{\sigma}_i^T \boldsymbol{\Lambda}_{\boldsymbol{P}} \boldsymbol{\sigma}_j \boldsymbol{\sigma}_j^T \boldsymbol{U}_n \boldsymbol{Q}_{-j}}{1 + \frac{1}{m} \boldsymbol{\sigma}_j^T \boldsymbol{U}_n \boldsymbol{Q}_{-j} (\boldsymbol{U}_n - \gamma \boldsymbol{V}_n)^T \boldsymbol{\sigma}_j} \boldsymbol{r} \right]$$

$$= \underbrace{\frac{1}{nm^2} \sum_{i=1}^{N} \sum_{j \neq i} \mathbb{E}\left[ \boldsymbol{r}^T \frac{\boldsymbol{Q}_{-j}^T \boldsymbol{U}_n^T \boldsymbol{\sigma}_i \boldsymbol{\sigma}_i^T \boldsymbol{\Lambda}_{\boldsymbol{P}} \boldsymbol{\sigma}_j \boldsymbol{\sigma}_j^T \boldsymbol{U}_n \boldsymbol{Q}_{-j}}{1 + \frac{1}{m} \boldsymbol{\sigma}_j^T \boldsymbol{U}_n \boldsymbol{Q}_{-j} (\boldsymbol{U}_n - \gamma \boldsymbol{V}_n)^T \boldsymbol{\sigma}_j} \boldsymbol{r} \right]}_{=Z_{311}}$$

$$- \underbrace{\frac{1}{nm^3} \sum_{i=1}^{N} \sum_{j \neq i} \mathbb{E}\left[ \boldsymbol{r}^T \frac{\boldsymbol{Q}_{-j}^T \boldsymbol{U}_n^T \boldsymbol{\sigma}_j \boldsymbol{\sigma}_j^T (\boldsymbol{U}_n - \gamma \boldsymbol{V}_n) \boldsymbol{Q}_{-j}^T \boldsymbol{U}_n^T \boldsymbol{\sigma}_i \boldsymbol{\sigma}_i^T \boldsymbol{\Lambda}_{\boldsymbol{P}} \boldsymbol{\sigma}_j \boldsymbol{\sigma}_j^T \boldsymbol{U}_n \boldsymbol{Q}_{-j}}{\left(1 + \frac{1}{m} \boldsymbol{\sigma}_j^T \boldsymbol{U}_n \boldsymbol{Q}_{-j} (\boldsymbol{U}_n - \gamma \boldsymbol{V}_n)^T \boldsymbol{\sigma}_j\right)^2} \boldsymbol{r} \right]}_{=Z_{312}}.$$

We want to find an asymptotic equivalent for both $Z_{311}$ and $Z_{312}$. For $Z_{312}$, we have

$$Z_{312} = \frac{1}{nm^3} \sum_{i=1}^{N} \sum_{j \neq i} \mathbb{E}\left[ \boldsymbol{r}^T \frac{\boldsymbol{Q}_{-j}^T \boldsymbol{U}_n^T \boldsymbol{\sigma}_j \boldsymbol{\sigma}_j^T (\boldsymbol{U}_n - \gamma \boldsymbol{V}_n) \boldsymbol{Q}_{-j}^T \boldsymbol{U}_n^T \boldsymbol{\sigma}_i \boldsymbol{\sigma}_i^T \boldsymbol{\Lambda}_{\boldsymbol{P}} \boldsymbol{\sigma}_j \boldsymbol{\sigma}_j^T \boldsymbol{U}_n \boldsymbol{Q}_{-j}}{\left(1 + \frac{1}{m} \boldsymbol{\sigma}_j^T \boldsymbol{U}_n \boldsymbol{Q}_{-j} (\boldsymbol{U}_n - \gamma \boldsymbol{V}_n)^T \boldsymbol{\sigma}_j\right)^2} \boldsymbol{r} \right]$$

$$= \frac{1}{nm} \sum_{j=1}^{N} \mathbb{E}\left[ \boldsymbol{r}^T \frac{\boldsymbol{Q}_{-j}^T \boldsymbol{U}_n^T \boldsymbol{\sigma}_j \boldsymbol{\sigma}_j^T \boldsymbol{U}_n \boldsymbol{Q}_{-j} \left(\frac{1}{m^2} \boldsymbol{\sigma}_j^T (\boldsymbol{U}_n - \gamma \boldsymbol{V}_n) \boldsymbol{Q}_{-j}^T \boldsymbol{U}_n^T \boldsymbol{\Sigma}_{\mathcal{S}}^{-jT} \boldsymbol{\Sigma}_{\mathcal{S}}^{-j} \boldsymbol{\Lambda}_{\boldsymbol{P}} \boldsymbol{\sigma}_j\right)}{\left(1 + \frac{1}{m} \boldsymbol{\sigma}_j^T \boldsymbol{U}_n \boldsymbol{Q}_{-j} (\boldsymbol{U}_n - \gamma \boldsymbol{V}_n)^T \boldsymbol{\sigma}_j\right)^2} \boldsymbol{r} \right],$$

where $\boldsymbol{\Sigma}_{\mathcal{S}}^{-j} = \sigma(\boldsymbol{W}^{-j} \boldsymbol{S}) \in \mathbb{R}^{(N-1) \times n}$ for which $\boldsymbol{W}^{-j} \in \mathbb{R}^{(N-1) \times d}$ depicts the same matrix than the weight matrix $\boldsymbol{W}$ defined in equation 5 without the $j^{\text{th}}$ row. Let $\boldsymbol{D}_{312} \in \mathbb{R}^{N \times N}$ be a diagonal matrix for which, for all $j \in [N]$, we have

$$[\boldsymbol{D}_{312}]_j = \frac{1}{m^2} \boldsymbol{\sigma}_j^T (\boldsymbol{U}_n - \gamma \boldsymbol{V}_n) \boldsymbol{Q}_{-j}^T \boldsymbol{U}_n^T \boldsymbol{\Sigma}_{\mathcal{S}}^{-jT} \boldsymbol{\Sigma}_{\mathcal{S}}^{-j} \boldsymbol{\Lambda}_{\boldsymbol{P}} \boldsymbol{\sigma}_j - \frac{1}{m^2} \operatorname{Tr}\left((\boldsymbol{U}_n - \gamma \boldsymbol{V}_n) \boldsymbol{Q}_{-j}^T \boldsymbol{U}_n^T \boldsymbol{\Sigma}_{\mathcal{S}}^{-jT} \boldsymbol{\Sigma}_{\mathcal{S}}^{-j} \boldsymbol{\Lambda}_{\boldsymbol{P}} \boldsymbol{\Phi}_{\mathcal{S}}\right).$$

From Lemma G.1, we know there exists a real $K_1 > 0$ such that, for all $m$, we have $\|\boldsymbol{D}_{\boldsymbol{\pi}}[\boldsymbol{I}_p - \gamma \boldsymbol{P}] \boldsymbol{\Sigma}_{\mathcal{S}}^T \boldsymbol{\Sigma}_{\mathcal{S}} \boldsymbol{U}_n \boldsymbol{Q}_m\| \leq K_1$. Therefore, we deduce that

$$\left\| \frac{1}{m} (\boldsymbol{U}_n - \gamma \boldsymbol{V}_n) \boldsymbol{Q}_{-j}^T \boldsymbol{U}_n^T \boldsymbol{\Sigma}_{\mathcal{S}}^{-jT} \boldsymbol{\Sigma}_{\mathcal{S}}^{-j} \boldsymbol{\Lambda}_{\boldsymbol{P}} \right\| = \mathcal{O}(1).$$

From Lemma K.2, we deduce that $\mathbb{E}\big[\|\boldsymbol{D}_{312}\|\big] = \mathcal{O}\left(\frac{1}{\sqrt{m}}\right)$. Therefore, we get

$$
\begin{aligned}
Z_{312} &= \frac{1}{nm} \sum_{j=1}^{N} \mathbb{E}\left[\boldsymbol{r}^T \frac{\boldsymbol{Q}_{-j}^T \boldsymbol{U}_n^T \boldsymbol{\sigma}_j \boldsymbol{\sigma}_j^T \boldsymbol{U}_n \boldsymbol{Q}_{-j} \frac{1}{m^2} \operatorname{Tr}\big((\boldsymbol{U}_n - \gamma \boldsymbol{V}_n)\boldsymbol{Q}_{-j}^T \boldsymbol{U}_n^T \boldsymbol{\Sigma}_{\mathcal{S}}^{-jT} \boldsymbol{\Sigma}_{\mathcal{S}}^{-j} \boldsymbol{\Lambda}_P \boldsymbol{\Phi}_{\mathcal{S}}\big)}{\left(1 + \frac{1}{m}\boldsymbol{\sigma}_j^T \boldsymbol{U}_n \boldsymbol{Q}_{-j}(\boldsymbol{U}_n - \gamma \boldsymbol{V}_n)^T \boldsymbol{\sigma}_j\right)^2} \boldsymbol{r}\right] \\
&\quad + \frac{1}{nm} \sum_{j=1}^{N} \mathbb{E}\left[\boldsymbol{r}^T \frac{\boldsymbol{Q}_{-j}^T \boldsymbol{U}_n^T \boldsymbol{\sigma}_j \boldsymbol{\sigma}_j^T \boldsymbol{U}_n \boldsymbol{Q}_{-j}[\boldsymbol{D}_{312}]_j}{\left(1 + \frac{1}{m}\boldsymbol{\sigma}_j^T \boldsymbol{U}_n \boldsymbol{Q}_{-j}(\boldsymbol{U}_n - \gamma \boldsymbol{V}_n)^T \boldsymbol{\sigma}_j\right)^2} \boldsymbol{r}\right] \\
&= \frac{1}{nm} \sum_{j=1}^{N} \mathbb{E}\left[\boldsymbol{r}^T \frac{\boldsymbol{Q}_{-j}^T \boldsymbol{U}_n^T \boldsymbol{\sigma}_j \boldsymbol{\sigma}_j^T \boldsymbol{U}_n \boldsymbol{Q}_{-j} \frac{1}{m^2} \operatorname{Tr}\big((\boldsymbol{U}_n - \gamma \boldsymbol{V}_n)\boldsymbol{Q}_{-j}^T \boldsymbol{U}_n^T \boldsymbol{\Sigma}_{\mathcal{S}}^{-jT} \boldsymbol{\Sigma}_{\mathcal{S}}^{-j} \boldsymbol{\Lambda}_P \boldsymbol{\Phi}_{\mathcal{S}}\big)}{\left(1 + \frac{1}{m}\boldsymbol{\sigma}_j^T \boldsymbol{U}_n \boldsymbol{Q}_{-j}(\boldsymbol{U}_n - \gamma \boldsymbol{V}_n)^T \boldsymbol{\sigma}_j\right)^2} \boldsymbol{r}\right] \\
&\quad + \frac{1}{nm} \mathbb{E}\left[\boldsymbol{r}^T \boldsymbol{Q}_m^T \boldsymbol{U}_n^T \boldsymbol{\Sigma}_{\mathcal{S}}^T \boldsymbol{D}_{312} \boldsymbol{\Sigma}_{\mathcal{S}} \boldsymbol{U}_n \boldsymbol{Q}_m \boldsymbol{r}\right] \\
&= \frac{1}{nm} \sum_{j=1}^{N} \mathbb{E}\left[\boldsymbol{r}^T \frac{\boldsymbol{Q}_{-j}^T \boldsymbol{U}_n^T \boldsymbol{\sigma}_j \boldsymbol{\sigma}_j^T \boldsymbol{U}_n \boldsymbol{Q}_{-j} \frac{1}{m^2} \operatorname{Tr}\big((\boldsymbol{U}_n - \gamma \boldsymbol{V}_n)\boldsymbol{Q}_{-j}^T \boldsymbol{U}_n^T \boldsymbol{\Sigma}_{\mathcal{S}}^{-jT} \boldsymbol{\Sigma}_{\mathcal{S}}^{-j} \boldsymbol{\Lambda}_P \boldsymbol{\Phi}_{\mathcal{S}}\big)}{\left(1 + \frac{1}{m}\boldsymbol{\sigma}_j^T \boldsymbol{U}_n \boldsymbol{Q}_{-j}(\boldsymbol{U}_n - \gamma \boldsymbol{V}_n)^T \boldsymbol{\sigma}_j\right)^2} \boldsymbol{r}\right] \\
&\quad + \mathcal{O}\left(\frac{1}{\sqrt{m}}\right),
\end{aligned}
\tag{73}
$$

where the last equality is obtained since $\mathbb{E}\big[\|\boldsymbol{D}_{312}\|\big] = \mathcal{O}\left(\frac{1}{\sqrt{m}}\right)$, and since we know there exists a real $K_{\boldsymbol{Q}}' > 0$ such that, for all $m$, we have

$$
\left\|\frac{1}{\sqrt{m}} \boldsymbol{\Sigma}_{\mathcal{S}} \boldsymbol{U}_n \boldsymbol{Q}_m\right\| \leq K_{\boldsymbol{Q}}'
$$

and

$$
\left\|\frac{1}{\sqrt{m}} \boldsymbol{Q}_m (\boldsymbol{U}_n - \gamma \boldsymbol{V}_n)^T \boldsymbol{\Sigma}_{\mathcal{S}}^T\right\| \leq 2 K_{\boldsymbol{Q}}'.
$$

from Lemma I.4. We replace $1 + \frac{1}{m}\boldsymbol{\sigma}_j^T \boldsymbol{U}_n \boldsymbol{Q}_{-j}(\boldsymbol{U}_n - \gamma \boldsymbol{V}_n)^T \boldsymbol{\sigma}_j$ by $1 + \delta$ in $Z_{312}$ as following

$$
\begin{aligned}
Z_{312} &= \frac{1}{nm^3} \frac{1}{(1+\delta)^2} \sum_{j=1}^{N} \mathbb{E}\left[\boldsymbol{r}^T \boldsymbol{Q}_{-j}^T \boldsymbol{U}_n^T \boldsymbol{\sigma}_j \boldsymbol{\sigma}_j^T \boldsymbol{U}_n \boldsymbol{Q}_{-j} \operatorname{Tr}\big((\boldsymbol{U}_n - \gamma \boldsymbol{V}_n)\boldsymbol{Q}_{-j}^T \boldsymbol{U}_n^T \boldsymbol{\Sigma}_{\mathcal{S}}^{-jT} \boldsymbol{\Sigma}_{\mathcal{S}}^{-j} \boldsymbol{\Lambda}_P \boldsymbol{\Phi}_{\mathcal{S}}\big) \boldsymbol{r}\right] \\
&\quad + \frac{1}{nm^3} \frac{1}{(1+\delta)^2} \sum_{j=1}^{N} \mathbb{E}\left[\boldsymbol{r}^T \frac{\boldsymbol{Q}_{-j}^T \boldsymbol{U}_n^T \boldsymbol{\sigma}_j \boldsymbol{D}_j' \boldsymbol{\sigma}_j^T \boldsymbol{U}_n \boldsymbol{Q}_{-j} \operatorname{Tr}\big((\boldsymbol{U}_n - \gamma \boldsymbol{V}_n)\boldsymbol{Q}_{-j}^T \boldsymbol{U}_n^T \boldsymbol{\Sigma}_{\mathcal{S}}^{-jT} \boldsymbol{\Sigma}_{\mathcal{S}}^{-j} \boldsymbol{\Lambda}_P \boldsymbol{\Phi}_{\mathcal{S}}\big)}{\left(1 + \frac{1}{m}\boldsymbol{\sigma}_j^T \boldsymbol{U}_n \boldsymbol{Q}_{-j}(\boldsymbol{U}_n - \gamma \boldsymbol{V}_n)^T \boldsymbol{\sigma}_j\right)^2} \boldsymbol{r}\right] \\
&\quad + \mathcal{O}\left(\frac{1}{\sqrt{m}}\right) \\
&= \underbrace{\frac{N}{nm^3} \frac{1}{(1+\delta)^2} \mathbb{E}\left[\boldsymbol{r}^T \boldsymbol{Q}_-^T \boldsymbol{U}_n^T \boldsymbol{\Phi}_{\mathcal{S}} \boldsymbol{U}_n \boldsymbol{Q}_- \boldsymbol{r} \operatorname{Tr}\big((\boldsymbol{U}_n - \gamma \boldsymbol{V}_n)\boldsymbol{Q}_-^T \boldsymbol{U}_n^T \boldsymbol{\Sigma}_{\mathcal{S}}^{-T} \boldsymbol{\Sigma}_{\mathcal{S}}^{-} \boldsymbol{\Lambda}_P \boldsymbol{\Phi}_{\mathcal{S}}\big)\right]}_{=Z_{3121}} \\
&\quad + \underbrace{\frac{1}{nm^3} \frac{1}{(1+\delta)^2} \mathbb{E}\left[\boldsymbol{r}^T \boldsymbol{Q}_m^T \boldsymbol{U}_n^T \boldsymbol{\Sigma}_{\mathcal{S}}^T \boldsymbol{D}' \boldsymbol{\Sigma}_{\mathcal{S}} \boldsymbol{U}_n \boldsymbol{Q}_m \boldsymbol{r} \operatorname{Tr}\big((\boldsymbol{U}_n - \gamma \boldsymbol{V}_n)\boldsymbol{Q}_-^T \boldsymbol{U}_n^T \boldsymbol{\Sigma}_{\mathcal{S}}^{-T} \boldsymbol{\Sigma}_{\mathcal{S}}^{-} \boldsymbol{\Lambda}_P \boldsymbol{\Phi}_{\mathcal{S}}\big)\right]}_{=Z_{3122}} \\
&\quad + \mathcal{O}\left(\frac{1}{\sqrt{m}}\right),
\end{aligned}
$$

where $\boldsymbol{D}' \in \mathbb{R}^{N \times N}$ is a diagonal matrix for which, for all $j \in [N]$, we have

$$
\boldsymbol{D}_j' = (1+\delta)^2 - \left(1 + \frac{1}{m}\boldsymbol{\sigma}_j^T \boldsymbol{U}_n \boldsymbol{Q}_{-j}(\boldsymbol{U}_n - \gamma \boldsymbol{V}_n)^T \boldsymbol{\sigma}_j\right)^2.
$$

With a similar proof than for equation 30, we can show $\frac{1}{m}\operatorname{Tr}(\boldsymbol{\Phi}_{\mathcal{S}}) = \frac{p}{m}\frac{1}{p}\operatorname{Tr}(\boldsymbol{\Phi}_{\mathcal{S}})$ is uniformly bounded under Assumption 3. Combining $|\operatorname{Tr}(\boldsymbol{AB})| \le \|\boldsymbol{A}\|\operatorname{Tr}(\boldsymbol{B})$ for non-negative definite matrix $\boldsymbol{B}$ and Lemma G.1, we have $\frac{1}{m^2}\operatorname{Tr}\big((\boldsymbol{U}_n - \gamma\boldsymbol{V}_n)\boldsymbol{Q}_-^T\boldsymbol{U}_n^T\boldsymbol{\Sigma}_{\mathcal{S}}^{-T}\boldsymbol{\Sigma}_{\mathcal{S}}^{-}\boldsymbol{\Lambda}_{\boldsymbol{P}}\boldsymbol{\Phi}_{\mathcal{S}}\big) = \mathcal{O}(1)$. From all these upper bounds, and since it can be shown that $\mathbb{E}\big[\|\boldsymbol{D}'\|\big] = \mathcal{O}\left(\frac{1}{\sqrt{m}}\right)$, we deduce the second term, $Z_{3122}$, vanishes and thus

$$Z_{312} = \frac{1}{nm^2}\frac{1}{1+\delta}\mathbb{E}\left[\boldsymbol{r}^T\boldsymbol{Q}_-^T\boldsymbol{\Psi}_1\boldsymbol{Q}_-\boldsymbol{r}\operatorname{Tr}\big((\boldsymbol{U}_n - \gamma\boldsymbol{V}_n)\boldsymbol{Q}_-^T\boldsymbol{U}_n^T\boldsymbol{\Sigma}_{\mathcal{S}}^{-T}\boldsymbol{\Sigma}_{\mathcal{S}}^{-}\boldsymbol{\Lambda}_{\boldsymbol{P}}\boldsymbol{\Phi}_{\mathcal{S}}\big)\right]$$
$$+ \mathcal{O}\left(\frac{1}{\sqrt{m}}\right).$$

Let $\boldsymbol{Q}_{-ij} \in \mathbb{R}^{n \times n}$ be the resolvent defined as

$$\boldsymbol{Q}_{-ij} = \left[\frac{1}{m}(\boldsymbol{U}_n - \gamma\boldsymbol{V}_n)^T\boldsymbol{\Sigma}_{\mathcal{S}}^{-ijT}\boldsymbol{\Sigma}_{\mathcal{S}}^{-ij}\boldsymbol{U}_n + \lambda\boldsymbol{I}_n\right]^{-1}, \tag{74}$$

where $\boldsymbol{\Sigma}_{\mathcal{S}}^{-ij} = \sigma(\boldsymbol{W}^{-ij}\boldsymbol{S}) \in \mathbb{R}^{(N-2)\times n}$ for which $\boldsymbol{W}^{-ij} \in \mathbb{R}^{(N-2)\times d}$ depicts the same matrix than the weight matrix $\boldsymbol{W}$ defined in equation 5 without the $i^{\text{th}}$ and $j^{\text{th}}$ row. Using the Sherman identity (Lemma L.4), the term $\frac{1}{m^2}\operatorname{Tr}\big((\boldsymbol{U}_n - \gamma\boldsymbol{V}_n)\boldsymbol{Q}_-^T\boldsymbol{U}_n^T\boldsymbol{\Sigma}_{\mathcal{S}}^{-T}\boldsymbol{\Sigma}_{\mathcal{S}}^{-}\boldsymbol{\Lambda}_{\boldsymbol{P}}\boldsymbol{\Phi}_{\mathcal{S}}\big)$ in $Z_{312}$ can be rewritten as

$$\frac{1}{m^2}\operatorname{Tr}\big((\boldsymbol{U}_n - \gamma\boldsymbol{V}_n)\boldsymbol{Q}_-^T\boldsymbol{U}_n^T\boldsymbol{\Sigma}_{\mathcal{S}}^{-T}\boldsymbol{\Sigma}_{\mathcal{S}}^{-}\boldsymbol{\Lambda}_{\boldsymbol{P}}\boldsymbol{\Phi}_{\mathcal{S}}\big)$$

$$= \frac{1}{m^2}\operatorname{Tr}\left(\sum_{i\ne j}(\boldsymbol{U}_n - \gamma\boldsymbol{V}_n)\boldsymbol{Q}_{-j}^T\boldsymbol{U}_n^T\boldsymbol{\sigma}_i\boldsymbol{\sigma}_i^T\boldsymbol{\Lambda}_{\boldsymbol{P}}\boldsymbol{\Phi}_{\mathcal{S}}\right)$$

$$= \frac{1}{m^2}\operatorname{Tr}\left(\sum_{i\ne j}\frac{(\boldsymbol{U}_n - \gamma\boldsymbol{V}_n)\boldsymbol{Q}_{-ij}^T\boldsymbol{U}_n^T\boldsymbol{\sigma}_i\boldsymbol{\sigma}_i^T\boldsymbol{\Lambda}_{\boldsymbol{P}}\boldsymbol{\Phi}_{\mathcal{S}}}{1 + \frac{1}{m}\boldsymbol{\sigma}_i^T\boldsymbol{U}_n\boldsymbol{Q}_{-ij}(\boldsymbol{U}_n - \gamma\boldsymbol{V}_n)^T\boldsymbol{\sigma}_i}\right)$$

$$= \frac{1}{m^2}\frac{1}{1+\delta}\operatorname{Tr}\left(\sum_{i\ne j}(\boldsymbol{U}_n - \gamma\boldsymbol{V}_n)\boldsymbol{Q}_{-ij}^T\boldsymbol{U}_n^T\boldsymbol{\sigma}_i\boldsymbol{\sigma}_i^T\boldsymbol{\Lambda}_{\boldsymbol{P}}\boldsymbol{\Phi}_{\mathcal{S}}\right)$$

$$+ \frac{1}{m^2}\frac{1}{1+\delta}\operatorname{Tr}\left(\sum_{i\ne j}\frac{(\boldsymbol{U}_n - \gamma\boldsymbol{V}_n)\boldsymbol{Q}_{-ij}^T\boldsymbol{U}_n^T\boldsymbol{\sigma}_i\boldsymbol{\sigma}_i^T\boldsymbol{\Lambda}_{\boldsymbol{P}}\boldsymbol{\Phi}_{\mathcal{S}}\left(\delta - \frac{1}{m}\boldsymbol{\sigma}_i^T\boldsymbol{U}_n\boldsymbol{Q}_{-ij}(\boldsymbol{U}_n - \gamma\boldsymbol{V}_n)^T\boldsymbol{\sigma}_i\right)}{1 + \frac{1}{m}\boldsymbol{\sigma}_i^T\boldsymbol{U}_n\boldsymbol{Q}_{-ij}(\boldsymbol{U}_n - \gamma\boldsymbol{V}_n)^T\boldsymbol{\sigma}_i}\right)$$

$$= \frac{N}{m^2}\frac{1}{1+\delta}\operatorname{Tr}\big((\boldsymbol{U}_n - \gamma\boldsymbol{V}_n)\boldsymbol{Q}_{--}^T\boldsymbol{U}_n^T\boldsymbol{\Phi}_{\mathcal{S}}\boldsymbol{\Lambda}_{\boldsymbol{P}}\boldsymbol{\Phi}_{\mathcal{S}}\big)$$

$$+ \frac{1}{m^2}\frac{1}{1+\delta}\operatorname{Tr}\big((\boldsymbol{U}_n - \gamma\boldsymbol{V}_n)\boldsymbol{Q}_{-j}^T\boldsymbol{U}_n^T\boldsymbol{\Sigma}_{\mathcal{S}}^{-jT}\boldsymbol{D}\boldsymbol{\Sigma}_{\mathcal{S}}^{-j}\boldsymbol{\Lambda}_{\boldsymbol{P}}\boldsymbol{\Phi}_{\mathcal{S}}\big),$$

where $\boldsymbol{D} \in \mathbb{R}^{N\times N}$ is a diagonal matrix for which, for all $i \in [N]$, we have

$$\boldsymbol{D}_i = \delta - \frac{1}{m}\boldsymbol{\sigma}_i^T\boldsymbol{U}_n\boldsymbol{Q}_{-ij}(\boldsymbol{U}_n - \gamma\boldsymbol{V}_n)^T\boldsymbol{\sigma}_i.$$

From the uniform boundness of $\frac{1}{m}\operatorname{Tr}(\boldsymbol{\Phi}_{\mathcal{S}}) = \frac{1}{K_r}\frac{1}{p}\operatorname{Tr}(\boldsymbol{\Phi}_{\mathcal{S}})$, from Lemma G.1, we have $\frac{1}{m^2}\operatorname{Tr}\big((\boldsymbol{U}_n - \gamma\boldsymbol{V}_n)\boldsymbol{Q}_-^T\boldsymbol{U}_n^T\boldsymbol{\Sigma}_{\mathcal{S}}^{-T}\boldsymbol{\Sigma}_{\mathcal{S}}^{-}\boldsymbol{\Lambda}_{\boldsymbol{P}}\boldsymbol{\Phi}_{\mathcal{S}}\big) = \mathcal{O}(1)$. Since the operator norm of $\mathbb{E}\big[\|\boldsymbol{D}\|\big]$ is of order $\mathcal{O}\left(\frac{1}{\sqrt{m}}\right)$, we deduce the second term vanishes, and thus

$$\frac{1}{m^2}\operatorname{Tr}\big((\boldsymbol{U}_n - \gamma\boldsymbol{V}_n)\boldsymbol{Q}_-^T\boldsymbol{U}_n^T\boldsymbol{\Sigma}_{\mathcal{S}}^{-T}\boldsymbol{\Sigma}_{\mathcal{S}}^{-}\boldsymbol{\Lambda}_{\boldsymbol{P}}\boldsymbol{\Phi}_{\mathcal{S}}\big)$$

$$= \frac{N}{m^2}\frac{1}{1+\delta}\operatorname{Tr}\big((\boldsymbol{U}_n - \gamma\boldsymbol{V}_n)\boldsymbol{Q}_{--}^T\boldsymbol{U}_n^T\boldsymbol{\Phi}_{\mathcal{S}}\boldsymbol{\Lambda}_{\boldsymbol{P}}\boldsymbol{\Phi}_{\mathcal{S}}\big) + \mathcal{O}\left(\frac{1}{\sqrt{m}}\right).$$

Applying Lemma F.4 and Lemma F.5, we deduce for $Z_{312}$ that

$$
\begin{aligned}
Z_{312} &= \frac{1}{n} \frac{\frac{1}{N} \operatorname{Tr}\big((\boldsymbol{U}_n - \gamma \boldsymbol{V}_n) \bar{\boldsymbol{Q}}_m^T \boldsymbol{U}_n^T \boldsymbol{\Psi}_{\mathcal{S}} \boldsymbol{\Lambda}_{\boldsymbol{P}} \boldsymbol{\Psi}_{\mathcal{S}}\big)}{1 - \frac{1}{N} \operatorname{Tr}\big(\boldsymbol{\Psi}_2 \bar{\boldsymbol{Q}}_m^T \boldsymbol{\Psi}_1 \bar{\boldsymbol{Q}}_m\big)} \|\bar{\boldsymbol{Q}}_m \boldsymbol{r}\|_{\boldsymbol{\Psi}_1}^2 + \mathcal{O}\left(\frac{1}{\sqrt{m}}\right) \\
&= \frac{1}{n} \frac{\frac{1}{N} \operatorname{Tr}\big(\boldsymbol{\Psi}_{\mathcal{S}} (\boldsymbol{U}_n - \gamma \boldsymbol{V}_n) \boldsymbol{\Theta}_{\mathcal{S}}^T \boldsymbol{\Lambda}_{\boldsymbol{P}}\big)}{1 - \frac{1}{N} \operatorname{Tr}\big(\boldsymbol{\Psi}_2 \bar{\boldsymbol{Q}}_m^T \boldsymbol{\Psi}_1 \bar{\boldsymbol{Q}}_m\big)} \|\bar{\boldsymbol{Q}}_m \boldsymbol{r}\|_{\boldsymbol{\Psi}_1}^2 + \mathcal{O}\left(\frac{1}{\sqrt{m}}\right).
\end{aligned}
$$

Now, we want to find an asymptotic equivalent for $Z_{311}$. We replace $1 + \frac{1}{m} \boldsymbol{\sigma}_j^T \boldsymbol{U}_n \boldsymbol{Q}_{-j} (\boldsymbol{U}_n - \gamma \boldsymbol{V}_n)^T \boldsymbol{\sigma}_j$ by $1 + \delta$ in $Z_{311}$ as following

$$
\begin{aligned}
Z_{311} &= \frac{1}{nm^2} \sum_{i=1}^{N} \sum_{j \neq i} \mathbb{E}\left[\boldsymbol{r}^T \frac{\boldsymbol{Q}_{-j}^T \boldsymbol{U}_n^T \boldsymbol{\sigma}_i \boldsymbol{\sigma}_i^T \boldsymbol{\Lambda}_{\boldsymbol{P}} \boldsymbol{\sigma}_j \boldsymbol{\sigma}_j^T \boldsymbol{U}_n \boldsymbol{Q}_{-j}}{1 + \frac{1}{m} \boldsymbol{\sigma}_j^T \boldsymbol{U}_n \boldsymbol{Q}_{-j} (\boldsymbol{U}_n - \gamma \boldsymbol{V}_n)^T \boldsymbol{\sigma}_j} \boldsymbol{r}\right] \\
&= \frac{1}{nm^2} \sum_{j=1}^{N} \mathbb{E}\left[\boldsymbol{r}^T \frac{\boldsymbol{Q}_{-j}^T \boldsymbol{U}_n^T \boldsymbol{\Sigma}_{\mathcal{S}}^{-jT} \boldsymbol{\Sigma}_{\mathcal{S}}^{-j} \boldsymbol{\Lambda}_{\boldsymbol{P}} \boldsymbol{\sigma}_j \boldsymbol{\sigma}_j^T \boldsymbol{U}_n \boldsymbol{Q}_{-j}}{1 + \frac{1}{m} \boldsymbol{\sigma}_j^T \boldsymbol{U}_n \boldsymbol{Q}_{-j} (\boldsymbol{U}_n - \gamma \boldsymbol{V}_n)^T \boldsymbol{\sigma}_j} \boldsymbol{r}\right] \\
&= \underbrace{\frac{1}{nm^2} \frac{1}{1 + \delta} \sum_{j=1}^{N} \mathbb{E}\left[\boldsymbol{r}^T \boldsymbol{Q}_{-j}^T \boldsymbol{U}_n^T \boldsymbol{\Sigma}_{\mathcal{S}}^{-jT} \boldsymbol{\Sigma}_{\mathcal{S}}^{-j} \boldsymbol{\Lambda}_{\boldsymbol{P}} \boldsymbol{\sigma}_j \boldsymbol{\sigma}_j^T \boldsymbol{U}_n \boldsymbol{Q}_{-j} \boldsymbol{r}\right]}_{=Z_{3111}} \\
&\quad + \underbrace{\frac{1}{nm^2} \frac{1}{1 + \delta} \sum_{j=1}^{N} \mathbb{E}\left[\boldsymbol{r}^T \frac{\boldsymbol{Q}_{-j}^T \boldsymbol{U}_n^T \boldsymbol{\Sigma}_{\mathcal{S}}^{-jT} \boldsymbol{\Sigma}_{\mathcal{S}}^{-j} \boldsymbol{\Lambda}_{\boldsymbol{P}} \boldsymbol{\sigma}_j \boldsymbol{\sigma}_j^T \boldsymbol{U}_n \boldsymbol{Q}_{-j} \boldsymbol{D}_j}{1 + \frac{1}{m} \boldsymbol{\sigma}_j^T \boldsymbol{U}_n \boldsymbol{Q}_{-j} (\boldsymbol{U}_n - \gamma \boldsymbol{V}_n)^T \boldsymbol{\sigma}_j} \boldsymbol{r}\right]}_{=Z_{3112}},
\end{aligned}
$$

where $\boldsymbol{D} \in \mathbb{R}^{N \times N}$ is a diagonal matrix for which, for all $j \in [N]$, we have

$$
\boldsymbol{D}_j = \delta - \frac{1}{m} \boldsymbol{\sigma}_j^T \boldsymbol{U}_n \boldsymbol{Q}_{-j} (\boldsymbol{U}_n - \gamma \boldsymbol{V}_n)^T \boldsymbol{\sigma}_j.
$$

We observe that

$$
\begin{aligned}
\boldsymbol{Q}_{-j}^T \boldsymbol{U}_n^T \frac{\boldsymbol{\Sigma}_{\mathcal{S}}^{-jT} \boldsymbol{\Sigma}_{\mathcal{S}}^{-j}}{m} &= \boldsymbol{Q}_{-j}^T \boldsymbol{U}_n^T \frac{\boldsymbol{\Sigma}_{\mathcal{S}}^T \boldsymbol{\Sigma}_{\mathcal{S}}}{m} - \boldsymbol{Q}_{-j}^T \boldsymbol{U}_n^T \frac{\boldsymbol{\sigma}_j \boldsymbol{\sigma}_j^T}{m} \\
&= \boldsymbol{Q}_m^T \boldsymbol{U}_n^T \frac{\boldsymbol{\Sigma}_{\mathcal{S}}^T \boldsymbol{\Sigma}_{\mathcal{S}}}{m} + \frac{\frac{1}{m} \boldsymbol{Q}_{-j}^T \boldsymbol{U}_n^T \boldsymbol{\sigma}_j \boldsymbol{\sigma}_j^T (\boldsymbol{U}_n - \gamma \boldsymbol{V}_n) \boldsymbol{Q}_{-j}^T}{1 + \frac{1}{m} \boldsymbol{\sigma}_j^T \boldsymbol{U}_n^T \boldsymbol{Q}_{-j} (\boldsymbol{U}_n - \gamma \boldsymbol{V}_n)^T \boldsymbol{\sigma}_j} \boldsymbol{U}_n^T \frac{\boldsymbol{\Sigma}_{\mathcal{S}}^T \boldsymbol{\Sigma}_{\mathcal{S}}}{m} - \boldsymbol{Q}_{-j}^T \boldsymbol{U}_n^T \frac{\boldsymbol{\sigma}_j \boldsymbol{\sigma}_j^T}{m} \\
&= \boldsymbol{Q}_m^T \boldsymbol{U}_n^T \frac{\boldsymbol{\Sigma}_{\mathcal{S}}^T \boldsymbol{\Sigma}_{\mathcal{S}}}{m} \\
&\quad + \boldsymbol{Q}_m^T \boldsymbol{U}_n^T \boldsymbol{\sigma}_j \boldsymbol{\sigma}_j^T (\boldsymbol{U}_n - \gamma \boldsymbol{V}_n) \boldsymbol{Q}_m^T \left(1 + \frac{1}{m} \boldsymbol{\sigma}_j^T \boldsymbol{U}_n^T \boldsymbol{Q}_{-j} (\boldsymbol{U}_n - \gamma \boldsymbol{V}_n)^T \boldsymbol{\sigma}_j\right) \boldsymbol{U}_n^T \frac{\boldsymbol{\Sigma}_{\mathcal{S}}^T \boldsymbol{\Sigma}_{\mathcal{S}}}{m} \\
&\quad - \boldsymbol{Q}_{-j}^T \boldsymbol{U}_n^T \frac{\boldsymbol{\sigma}_j \boldsymbol{\sigma}_j^T}{m} \\
&= \boldsymbol{Q}_m^T \boldsymbol{U}_n^T \frac{\boldsymbol{\Sigma}_{\mathcal{S}}^T \boldsymbol{\Sigma}_{\mathcal{S}}}{m} + \frac{\boldsymbol{Q}_m^T \boldsymbol{U}_n^T \boldsymbol{\sigma}_j \boldsymbol{\sigma}_j^T (\boldsymbol{U}_n - \gamma \boldsymbol{V}_n) \boldsymbol{Q}_m^T}{1 - \frac{1}{m} \boldsymbol{\sigma}_j^T \boldsymbol{U}_n^T \boldsymbol{Q}_m (\boldsymbol{U}_n - \gamma \boldsymbol{V}_n)^T \boldsymbol{\sigma}_j} \boldsymbol{U}_n^T \frac{\boldsymbol{\Sigma}_{\mathcal{S}}^T \boldsymbol{\Sigma}_{\mathcal{S}}}{m} - \boldsymbol{Q}_{-j}^T \boldsymbol{U}_n^T \frac{\boldsymbol{\sigma}_j \boldsymbol{\sigma}_j^T}{m}.
\end{aligned}
$$

From above, we expand $Z_{3112}$ as

$$Z_{3112}$$

$$= \frac{1}{nm^2} \frac{1}{1+\delta} \sum_{j=1}^{N} \mathbb{E}\left[ r^T \frac{Q_{-j}^T U_n^T \Sigma_{\mathcal{S}}^{-jT} \Sigma_{\mathcal{S}}^{-j} \Lambda_P \sigma_j \sigma_j^T U_n Q_{-j} D_j}{1 + \frac{1}{m}\sigma_j^T U_n Q_{-j}(U_n - \gamma V_n)^T \sigma_j} r \right]$$

$$= \underbrace{\frac{1}{nm^2} \frac{1}{1+\delta} \mathbb{E}\left[ r^T Q_m^T U_n^T \Sigma_{\mathcal{S}}^T \Sigma_{\mathcal{S}} \Lambda_P \Sigma_{\mathcal{S}}^T D \Sigma_{\mathcal{S}} U_n Q_m r \right]}_{=Z_{31121}}$$

$$+ \underbrace{\sum_{j=1}^{N} \mathbb{E}\left[ r^T \frac{Q_m^T U_n^T \sigma_j \sigma_j^T (U_n - \gamma V_n) Q_m^T U_n^T \Sigma_{\mathcal{S}}^T \Sigma_{\mathcal{S}} \Lambda_P \sigma_j \sigma_j^T U_n Q_m D_j}{nm^2(1+\delta)\left(1 - \frac{1}{m}\sigma_j^T U_n^T Q_m(U_n - \gamma V_n)^T \sigma_j\right)} r \right]}_{=Z_{31122}}$$

$$- \underbrace{\frac{1}{nm^2} \frac{1}{1+\delta} \sum_{j=1}^{N} \mathbb{E}\left[ r^T Q_{-j}^T U_n^T \sigma_j \sigma_j^T \Lambda_P \sigma_j \sigma_j^T U_n Q_m D_j \left(1 + \frac{1}{m}\sigma_j^T U_n Q_{-j}(U_n - \gamma V_n)^T \sigma_j\right) r \right]}_{=Z_{31123}}.$$

We have $Z_{31121} = \mathcal{O}\left(\frac{1}{\sqrt{m}}\right)$ since $\mathbb{E}\left[\|D\|\right] = \mathcal{O}\left(\frac{1}{\sqrt{m}}\right)$ and from Lemma G.1. Subsequently, we rewrite $Z_{31122}$ as

$$Z_{31122} = \frac{1}{nm} \frac{1}{1+\delta} \mathbb{E}\left[ r^T Q_m^T U_n^T \Sigma_{\mathcal{S}}^T D_{31122} \Sigma_{\mathcal{S}} U_n Q_m r \right],$$

with $D_{31122} \in \mathbb{R}^{N \times N}$ a diagonal matrix for which, for all $j \in [N]$, we have

$$[D_{31122}]_j = \frac{1}{m} \frac{D_j \sigma_j^T (U_n - \gamma V_n) Q_m^T U_n^T \Sigma_{\mathcal{S}}^T \Sigma_{\mathcal{S}} \Lambda_P \sigma_j}{1 - \frac{1}{m}\sigma_j^T U_n^T Q_m(U_n - \gamma V_n)^T \sigma_j}.$$

It can be shown that $\mathbb{E}\left[\|D_{31122}\|\right] = \mathcal{O}\left(\frac{1}{\sqrt{m}}\right)$, and we can deduce $Z_{31122} = \mathcal{O}\left(\frac{1}{\sqrt{m}}\right)$. Similarly, we have $Z_{31123} = \mathcal{O}\left(\frac{1}{\sqrt{m}}\right)$. $Z_{3112}$ vanishes, and thus

$$Z_{311} = Z_{3111} + \mathcal{O}\left(\frac{1}{\sqrt{m}}\right)$$

$$= \frac{1}{nm^2} \frac{1}{1+\delta} \sum_{j=1}^{N} \mathbb{E}\left[ r^T Q_{-j}^T U_n^T \Sigma_{\mathcal{S}}^{-jT} \Sigma_{\mathcal{S}}^{-j} \Lambda_P \sigma_j \sigma_j^T U_n Q_{-j} r \right] + \mathcal{O}\left(\frac{1}{\sqrt{m}}\right).$$

It remains to handle $Z_{3111}$ for which we have from the Sherman identity (Lemma L.4),

$$Z_{3111} = \frac{1}{nm^2} \frac{1}{1+\delta} \sum_{j=1}^{N} \mathbb{E}\left[ r^T Q_{-j}^T U_n^T \Sigma_{\mathcal{S}}^{-jT} \Sigma_{\mathcal{S}}^{-j} \Lambda_P \sigma_j \sigma_j^T U_n Q_{-j} r \right]$$

$$= \frac{1}{nm^2} \frac{1}{1+\delta} \sum_{j=1}^{N} \sum_{i \neq j} \mathbb{E}\left[ r^T Q_{-j}^T U_n^T \sigma_i \sigma_i^T \Lambda_P \Phi_{\mathcal{S}} U_n Q_{-j} r \right]$$

$$= \underbrace{\frac{1}{nm^2} \frac{1}{1+\delta} \sum_{j=1}^{N} \sum_{i \neq j} \mathbb{E}\left[ r^T \frac{Q_{-ij}^T U_n^T \sigma_i \sigma_i^T \Lambda_P \Phi_{\mathcal{S}} U_n Q_{-ij}}{1 + \frac{1}{m}\sigma_i^T U_n Q_{-ij}(U_n - \gamma V_n)^T \sigma_i} r \right]}_{=Z_{31111}}$$

$$- \underbrace{\frac{1}{nm^3} \frac{1}{1+\delta} \sum_{j=1}^{N} \sum_{i \neq j} \mathbb{E}\left[ r^T \frac{Q_{-ij}^T U_n^T \sigma_i \sigma_i^T \Lambda_P \Phi_{\mathcal{S}} U_n Q_{-ij}(U_n - \gamma V_n)^T \sigma_i \sigma_i^T U_n Q_{-ij}}{\left(1 + \frac{1}{m}\sigma_i^T U_n Q_{-ij}(U_n - \gamma V_n)^T \sigma_i\right)^2} r \right]}_{=Z_{31112}}.$$

Again, we replace $1 + \frac{1}{m}\boldsymbol{\sigma}_i^T \boldsymbol{U}_n \boldsymbol{Q}_{-ij}(\boldsymbol{U}_n - \gamma \boldsymbol{V}_n)^T \boldsymbol{\sigma}_i$ by $1 + \delta$ in $Z_{31111}$ as following

$$
\begin{aligned}
Z_{31111} &= \frac{1}{nm^2}\frac{1}{1+\delta}\sum_{j=1}^{N}\sum_{i\neq j}\mathbb{E}\left[\boldsymbol{r}^T \frac{\boldsymbol{Q}_{-ij}^T \boldsymbol{U}_n^T \boldsymbol{\sigma}_i \boldsymbol{\sigma}_i^T \boldsymbol{\Lambda_P}\boldsymbol{\Phi_S}\boldsymbol{U}_n \boldsymbol{Q}_{-ij}}{1 + \frac{1}{m}\boldsymbol{\sigma}_i^T \boldsymbol{U}_n \boldsymbol{Q}_{-ij}(\boldsymbol{U}_n - \gamma\boldsymbol{V}_n)^T\boldsymbol{\sigma}_i}\boldsymbol{r}\right] \\
&= \frac{1}{nm^2}\frac{1}{(1+\delta)^2}\sum_{j=1}^{N}\sum_{i\neq j}\mathbb{E}\left[\boldsymbol{r}^T\boldsymbol{Q}_{-ij}^T\boldsymbol{U}_n^T\boldsymbol{\sigma}_i\boldsymbol{\sigma}_i^T\boldsymbol{\Lambda_P}\boldsymbol{\Phi_S}\boldsymbol{U}_n\boldsymbol{Q}_{-ij}\boldsymbol{r}\right] \\
&\quad + \frac{1}{nm^2}\frac{1}{(1+\delta)^2}\sum_{j=1}^{N}\sum_{i\neq j}\mathbb{E}\left[\boldsymbol{r}^T\frac{\boldsymbol{Q}_{-ij}^T\boldsymbol{U}_n^T\boldsymbol{\sigma}_i\boldsymbol{\sigma}_i^T\boldsymbol{\Lambda_P}\boldsymbol{\Phi_S}\boldsymbol{U}_n\boldsymbol{Q}_{-ij}\boldsymbol{D}_i}{1 + \frac{1}{m}\boldsymbol{\sigma}_i^T\boldsymbol{U}_n\boldsymbol{Q}_{-ij}(\boldsymbol{U}_n - \gamma\boldsymbol{V}_n)^T\boldsymbol{\sigma}_i}\boldsymbol{r}\right] \\
&= \frac{1}{n}\frac{N^2}{m^2}\frac{1}{(1+\delta)^2}\mathbb{E}\left[\boldsymbol{r}^T\boldsymbol{Q}_{--}^T\boldsymbol{U}_n^T\boldsymbol{\Phi_S}\boldsymbol{\Lambda_P}\boldsymbol{\Phi_S}\boldsymbol{U}_n\boldsymbol{Q}_{--}\boldsymbol{r}\right] \\
&\quad + \frac{1}{nm^2}\frac{1}{(1+\delta)^2}\sum_{j=1}^{N}\sum_{i\neq j}\mathbb{E}\left[\boldsymbol{r}^T\boldsymbol{Q}_{-j}^T\boldsymbol{U}_n^T\boldsymbol{\sigma}_i\boldsymbol{\sigma}_i^T\boldsymbol{\Lambda_P}\boldsymbol{\Phi_S}\boldsymbol{U}_n\boldsymbol{Q}_{-j}\boldsymbol{D}_i\boldsymbol{r}\right] \\
&\quad + \frac{1}{nm^3}\frac{1}{(1+\delta)^2}\sum_{j=1}^{N}\sum_{i\neq j}\mathbb{E}\left[\boldsymbol{r}^T\frac{\boldsymbol{Q}_{-j}^T\boldsymbol{U}_n^T\boldsymbol{\sigma}_i\boldsymbol{\sigma}_i^T\boldsymbol{\Lambda_P}\boldsymbol{\Phi_S}\boldsymbol{U}_n\boldsymbol{Q}_{-j}(\boldsymbol{U}_n - \gamma\boldsymbol{V}_n)^T\boldsymbol{\sigma}_i\boldsymbol{\sigma}_i^T\boldsymbol{U}_n\boldsymbol{Q}_{-j}}{1 - \frac{1}{m}\boldsymbol{\sigma}_i^T\boldsymbol{U}_n\boldsymbol{Q}_{-j}(\boldsymbol{U}_n - \gamma\boldsymbol{V}_n)^T\boldsymbol{\sigma}_i}\boldsymbol{D}_i\boldsymbol{r}\right] \\
&= \frac{1}{n}\mathbb{E}\left[\boldsymbol{r}^T\boldsymbol{Q}_{--}^T\boldsymbol{U}_n^T\boldsymbol{\Psi_S}\boldsymbol{\Lambda_P}\boldsymbol{\Psi_S}\boldsymbol{U}_n\boldsymbol{Q}_{--}\boldsymbol{r}\right] \\
&\quad + \frac{1}{nm^2}\frac{1}{(1+\delta)^2}\sum_{j=1}^{N}\mathbb{E}\left[\boldsymbol{r}^T\boldsymbol{Q}_{-j}^T\boldsymbol{U}_n^T\boldsymbol{\Sigma}_S^{-jT}\boldsymbol{D}\boldsymbol{\Sigma}_S^{-j}\boldsymbol{\Lambda_P}\boldsymbol{\Phi_S}\boldsymbol{U}_n\boldsymbol{Q}_{-j}\boldsymbol{r}\right] \\
&\quad + \frac{1}{nm}\frac{1}{(1+\delta)^2}\sum_{j=1}^{N}\mathbb{E}\left[\boldsymbol{r}^T\boldsymbol{Q}_{-j}^T\boldsymbol{U}_n^T\boldsymbol{\Sigma}_S^{-jT}\boldsymbol{D}_{31111}\boldsymbol{\Sigma}_S^{-j}\boldsymbol{U}_n\boldsymbol{Q}_{-j}\boldsymbol{r}\right],
\end{aligned}
$$

where $\boldsymbol{D}_{31111} \in \mathbb{R}^{N\times N}$ is a diagonal matrix for which, for all $i \in [N]$, we have

$$
[\boldsymbol{D}_{31111}]_i = \frac{1}{m}\frac{\boldsymbol{\sigma}_i^T\boldsymbol{\Lambda_P}\boldsymbol{\Phi_S}\boldsymbol{U}_n\boldsymbol{Q}_{-j}(\boldsymbol{U}_n - \gamma\boldsymbol{V}_n)^T\boldsymbol{\sigma}_i\left(\delta - \frac{1}{m}\boldsymbol{\sigma}_i^T\boldsymbol{U}_n\boldsymbol{Q}_{-ij}(\boldsymbol{U}_n - \gamma\boldsymbol{V}_n)^T\boldsymbol{\sigma}_i\right)}{1 - \frac{1}{m}\boldsymbol{\sigma}_i^T\boldsymbol{U}_n\boldsymbol{Q}_{-j}(\boldsymbol{U}_n - \gamma\boldsymbol{V}_n)^T\boldsymbol{\sigma}_i},
$$

and $\boldsymbol{Q}_{--}$ is a resolvent with the same law than $\boldsymbol{Q}_{-ij}$. With similar arguments that before, we can show that $\mathbb{E}\big[\|\boldsymbol{D}\|\big]$ and $\mathbb{E}\big[\|\boldsymbol{D}_{31111}\|\big]$ are of order $\mathcal{O}\left(\frac{1}{\sqrt{m}}\right)$, and therefore

$$
Z_{31111} = \frac{1}{n}\mathbb{E}\left[\boldsymbol{r}^T\boldsymbol{Q}_{--}^T\boldsymbol{U}_n^T\boldsymbol{\Psi_S}\boldsymbol{\Lambda_P}\boldsymbol{\Psi_S}\boldsymbol{U}_n\boldsymbol{Q}_{--}\boldsymbol{r}\right] + \mathcal{O}\left(\frac{1}{\sqrt{m}}\right).
$$

By extending Lemma F.2 to the matrix $\boldsymbol{\Lambda'_P} = \boldsymbol{U}_n^T\boldsymbol{\Psi_S}\boldsymbol{\Lambda_P}\boldsymbol{\Psi_S}\boldsymbol{U}_n$, and from Lemma F.4 we obtain

$$
\begin{aligned}
Z_{31111} &= \frac{1}{n}\boldsymbol{r}^T\bar{\boldsymbol{Q}}_m^T\boldsymbol{U}_n^T\boldsymbol{\Psi_S}\boldsymbol{\Lambda_P}\boldsymbol{\Psi_S}\boldsymbol{U}_n\bar{\boldsymbol{Q}}_m\boldsymbol{r} \\
&\quad + \frac{1}{n}\frac{\frac{1}{N}\text{Tr}\left(\boldsymbol{\Lambda_P}\boldsymbol{\Theta_S}\boldsymbol{\Psi_2}\boldsymbol{\Theta_S^T}\right)}{1 - \frac{1}{N}\text{Tr}\left(\boldsymbol{\Psi_2}\bar{\boldsymbol{Q}}_m^T\boldsymbol{\Psi_1}\bar{\boldsymbol{Q}}_m\right)}\|\bar{\boldsymbol{Q}}_m\boldsymbol{r}\|_{\boldsymbol{\Psi_1}}^2 \\
&\quad + \mathcal{O}\left(\frac{1}{\sqrt{m}}\right).
\end{aligned}
$$

Following the same reasoning for $Z_{31112}$, and from Lemma K.2, we have

$$Z_{31112} = \frac{1}{nm^3} \frac{1}{1+\delta} \sum_{j=1}^{N} \sum_{i \neq j} \mathbb{E}\left[ r^T \frac{Q_{-ij}^T U_n^T \sigma_i \sigma_i^T \Lambda_P \Phi_S U_n Q_{-ij}(U_n - \gamma V_n)^T \sigma_i \sigma_i^T U_n Q_{-ij}}{\left(1 + \frac{1}{m}\sigma_i^T U_n Q_{-ij}(U_n - \gamma V_n)^T \sigma_i\right)^2} r \right]$$

$$= \frac{1}{nm^3} \frac{1}{(1+\delta)^3} \sum_{j=1}^{N} \sum_{i \neq j} \mathbb{E}\left[ r^T Q_{-ij}^T U_n^T \sigma_i \sigma_i^T \Lambda_P \Phi_S U_n Q_{-ij}(U_n - \gamma V_n)^T \sigma_i \sigma_i^T U_n Q_{-ij} r \right]$$

$$+ \mathcal{O}\left(\frac{1}{\sqrt{m}}\right)$$

$$= \frac{1}{nm^2} \frac{1}{(1+\delta)^3} \sum_{j=1}^{N} \sum_{i \neq j} \mathbb{E}\left[ r^T Q_{-ij}^T U_n^T \sigma_i \sigma_i^T U_n Q_{-ij} r \left( \frac{1}{m}\sigma_i^T \Lambda_P \Phi_S U_n Q_{-ij}(U_n - \gamma V_n)^T \sigma_i \right) \right]$$

$$+ \mathcal{O}\left(\frac{1}{\sqrt{m}}\right)$$

$$= \frac{1}{nm^2} \frac{1}{(1+\delta)^3} \sum_{j=1}^{N} \sum_{i \neq j} \mathbb{E}\left[ r^T Q_{-ij}^T U_n^T \sigma_i \sigma_i^T U_n Q_{-ij} r \frac{1}{m} \mathrm{Tr}(\Lambda_P \Phi_S U_n Q_{-ij}(U_n - \gamma V_n)^T \Phi_S) \right]$$

$$+ \mathcal{O}\left(\frac{1}{\sqrt{m}}\right)$$

$$= \frac{1}{n} \frac{1}{N} \mathbb{E}\left[ r^T Q_{--}^T \Psi_1 Q_{--} r \, \mathrm{Tr}(\Lambda_P \Psi_S U_n Q_{--}(U_n - \gamma V_n)^T \Psi_S) \right]$$

$$+ \mathcal{O}\left(\frac{1}{\sqrt{m}}\right),$$

where the last equality is obtained with a similar reasoning than for equation 73. From Lemma F.4 and Lemma F.5, we have

$$Z_{31112} = \frac{1}{n} \frac{\frac{1}{N} \mathrm{Tr}(\Lambda_P \Theta_S (U_n - \gamma V_n)^T \Psi_S)}{1 - \frac{1}{N} \mathrm{Tr}\left( \Psi_2 \bar{Q}_m^T \Psi_1 \bar{Q}_m \right)} \|\bar{Q}_m r\|_{\Psi_1}^2 + \mathcal{O}\left(\frac{1}{\sqrt{m}}\right).$$

We conclude for $Z_{311}$ that

$$Z_{311} = \frac{1}{n} r^T \bar{Q}_m^T U_n^T \Psi_S \Lambda_P \Psi_S U_n \bar{Q}_m r$$

$$+ \frac{1}{n} \frac{\frac{1}{N} \mathrm{Tr}\left( \Lambda_P \left[ \Theta_S \Psi_2 \Theta_S^T - \Theta_S (U_n - \gamma V_n)^T \Psi_S \right] \right)}{1 - \frac{1}{N} \mathrm{Tr}\left( \Psi_2 \bar{Q}_m^T \Psi_1 \bar{Q}_m \right)} \|\bar{Q}_m r\|_{\Psi_1}^2$$

$$+ \mathcal{O}\left(\frac{1}{\sqrt{m}}\right),$$

and for $Z_{31}$ that

$$Z_{31} = \frac{1}{n} r^T \bar{Q}_m^T U_n^T \Psi_S \Lambda_P \Psi_S U_n \bar{Q}_m r$$

$$+ \frac{1}{n} \frac{\frac{1}{N} \mathrm{Tr}\left( \Lambda_P \left[ \Theta_S \Psi_2 \Theta_S^T - 2\Theta_S (U_n - \gamma V_n)^T \Psi_S \right] \right)}{1 - \frac{1}{N} \mathrm{Tr}\left( \Psi_2 \bar{Q}_m^T \Psi_1 \bar{Q}_m \right)} \|\bar{Q}_m r\|_{\Psi_1}^2$$

$$+ \mathcal{O}\left(\frac{1}{\sqrt{m}}\right).$$

$\square$

**Lemma G.6.** *Under Assumptions 1, 2 and 3, let $Z_{32} \in \mathbb{R}$ defined in equation 72 as*

$$Z_{32} = \sum_{i=1}^{N} \mathbb{E}\left[ \frac{1}{nm^2} r^T \frac{Q_{-i}^T U_n^T \sigma_i \sigma_i^T \Lambda_P \sigma_i \sigma_i^T U_n Q_{-i}}{\left(1 + \frac{1}{m}\sigma_i^T U_n Q_{-i}(U_n - \gamma V_n)^T \sigma_i\right)^2} r \right].$$

*Then*

$$\left| Z_{32} - \frac{1}{n} \frac{\frac{1}{N} \operatorname{Tr}(\mathbf{\Lambda}_P \mathbf{\Psi}_S)}{1 - \frac{1}{N} \operatorname{Tr}\left(\mathbf{\Psi}_2 \bar{\mathbf{Q}}_m^T \mathbf{\Psi}_1 \bar{\mathbf{Q}}_m\right)} \|\bar{\mathbf{Q}}_m \mathbf{r}\|_{\mathbf{\Psi}_1}^2 \right| = \mathcal{O}\left(\frac{1}{\sqrt{m}}\right),$$

*where $\bar{\mathbf{Q}}_m$ is the deterministic resolent defined in equation 15, $\mathbf{\Psi}_1, \mathbf{\Psi}_2 \in \mathbb{R}^{n \times n}$ are defined in equation 19, and $\mathbf{\Psi}_S \in \mathbb{R}^{p \times p}$ is defined in equation 21.*

*Proof.* We decompose $Z_{32}$ as

$$
\begin{aligned}
Z_{32} &= \frac{1}{nm^2} \sum_{i=1}^N \mathbb{E}\left[\mathbf{r}^T \frac{\mathbf{Q}_{-i}^T \mathbf{U}_n^T \boldsymbol{\sigma}_i \boldsymbol{\sigma}_i^T \mathbf{\Lambda}_P \boldsymbol{\sigma}_i \boldsymbol{\sigma}_i^T \mathbf{U}_n \mathbf{Q}_{-i}}{\left(1 + \frac{1}{m} \boldsymbol{\sigma}_i^T \mathbf{U}_n \mathbf{Q}_{-i}(\mathbf{U}_n - \gamma \mathbf{V}_n)^T \boldsymbol{\sigma}_i\right)^2} \mathbf{r}\right] \\
&= \frac{1}{nm} \sum_{i=1}^N \mathbb{E}\left[\mathbf{r}^T \frac{\mathbf{Q}_{-i}^T \mathbf{U}_n^T \boldsymbol{\sigma}_i \boldsymbol{\sigma}_i^T \mathbf{U}_n \mathbf{Q}_{-i} \frac{1}{m} \operatorname{Tr}\left(\mathbf{\Phi}_S \mathbf{\Lambda}_P\right)}{\left(1 + \frac{1}{m} \boldsymbol{\sigma}_i^T \mathbf{U}_n \mathbf{Q}_{-i}(\mathbf{U}_n - \gamma \mathbf{V}_n)^T \boldsymbol{\sigma}_i\right)^2} \mathbf{r}\right] \\
&\quad + \frac{1}{nm} \sum_{i=1}^N \mathbb{E}\left[\mathbf{r}^T \frac{\mathbf{Q}_{-i}^T \mathbf{U}_n^T \boldsymbol{\sigma}_i \boldsymbol{\sigma}_i^T \mathbf{U}_n \mathbf{Q}_{-i} \frac{1}{m}\left(\boldsymbol{\sigma}_i^T \mathbf{\Lambda}_P \boldsymbol{\sigma}_i - \operatorname{Tr}\left(\mathbf{\Phi}_S \mathbf{\Lambda}_P\right)\right)}{\left(1 + \frac{1}{m} \boldsymbol{\sigma}_i^T \mathbf{U}_n \mathbf{Q}_{-i}(\mathbf{U}_n - \gamma \mathbf{V}_n)^T \boldsymbol{\sigma}_i\right)^2} \mathbf{r}\right] \\
&= \frac{1}{n} \frac{\operatorname{Tr}\left(\mathbf{\Phi}_S \mathbf{\Lambda}_P\right)}{m} \mathbf{r}^T \underbrace{\sum_{i=1}^N \mathbb{E}\left[\frac{1}{m} \frac{\mathbf{Q}_{-i}^T \mathbf{U}_n^T \boldsymbol{\sigma}_i \boldsymbol{\sigma}_i^T \mathbf{U}_n \mathbf{Q}_{-i}}{\left(1 + \frac{1}{m} \boldsymbol{\sigma}_i^T \mathbf{U}_n \mathbf{Q}_{-i}(\mathbf{U}_n - \gamma \mathbf{V}_n)^T \boldsymbol{\sigma}_i\right)^2}\right]}_{=\mathbf{Z}_{321}} \mathbf{r} \\
&\quad + \frac{1}{n} \mathbf{r}^T \underbrace{\sum_{i=1}^N \mathbb{E}\left[\frac{1}{m} \mathbf{Q}_m^T \mathbf{U}_n^T \boldsymbol{\sigma}_i \boldsymbol{\sigma}_i^T \mathbf{U}_n \mathbf{Q}_m \frac{1}{m}\left(\boldsymbol{\sigma}_i^T \mathbf{\Lambda}_P \boldsymbol{\sigma}_i - \operatorname{Tr}\left(\mathbf{\Phi}_S \mathbf{\Lambda}_P\right)\right)\right]}_{\mathbf{Z}_{322}} \mathbf{r}.
\end{aligned}
$$

We want to show $\mathbf{Z}_{322}$ vanishes and find an asymptotic equivalent for $\mathbf{Z}_{321}$. Let $\mathbf{D}_{322} \in \mathbb{R}^{N \times N}$ be a diagonal matrix for which, for all $i \in [N]$, we have

$$\left[\mathbf{D}_{322}\right]_i = \frac{1}{m} \boldsymbol{\sigma}_i^T \mathbf{\Lambda}_P \boldsymbol{\sigma}_i - \frac{1}{m} \operatorname{Tr}\left(\mathbf{\Phi}_S \mathbf{\Lambda}_P\right).$$

We rewrite $\mathbf{Z}_{322}$ as

$$
\begin{aligned}
\mathbf{Z}_{322} &= \sum_{i=1}^N \mathbb{E}\left[\frac{1}{m} \mathbf{Q}_m^T \mathbf{U}_n^T \boldsymbol{\sigma}_i \boldsymbol{\sigma}_i^T \mathbf{U}_n \mathbf{Q}_m \frac{1}{m}\left(\boldsymbol{\sigma}_i^T \mathbf{\Lambda}_P \boldsymbol{\sigma}_i - \operatorname{Tr}\left(\mathbf{\Phi}_S \mathbf{\Lambda}_P\right)\right)\right] \\
&= \mathbb{E}\left[\frac{1}{m} \mathbf{Q}_m^T \mathbf{U}_n^T \mathbf{\Sigma}_S^T \mathbf{D}_{322} \mathbf{\Sigma}_S \mathbf{U}_n \mathbf{Q}_m\right]
\end{aligned}
$$

From Lemma I.4, we know there exists a real $K'_{\mathbf{Q}} > 0$ such that, for all $m$, we have

$$\left\|\frac{1}{\sqrt{m}} \mathbf{\Sigma}_S \mathbf{U}_n \mathbf{Q}_m\right\| \le K'_{\mathbf{Q}}$$

and

$$\left\|\frac{1}{\sqrt{m}} \mathbf{Q}_m (\mathbf{U}_n - \gamma \mathbf{V}_n)^T \mathbf{\Sigma}_S^T\right\| \le 2K'_{\mathbf{Q}}.$$

Using Lemma K.2 we show that $\mathbb{E}\left[\|\mathbf{D}_{322}\|\right] = \mathcal{O}\left(\frac{1}{\sqrt{m}}\right)$, and we deduce that

$$\|\mathbf{Z}_{322}\| = \mathcal{O}\left(\frac{1}{\sqrt{m}}\right).$$

We want to find an asymptotic equivalent for $\mathbf{Z}_{321}$. Let $\mathbf{D}_{321} \in \mathbb{R}^{N \times N}$ be a diagonal matrix for which, for all $i \in [N]$, we have

$$\left[\mathbf{D}_{321}\right]_i = (1 + \delta)^2 - \left(1 + \frac{1}{m} \boldsymbol{\sigma}_i^T \mathbf{U}_n \mathbf{Q}_{-i}(\mathbf{U}_n - \gamma \mathbf{V}_n)^T \boldsymbol{\sigma}_i\right)^2.$$

We replace $1 + \frac{1}{m}\boldsymbol{\sigma}_i^T \boldsymbol{U}_n \boldsymbol{Q}_{-i}(\boldsymbol{U}_n - \gamma\boldsymbol{V}_n)^T\boldsymbol{\sigma}_i$ by $1 + \delta$ as following

$$
\begin{aligned}
\boldsymbol{Z}_{321} &= \sum_{i=1}^{N}\mathbb{E}\left[\frac{1}{m}\frac{\boldsymbol{Q}_{-i}^T\boldsymbol{U}_n^T\boldsymbol{\sigma}_i\boldsymbol{\sigma}_i^T\boldsymbol{U}_n\boldsymbol{Q}_{-i}}{\left(1 + \frac{1}{m}\boldsymbol{\sigma}_i^T\boldsymbol{U}_n\boldsymbol{Q}_{-i}(\boldsymbol{U}_n - \gamma\boldsymbol{V}_n)^T\boldsymbol{\sigma}_i\right)^2}\right] \\
&= \frac{1}{(1+\delta)^2}\sum_{i=1}^{N}\mathbb{E}\left[\frac{1}{m}\boldsymbol{Q}_{-i}^T\boldsymbol{U}_n^T\boldsymbol{\sigma}_i\boldsymbol{\sigma}_i^T\boldsymbol{U}_n\boldsymbol{Q}_{-i}\right] \\
&+ \frac{1}{(1+\delta)^2}\sum_{i=1}^{N}\mathbb{E}\left[\frac{1}{m}\boldsymbol{Q}_m^T\boldsymbol{U}_n^T\boldsymbol{\sigma}_i\boldsymbol{\sigma}_i^T\boldsymbol{U}_n\boldsymbol{Q}_m\left((1+\delta)^2 - \left(1 + \frac{1}{m}\boldsymbol{\sigma}_i^T\boldsymbol{U}_n\boldsymbol{Q}_{-i}(\boldsymbol{U}_n - \gamma\boldsymbol{V}_n)^T\boldsymbol{\sigma}_i\right)^2\right)\right] \\
&= \frac{N}{m}\frac{1}{(1+\delta)^2}\mathbb{E}\left[\boldsymbol{Q}_-^T\boldsymbol{U}_n^T\boldsymbol{\Phi}_{\mathcal{S}}\boldsymbol{U}_n\boldsymbol{Q}_-\right] + \frac{1}{(1+\delta)^2}\mathbb{E}\left[\frac{1}{m}\boldsymbol{Q}_m\boldsymbol{U}_n^T\boldsymbol{\Sigma}_{\mathcal{S}}^T\boldsymbol{D}_{321}\boldsymbol{\Sigma}_{\mathcal{S}}\boldsymbol{U}_n\boldsymbol{Q}_m\right] \\
&= \frac{1}{1+\delta}\mathbb{E}\left[\boldsymbol{Q}_-^T\boldsymbol{\Psi}_1\boldsymbol{Q}_-\right] + \mathcal{O}_{\|\cdot\|}\left(\frac{1}{\sqrt{m}}\right).
\end{aligned}
$$

The last equality is obtained since we can show that $\mathbb{E}\left[\|\boldsymbol{D}_{321}\|\right] = \mathcal{O}\left(\frac{1}{\sqrt{m}}\right)$. We have from Lemma F.4

$$
\left\|\mathbb{E}\left[\boldsymbol{Q}_m^T\boldsymbol{\Psi}_1\boldsymbol{Q}_m\right] - \mathbb{E}\left[\boldsymbol{Q}_-^T\boldsymbol{\Psi}_1\boldsymbol{Q}_-\right]\right\| = \mathcal{O}\left(\frac{1}{\sqrt{m}}\right),
$$

and from Lemma F.5

$$
\boldsymbol{Z}_{321} = \frac{1}{1+\delta}\mathbb{E}\left[\boldsymbol{Q}_m^T\boldsymbol{\Psi}_1\boldsymbol{Q}_m\right] = \frac{1}{1+\delta}\frac{1}{1 - \frac{1}{N}\text{Tr}\left(\boldsymbol{\Psi}_2\bar{\boldsymbol{Q}}_m^T\boldsymbol{\Psi}_1\bar{\boldsymbol{Q}}_m\right)}\bar{\boldsymbol{Q}}_m^T\boldsymbol{\Psi}_1\bar{\boldsymbol{Q}}_m + \mathcal{O}_{\|\cdot\|}\left(\frac{1}{\sqrt{m}}\right).
$$

We conclude that

$$
Z_{32} = \frac{1}{n}\frac{\frac{1}{N}\text{Tr}(\boldsymbol{\Lambda}_{\boldsymbol{P}}\boldsymbol{\Psi}_{\mathcal{S}})}{1 - \frac{1}{N}\text{Tr}\left(\boldsymbol{\Psi}_2\bar{\boldsymbol{Q}}_m^T\boldsymbol{\Psi}_1\bar{\boldsymbol{Q}}_m\right)}\|\bar{\boldsymbol{Q}}_m\boldsymbol{r}\|_{\boldsymbol{\Psi}_1}^2 + \mathcal{O}\left(\frac{1}{\sqrt{m}}\right).
$$

$\square$

**Lemma G.7.** *When all states have been visited, the empirical transition model matrix $\hat{\boldsymbol{A}}_m = \hat{\boldsymbol{U}}_n(\hat{\boldsymbol{U}}_n - \gamma\hat{\boldsymbol{V}}_n) = \boldsymbol{U}_n(\boldsymbol{U}_n - \gamma\boldsymbol{V}_n)$ defined in equation 14 is invertible.*

*Proof.* Let $c : \mathcal{S} \to \mathbb{N}$ and $c' : \mathcal{S} \to \mathbb{N}$ be defined such that, for all $i \in [p]$, $c(\boldsymbol{S}_i)$ and $c'(\boldsymbol{S}_i)$ represent the number of times $\boldsymbol{S}_i$ occurs in $\boldsymbol{X}_n$ and $\boldsymbol{X}_n'$, respectively. If all states have been visited ($m = p$), for all i in $[p]$, we have thus $c(\boldsymbol{S}_i) > 0$. The structure of $\sqrt{n}\boldsymbol{U}_n \in \mathbb{R}^{m \times n}$ indicates each column $i$ of $\boldsymbol{U}_n$ is a one-hot vector, where its $j$-th element is 1 if the $i$-th state $\boldsymbol{s}_i$ of $\boldsymbol{X}_n$ is $\boldsymbol{S}_j$. Conversely, each row $i$ of $\sqrt{n}\boldsymbol{U}_n$ has a $j$-th element is one if the $j$-th state $\boldsymbol{s}_j$ of $\boldsymbol{X}_n$ is $\boldsymbol{S}_i$. A similar correspondence holds for $\sqrt{n}\boldsymbol{V}_n$ and $\boldsymbol{X}_n'$. From interpretations of $\boldsymbol{U}_n$ and $\boldsymbol{V}_n$, we deduce $n\boldsymbol{U}_n\boldsymbol{U}_n^T \in \mathbb{R}^{m \times m}$ and $n\boldsymbol{V}_n\boldsymbol{V}_n^T \in \mathbb{R}^{m \times m}$ are diagonal matrices where the $i$-th element of its diagonal are $c(\boldsymbol{S}_i)$ and $c'(\boldsymbol{S}_i)$, respectively. In the same way, $n\boldsymbol{U}_n\boldsymbol{V}_n^T \in \mathbb{R}^{m \times m}$ is matrix for which $[n\boldsymbol{U}_n\boldsymbol{V}_n^T]_{ij}$ is $c(\boldsymbol{S}_i \to \boldsymbol{S}_j)$ which represents the number of times the state $\boldsymbol{S}_i$ follows $\boldsymbol{S}_j$ in $\mathcal{D}_{\text{train}}$. We are going to prove $\hat{\boldsymbol{A}}_m$ is invertible by using the Gershgorin circle theorem to show $\hat{\boldsymbol{A}}_m$ is strictly diagonally dominant, i.e., $|[\hat{\boldsymbol{A}}_m]_{ii}| > \sum_{i \neq j}|[\hat{\boldsymbol{A}}_m]_{ij}|$. From the interpretations of $\boldsymbol{U}_n\boldsymbol{U}_n^T$ and $\boldsymbol{U}_n\boldsymbol{V}_n^T$, we have

$$
[\hat{\boldsymbol{A}}_m]_{ii} = [\boldsymbol{U}_n\boldsymbol{U}_n^T]_{ii} - \gamma[\boldsymbol{U}_n\boldsymbol{V}_n^T]_{ii} = \frac{c(\boldsymbol{S}_i) - \gamma c(\boldsymbol{S}_i \to \boldsymbol{S}_i)}{n} > 0, \qquad \forall i \in [n],
$$

and

$$
[\hat{\boldsymbol{A}}_m]_{ij} = -\gamma[\boldsymbol{U}_n\boldsymbol{V}_n^T]_{ij} = \frac{-\gamma c(\boldsymbol{S}_i \to \boldsymbol{S}_j)}{n} < 0, \qquad \forall i \neq j.
$$

To prove $\hat{\boldsymbol{A}}_m$ is invertible it remains to show $\sum_j[\hat{\boldsymbol{A}}_m]_{ij} = \sum_j[\boldsymbol{U}_n(\boldsymbol{U}_n - \gamma\boldsymbol{V}_n)^T]_{ij} > 0$ for all $i \in [m]$. Let $i \in [m]$, we have

$$
\sum_j[\boldsymbol{U}_n(\boldsymbol{U}_n - \gamma\boldsymbol{V}_n)^T]_{ij} = \frac{c(\boldsymbol{S}_i)}{n} - \gamma\sum_j\frac{c(\boldsymbol{S}_i \to \boldsymbol{S}_j)}{n} = (1 - \gamma)\frac{c(\boldsymbol{S}_i)}{n} > 0,
$$

which concludes the proof. $\square$

**Lemma G.8.** *Let $\Delta$ be the second-order correction factor of $\overline{\mathrm{MSBE}}(\hat{\boldsymbol{\theta}})$ defined in equation 20. If all states have been visited then*

$$\Delta = \frac{\lambda^2}{n} \frac{\frac{1}{N}\operatorname{Tr}\left(\boldsymbol{U}_n^T \hat{\boldsymbol{A}}_m^{-1T} \boldsymbol{\Lambda}_{\boldsymbol{P}} \hat{\boldsymbol{A}}_m^{-1} \boldsymbol{U}_n \bar{\boldsymbol{Q}}_m \boldsymbol{\Psi}_2 \bar{\boldsymbol{Q}}_m^T\right)}{1 - \frac{1}{N}\operatorname{Tr}\left(\boldsymbol{\Psi}_2 \bar{\boldsymbol{Q}}_m(\lambda)^T \boldsymbol{\Psi}_1 \bar{\boldsymbol{Q}}_m(\lambda)\right)} \|\bar{\boldsymbol{Q}}_m(\lambda)\boldsymbol{r}\|^2_{\boldsymbol{\Psi}_1},$$

*where $\hat{\boldsymbol{A}}_m = \hat{\boldsymbol{U}}_n(\hat{\boldsymbol{U}}_n - \gamma\hat{\boldsymbol{V}}_n) = \boldsymbol{U}_n(\boldsymbol{U}_n - \gamma\boldsymbol{V}_n)$ is the empirical transition model matrix defined in equation 14.*

*Proof.* When all states have been visited, we have $\boldsymbol{U}_n = \hat{\boldsymbol{U}}_n$, $\boldsymbol{V}_n = \hat{\boldsymbol{V}}_n$ and $\boldsymbol{\Sigma}_{\mathcal{S}} = \boldsymbol{\Sigma}_{\hat{\mathcal{S}}}$. Furthermore, from Lemma G.7, $\hat{\boldsymbol{A}}_m = \hat{\boldsymbol{U}}_n(\hat{\boldsymbol{U}}_n - \gamma\hat{\boldsymbol{V}}_n) = \boldsymbol{U}_n(\boldsymbol{U}_n - \gamma\boldsymbol{V}_n)$ is invertible. We write

$$\boldsymbol{\Theta}_{\mathcal{S}} = \boldsymbol{\Psi}_{\mathcal{S}}\boldsymbol{U}_n\bar{\boldsymbol{Q}}_m(\lambda) = \hat{\boldsymbol{A}}_m^{-1}\boldsymbol{U}_n(\boldsymbol{U}_n - \gamma\boldsymbol{V}_n)^T\boldsymbol{\Psi}_{\mathcal{S}}\boldsymbol{U}_n\bar{\boldsymbol{Q}}_m(\lambda)$$
$$= \hat{\boldsymbol{A}}_m^{-1}\boldsymbol{U}_n\big[\boldsymbol{I}_n - \lambda\bar{\boldsymbol{Q}}_m\big]$$

Using the equality above and the cyclic properties of the trace we conclude that

$$\operatorname{Tr}\Big(\boldsymbol{\Lambda}_{\boldsymbol{P}}\big[\boldsymbol{\Theta}_{\mathcal{S}}\boldsymbol{\Psi}_2\boldsymbol{\Theta}_{\mathcal{S}}^T - 2\boldsymbol{\Theta}_{\mathcal{S}}(\boldsymbol{U}_n - \gamma\boldsymbol{V}_n)^T\boldsymbol{\Psi}_{\mathcal{S}} + \boldsymbol{\Psi}_{\mathcal{S}}\big]\Big)$$
$$= \operatorname{Tr}\Big(\boldsymbol{\Lambda}_{\boldsymbol{P}}\big[\hat{\boldsymbol{A}}_m^{-1}\boldsymbol{U}_n\big[\boldsymbol{I}_n - \lambda\bar{\boldsymbol{Q}}_m\big]\boldsymbol{\Psi}_2\big[\boldsymbol{I}_n - \lambda\bar{\boldsymbol{Q}}_m\big]^T\boldsymbol{U}_n^T\hat{\boldsymbol{A}}_m^{-1T}$$
$$\quad - 2\hat{\boldsymbol{A}}_m^{-1}\boldsymbol{U}_n\big[\boldsymbol{I}_n - \lambda\bar{\boldsymbol{Q}}_m\big](\boldsymbol{U}_n - \gamma\boldsymbol{V}_n)^T\boldsymbol{\Psi}_{\mathcal{S}} + \boldsymbol{\Psi}_{\mathcal{S}}\big]\Big)$$
$$= \lambda^2\operatorname{Tr}\Big(\boldsymbol{\Lambda}_{\boldsymbol{P}}\hat{\boldsymbol{A}}_m^{-1}\boldsymbol{U}_n\bar{\boldsymbol{Q}}_m\boldsymbol{\Psi}_2\bar{\boldsymbol{Q}}_m^T\boldsymbol{U}_n^T\hat{\boldsymbol{A}}_m^{-1T}\Big) - \lambda\operatorname{Tr}\Big(\boldsymbol{\Lambda}_{\boldsymbol{P}}\hat{\boldsymbol{A}}_m^{-1}\boldsymbol{U}_n\bar{\boldsymbol{Q}}_m(\boldsymbol{U}_n - \gamma\boldsymbol{V}_n)^T\boldsymbol{\Psi}_{\mathcal{S}}\Big)$$
$$\quad - \lambda\operatorname{Tr}\Big(\boldsymbol{\Lambda}_{\boldsymbol{P}}\boldsymbol{\Psi}_{\mathcal{S}}(\boldsymbol{U}_n - \gamma\boldsymbol{V}_n)\boldsymbol{Q}_m^T\boldsymbol{U}_n^T\hat{\boldsymbol{A}}_m^{-1T}\Big) + 2\lambda\operatorname{Tr}\Big(\boldsymbol{\Lambda}_{\boldsymbol{P}}\hat{\boldsymbol{A}}_m^{-1}\boldsymbol{U}_n\bar{\boldsymbol{Q}}_m(\boldsymbol{U}_n - \gamma\boldsymbol{V}_n)^T\boldsymbol{\Psi}_{\mathcal{S}}\Big)$$
$$= \lambda^2\operatorname{Tr}\Big(\boldsymbol{\Lambda}_{\boldsymbol{P}}\hat{\boldsymbol{A}}_m^{-1}\boldsymbol{U}_n\bar{\boldsymbol{Q}}_m\boldsymbol{\Psi}_2\bar{\boldsymbol{Q}}_m^T\boldsymbol{U}_n^T\hat{\boldsymbol{A}}_m^{-1T}\Big)$$
$$= \lambda^2\operatorname{Tr}\Big(\boldsymbol{U}_n^T\hat{\boldsymbol{A}}_m^{-1T}\boldsymbol{\Lambda}_{\boldsymbol{P}}\hat{\boldsymbol{A}}_m^{-1}\boldsymbol{U}_n\bar{\boldsymbol{Q}}_m\boldsymbol{\Psi}_2\bar{\boldsymbol{Q}}_m^T\Big).$$

$\square$

## H  EXISTENCE OF THE RESOLVENT $\boldsymbol{Q}_m(\lambda)$

In this section, we show that Assumption 2 guarantees the existence of the resolvent $\boldsymbol{Q}_m(\lambda)$ (Lemma H.1), but also that Assumption 2 may be true in practice under certain conditions (Lemma H.2).

**Lemma H.1.** *Under Assumption 2, for any $\lambda > 0$, the resolvent $\boldsymbol{Q}_m(\lambda)$ defined in equation 13 exists.*

*Proof.* From Assumption 2, we know that $\nu_{\min}\big(H(\hat{\boldsymbol{A}}_m)\big) > \xi_{\min} > 0$, and thus $H(\boldsymbol{\Sigma}_{\hat{\mathcal{S}}}\hat{\boldsymbol{A}}_m\boldsymbol{\Sigma}_{\hat{\mathcal{S}}}^T)$ is at least semi-positive-definite. From the Min-Max theorem, we deduce that the eigenvalues of $\boldsymbol{\Sigma}_{\hat{\mathcal{S}}}\hat{\boldsymbol{A}}_m\boldsymbol{\Sigma}_{\hat{\mathcal{S}}}^T$ have nonnegative real-parts. Consequently, the eigenvalues of $\frac{1}{m}(\hat{\boldsymbol{U}}_n - \gamma\hat{\boldsymbol{V}}_n)^T\boldsymbol{\Sigma}_{\hat{\mathcal{S}}}^T\boldsymbol{\Sigma}_{\hat{\mathcal{S}}}\hat{\boldsymbol{U}}_n$ have nonnegative real parts since both $\frac{1}{m}(\hat{\boldsymbol{U}}_n - \gamma\hat{\boldsymbol{V}}_n)^T\boldsymbol{\Sigma}_{\hat{\mathcal{S}}}^T\boldsymbol{\Sigma}_{\hat{\mathcal{S}}}\hat{\boldsymbol{U}}_n$ and $\boldsymbol{\Sigma}_{\hat{\mathcal{S}}}\hat{\boldsymbol{A}}_m\boldsymbol{\Sigma}_{\hat{\mathcal{S}}}^T$ share the same nonzero eigenvalues from the Weinstein–Aronszajn identity (Lemma L.6). $\square$

**Lemma H.2.** *Let $c : \hat{\mathcal{S}} \to \mathbb{N}$ and $c' : \hat{\mathcal{S}} \to \mathbb{N}$ be defined such that, for all $i \in [m]$, $c(\hat{\boldsymbol{S}}_i)$ and $c'(\hat{\boldsymbol{S}}_i)$ represent the number of times $\hat{\boldsymbol{S}}_i$ occurs in $\boldsymbol{X}_n$ and $\boldsymbol{X}_n'$, respectively. If for all $i \in [m]$, $c(\hat{\boldsymbol{S}}_i) \geq \gamma c'(\hat{\boldsymbol{S}}_i)$ then the symmetric part of the empirical transition model matrix $\hat{\boldsymbol{A}}_m$ (defined in equation 14) is positive-definite.*

*Proof.* The structure of $\sqrt{n}\hat{\boldsymbol{U}}_n \in \mathbb{R}^{m \times n}$ indicates each column $i$ of $\hat{\boldsymbol{U}}_n$ is a one-hot vector, where its $j$-th element is 1 if the $i$-th state $\boldsymbol{s}_i$ of $\boldsymbol{X}_n$ is $\hat{\boldsymbol{S}}_j$. Conversely, each row $i$ of $\sqrt{n}\hat{\boldsymbol{U}}_n$ has a $j$-th element equal to one if the $j$-th state $\boldsymbol{s}_j$ of $\boldsymbol{X}_n$ is $\hat{\boldsymbol{S}}_i$. A similar correspondence holds for $\sqrt{n}\hat{\boldsymbol{V}}_n$ and $\boldsymbol{X}_n'$.

From interpretations of $\hat{\boldsymbol{U}}_n$ and $\hat{\boldsymbol{V}}_n$, we deduce that $n\hat{\boldsymbol{U}}_n\hat{\boldsymbol{U}}_n^T \in \mathbb{R}^{m \times m}$ and $n\hat{\boldsymbol{V}}_n\hat{\boldsymbol{V}}_n^T \in \mathbb{R}^{m \times m}$ are diagonal matrices where the $i$-th element of its diagonal is equal to $c(\hat{\boldsymbol{S}}_i)$ and $c'(\hat{\boldsymbol{S}}_i)$, respectively. In the same way, $n\hat{\boldsymbol{U}}_n\hat{\boldsymbol{V}}_n^T \in \mathbb{R}^{m \times m}$ is matrix for which $[n\hat{\boldsymbol{U}}_n\hat{\boldsymbol{V}}_n^T]_{ij}$ is $c(\hat{\boldsymbol{S}}_i \to \hat{\boldsymbol{S}}_j)$, i.e., the number of times the state $\hat{\boldsymbol{S}}_i$ follows $\hat{\boldsymbol{S}}_j$ in $\mathcal{D}_{\text{train}}$. We want to prove $H(\hat{\boldsymbol{A}}_m) = \frac{\hat{\boldsymbol{A}}_m + \hat{\boldsymbol{A}}_m^T}{2}$ is positive-definite by using the Gershgorin circle theorem and by showing $H(\hat{\boldsymbol{A}}_m)$ is strictly diagonally dominant, i.e., $|[H(\hat{\boldsymbol{A}}_m)]_{ii}| > \sum_{i \neq j}|[H(\hat{\boldsymbol{A}}_m)]_{ij}|$. From interpretations of $\hat{\boldsymbol{U}}_n\hat{\boldsymbol{U}}_n^T$ and $\hat{\boldsymbol{U}}_n\hat{\boldsymbol{V}}_n^T$, we have for all $i \in [n]$

$$[H(\hat{\boldsymbol{A}}_m)]_{ii} = [\hat{\boldsymbol{U}}_n\hat{\boldsymbol{U}}_n^T]_{ii} - \gamma[\hat{\boldsymbol{U}}_n\hat{\boldsymbol{V}}_n^T]_{ii} = \frac{c(\hat{\boldsymbol{S}}_i) - \gamma c(\hat{\boldsymbol{S}}_i \to \hat{\boldsymbol{S}}_i)}{n} > 0,$$

and for all $i \neq j$

$$[H(\hat{\boldsymbol{A}}_m)]_{ij} = \frac{-\gamma[\hat{\boldsymbol{U}}_n\hat{\boldsymbol{V}}_n^T]_{ij} - \gamma[\hat{\boldsymbol{U}}_n\hat{\boldsymbol{V}}_n^T]_{ji}}{2} = \frac{-\gamma c(\hat{\boldsymbol{S}}_i \to \hat{\boldsymbol{S}}_j) - \gamma c(\hat{\boldsymbol{S}}_j \to \hat{\boldsymbol{S}}_i)}{2n} < 0.$$

To prove that $H(\hat{\boldsymbol{A}}_m)$ is positive-definite it remains to show that

$$\sum_j [H(\hat{\boldsymbol{A}}_m)]_{ij} = \sum_{j \neq i}[H(\hat{\boldsymbol{A}}_m)]_{ij} + [H(\hat{\boldsymbol{A}}_m)]_{ii}$$

$$= \sum_j \left[\frac{\hat{\boldsymbol{U}}_n(\hat{\boldsymbol{U}}_n - \gamma\hat{\boldsymbol{V}}_n)^T}{2}\right]_{ij} + \sum_j \left[\frac{\hat{\boldsymbol{U}}_n(\hat{\boldsymbol{U}}_n - \gamma\hat{\boldsymbol{V}}_n)^T}{2}\right]_{ji} > 0$$

for all $i \in [m]$. Let $i \in [m]$, we have

$$\sum_j [\hat{\boldsymbol{U}}_n(\hat{\boldsymbol{U}}_n - \gamma\hat{\boldsymbol{V}}_n)^T]_{ij} = \frac{c(\hat{\boldsymbol{S}}_i)}{n} - \gamma \sum_j \frac{c(\hat{\boldsymbol{S}}_i \to \hat{\boldsymbol{S}}_j)}{n} = (1 - \gamma)\frac{c(\hat{\boldsymbol{S}}_i)}{n} > 0$$

and

$$\sum_j [\hat{\boldsymbol{U}}_n(\hat{\boldsymbol{U}}_n - \gamma\hat{\boldsymbol{V}}_n)^T]_{ji} = \frac{c(\hat{\boldsymbol{S}}_i)}{n} - \gamma \sum_j \frac{c(\hat{\boldsymbol{S}}_j \to \hat{\boldsymbol{S}}_i)}{n} = \frac{c(\hat{\boldsymbol{S}}_i) - \gamma c'(\hat{\boldsymbol{S}}_i)}{n} > 0,$$

since $c(\hat{\boldsymbol{S}}_i) \geq \gamma c'(\hat{\boldsymbol{S}}_i)$ for all $i \in [m]$. We deduce for all $i \in [m]$ that

$$\sum_j [H(\hat{\boldsymbol{A}}_m)]_{ij} = \sum_j \left[\frac{\hat{\boldsymbol{U}}_n(\hat{\boldsymbol{U}}_n - \gamma\hat{\boldsymbol{V}}_n)^T}{2}\right]_{ij} + \sum_j \left[\frac{\hat{\boldsymbol{U}}_n(\hat{\boldsymbol{U}}_n - \gamma\hat{\boldsymbol{V}}_n)^T}{2}\right]_{ji} > 0,$$

and thus $H(\hat{\boldsymbol{A}}_m)$ is strictly diagonally dominant and positive-definite. $\qquad\square$

**Remark 19.** *Conditions of Lemma H.2 may hold in practice. If $\mathcal{D}_{train}$ is derived from a sample path of the MRP, where $\boldsymbol{s}'_{i+1} = \boldsymbol{s}_i$ for all $i \in [n-1]$, and if $\hat{\boldsymbol{S}}_l$ depicts the distinct visited state corresponding to the last next state visited $\boldsymbol{s}'_n$ in $\mathcal{D}_{train}$, then we have $c(\hat{\boldsymbol{S}}_i) = c'(\hat{\boldsymbol{S}}_i)$ for all $i \neq l$ and $c(\hat{\boldsymbol{S}}_l) = c'(\hat{\boldsymbol{S}}_l) - 1$. For sufficiently large $n$, we may have $c(\hat{\boldsymbol{S}}_l) \geq \frac{\gamma}{1-\gamma}$ which satisfies conditions of Lemma H.2. Similarly, conditions of Lemma H.2 are satisfied for the pathwise LSTD algorithm, where $\mathcal{D}_{train}$ is perturbed slightly by setting the feature of the next state of the last transition to zero (Lazaric et al., 2012) to get $c(\hat{\boldsymbol{S}}_l) \geq c'(\hat{\boldsymbol{S}}_l)$.*

## I   TECHNICAL DETAILS ON THE RESOLVENT $\boldsymbol{Q}_m(\lambda)$

The objective of this section is to prove that the operator norm of $\boldsymbol{Q}_m(\lambda)$ is uniformly upper bounded under Assumption 2. Indeed, controlling the operator norm of $\boldsymbol{Q}_m(\lambda)$ is crucial for proving the theorems in Section 5. When $\gamma = 0$, which corresponds to the supervised learning case on the reward function, the result is straightforward with Lemma L.7 since $\frac{1}{m}\boldsymbol{\Sigma}_{\boldsymbol{X}_n}^T\boldsymbol{\Sigma}_{\boldsymbol{X}_n}$ is positive-definite (Louart et al., 2018; Liao et al., 2020). In the RL setting, the conclusion is less straightforward as the resolvent is no longer that of a symmetric positive-definite matrix. This issue is further exacerbated by the lack of results in the literature concerning the upper bounds for operator norm of resolvents of non-positive-definite matrices. Lemma I.1 aims to propose a solution for the RL setting under Assumptions 1 and 2. Proof of the widely used Lemma I.4 is also presented at the end of this section.

**Lemma I.1.** *Under Assumptions 1 and 2, let $\lambda > 0$ and let $\boldsymbol{Q}_m(\lambda) \in \mathbb{R}^{n \times n}$ be the resolvent defined in equation 13 as*

$$\boldsymbol{Q}_m(\lambda) = \left[ \frac{1}{m}(\hat{\boldsymbol{U}}_n - \gamma\hat{\boldsymbol{V}}_n)^T \boldsymbol{\Sigma}_{\hat{\mathcal{S}}}^T \boldsymbol{\Sigma}_{\hat{\mathcal{S}}} \hat{\boldsymbol{U}}_n + \lambda \boldsymbol{I}_n \right]^{-1}.$$

*Then there exists a real $K > 0$ such that, for all $m$, we have*

$$\|\boldsymbol{Q}_m(\lambda)\| \leq K.$$

*Proof.* Under Assumption 2, the empirical transition model matrix $\hat{\boldsymbol{A}}_m = \hat{\boldsymbol{U}}_n(\hat{\boldsymbol{U}}_n - \gamma\hat{\boldsymbol{V}}_n)^T$ (equation 14) is invertible since the symmetric part of $\hat{\boldsymbol{A}}_m$ is positive-definite. Let

$$0 < \epsilon < \lambda \min\left\{ \frac{1}{\xi_{\max}}, \frac{\xi_{\min}}{4} \right\},$$

for $\xi_{\min}, \xi_{\max} > 0$ defined in Assumption 2. We rewrite equation 13 as

$$\boldsymbol{Q}_m(\lambda) = \left[ \frac{1}{m}(\hat{\boldsymbol{U}}_n - \gamma\hat{\boldsymbol{V}}_n)^T \boldsymbol{\Sigma}_{\hat{\mathcal{S}}}^T \boldsymbol{\Sigma}_{\hat{\mathcal{S}}} \hat{\boldsymbol{U}}_n + \lambda \boldsymbol{I}_n \right]^{-1}$$

$$= \left[ (\hat{\boldsymbol{U}}_n - \gamma\hat{\boldsymbol{V}}_n)^T \left[ \frac{1}{m}\boldsymbol{\Sigma}_{\hat{\mathcal{S}}}^T \boldsymbol{\Sigma}_{\hat{\mathcal{S}}} + \epsilon \boldsymbol{I}_m \right] \hat{\boldsymbol{U}}_n + \underbrace{\lambda \boldsymbol{I}_n - \epsilon(\hat{\boldsymbol{U}}_n - \gamma\hat{\boldsymbol{V}}_n)^T \hat{\boldsymbol{U}}_n}_{=\boldsymbol{B}_n} \right]^{-1}.$$

To apply the Woodbury identity (Lemma L.3) on $\boldsymbol{Q}_m(\lambda)$, we check that both

$$\boldsymbol{B}_n = \lambda \boldsymbol{I}_n - \epsilon(\hat{\boldsymbol{U}}_n - \gamma\hat{\boldsymbol{V}}_n)^T \hat{\boldsymbol{U}}_n,$$

and

$$\boldsymbol{M}_m = \left[ \frac{1}{m}\boldsymbol{\Sigma}_{\hat{\mathcal{S}}}^T \boldsymbol{\Sigma}_{\hat{\mathcal{S}}} + \epsilon \boldsymbol{I}_m \right]^{-1} + \hat{\boldsymbol{U}}_n \boldsymbol{B}_n^{-1}(\hat{\boldsymbol{U}}_n - \gamma\hat{\boldsymbol{V}}_n)^T$$

$$= \left[ \frac{1}{m}\boldsymbol{\Sigma}_{\hat{\mathcal{S}}}^T \boldsymbol{\Sigma}_{\hat{\mathcal{S}}} + \epsilon \boldsymbol{I}_m \right]^{-1} + \left[ \lambda \boldsymbol{I}_n - \epsilon\hat{\boldsymbol{A}}_m \right]^{-1} \hat{\boldsymbol{A}}_m$$

$$= \left[ \frac{1}{m}\boldsymbol{\Sigma}_{\hat{\mathcal{S}}}^T \boldsymbol{\Sigma}_{\hat{\mathcal{S}}} + \epsilon \boldsymbol{I}_m \right]^{-1} + \left[ \lambda \hat{\boldsymbol{A}}_m^{-1} - \epsilon \boldsymbol{I}_m \right]^{-1}$$

are non-singular, since $\frac{1}{m}\boldsymbol{\Sigma}_{\hat{\mathcal{S}}}^T \boldsymbol{\Sigma}_{\hat{\mathcal{S}}} + \epsilon \boldsymbol{I}_m$ is non-singular. Given that $H(\hat{\boldsymbol{A}}_m)$ is positive-definite, $\hat{\boldsymbol{A}}_m$ has eigenvalues with positive real parts. Consequently, by the Weinstein–Aronszajn identity (Lemma L.6), $(\hat{\boldsymbol{U}}_n - \gamma\hat{\boldsymbol{V}}_n)^T \hat{\boldsymbol{U}}_n$ has non-zero eigenvalues with positive real parts. As $\epsilon < \frac{\lambda}{\xi_{\max}} \leq \frac{\lambda}{\nu_{\max}(H(\hat{\boldsymbol{A}}_m))} \leq \frac{\lambda}{\text{Re}(\nu_{\max}(\hat{\boldsymbol{A}}_m))}$, we deduce that the matrix $\boldsymbol{B}_n = \lambda \boldsymbol{I}_n - \epsilon(\hat{\boldsymbol{U}}_n - \gamma\hat{\boldsymbol{V}}_n)^T \hat{\boldsymbol{U}}_n$ has eigenvalues with positive real parts and is non-singular. In order to prove that the matrix $\boldsymbol{M}_m$ is non-singular, we propose to show $\boldsymbol{x}^T \boldsymbol{M}_m \boldsymbol{x} > 0$ for all non-zero $\boldsymbol{x} \in \mathbb{R}^m$. Since $\left[ \frac{1}{m}\boldsymbol{\Sigma}_{\hat{\mathcal{S}}}^T \boldsymbol{\Sigma}_{\hat{\mathcal{S}}} + \epsilon \boldsymbol{I}_m \right]^{-1}$ is at least positive-semi-definite, the statement $\boldsymbol{x}^T \boldsymbol{M}_m \boldsymbol{x} > 0$ for all non-zero $\boldsymbol{x} \in \mathbb{R}^m$ may be restated as

$$\text{for all non-zero } \boldsymbol{x} \in \mathbb{R}^m, \quad \boldsymbol{x}^T \left[ \lambda \hat{\boldsymbol{A}}_m^{-1} - \epsilon \boldsymbol{I}_m \right]^{-1} \boldsymbol{x} > 0$$

$$\text{iff} \quad \text{for all non-zero } \boldsymbol{x} \in \mathbb{R}^m, \quad \boldsymbol{x}^T \left[ \lambda \hat{\boldsymbol{A}}_m^{-1} - \epsilon \boldsymbol{I}_m \right] \boldsymbol{x} > 0$$

$$\text{iff} \quad \text{for all non-zero } \boldsymbol{x} \in \mathbb{R}^m, \quad \boldsymbol{x}^T H\left( \hat{\boldsymbol{A}}_m^{-1} \right) \boldsymbol{x} - \frac{\epsilon}{\lambda}\boldsymbol{x}^T \boldsymbol{x} > 0$$

$$\text{iff} \quad \nu_{\min}\left( H(\hat{\boldsymbol{A}}_m^{-1}) \right) > \frac{\epsilon}{\lambda}.$$

By construction of $\hat{\boldsymbol{U}}_n$ and $\hat{\boldsymbol{V}}_n$, we have both $\|\hat{\boldsymbol{U}}_n\| \leq 1$ and $\|\hat{\boldsymbol{V}}_n\| \leq 1$. We deduce thus

$$\left\| \hat{\boldsymbol{A}}_m \right\| = \left\| \hat{\boldsymbol{U}}_n(\hat{\boldsymbol{U}}_n - \gamma\hat{\boldsymbol{V}}_n)^T \right\| < 2.$$

Since $H(\hat{A}_m^{-1}) = [\hat{A}_m^{-1}]^T H(\hat{A}_m) \hat{A}_m^{-1}$, we deduce from Ostrowski's Theorem (Lemma L.5) that

$$\nu_{\min}\big(H(\hat{A}_m^{-1})\big) \geq \frac{\nu_{\min}\big(H(\hat{A}_m)\big)}{\|\hat{A}_m\|^2} \geq \frac{\xi_{\min}}{4}.$$

Since $\epsilon < \frac{\lambda \xi_{\min}}{4}$, we have $\boldsymbol{x}^T \boldsymbol{M}_m \boldsymbol{x} > 0$ for all non-zero $\boldsymbol{x} \in \mathbb{R}^m$, and thus $\boldsymbol{M}_m$ is non-singular. As a consequence, we apply the Woodbury identity (Lemma L.3) on the resolvent $\boldsymbol{Q}_m(\lambda)$ to get

$$\boldsymbol{Q}_m(\lambda) = \left[ (\hat{\boldsymbol{U}}_n - \gamma \hat{\boldsymbol{V}}_n)^T \left[ \frac{1}{m} \boldsymbol{\Sigma}_{\hat{\mathcal{S}}}^T \boldsymbol{\Sigma}_{\hat{\mathcal{S}}} + \epsilon \boldsymbol{I}_m \right] \hat{\boldsymbol{U}}_n + \boldsymbol{B}_n \right]^{-1}$$
$$= \boldsymbol{B}_n^{-1} - \boldsymbol{B}_n^{-1}(\hat{\boldsymbol{U}}_n - \gamma \hat{\boldsymbol{V}}_n)^T \boldsymbol{M}_m^{-1} \hat{\boldsymbol{U}}_n \boldsymbol{B}_n^{-1}.$$

Multiplying the equation above by $\boldsymbol{B}_n = \lambda \boldsymbol{I}_n - \epsilon(\hat{\boldsymbol{U}}_n - \gamma \hat{\boldsymbol{V}}_n)^T \hat{\boldsymbol{U}}_n$ on both sides, and after manipulating terms to isolate $\boldsymbol{Q}_n$ on the left-hand side gives

$$\boldsymbol{Q}_m(\lambda) = \frac{1}{\lambda^2} \Bigg[ \boldsymbol{B}_n - (\hat{\boldsymbol{U}}_n - \gamma \hat{\boldsymbol{V}}_n)^T \boldsymbol{M}_m^{-1} \hat{\boldsymbol{U}}_n$$
$$+ \lambda\epsilon \Big[ (\hat{\boldsymbol{U}}_n - \gamma \hat{\boldsymbol{V}}_n)^T \hat{\boldsymbol{U}}_n \boldsymbol{Q}_m(\lambda) + \boldsymbol{Q}_m(\lambda)(\hat{\boldsymbol{U}}_n - \gamma \hat{\boldsymbol{V}}_n)^T \hat{\boldsymbol{U}}_n \Big]$$
$$- \epsilon^2 (\hat{\boldsymbol{U}}_n - \gamma \hat{\boldsymbol{V}}_n)^T \hat{\boldsymbol{U}}_n \boldsymbol{Q}_m(\lambda)(\hat{\boldsymbol{U}}_n - \gamma \hat{\boldsymbol{V}}_n)^T \hat{\boldsymbol{U}}_n \Bigg]$$

$$= \frac{1}{\lambda^2} \Bigg[ \boldsymbol{B}_n - (\hat{\boldsymbol{U}}_n - \gamma \hat{\boldsymbol{V}}_n)^T \boldsymbol{M}_m^{-1} \hat{\boldsymbol{U}}_n$$
$$+ \lambda\epsilon(\hat{\boldsymbol{U}}_n - \gamma \hat{\boldsymbol{V}}_n)^T \left[ \frac{1}{m} \hat{\boldsymbol{A}}_m \boldsymbol{\Sigma}_{\hat{\mathcal{S}}}^T \boldsymbol{\Sigma}_{\hat{\mathcal{S}}} + \lambda \boldsymbol{I}_m \right]^{-1} \hat{\boldsymbol{U}}_n$$
$$+ \lambda\epsilon(\hat{\boldsymbol{U}}_n - \gamma \hat{\boldsymbol{V}}_n)^T \left[ \frac{1}{m} \boldsymbol{\Sigma}_{\hat{\mathcal{S}}}^T \boldsymbol{\Sigma}_{\hat{\mathcal{S}}} \hat{\boldsymbol{A}}_m + \lambda \boldsymbol{I}_m \right]^{-1} \hat{\boldsymbol{U}}_n$$
$$- \epsilon^2 (\hat{\boldsymbol{U}}_n - \gamma \hat{\boldsymbol{V}}_n)^T \left[ \frac{1}{m} \hat{\boldsymbol{A}}_m \boldsymbol{\Sigma}_{\hat{\mathcal{S}}}^T \boldsymbol{\Sigma}_{\hat{\mathcal{S}}} + \lambda \boldsymbol{I}_m \right]^{-1} \hat{\boldsymbol{U}}_n(\hat{\boldsymbol{U}}_n - \gamma \hat{\boldsymbol{V}}_n)^T \hat{\boldsymbol{U}}_n \Bigg].$$

Applying the operator nom on the equality above, we find

$$\|\boldsymbol{Q}_m(\lambda)\| \leq \frac{1}{\lambda^2} \Bigg[ \lambda + 2\epsilon + 2\|\boldsymbol{M}_m^{-1}\|$$
$$+ 2\lambda\epsilon \left\| \left[ \frac{1}{m} \hat{\boldsymbol{A}}_m \boldsymbol{\Sigma}_{\hat{\mathcal{S}}}^T \boldsymbol{\Sigma}_{\hat{\mathcal{S}}} + \lambda \boldsymbol{I}_m \right]^{-1} \right\| + 2\lambda\epsilon \left\| \left[ \frac{1}{m} \boldsymbol{\Sigma}_{\hat{\mathcal{S}}}^T \boldsymbol{\Sigma}_{\hat{\mathcal{S}}} \hat{\boldsymbol{A}}_m + \lambda \boldsymbol{I}_m \right]^{-1} \right\|$$
$$+ 4\epsilon^2 \left\| \left[ \frac{1}{m} \hat{\boldsymbol{A}}_m \boldsymbol{\Sigma}_{\hat{\mathcal{S}}}^T \boldsymbol{\Sigma}_{\hat{\mathcal{S}}} + \lambda \boldsymbol{I}_m \right]^{-1} \right\| \Bigg], \tag{75}$$

since

$$\|\boldsymbol{B}_n\| = \left\| \lambda \boldsymbol{I}_n - \epsilon(\hat{\boldsymbol{U}}_n - \gamma \hat{\boldsymbol{V}}_n)^T \hat{\boldsymbol{U}}_n \right\| \leq \lambda + 2\epsilon.$$

From Lemma I.2, we have

$$\left\| \left[ \frac{1}{m} \hat{\boldsymbol{A}}_m \boldsymbol{\Sigma}_{\hat{\mathcal{S}}}^T \boldsymbol{\Sigma}_{\hat{\mathcal{S}}} + \lambda \boldsymbol{I}_m \right]^{-1} \right\| \leq \frac{1}{\lambda} \frac{4}{\xi_{\min}^2},$$

and

$$\left\| \left[ \frac{1}{m} \boldsymbol{\Sigma}_{\hat{\mathcal{S}}}^T \boldsymbol{\Sigma}_{\hat{\mathcal{S}}} \hat{\boldsymbol{A}}_m + \lambda \boldsymbol{I}_m \right]^{-1} \right\| \leq \frac{1}{\lambda} \frac{4}{\xi_{\min}^2}.$$

To finish the proof, we find an upper bound for $\|\boldsymbol{M}_m^{-1}\|$. By denoting by $\boldsymbol{Z}^T\boldsymbol{Z}$ the Cholesky decomposition of the positive-semi-definite matrix $\left[\frac{1}{m}\boldsymbol{\Sigma}_{\hat{\mathcal{S}}}^T\boldsymbol{\Sigma}_{\hat{\mathcal{S}}} + \epsilon\boldsymbol{I}_m\right]^{-1}$, we reuse the Woodbury identity (Lemma L.3) to rewrite $\boldsymbol{M}_m^{-1}$ as

$$
\begin{aligned}
\boldsymbol{M}_m^{-1} &= \left[\left[\frac{1}{m}\boldsymbol{\Sigma}_{\hat{\mathcal{S}}}^T\boldsymbol{\Sigma}_{\hat{\mathcal{S}}} + \epsilon\boldsymbol{I}_m\right]^{-1} + \left[\lambda\hat{\boldsymbol{A}}_m^{-1} - \epsilon\boldsymbol{I}_m\right]^{-1}\right]^{-1} \\
&= \left[\boldsymbol{Z}^T\boldsymbol{Z} + \left[\lambda\hat{\boldsymbol{A}}_m^{-1} - \epsilon\boldsymbol{I}_m\right]^{-1}\right]^{-1} \\
&= \left[\lambda\hat{\boldsymbol{A}}_m^{-1} - \epsilon\boldsymbol{I}_m\right] - \left[\lambda\hat{\boldsymbol{A}}_m^{-1} - \epsilon\boldsymbol{I}_m\right]\boldsymbol{Z}^T\left[\boldsymbol{Z}\left[\lambda\hat{\boldsymbol{A}}_m^{-1} - \epsilon\boldsymbol{I}_m\right]\boldsymbol{Z}^T + \boldsymbol{I}_m\right]^{-1}\boldsymbol{Z}\left[\lambda\hat{\boldsymbol{A}}_m^{-1} - \epsilon\boldsymbol{I}_m\right].
\end{aligned}
$$

From Lemma L.7,

$$
\left\|\left[\boldsymbol{Z}\left[\lambda\hat{\boldsymbol{A}}_m^{-1} - \epsilon\boldsymbol{I}_m\right]\boldsymbol{Z}^T + \boldsymbol{I}_m\right]^{-1}\right\| \le 1,
$$

since $H\left(\boldsymbol{Z}\left[\lambda\hat{\boldsymbol{A}}_m^{-1} - \epsilon\boldsymbol{I}_m\right]\boldsymbol{Z}^T\right)$ is positive-semi-definite, and from Lemma I.3 we have

$$
\|\hat{\boldsymbol{A}}_m^{-1}\| \le \frac{1}{\xi_{\min}}.
$$

Besides,

$$
\|\boldsymbol{Z}\|^2 = \nu_{\max}\left(\left[\frac{1}{m}\boldsymbol{\Sigma}_{\hat{\mathcal{S}}}^T\boldsymbol{\Sigma}_{\hat{\mathcal{S}}} + \epsilon\boldsymbol{I}_m\right]^{-1}\right) \le \frac{1}{\epsilon}.
$$

We deduce for the operator norm of $\boldsymbol{M}_m^{-1}$ that

$$
\|\boldsymbol{M}_m^{-1}\| \le \left(\frac{\lambda}{\xi_{\min}} + \epsilon\right) + \frac{1}{\epsilon}\left(\frac{\lambda}{\xi_{\min}} + \epsilon\right)^2.
$$

Setting $\epsilon = \frac{\lambda}{2\epsilon'} < \lambda\min\left\{\frac{1}{\xi_{\max}}, \frac{\xi_{\min}}{4}\right\}$ for $\epsilon' > \frac{1}{2}\min\left\{\frac{1}{\xi_{\max}}, \frac{\xi_{\min}}{4}\right\}$ and putting upper bounds of $\|\boldsymbol{M}_m^{-1}\|, \left\|\left[\frac{1}{m}\hat{\boldsymbol{A}}_m\boldsymbol{\Sigma}_{\hat{\mathcal{S}}}^T\boldsymbol{\Sigma}_{\hat{\mathcal{S}}} + \lambda\boldsymbol{I}_m\right]^{-1}\right\|, \left\|\left[\frac{1}{m}\boldsymbol{\Sigma}_{\hat{\mathcal{S}}}^T\boldsymbol{\Sigma}_{\hat{\mathcal{S}}}\hat{\boldsymbol{A}}_m + \lambda\boldsymbol{I}_m\right]^{-1}\right\|$ into equation 75 give

$$
\begin{aligned}
\|\boldsymbol{Q}_m(\lambda)\| &\le \frac{1}{\lambda^2}\left[\lambda + \frac{\lambda}{\epsilon'} + \lambda\left(\frac{2}{\xi_{\min}} + \frac{1}{\epsilon'}\right) + \lambda\epsilon'\left(\frac{2}{\xi_{\min}} + \frac{1}{\epsilon'}\right)^2 + \lambda\frac{8}{\xi_{\min}^2\epsilon'} + \lambda\frac{4}{\xi_{\min}^2\epsilon'^2}\right] \\
&= \frac{1}{\lambda}\left[1 + \frac{1}{\epsilon'} + \left(\frac{2}{\xi_{\min}} + \frac{1}{\epsilon'}\right) + \epsilon'\left(\frac{2}{\xi_{\min}} + \frac{1}{\epsilon'}\right)^2 + \frac{8}{\xi_{\min}^2\epsilon'} + \frac{4}{\xi_{\min}^2\epsilon'^2}\right].
\end{aligned}
$$

$\square$

**Remark 20.** *From the proof of Lemma I.1, eigenspectrum constraints on the empirical transition model matrix $\hat{\boldsymbol{A}}_m$ in Assumption 2 ensure the resolvent $\boldsymbol{Q}_m(\lambda)$ is uniformly bounded.*

**Lemma I.2.** *Under Assumptions 1 and 2, let $\lambda > 0$ and let $\boldsymbol{Q}'_m(\lambda), \boldsymbol{Q}''_m(\lambda) \in \mathbb{R}^{m\times m}$ be the following resolvents*

$$
\boldsymbol{Q}'_m(\lambda) = \left[\frac{1}{m}\hat{\boldsymbol{A}}_m\boldsymbol{\Sigma}_{\hat{\mathcal{S}}}^T\boldsymbol{\Sigma}_{\hat{\mathcal{S}}} + \lambda\boldsymbol{I}_m\right]^{-1}
$$

*and*

$$
\boldsymbol{Q}''_m(\lambda) = \left[\frac{1}{m}\boldsymbol{\Sigma}_{\hat{\mathcal{S}}}^T\boldsymbol{\Sigma}_{\hat{\mathcal{S}}}\hat{\boldsymbol{A}}_m + \lambda\boldsymbol{I}_m\right]^{-1},
$$

*where $\hat{\boldsymbol{A}}_m = \hat{\boldsymbol{U}}_n(\hat{\boldsymbol{U}}_n - \gamma\hat{\boldsymbol{V}}_n)^T \in \mathbb{R}^{m\times m}$ is the empirical transition model matrix (equation 14). Then, for all $m$, we have*

$$
\|\boldsymbol{Q}'_m(\lambda)\| \le \frac{1}{\lambda}\frac{4}{\xi_{\min}^2} \quad and \quad \|\boldsymbol{Q}''_m(\lambda)\| \le \frac{1}{\lambda}\frac{4}{\xi_{\min}^2}.
$$

*Proof.* Since the symmetric part of the empirical transition model matrix $\hat{\boldsymbol{A}}_m$ is positive-definite under Assumption 2, the matrix $\hat{\boldsymbol{A}}_m$ is non-singular. We write thus

$$
\begin{aligned}
\|\boldsymbol{Q}_m'(\lambda)\| &= \left\| \left[ \frac{1}{m}\hat{\boldsymbol{A}}_m \boldsymbol{\Sigma}_{\hat{\mathcal{S}}}^T \boldsymbol{\Sigma}_{\hat{\mathcal{S}}} + \lambda \boldsymbol{I}_m \right]^{-1} \right\| \\
&= \left\| \left[ \frac{1}{m}\boldsymbol{\Sigma}_{\hat{\mathcal{S}}}^T \boldsymbol{\Sigma}_{\hat{\mathcal{S}}} + \lambda \hat{\boldsymbol{A}}_m^{-1} \right]^{-1} \hat{\boldsymbol{A}}_m^{-1} \right\| \\
&\leq \left\| \left[ \frac{1}{m}\boldsymbol{\Sigma}_{\hat{\mathcal{S}}}^T \boldsymbol{\Sigma}_{\hat{\mathcal{S}}} + \lambda \hat{\boldsymbol{A}}_m^{-1} - \lambda \nu_{\min}\big(H(\hat{\boldsymbol{A}}_m^{-1})\big)\boldsymbol{I}_m + \lambda \nu_{\min}\big(H(\hat{\boldsymbol{A}}_m^{-1})\big)\boldsymbol{I}_m \right]^{-1} \right\| \|\hat{\boldsymbol{A}}_m^{-1}\| \\
&= \frac{1}{\lambda}\frac{1}{\nu_{\min}\big(H(\hat{\boldsymbol{A}}_m^{-1})\big)}\|\hat{\boldsymbol{A}}_m^{-1}\|.
\end{aligned}
$$

The last inequality is obtained with Lemma L.7 since $H\left( \frac{1}{m}\boldsymbol{\Sigma}_{\hat{\mathcal{S}}}^T \boldsymbol{\Sigma}_{\hat{\mathcal{S}}} + \lambda \hat{\boldsymbol{A}}_m^{-1} - \lambda \nu_{\min}\big(H(\hat{\boldsymbol{A}}_m^{-1})\big)\boldsymbol{I}_m \right)$ is positive-semi-definite. By construction of both $\hat{\boldsymbol{U}}_n$ and $\hat{\boldsymbol{V}}_n$, we have $\|\hat{\boldsymbol{U}}_n\| \leq 1$ and $\|\hat{\boldsymbol{V}}_n\| \leq 1$. We deduce that

$$
\big\|\hat{\boldsymbol{A}}_m\big\| = \big\|\hat{\boldsymbol{U}}_n(\hat{\boldsymbol{U}}_n - \gamma\hat{\boldsymbol{V}}_n)^T\big\| < 2.
$$

Since $H(\hat{\boldsymbol{A}}_m^{-1}) = [\hat{\boldsymbol{A}}_m^{-1}]^T H(\hat{\boldsymbol{A}}_m)\hat{\boldsymbol{A}}_m^{-1}$, we deduce from Ostrowski's theorem (Lemma L.5) that

$$
\nu_{\min}\big(H(\hat{\boldsymbol{A}}_m^{-1})\big) \geq \frac{\nu_{\min}\big(H(\hat{\boldsymbol{A}}_m)\big)}{\|\hat{\boldsymbol{A}}_m\|^2} \geq \frac{\xi_{\min}}{4}.
$$

Furthermore, from Lemma I.3, we have $\|\hat{\boldsymbol{A}}_m^{-1}\| \leq \frac{1}{\xi_{\min}}$. We conclude that

$$
\|\boldsymbol{Q}_m'(\lambda)\| \leq \frac{1}{\lambda}\frac{4}{\xi_{\min}^2}.
$$

With a similar reasoning, we can find the same upper bound for $\|\boldsymbol{Q}_m''(\lambda)\|$. $\qquad\square$

**Lemma I.3.** *Let $\hat{\boldsymbol{A}}_m = \hat{\boldsymbol{U}}_n(\hat{\boldsymbol{U}}_n - \gamma\hat{\boldsymbol{V}}_n)^T$ be the empirical transition model matrix defined in equation 14. Under Assumption 2, for all $m$, we have*

$$
\|\hat{\boldsymbol{A}}_m^{-1}\| \leq \frac{1}{\xi_{\min}}.
$$

*Proof.* We rewrite $\hat{\boldsymbol{A}}_m$ as

$$
\hat{\boldsymbol{A}}_m^{-1} = \Big[ \big[ \hat{\boldsymbol{A}}_m - \nu_{\min}\big(H(\hat{\boldsymbol{A}}_m)\big)\boldsymbol{I}_m \big] + \nu_{\min}\big(H(\hat{\boldsymbol{A}}_m)\big)\boldsymbol{I}_m \Big]^{-1}.
$$

Since the matrix $H\Big( \big[ \hat{\boldsymbol{A}}_m - \nu_{\min}\big(H(\hat{\boldsymbol{A}}_m)\big)\boldsymbol{I}_m \big] \Big)$ is positive-semi-definite, we apply Lemma L.7 on $\hat{\boldsymbol{A}}_m^{-1}$ to get

$$
\|\hat{\boldsymbol{A}}_m^{-1}\| \leq \frac{1}{\nu_{\min}\big(H(\hat{\boldsymbol{A}}_m)\big)} \leq \frac{1}{\xi_{\min}}.
$$

$\qquad\square$

**Lemma I.4.** *Under Assumption 1 and 2, let $\lambda > 0$ and let $\boldsymbol{Q}_m(\lambda) \in \mathbb{R}^{n \times n}$ be the resolvent defined in equation 13 as*

$$
\boldsymbol{Q}_m(\lambda) = \left[ \frac{1}{m}(\hat{\boldsymbol{U}}_n - \gamma\hat{\boldsymbol{V}}_n)^T\boldsymbol{\Sigma}_{\hat{\mathcal{S}}}^T\boldsymbol{\Sigma}_{\hat{\mathcal{S}}}\hat{\boldsymbol{U}}_n + \lambda\boldsymbol{I}_n \right]^{-1}.
$$

*Then there exists a real $K > 0$ such that, for all $m$, we have*

$$
\left\| \frac{1}{\sqrt{m}}\boldsymbol{\Sigma}_{\hat{\mathcal{S}}}\hat{\boldsymbol{U}}_n\boldsymbol{Q}_m(\lambda) \right\| \leq K
$$

*and*

$$
\left\| \frac{1}{\sqrt{m}}\boldsymbol{Q}_m(\lambda)(\hat{\boldsymbol{U}}_n - \gamma\hat{\boldsymbol{V}}_n)^T\boldsymbol{\Sigma}_{\hat{\mathcal{S}}}^T \right\| \leq 2K.
$$

*Proof.* From Lemma I.1, we know there exists a real $K > 0$ such that, for all $m$, we have $\|\boldsymbol{Q}_m(\lambda)\| \leq K$. Since the symmetric part of the empirical transition model matrix $\hat{\boldsymbol{A}}_m = \hat{\boldsymbol{U}}_n(\hat{\boldsymbol{U}}_n - \gamma\hat{\boldsymbol{V}}_n)^T$ (equation 14) is positive-definite under Assumption 2, the matrix $\hat{\boldsymbol{A}}_m$ is non-singular. Furthermore, from Lemma I.3 we have $\|\hat{\boldsymbol{A}}_m^{-1}\| \leq \frac{1}{\xi_{\min}}$, and both $\|\hat{\boldsymbol{U}}_n\|$ and $\|\hat{\boldsymbol{V}}_n\|$ are upper bounded by 1. We deduce that

$$
\begin{aligned}
\left\|\frac{1}{\sqrt{m}}\boldsymbol{\Sigma}_{\hat{\mathcal{S}}}\hat{\boldsymbol{U}}_n\boldsymbol{Q}_m(\lambda)\right\| &= \left\|\frac{1}{m}\boldsymbol{Q}_m(\lambda)^T\hat{\boldsymbol{U}}_n^T\boldsymbol{\Sigma}_{\hat{\mathcal{S}}}^T\boldsymbol{\Sigma}_{\hat{\mathcal{S}}}\hat{\boldsymbol{U}}_n\boldsymbol{Q}_m(\lambda)\right\|^{\frac{1}{2}} \\
&= \left\|\frac{1}{m}\boldsymbol{Q}_m(\lambda)^T\hat{\boldsymbol{U}}_n^T\hat{\boldsymbol{A}}_m^{-1}\hat{\boldsymbol{U}}_n(\hat{\boldsymbol{U}}_n - \gamma\hat{\boldsymbol{V}}_n)^T\boldsymbol{\Sigma}_{\hat{\mathcal{S}}}^T\boldsymbol{\Sigma}_{\hat{\mathcal{S}}}\hat{\boldsymbol{U}}_n\boldsymbol{Q}_m(\lambda)\right\|^{\frac{1}{2}} \\
&= \left\|\boldsymbol{Q}_m(\lambda)^T\hat{\boldsymbol{U}}_n^T\hat{\boldsymbol{A}}_m^{-1}\hat{\boldsymbol{U}}_n[\boldsymbol{I}_n - \lambda\boldsymbol{Q}_m(\lambda)]\right\|^{\frac{1}{2}} \\
&\leq \sqrt{\frac{K(1+K)}{\xi_{\min}}}.
\end{aligned}
$$

Similarly, we have

$$
\begin{aligned}
\left\|\frac{1}{\sqrt{m}}\boldsymbol{Q}_m(\lambda)(\hat{\boldsymbol{U}}_n - \gamma\hat{\boldsymbol{V}}_n)^T\boldsymbol{\Sigma}_{\hat{\mathcal{S}}}^T\right\| &\\
= \left\|\frac{1}{m}\boldsymbol{Q}_m(\lambda)(\hat{\boldsymbol{U}}_n - \gamma\hat{\boldsymbol{V}}_n)^T\boldsymbol{\Sigma}_{\hat{\mathcal{S}}}^T\boldsymbol{\Sigma}_{\hat{\mathcal{S}}}\hat{\boldsymbol{U}}_n(\hat{\boldsymbol{U}}_n - \gamma\hat{\boldsymbol{V}}_n)^T\hat{\boldsymbol{A}}_m^{-1}(\hat{\boldsymbol{U}}_n - \gamma\hat{\boldsymbol{V}}_n)\boldsymbol{Q}_m(\lambda)^T\right\|^{\frac{1}{2}} & \\
= \left\|[\boldsymbol{I}_n - \lambda\boldsymbol{Q}_m(\lambda)](\hat{\boldsymbol{U}}_n - \gamma\hat{\boldsymbol{V}}_n)^T\hat{\boldsymbol{A}}_m^{-1}(\hat{\boldsymbol{U}}_n - \gamma\hat{\boldsymbol{V}}_n)\boldsymbol{Q}_m(\lambda)^T\right\|^{\frac{1}{2}} & \\
\leq 2\sqrt{\frac{K(1+K)}{\xi_{\min}}}. &
\end{aligned}
$$

$\square$

## J  ABOUT THE EXISTENCE, POSITIVENESS, AND UNIQUENESS OF $\delta$

This section is dedicated to prove that the fixed-point solution $\delta$ of equation 17 is unique and positive under Assumptions 1 and 2. This result is proven in the following Lemma.

**Lemma J.1.** *Under Assumptions 1 and 2, for all $m$, let $\delta$ be the solution to the fixed-point equation 17 defined as*

$$
\delta = \frac{1}{m}\operatorname{Tr}\left((\hat{\boldsymbol{U}}_n - \gamma\hat{\boldsymbol{V}}_n)^T\boldsymbol{\Phi}_{\hat{\mathcal{S}}}\hat{\boldsymbol{U}}_n\left[\frac{N}{m}\frac{1}{1+\delta}(\hat{\boldsymbol{U}}_n - \gamma\hat{\boldsymbol{V}}_n)^T\boldsymbol{\Phi}_{\hat{\mathcal{S}}}\hat{\boldsymbol{U}}_n + \lambda\boldsymbol{I}_n\right]^{-1}\right).
$$

*Then $\delta$ exists, is positive, and is unique.*

*Proof.* For ease of notations, we define the matrix $\boldsymbol{B}_n = (\hat{\boldsymbol{U}}_n - \gamma\hat{\boldsymbol{V}}_n)^T\boldsymbol{\Phi}_{\hat{\mathcal{S}}}\hat{\boldsymbol{U}}_n$. The proof is based on the use of Lemma L.8 on the mapping $f : \delta \mapsto \frac{1}{m}\operatorname{Tr}(\boldsymbol{B}_n\bar{\boldsymbol{Q}}_m(\delta))$. To apply Lemma L.8, we need to show *i.* $f$ is positive on $[0, \infty)$, *ii.* $f$ is monotonically increasing, *iii.* $f$ is scalable, and *iv.* $f$ admits $x_0 \in [0, \infty)$ such that $x_0 \geq f(x_0)$. Following this plan, we are going to show first *i.*, i.e., $f(\delta) > 0$ for all $\delta > 0$. By denoting $\nu_j(\boldsymbol{B}_n\bar{\boldsymbol{Q}}_m(\delta))$ the j-th eigenvalues of the matrix $\boldsymbol{B}_n\bar{\boldsymbol{Q}}_m(\delta)$,

we have

$$\nu_j\big(\boldsymbol{B}_n\bar{\boldsymbol{Q}}_m(\delta)\big) = \nu_j\left(\boldsymbol{B}_n\left[\frac{N}{m}\frac{1}{1+\delta}\boldsymbol{B}_n + \lambda\boldsymbol{I}_n\right]^{-1}\right)$$

$$= \nu_j(\boldsymbol{B}_n)\nu_j\left(\left[\frac{N}{m}\frac{1}{1+\delta}\boldsymbol{B}_n + \lambda\boldsymbol{I}_n\right]^{-1}\right) \qquad \text{(from the Schur decomposition of } \boldsymbol{B}_n\text{)}$$

$$= \frac{\nu_j(\boldsymbol{B}_n)}{\frac{N}{m}\frac{1}{1+\delta}\nu_j(\boldsymbol{B}_n) + \lambda}$$

$$= \frac{1}{\left|\frac{N}{m}\frac{1}{1+\delta}\nu_j(\boldsymbol{B}_n) + \lambda\right|^2}\left(\frac{N}{m}\frac{1}{1+\delta}|\nu_j(\boldsymbol{B}_n)|^2 + \lambda\nu_j(\boldsymbol{B}_n)\right).$$

$$(76)$$

Let $\hat{\boldsymbol{A}}_m = \hat{\boldsymbol{U}}_n(\hat{\boldsymbol{U}}_n - \gamma\hat{\boldsymbol{V}}_n)^T$ be the transition model matrix defined in equation 14, and $\bar{\boldsymbol{Z}}\bar{\boldsymbol{Z}}^T$ be the Cholesky decompositon of $\boldsymbol{\Phi}_{\hat{\mathcal{S}}}$. From the Weinstein–Aronszajn identity (Lemma L.6), the matrices $\boldsymbol{B}_n = (\hat{\boldsymbol{U}}_n - \gamma\hat{\boldsymbol{V}}_n)^T\boldsymbol{\Phi}_{\hat{\mathcal{S}}}\hat{\boldsymbol{U}}_n$ and $\bar{\boldsymbol{Z}}^T\hat{\boldsymbol{A}}_m\bar{\boldsymbol{Z}}$ share the same non-zero eigenvalues. Under Assumption 2, the matrix $H(\bar{\boldsymbol{Z}}^T\hat{\boldsymbol{A}}_m\bar{\boldsymbol{Z}})$ is at least semi-positive-definite, which implies that non-zero real parts of eigenvalues of $\bar{\boldsymbol{Z}}^T\hat{\boldsymbol{A}}_m\bar{\boldsymbol{Z}}$ are positive. We deduce that $\mathrm{Re}\big(\nu_j(\boldsymbol{B}_n)\big) \geq 0$, for all $j \in [m]$. As a consequence,

$$f(\delta) = \frac{1}{m}\,\mathrm{Tr}\big(\boldsymbol{B}_n\bar{\boldsymbol{Q}}_m(\delta)\big)$$

$$= \frac{1}{m}\sum_{j=1}^{n}\frac{1}{\left|\frac{N}{m}\frac{1}{1+\delta}\nu_j(\boldsymbol{B}_n) + \lambda\right|^2}\left(\frac{N}{m}\frac{1}{1+\delta}|\nu_j(\boldsymbol{B}_n)|^2 + \lambda\nu_j(\boldsymbol{B}_n)\right)$$

$$= \frac{1}{m}\sum_{j=1}^{n}\frac{1}{\left|\frac{N}{m}\frac{1}{1+\delta}\nu_j(\boldsymbol{B}_n) + \lambda\right|^2}\left(\frac{N}{m}\frac{1}{1+\delta}|\nu_j(\boldsymbol{B}_n)|^2 + \lambda\,\mathrm{Re}\big(\nu_j(\boldsymbol{B}_n)\big)\right) \qquad (77)$$

$$> 0.$$

To prove $ii.$, i.e., $f$ is monotonically increasing on $[0,\infty)$, we show the derivative $f'$ of $f$ is positive on $[0,\infty)$. Let $\delta > 0$,

$$f'(\delta) = \frac{1}{m}\left(\sum_{j=1}^{n}\frac{\nu_j(\boldsymbol{B}_n)}{\frac{N}{m}\frac{1}{1+\delta}\nu_j(\boldsymbol{B}_n) + \lambda}\right)'$$

$$= \frac{1}{m}\sum_{j=1}^{n}\frac{\frac{N}{m}\frac{1}{(1+\delta)^2}}{\left(\frac{N}{m}\frac{1}{1+\delta}\nu_j(\boldsymbol{B}_n) + \lambda\right)^2}\nu_j(\boldsymbol{B}_n)^2$$

$$= \frac{1}{m}\sum_{j=1}^{n}\frac{\frac{N}{m}\frac{1}{(1+\delta)^2}}{\left|\frac{N}{m}\frac{1}{1+\delta}\nu_j(\boldsymbol{B}_n) + \lambda\right|^4}\left(\frac{N}{m}\frac{1}{(1+\delta)}|\nu_j(\boldsymbol{B}_n)|^2 + \lambda\nu_j(\boldsymbol{B}_n)\right)^2$$

$$= \frac{1}{m}\sum_{j=1}^{n}\frac{\frac{N}{m}\frac{1}{(1+\delta)^2}}{\left|\frac{N}{m}\frac{1}{1+\delta}\nu_j(\boldsymbol{B}_n) + \lambda\right|^4}\left(\frac{N^2}{m^2}\frac{1}{(1+\delta)^2}|\nu_j(\boldsymbol{B}_n)|^4 + 2\lambda\frac{N}{m}\frac{1}{1+\delta}|\nu_j(\boldsymbol{B}_n)|^2\nu_j(\boldsymbol{B}_n) + \lambda^2\nu_j(\boldsymbol{B}_n)^2\right)$$

$$= \underbrace{\frac{1}{m}\sum_{j=1}^{n}\frac{\frac{N}{m}\frac{1}{(1+\delta)^2}}{\left|\frac{N}{m}\frac{1}{1+\delta}\nu_j(\boldsymbol{B}_n) + \lambda\right|^4}\left(\frac{N^2}{m^2}\frac{1}{(1+\delta)^2}|\nu_j(\boldsymbol{B}_n)|^4 + 2\lambda\frac{N}{m}\frac{1}{1+\delta}|\nu_j(\boldsymbol{B}_n)|^2\,\mathrm{Re}\big(\nu_j(\boldsymbol{B}_n)\big)\right)}_{(1)}$$

$$+ \underbrace{\sum_{j=1}^{n}\lambda^2\frac{\frac{N}{m}\frac{1}{(1+\delta)^2}}{\left|\frac{N}{m}\frac{1}{1+\delta}\nu_j(\boldsymbol{B}_n) + \lambda\right|^4}\,\mathrm{Re}\big(\nu_j(\boldsymbol{B}_n^2)\big)}_{(2)}$$

Since real parts of eigenvalues of $\boldsymbol{B}_n$ are positive, $(1)$ is clearly positive. Since $\mathrm{Tr}(\boldsymbol{B}_n^2) > 0$ (Lemma J.2) and thus $(2)$ is positive, we can conclude $ii.$. We can use a similar proof for the scalability in $iii.$, i.e., $\alpha f(\delta) > f(\alpha \delta),\ \forall \alpha > 1$. Let $\alpha > 1$ and $\delta > 0$,

$$\alpha f(\delta) - f(\alpha \delta) = \alpha \frac{1}{m} \mathrm{Tr}\big(\boldsymbol{B}_n \bar{\boldsymbol{Q}}_m(\delta)\big) - \frac{1}{m} \mathrm{Tr}\big(\boldsymbol{B}_n \bar{\boldsymbol{Q}}_m(\alpha \delta)\big) \tag{78}$$

$$= \frac{1}{m} \mathrm{Tr}\big(\boldsymbol{B}_n \big[\alpha \bar{\boldsymbol{Q}}_m(\delta) - \bar{\boldsymbol{Q}}_m(\alpha \delta)\big]\big) \tag{79}$$

$$= \frac{1}{m} \mathrm{Tr}\left(\alpha \boldsymbol{B}_n \bar{\boldsymbol{Q}}_m(\delta)\left[\frac{N}{m}\left(\frac{1}{1+\alpha\delta} - \frac{1}{\alpha(1+\delta)}\right)\boldsymbol{B}_n + \left(\lambda - \frac{\lambda}{\alpha}\right)\boldsymbol{I}_n\right]\bar{\boldsymbol{Q}}_m(\alpha\delta)\right) \tag{80}$$

$$= \underbrace{\alpha \frac{1}{m}\frac{N}{m}\left(\frac{1}{1+\alpha\delta} - \frac{1}{\alpha(1+\delta)}\right)}_{>0} \underbrace{\mathrm{Tr}\big(\boldsymbol{B}_n \bar{\boldsymbol{Q}}_m(\delta)\boldsymbol{B}_n \bar{\boldsymbol{Q}}_m(\alpha\delta)\big)}_{(1)}$$

$$+ \underbrace{\alpha \frac{1}{m}\left(\lambda - \frac{\lambda}{\alpha}\right)}_{>0}\underbrace{\mathrm{Tr}\big(\boldsymbol{B}_n \bar{\boldsymbol{Q}}_m(\delta)\bar{\boldsymbol{Q}}_m(\alpha\delta)\big)}_{(2)}. \tag{81}$$

To prove $iii.$, we can show that both $(1)$ and $(2)$ in equation 81 are positive. We prove in $ii.$ that $\mathrm{Tr}\big(\boldsymbol{B}_n \bar{\boldsymbol{Q}}_m(\delta')\boldsymbol{B}_n \bar{\boldsymbol{Q}}_m(\delta)\big) > 0$ for any $\delta' > \delta$. Since $\alpha\delta > \delta$, we also deduce $(1)$ is positive. For $(2)$, we can write

$$\mathrm{Tr}\big(\boldsymbol{B}_n \bar{\boldsymbol{Q}}_m(\delta)\bar{\boldsymbol{Q}}_m(\alpha\delta)\big)$$

$$= \sum_{j=1}^{n} \nu_j\big(\boldsymbol{B}_n \bar{\boldsymbol{Q}}_m(\delta)\bar{\boldsymbol{Q}}_m(\alpha\delta)\big)$$

$$= \sum_{j=1}^{n} \frac{\nu_j(\boldsymbol{B}_n)}{\left(\frac{N}{m}\frac{1}{1+\delta}\nu_j(\boldsymbol{B}_n) + \lambda\right)\left(\frac{N}{m}\frac{1}{1+\alpha\delta}\nu_j(\boldsymbol{B}_n) + \lambda\right)}$$

$$= \sum_{j=1}^{n} c_j \left(\left(\frac{N^2}{m^2}\frac{|\nu_j(\boldsymbol{B}_n)|^2}{(1+\delta)(1+\alpha\delta)} + \lambda^2\right)\mathrm{Re}\big(\nu_j(\boldsymbol{B}_n)\big) + \frac{N}{m}\left(\frac{\lambda}{1+\delta} + \frac{\lambda}{1+\alpha\delta}\right)|\nu_j(\boldsymbol{B}_n)|^2\right)$$

$$> 0,$$

where

$$c_j = \frac{1}{\left|\left(\frac{N}{m}\frac{1}{1+\delta}\nu_j(\boldsymbol{B}_n) + \lambda\right)\left(\frac{N}{m}\frac{1}{1+\alpha\delta}\nu_j(\boldsymbol{B}_n) + \lambda\right)\right|^2}.$$

In order to apply Lemma L.8, we still need to demonstrate $iv.$, i.e., $f$ admits $x_0 \in [0, \infty)$ such that $x_0 \geq f(x_0)$. To prove $iv.$, it is sufficient to notice that if $f$ is bounded, i.e., $\forall \delta, f(\delta) \leq C$. Let $\delta > 0$, we have

$$f(\delta) = \frac{1}{m}\mathrm{Tr}\big(\boldsymbol{B}_n \bar{\boldsymbol{Q}}_m(\delta)\big) = \frac{1}{m}\mathrm{Tr}\big((\hat{\boldsymbol{U}}_n - \gamma\hat{\boldsymbol{V}}_n)^T \boldsymbol{\Phi}_{\hat{\mathcal{S}}}\hat{\boldsymbol{U}}_n \bar{\boldsymbol{Q}}_m(\delta)\big)$$

$$= \frac{1}{m}\mathrm{Tr}\big(\boldsymbol{\Phi}_{\hat{\mathcal{S}}}\hat{\boldsymbol{U}}_n \bar{\boldsymbol{Q}}_m(\delta)(\hat{\boldsymbol{U}}_n - \gamma\hat{\boldsymbol{V}}_n)^T\big)$$

$$\leq \frac{1}{m}\mathrm{Tr}(\boldsymbol{\Phi}_{\hat{\mathcal{S}}})\|\hat{\boldsymbol{U}}_n \bar{\boldsymbol{Q}}_m(\delta)(\hat{\boldsymbol{U}}_n - \gamma\hat{\boldsymbol{V}}_n)^T\|$$

$$\leq \frac{2}{n}\mathrm{Tr}(\boldsymbol{\Phi}_{\hat{\mathcal{S}}})\|\bar{\boldsymbol{Q}}_m(\delta)\|$$

$$= \mathcal{O}(1),$$

where we used for the first inequality $|\mathrm{Tr}(\boldsymbol{AB})| \leq \|\boldsymbol{B}\|\mathrm{Tr}(\boldsymbol{A})$ for non-negative definite matrix $\boldsymbol{A}$. The last inequality is obtained since $\frac{1}{m}\mathrm{Tr}(\boldsymbol{\Phi}_{\hat{\mathcal{S}}})$ is uniformly bounded under Assumptions 1 and 2 (see equation 30). Furthermore, both $\|\hat{\boldsymbol{U}}_n\|$ and $\|\hat{\boldsymbol{V}}_n\|$ are upper bounded by 1 and, with a similar proof than for Lemma I.1, we can show there exists a real $K_{\bar{\boldsymbol{Q}}} > 0$ such that, for all $m$ and for all $\delta \in [0, \infty)$, we have $\|\bar{\boldsymbol{Q}}_m(\delta)\| \leq K_{\bar{\boldsymbol{Q}}}$. Since all hypotheses required on $f$ to apply Lemma L.8 are satisfied, we can apply this Lemma which concludes the proof. $\qquad \square$

**Lemma J.2.** *We have*

$$\mathrm{Tr}\big((\hat{\boldsymbol{U}}_n - \gamma\hat{\boldsymbol{V}}_n)^T\boldsymbol{\Phi}_{\hat{\mathcal{S}}}\hat{\boldsymbol{U}}_n(\hat{\boldsymbol{U}}_n - \gamma\hat{\boldsymbol{V}}_n)^T\boldsymbol{\Phi}_{\hat{\mathcal{S}}}\hat{\boldsymbol{U}}_n\big) > 0.$$

*Proof.* Let $\boldsymbol{A} = \hat{\boldsymbol{U}}_n(\hat{\boldsymbol{U}}_n - \gamma\hat{\boldsymbol{V}}_n)^T$. We denote by $S(\boldsymbol{A}) = \frac{\boldsymbol{A}-\boldsymbol{A}^T}{2}$ the skew-symmetric part of $\boldsymbol{A}$. We have

$$\mathrm{Tr}\big((\hat{\boldsymbol{U}}_n - \gamma\hat{\boldsymbol{V}}_n)^T\boldsymbol{\Phi}_{\hat{\mathcal{S}}}\hat{\boldsymbol{U}}_n(\hat{\boldsymbol{U}}_n - \gamma\hat{\boldsymbol{V}}_n)^T\boldsymbol{\Phi}_{\hat{\mathcal{S}}}\hat{\boldsymbol{U}}_n\big)$$
$$= \mathrm{Tr}\big(\boldsymbol{\Phi}_{\hat{\mathcal{S}}}\boldsymbol{A}\boldsymbol{\Phi}_{\hat{\mathcal{S}}}\boldsymbol{A}\big)$$
$$= \mathrm{Tr}\big(\boldsymbol{\Phi}_{\hat{\mathcal{S}}}\boldsymbol{A}\boldsymbol{\Phi}_{\hat{\mathcal{S}}}H(\boldsymbol{A})\big) + \mathrm{Tr}\big(\boldsymbol{\Phi}_{\hat{\mathcal{S}}}\boldsymbol{A}\boldsymbol{\Phi}_{\hat{\mathcal{S}}}S(\boldsymbol{A})\big)$$
$$= \mathrm{Tr}\big(\boldsymbol{\Phi}_{\hat{\mathcal{S}}}H(\boldsymbol{A})\boldsymbol{\Phi}_{\hat{\mathcal{S}}}H(\boldsymbol{A})\big) + \mathrm{Tr}\big(\boldsymbol{\Phi}_{\hat{\mathcal{S}}}S(\boldsymbol{A})\boldsymbol{\Phi}_{\hat{\mathcal{S}}}H(\boldsymbol{A})\big) + \mathrm{Tr}\big(\boldsymbol{\Phi}_{\hat{\mathcal{S}}}H(\boldsymbol{A})\boldsymbol{\Phi}_{\hat{\mathcal{S}}}S(\boldsymbol{A})\big) + \mathrm{Tr}\big(\boldsymbol{\Phi}_{\hat{\mathcal{S}}}S(\boldsymbol{A})\boldsymbol{\Phi}_{\hat{\mathcal{S}}}S(\boldsymbol{A})\big)$$
$$= \mathrm{Tr}\big(\boldsymbol{\Phi}_{\hat{\mathcal{S}}}H(\boldsymbol{A})\boldsymbol{\Phi}_{\hat{\mathcal{S}}}H(\boldsymbol{A})\big) + \mathrm{Tr}\big(\boldsymbol{\Phi}_{\hat{\mathcal{S}}}S(\boldsymbol{A})\boldsymbol{\Phi}_{\hat{\mathcal{S}}}S(\boldsymbol{A})\big) > 0.$$

$\square$

**Lemma J.3.** *Under Assumptions 1 and 2, let $\delta$ be the correction factor defined in equation 17. $\delta$ is a decreasing function with respect to $N$.*

*Proof.* For ease of notations, we define the matrix $\boldsymbol{B}_n = (\hat{\boldsymbol{U}}_n - \gamma\hat{\boldsymbol{V}}_n)^T\boldsymbol{\Phi}_{\hat{\mathcal{S}}}\hat{\boldsymbol{U}}_n$ and we denote by $\bar{\boldsymbol{Q}}_m$ the resolvent $\bar{\boldsymbol{Q}}_m(\lambda)$. The derivative of $\delta$ as function of $N$ is denoted as $\delta'(N)$ and defined as

$$\delta'(N) = -\frac{1}{m}\frac{\frac{\frac{1}{m}\mathrm{Tr}(\boldsymbol{B}_n\bar{\boldsymbol{Q}}_m\boldsymbol{B}_n\bar{\boldsymbol{Q}}_m)}{(1+\delta)}}{1 - \frac{N}{m}\frac{\frac{1}{m}\mathrm{Tr}(\boldsymbol{B}_n\bar{\boldsymbol{Q}}_m\boldsymbol{B}_n\bar{\boldsymbol{Q}}_m)}{(1+\delta)^2}}$$

For all $N$, we have $\delta'(N) \leq 0$ since $\frac{\frac{1}{m}\mathrm{Tr}(\boldsymbol{B}_n\bar{\boldsymbol{Q}}_m\boldsymbol{B}_n\bar{\boldsymbol{Q}}_m)}{(1+\delta)} > 0$ and $\frac{N}{m}\frac{\frac{1}{m}\mathrm{Tr}(\boldsymbol{B}_n\bar{\boldsymbol{Q}}_m\boldsymbol{B}_n\bar{\boldsymbol{Q}}_m)}{(1+\delta)^2} < 1$ using a similar reasoning than for equation 48. $\square$

**Lemma J.4.** *Under Assumptions 1 and 2, let $\delta$ be the correction factor defined in equation 17. $\delta$ is a decreasing function with respect to $\lambda$.*

*Proof.* For ease of notations, we define the matrix $\boldsymbol{B}_n = (\hat{\boldsymbol{U}}_n - \gamma\hat{\boldsymbol{V}}_n)^T\boldsymbol{\Phi}_{\hat{\mathcal{S}}}\hat{\boldsymbol{U}}_n$ and we denote by $\bar{\boldsymbol{Q}}_m$ the resolvent $\bar{\boldsymbol{Q}}_m(\lambda)$. The derivative of $\delta$ as function of $\lambda$ is denoted as $\delta'(\lambda)$ and defined as

$$\delta'(\lambda) = -\frac{1}{m}\mathrm{Tr}(\bar{\boldsymbol{Q}}_m\boldsymbol{B}_n\bar{\boldsymbol{Q}}_m)$$

For all $\lambda$, we have $\delta'(\lambda) \leq 0$ using a similar reasoning than for $iii.$ in Lemma J.1. $\square$

## K CONCENTRATION RESULTS

The following section is dedicated to a set of concentration results used for the proofs of Theorems. Preliminary results yield a concentration of measure properties for the random feature matrix $\boldsymbol{\Sigma}_{\hat{\mathcal{S}}} \in \mathbb{R}^{N\times m}$, which stem from the concentration inequality of Lemma L.1 for Lipschitz applications of a Gaussian vector. Essentially, the guideline of the proofs involves the following steps; given $\boldsymbol{W}_{ij} = \varphi(\tilde{\boldsymbol{W}}_{ij})$, for which $\tilde{\boldsymbol{W}}_{ij} \sim \mathcal{N}(0,1)$ and $\varphi$ a Lipschitz function, the normal concentration of $\tilde{\boldsymbol{W}}$ is transferred to $\boldsymbol{W}$. This process induces a normal concentration of the random vector $\sigma(\boldsymbol{w}^T\hat{\boldsymbol{S}})$, for $\boldsymbol{w} = \varphi(\tilde{\boldsymbol{w}})$ and $\boldsymbol{w} \sim \mathcal{N}(\boldsymbol{0}, \boldsymbol{I}_d)$, and of the matrix $\boldsymbol{\Sigma}_{\hat{\mathcal{S}}}$. This implies that Lipschitz functionals of $\sigma(\boldsymbol{w}^T\hat{\boldsymbol{S}})$ or $\boldsymbol{\Sigma}_{\hat{\mathcal{S}}}$ also concentrate. As highlighted earlier, these concentration results have multiple consequences on convergence of random variables, and are traditionally employed in Random Matrix theory and in Theorem 5.1. We start by revisiting Lemma K.1 and Lemma K.2, which are derived from Lemma L.1 and that were previously introduced in Louart et al. (2018). Subsequently, we provide intermediary Lemma K.3 and Lemma K.4 to reach the principal results of this section articulated by Lemma K.5 and Lemma K.6, which are employed in proof of Theorems. In the remainder of this section, we denote by $\|\cdot\|_F$ the Frobenius norm of a matrix.

**Lemma K.1.** *Let $\boldsymbol{\sigma} : \mathbb{R} \to \mathbb{R}$ be a $K_{\boldsymbol{\sigma}}$-Lipschitz continuous function, let $\boldsymbol{X} \in \mathbb{R}^{d \times m}$ be a matrix, and let $\boldsymbol{w} = \varphi(\tilde{\boldsymbol{w}})$ be a vector for which $\varphi : \mathbb{R} \to \mathbb{R}$ is a $K_{\varphi}$-Lipschitz continuous function and $\tilde{\boldsymbol{w}} \sim \mathcal{N}(\boldsymbol{0}, \boldsymbol{I}_d)$. Let*

$$t_0 = |\sigma(0)| + K_{\boldsymbol{\sigma}} K_{\varphi} \|\boldsymbol{X}\| \sqrt{\frac{d}{m}}.$$

*Then, for all $t \geq 4t_0$, we have*

$$\Pr\left( \left\| \frac{1}{\sqrt{m}} \sigma(\boldsymbol{w}^T \boldsymbol{X}) \right\| \geq t \right) \leq C e^{-\frac{cmt^2}{2K_{\boldsymbol{\sigma}}^2 K_{\varphi}^2 \|\boldsymbol{X}\|^2}},$$

*for some $C, c > 0$ are independent of all other parameters.*

*Proof.* The proof of this Lemma can be found in the first half of proof of Louart et al. (2018, Lemma 2), and is based on Lemma L.1. □

**Corollary K.1.1.** *(Louart et al., 2018, Remark 2) Let $\boldsymbol{X} \in \mathbb{R}^{d \times m}$ and let $\boldsymbol{\Sigma_X} = \sigma(\boldsymbol{WX}) \in \mathbb{R}^{N \times m}$ be its random features matrix defined as in equation 5. For all $t \geq 4t_0$, we have*

$$\Pr\left( \left\| \frac{1}{m} \boldsymbol{\Sigma_X} \right\| \geq t \right) \leq CN e^{-\frac{cm^2 t^2}{2N \|\boldsymbol{X}\|^2}},$$

*where $t_0 = |\sigma(0)| + \|\boldsymbol{X}\| \sqrt{\frac{d}{m}}$.*

From the previous Lemma, we deduce the following key concentration result.

**Lemma K.2.** *(Louart et al., 2018, Lemma 2) Let $\boldsymbol{\sigma} : \mathbb{R} \to \mathbb{R}$ be a $K_{\boldsymbol{\sigma}}$-Lipschitz continuous function, let $\boldsymbol{X} \in \mathbb{R}^{d \times m}$ be a matrix, and let $\boldsymbol{w} = \varphi(\tilde{\boldsymbol{w}})$ be a vector for which $\varphi : \mathbb{R} \to \mathbb{R}$ is a $K_{\varphi}$-Lipschitz continuous function and $\tilde{\boldsymbol{w}} \sim \mathcal{N}(\boldsymbol{0}, \boldsymbol{I}_d)$. Let $\boldsymbol{A} \in \mathbb{R}^{m \times m}$ be a matrix independent of $\boldsymbol{w}$ such that $\|\boldsymbol{A}\| \leq K_{\boldsymbol{A}}$. Then, we have*

$$\Pr\left( \left| \frac{1}{m} \sigma(\boldsymbol{w}^T \boldsymbol{X})^T \boldsymbol{A} \sigma(\boldsymbol{w}^T \boldsymbol{X}) - \frac{1}{m} \operatorname{Tr}\left( \boldsymbol{A} \mathbb{E}\left[ \sigma(\boldsymbol{w}^T \boldsymbol{X}) \sigma(\boldsymbol{w}^T \boldsymbol{X})^T \right] \right) \right| > t \right)$$

$$\leq C e^{-\frac{cm}{2K_{\boldsymbol{\sigma}}^2 K_{\varphi}^2 \|\boldsymbol{X}\|^2} \min\left( \frac{t^2}{2^6 t_0^2 K_{\boldsymbol{A}}^2}, \frac{t}{K_{\boldsymbol{A}}} \right)},$$

*for $t_0 = |\sigma(0)| + \sqrt{\frac{d}{m}} K_{\boldsymbol{\sigma}} K_{\varphi} \|\boldsymbol{X}\|$, and $c, C \in \mathbb{R}$ independent of all other parameters.*

**Lemma K.3.** *Let $f : \mathbb{R}^{N \times d} \to \mathbb{R}$, $\boldsymbol{W} \mapsto f(\boldsymbol{W})$ be a $K_f$-Lipschitz function with respect to the Frobenius norm for which $\boldsymbol{W} = \varphi(\tilde{\boldsymbol{W}})$ is the matrix defined in equation 5. Then, we have*

$$\Pr\left( \left| f(\boldsymbol{W}) - \mathbb{E}[f(\boldsymbol{W})] \right| > t \right) \leq C e^{-\frac{ct^2}{K_f^2 K_{\varphi}^2}},$$

*for some $C, c > 0$.*

*Proof.* The vectorization of $\tilde{\boldsymbol{W}}$, $\operatorname{vec}(\tilde{\boldsymbol{W}}) = \left[ \tilde{\boldsymbol{W}}_{11}, \cdots, \tilde{\boldsymbol{W}}_{nd} \right] \in \mathbb{R}^{N \times d}$ is a Gaussian vector. A $K_f$-Lipschitz function $f$ of $\boldsymbol{W}$ with respect to the Frobenius norm is also a $K_f$-Lipschitz function of $\operatorname{vec}(\boldsymbol{W})$ with respect to the Euclidean norm. Applying Lemma L.1 gives

$$\Pr\left( \left| f(\boldsymbol{W}) - \mathbb{E}[f(\boldsymbol{W})] \right| > t \right) = \Pr\left( \left| f\big(\varphi(\tilde{\boldsymbol{W}})\big) - \mathbb{E}\left[ f\big(\varphi(\tilde{\boldsymbol{W}})\big) \right] \right| > t \right) \leq C e^{-\frac{ct^2}{K_{\varphi}^2 K_f^2}},$$

for some $C, c > 0$. □

**Lemma K.4.** *Under Assumptions 1 and 2, let $\lambda > 0$, let $\boldsymbol{W} \in \mathbb{R}^{N \times d}$, and let the resolvent*

$$\boldsymbol{Q}_m(\boldsymbol{W}) = \left[ \frac{1}{m} (\hat{\boldsymbol{U}}_n - \gamma \hat{\boldsymbol{V}}_n)^T \boldsymbol{\Sigma}_{\hat{\mathcal{S}}}(\boldsymbol{W})^T \boldsymbol{\Sigma}_{\hat{\mathcal{S}}}(\boldsymbol{W}) \hat{\boldsymbol{U}}_n + \lambda \boldsymbol{I}_n \right]^{-1}$$

*defined as in equation 13. Let $\boldsymbol{\sigma} \in \mathbb{R}^m$ independent of $\boldsymbol{W}$ such that $\frac{1}{\sqrt{m}} \|\boldsymbol{\sigma}\| \leq \sqrt{K_v}$ for $K_v > 0$. Then*

$$\Pr\left( \left| \frac{1}{m} \boldsymbol{\sigma}^T \hat{\boldsymbol{U}}_n \boldsymbol{Q}_m(\boldsymbol{W})(\hat{\boldsymbol{U}}_n - \gamma \hat{\boldsymbol{V}}_n)^T \boldsymbol{\sigma} - \frac{1}{m} \boldsymbol{\sigma}^T \hat{\boldsymbol{U}}_n \mathbb{E}[\boldsymbol{Q}_m(\boldsymbol{W})](\hat{\boldsymbol{U}}_n - \gamma \hat{\boldsymbol{V}}_n)^T \boldsymbol{\sigma} \right| > t \right) \leq C e^{-cmt^2},$$

*for some $C, c > 0$ independent of $m$ and $N$.*

*Proof.* Let the function $f : \boldsymbol{W} \mapsto \frac{1}{m}\boldsymbol{\sigma}^T\hat{\boldsymbol{U}}_n\boldsymbol{Q}_m(\boldsymbol{W})(\hat{\boldsymbol{U}}_n - \gamma\hat{\boldsymbol{V}}_n)^T\boldsymbol{\sigma}$. We want to show $f$ is Lipschitz in order to apply Lemma K.3. From Lemma I.1, we know there exists a real $K > 0$ such that, for all $m$ and for any $\boldsymbol{W}$, we have

$$\|\boldsymbol{Q}_m(\boldsymbol{W})\| \le K.$$

Furthemore, both $\|\hat{\boldsymbol{U}}_n\|$ and $\|\hat{\boldsymbol{V}}_n\|$ are upper bounded by 1. Let $\boldsymbol{H} \in \mathbb{R}^{N \times d}$, we have

$$|f(\boldsymbol{W} + \boldsymbol{H}) - f(\boldsymbol{W})|$$

$$= \left| \frac{1}{m}\boldsymbol{\sigma}^T\hat{\boldsymbol{U}}_n\Big[\boldsymbol{Q}_m(\boldsymbol{W} + \boldsymbol{H}) - \boldsymbol{Q}_m(\boldsymbol{W})\Big](\hat{\boldsymbol{U}}_n - \gamma\hat{\boldsymbol{V}}_n)^T\boldsymbol{\sigma} \right|$$

$$= \left| \frac{1}{m^2}\boldsymbol{\sigma}^T\hat{\boldsymbol{U}}_n\boldsymbol{Q}_m(\boldsymbol{W} + \boldsymbol{H})(\hat{\boldsymbol{U}}_n - \gamma\hat{\boldsymbol{V}}_n)^T\Big[\boldsymbol{\Sigma}_{\hat{\mathcal{S}}}(\boldsymbol{W} + \boldsymbol{H})^T\boldsymbol{\Sigma}_{\hat{\mathcal{S}}}(\boldsymbol{W} + \boldsymbol{H}) \right.$$

$$\left. - \boldsymbol{\Sigma}_{\hat{\mathcal{S}}}(\boldsymbol{W})^T\boldsymbol{\Sigma}_{\hat{\mathcal{S}}}(\boldsymbol{W})\Big]\hat{\boldsymbol{U}}_n\boldsymbol{Q}_m(\boldsymbol{W})(\hat{\boldsymbol{U}}_n - \gamma\hat{\boldsymbol{V}}_n)^T\boldsymbol{\sigma} \right|$$

$$= \left| \frac{1}{m^2}\boldsymbol{\sigma}^T\hat{\boldsymbol{U}}_n\boldsymbol{Q}_m(\boldsymbol{W} + \boldsymbol{H})(\hat{\boldsymbol{U}}_n - \gamma\hat{\boldsymbol{V}}_n)^T\Big[\boldsymbol{\Sigma}_{\hat{\mathcal{S}}}(\boldsymbol{W} + \boldsymbol{H})^T\big[\boldsymbol{\Sigma}_{\hat{\mathcal{S}}}(\boldsymbol{W} + \boldsymbol{H}) - \boldsymbol{\Sigma}_{\hat{\mathcal{S}}}(\boldsymbol{W})\big] \right.$$

$$\left. + \big[\boldsymbol{\Sigma}_{\hat{\mathcal{S}}}(\boldsymbol{W} + \boldsymbol{H}) - \boldsymbol{\Sigma}_{\hat{\mathcal{S}}}(\boldsymbol{W})\big]^T\boldsymbol{\Sigma}_{\hat{\mathcal{S}}}(\boldsymbol{W})\Big]\hat{\boldsymbol{U}}_n\boldsymbol{Q}_m(\boldsymbol{W})(\hat{\boldsymbol{U}}_n - \gamma\hat{\boldsymbol{V}}_n)^T\boldsymbol{\sigma} \right|$$

$$\le \left| \frac{1}{m^2}\boldsymbol{\sigma}^T\hat{\boldsymbol{U}}_n\boldsymbol{Q}_m(\boldsymbol{W} + \boldsymbol{H})(\hat{\boldsymbol{U}}_n - \gamma\hat{\boldsymbol{V}}_n)^T\boldsymbol{\Sigma}_{\hat{\mathcal{S}}}(\boldsymbol{W} + \boldsymbol{H})^T\big[\boldsymbol{\Sigma}_{\hat{\mathcal{S}}}(\boldsymbol{W} + \boldsymbol{H}) \right.$$

$$\left. - \boldsymbol{\Sigma}_{\hat{\mathcal{S}}}(\boldsymbol{W})\big]\hat{\boldsymbol{U}}_n\boldsymbol{Q}_m(\boldsymbol{W})(\hat{\boldsymbol{U}}_n - \gamma\hat{\boldsymbol{V}}_n)^T\boldsymbol{\sigma} \right|$$

$$+ \left| \frac{1}{m^2}\boldsymbol{\sigma}^T\hat{\boldsymbol{U}}_n\boldsymbol{Q}_m(\boldsymbol{W} + \boldsymbol{H})(\hat{\boldsymbol{U}}_n - \gamma\hat{\boldsymbol{V}}_n)^T\big[\boldsymbol{\Sigma}_{\hat{\mathcal{S}}}(\boldsymbol{W} + \boldsymbol{H}) \right.$$

$$\left. - \boldsymbol{\Sigma}_{\hat{\mathcal{S}}}(\boldsymbol{W})\big]^T\boldsymbol{\Sigma}_{\hat{\mathcal{S}}}(\boldsymbol{W})\hat{\boldsymbol{U}}_n\boldsymbol{Q}_m(\boldsymbol{W})(\hat{\boldsymbol{U}}_n - \gamma\hat{\boldsymbol{V}}_n)^T\boldsymbol{\sigma} \right|$$

$$\le 2K_vK\left\| \frac{1}{\sqrt{m}}\boldsymbol{Q}_m(\boldsymbol{W} + \boldsymbol{H})(\hat{\boldsymbol{U}}_n - \gamma\hat{\boldsymbol{V}}_n)^T\boldsymbol{\Sigma}_{\hat{\mathcal{S}}}(\boldsymbol{W} + \boldsymbol{H})^T \right\| \left\| \frac{1}{\sqrt{m}}\big[\boldsymbol{\Sigma}_{\hat{\mathcal{S}}}(\boldsymbol{W} + \boldsymbol{H}) - \boldsymbol{\Sigma}_{\hat{\mathcal{S}}}(\boldsymbol{W})\big] \right\|$$

$$+ 4K_vK\left\| \frac{1}{\sqrt{m}}\big[\boldsymbol{\Sigma}_{\hat{\mathcal{S}}}(\boldsymbol{W} + \boldsymbol{H}) - \boldsymbol{\Sigma}_{\hat{\mathcal{S}}}(\boldsymbol{W})\big] \right\| \left\| \frac{1}{\sqrt{m}}\boldsymbol{\Sigma}_{\hat{\mathcal{S}}}(\boldsymbol{W})\hat{\boldsymbol{U}}_n\boldsymbol{Q}_m(\boldsymbol{W}) \right\|.$$

From Lemma I.4, we know there exists a real $K' > 0$ such that, for all $m$, we have

$$\left\| \frac{1}{\sqrt{m}}\boldsymbol{\Sigma}_{\hat{\mathcal{S}}}(\boldsymbol{W})\hat{\boldsymbol{U}}_n\boldsymbol{Q}_m(\boldsymbol{W}) \right\| \le K'$$

and

$$\left\| \frac{1}{\sqrt{m}}\boldsymbol{Q}_m(\boldsymbol{W} + \boldsymbol{H})(\hat{\boldsymbol{U}}_n - \gamma\hat{\boldsymbol{V}}_n)^T\boldsymbol{\Sigma}_{\hat{\mathcal{S}}}(\boldsymbol{W} + \boldsymbol{H})^T \right\| \le 2K'.$$

From those results, we conclude the Lipschitz continuity of $f$ since

$$|f(\boldsymbol{W} + \boldsymbol{H}) - f(\boldsymbol{W})| \le 8K_vKK'\left\| \frac{1}{\sqrt{m}}\big[\boldsymbol{\Sigma}_{\hat{\mathcal{S}}}(\boldsymbol{W} + \boldsymbol{H}) - \boldsymbol{\Sigma}_{\hat{\mathcal{S}}}(\boldsymbol{W})\big] \right\|$$

$$\le 8K_vKK'\left\| \frac{1}{\sqrt{m}}\big[\boldsymbol{\Sigma}_{\hat{\mathcal{S}}}(\boldsymbol{W} + \boldsymbol{H}) - \boldsymbol{\Sigma}_{\hat{\mathcal{S}}}(\boldsymbol{W})\big] \right\|_F$$

$$\le \frac{8K_vKK'K_{\boldsymbol{\sigma}}}{\sqrt{m}}\|\boldsymbol{H}\boldsymbol{S}\|_F$$

$$= \frac{8K_vKK'K_{\boldsymbol{\sigma}}}{\sqrt{m}}\sqrt{\mathrm{Tr}\,(\boldsymbol{H}\boldsymbol{S}\boldsymbol{S}^T\boldsymbol{H}^T)}$$

$$\le \frac{8K_vKK'K_{\boldsymbol{\sigma}}}{\sqrt{m}}\|\boldsymbol{S}\|\,\|\boldsymbol{H}\|_F.$$

The last inequality is obtained because $|\mathrm{Tr}(\boldsymbol{AB})| \leq \|\boldsymbol{B}\|\mathrm{Tr}(\boldsymbol{A})$ for some semi-positive-definite matrix $\boldsymbol{A}$. We prove that $f$ is Lipschitz with parameter $\frac{8K_v K K' K_\sigma}{\sqrt{m}}\|\boldsymbol{S}\|$, and applying Lemma L.1 gives

$$\Pr\left( \left| \frac{1}{m}\boldsymbol{\sigma}^T\hat{\boldsymbol{U}}_n\boldsymbol{Q}_m(\boldsymbol{W})(\hat{\boldsymbol{U}}_n - \gamma\hat{\boldsymbol{V}}_n)^T\boldsymbol{\sigma} - \frac{1}{m}\boldsymbol{\sigma}^T\hat{\boldsymbol{U}}_n\mathbb{E}[\boldsymbol{Q}_m(\boldsymbol{W})](\hat{\boldsymbol{U}}_n - \gamma\hat{\boldsymbol{V}}_n)^T\boldsymbol{\sigma} \right| > t \right)$$
$$\leq Ce^{-\frac{cmt^2}{2^6 K_v^2 K^2 K'^2 K_\sigma^2 K_\varphi^2 \|\boldsymbol{S}\|^2}},$$

for some $C, c > 0$ independent of other parameters. $\qquad\square$

**Lemma K.5.** *Under Assumptions 1 and 2, let $\boldsymbol{Q}_- \in \mathbb{R}^{n\times n}$ be the resolvent defined in equation 29, let $\boldsymbol{w}_i \sim \mathcal{N}(\boldsymbol{0}, \boldsymbol{I}_d)$ be a Gaussian vector independent of $\boldsymbol{Q}_-$, and let $\boldsymbol{\sigma} : \mathbb{R} \to \mathbb{R}$ be a real 1-Lipschitz function. Then*

$$\Pr\bigg( \bigg| \frac{1}{m}\sigma(\boldsymbol{w}_i^T\hat{\boldsymbol{S}})\hat{\boldsymbol{U}}_n\boldsymbol{Q}_{-i}(\hat{\boldsymbol{U}}_n - \gamma\hat{\boldsymbol{V}}_n)^T\sigma(\hat{\boldsymbol{S}}^T\boldsymbol{w}_i)$$
$$- \frac{1}{m}\mathrm{Tr}\left(\hat{\boldsymbol{U}}_n\mathbb{E}[\boldsymbol{Q}_{-i}](\hat{\boldsymbol{U}}_n - \gamma\hat{\boldsymbol{V}}_n)^T\mathbb{E}[\sigma(\hat{\boldsymbol{S}}^T\boldsymbol{w}_i)\sigma(\boldsymbol{w}_i^T\hat{\boldsymbol{S}})]\right) \bigg| > t \bigg)$$
$$\leq Ce^{-cm\max(t^2,t)},$$

*for some $C, c > 0$ independent of $N, m$.*

*Proof.* We can observe that

$$\Pr\bigg( \bigg| \frac{1}{m}\sigma(\boldsymbol{w}_i^T\hat{\boldsymbol{S}})\hat{\boldsymbol{U}}_n\boldsymbol{Q}_{-i}(\hat{\boldsymbol{U}}_n - \gamma\hat{\boldsymbol{V}}_n)^T\sigma(\hat{\boldsymbol{S}}^T\boldsymbol{w}_i)$$
$$- \frac{1}{m}\mathrm{Tr}\left(\hat{\boldsymbol{U}}_n\mathbb{E}[\boldsymbol{Q}_{-i}](\hat{\boldsymbol{U}}_n - \gamma\hat{\boldsymbol{V}}_n)^T\mathbb{E}[\sigma(\hat{\boldsymbol{S}}^T\boldsymbol{w}_i)\sigma(\boldsymbol{w}_i^T\hat{\boldsymbol{S}})]\right) \bigg| > t \bigg)$$
$$\leq \Pr\bigg( \bigg| \frac{1}{m}\sigma(\boldsymbol{w}^T\hat{\boldsymbol{S}})^T\hat{\boldsymbol{U}}_n\boldsymbol{Q}_{-i}(\hat{\boldsymbol{U}}_n - \gamma\hat{\boldsymbol{V}}_n)^T\sigma(\boldsymbol{w}^T\hat{\boldsymbol{S}})$$
$$- \frac{1}{m}\sigma(\boldsymbol{w}^T\hat{\boldsymbol{S}})^T\hat{\boldsymbol{U}}_n\mathbb{E}[\boldsymbol{Q}_{-i}](\hat{\boldsymbol{U}}_n - \gamma\hat{\boldsymbol{V}}_n)^T\sigma(\boldsymbol{w}^T\hat{\boldsymbol{S}}) \bigg| > \frac{t}{2} \bigg) \qquad (82)$$
$$+ \Pr\bigg( \bigg| \frac{1}{m}\sigma(\boldsymbol{w}^T\hat{\boldsymbol{S}})^T\hat{\boldsymbol{U}}_n\mathbb{E}[\boldsymbol{Q}_{-i}](\hat{\boldsymbol{U}}_n - \gamma\hat{\boldsymbol{V}}_n)^T\sigma(\boldsymbol{w}^T\hat{\boldsymbol{S}})$$
$$- \frac{1}{m}\mathrm{Tr}\left(\hat{\boldsymbol{U}}_n\mathbb{E}[\boldsymbol{Q}_{-i}](\hat{\boldsymbol{U}}_n - \gamma\hat{\boldsymbol{V}}_n)^T\mathbb{E}[\sigma(\hat{\boldsymbol{S}}^T\boldsymbol{w}_i)\sigma(\boldsymbol{w}_i^T\hat{\boldsymbol{S}})]\right) \bigg| > \frac{t}{2} \bigg).$$

From Lemma I.1, there exists a real $K > 0$ such that, for all $m$, we have

$$\|\boldsymbol{Q}_{-i}\| \leq K.$$

Besides, both $\|\hat{\boldsymbol{U}}_n\|$ and $\|\hat{\boldsymbol{V}}_n\|$ are upper bounded by 1. We thus bound the probability of the right-hand part with Lemma K.2 as

$$\Pr\bigg( \bigg| \frac{1}{m}\sigma(\boldsymbol{w}^T\hat{\boldsymbol{S}})^T\hat{\boldsymbol{U}}_n\mathbb{E}[\boldsymbol{Q}_{-i}](\hat{\boldsymbol{U}}_n - \gamma\hat{\boldsymbol{V}}_n)^T\sigma(\boldsymbol{w}^T\hat{\boldsymbol{S}})$$
$$- \frac{1}{m}\mathrm{Tr}\left(\hat{\boldsymbol{U}}_n\mathbb{E}[\boldsymbol{Q}_{-i}](\hat{\boldsymbol{U}}_n - \gamma\hat{\boldsymbol{V}}_n)^T\mathbb{E}[\sigma(\hat{\boldsymbol{S}}^T\boldsymbol{w}_i)\sigma(\boldsymbol{w}_i^T\hat{\boldsymbol{S}})]\right) \bigg| > t \bigg) \qquad (83)$$
$$\leq Ce^{-\frac{cm}{2K_\sigma^2 K_\varphi^2 \|\hat{\boldsymbol{S}}\|^2}\min\left(\frac{t^2}{2^8 t_0^2 K^2}, \frac{t}{2K}\right)},$$

for $t_0 = |\sigma(0)| + \sqrt{\frac{d}{m}}K_\sigma K_\varphi\|\hat{\boldsymbol{S}}\|$, and $c, C \in \mathbb{R}$ independent of all other parameters. Let define the real $K' > 0$ and let $\mathcal{A}_{K'}$ be the probability space defined as

$$\mathcal{A}_{K'} = \{\boldsymbol{w} \in \mathbb{R}^m, \|\sigma(\boldsymbol{w}^T\hat{\boldsymbol{S}})\| \leq K'\sqrt{m}\}.$$

From Lemma K.1, we bound the second term $\Pr(\mathcal{A}_{K'}^c)$ as

$$\Pr(\mathcal{A}_{K'}^c) = \Pr(\{\|\sigma(\boldsymbol{w}^T\hat{\boldsymbol{S}})\| > K'\sqrt{m}\}) \leq C'e^{-\frac{c'mK'^2}{2K_\sigma^2 K_\varphi^2 \|\boldsymbol{X}\|^2}},$$

for some $c', C' > 0$ independent of other parameters. Conditioning the random variable of interest with respect to $\mathcal{A}_{K'}$ and its complementary $\mathcal{A}_{K'}^c$ gives with Lemma K.4

$$\Pr\left(\left|\frac{1}{m}\sigma(\boldsymbol{w}^T\hat{\boldsymbol{S}})^T\hat{\boldsymbol{U}}_n\boldsymbol{Q}_-(\hat{\boldsymbol{U}}_n - \gamma\hat{\boldsymbol{V}}_n)^T\sigma(\boldsymbol{w}^T\hat{\boldsymbol{S}}) - \frac{1}{m}\sigma(\boldsymbol{w}^T\hat{\boldsymbol{S}})^T\hat{\boldsymbol{U}}_n\mathbb{E}[\boldsymbol{Q}_-](\hat{\boldsymbol{U}}_n - \gamma\hat{\boldsymbol{V}}_n)^T\sigma(\boldsymbol{w}^T\hat{\boldsymbol{S}})\right| > t\right)$$

$$\leq \Pr\Big(\Big|\frac{1}{m}\sigma(\boldsymbol{w}^T\hat{\boldsymbol{S}})^T\hat{\boldsymbol{U}}_n\boldsymbol{Q}_-(\hat{\boldsymbol{U}}_n - \gamma\hat{\boldsymbol{V}}_n)^T\sigma(\boldsymbol{w}^T\hat{\boldsymbol{S}})$$

$$- \frac{1}{m}\sigma(\boldsymbol{w}^T\hat{\boldsymbol{S}})^T\hat{\boldsymbol{U}}_n\mathbb{E}[\boldsymbol{Q}_-](\hat{\boldsymbol{U}}_n - \gamma\hat{\boldsymbol{V}}_n)^T\sigma(\boldsymbol{w}^T\hat{\boldsymbol{S}})\Big| > t \ \cap \ \mathcal{A}_{K'}\Big) + \Pr(\mathcal{A}_{K'}^c)$$

$$\leq C''e^{-c''mt^2} + C'e^{-\frac{c'mK'^2}{2K_\sigma^2 K_\varphi^2 \|\hat{\boldsymbol{S}}\|^2}},$$

(84)

where $c'', C'' > 0$. Combing both equation 83 and equation 84 with equation 82 gives

$$\Pr\Big(\Big|\frac{1}{m}\sigma(\boldsymbol{w}_i^T\hat{\boldsymbol{S}})\hat{\boldsymbol{U}}_n\boldsymbol{Q}_{-i}(\hat{\boldsymbol{U}}_n - \gamma\hat{\boldsymbol{V}}_n)^T\sigma(\hat{\boldsymbol{S}}^T\boldsymbol{w}_i)$$

$$- \frac{1}{m}\operatorname{Tr}\left(\hat{\boldsymbol{U}}_n\mathbb{E}[\boldsymbol{Q}_{-i}](\hat{\boldsymbol{U}}_n - \gamma\hat{\boldsymbol{V}}_n)^T\mathbb{E}[\sigma(\hat{\boldsymbol{S}}^T\boldsymbol{w}_i)\sigma(\boldsymbol{w}_i^T\hat{\boldsymbol{S}})]\right)\Big| > t\Big)$$

(85)

$$\leq Ce^{-\frac{cm}{2K_\sigma^2 K_\varphi^2 \|\hat{\boldsymbol{S}}\|^2}\min\left(\frac{t^2}{2^{10}t_0^2K^2}, \frac{t}{4K}\right)} + C''e^{-\frac{c''mt^2}{4}} + C'e^{-\frac{c'mK'^2}{2K_\sigma^2 K_\varphi^2 \|\boldsymbol{X}\|^2}}.$$

$\square$

**Lemma K.6.** *Under Assumptions 1 and 2, let $\lambda > 0$, let $\boldsymbol{W} \in \mathbb{R}^{N\times d}$, and let the resolvent*

$$\boldsymbol{Q}_m(\boldsymbol{W}) = \left[\frac{1}{m}(\hat{\boldsymbol{U}}_n - \gamma\hat{\boldsymbol{V}}_n)^T\boldsymbol{\Sigma}_{\hat{\mathcal{S}}}(\boldsymbol{W})^T\boldsymbol{\Sigma}_{\hat{\mathcal{S}}}(\boldsymbol{W})\hat{\boldsymbol{U}}_n + \lambda\boldsymbol{I}_n\right]^{-1}$$

*defined as in equation 13. Let $\boldsymbol{u} \in \mathbb{R}^n$ such that $\|\boldsymbol{u}\| \leq K_{\boldsymbol{u}}$ for $K_{\boldsymbol{u}} > 0$. Then*

$$\Pr\left(\left|\frac{\lambda^2}{n}\boldsymbol{u}^T\boldsymbol{Q}_m(\boldsymbol{W})^T\boldsymbol{Q}_m(\boldsymbol{W})\boldsymbol{u} - \frac{\lambda^2}{n}\boldsymbol{u}^T\mathbb{E}[\boldsymbol{Q}_m(\boldsymbol{W})^T\boldsymbol{Q}_m(\boldsymbol{W})]\boldsymbol{u}\right| > t\right) \leq Ce^{-cn^2mt^2},$$

*for some $C, c > 0$ independent of $m, n$ and $N$.*

*Proof.* Let the function $f : \boldsymbol{W} \mapsto \frac{\lambda^2}{n}\boldsymbol{u}^T\boldsymbol{Q}_m(\boldsymbol{W})^T\boldsymbol{Q}_m(\boldsymbol{W})\boldsymbol{u}$. We want to show $f$ is Lipschitz in order to apply Lemma K.3. From Lemma I.1, we know there exists a real $K > 0$ such that, for all $m$ and $\boldsymbol{W}$, we have

$$\|\boldsymbol{Q}_m(\boldsymbol{W})\| \leq K.$$

Furthermore, both $\|\hat{\boldsymbol{U}}_n\|$ and $\|\hat{\boldsymbol{V}}_n\|$ are upper bounded by 1. Let $\boldsymbol{H} \in \mathbb{R}^{N\times d}$, we have

$$|f(\boldsymbol{W} + \boldsymbol{H}) - f(\boldsymbol{W})| = \left|\frac{\lambda^2}{n}\boldsymbol{u}^T\big[\boldsymbol{Q}_m(\boldsymbol{W} + \boldsymbol{H})^T\boldsymbol{Q}_m(\boldsymbol{W} + \boldsymbol{H}) - \boldsymbol{Q}_m(\boldsymbol{W})^T\boldsymbol{Q}_m(\boldsymbol{W})\big]\boldsymbol{u}\right|$$

$$\leq \underbrace{\left|\frac{\lambda^2}{n}\boldsymbol{u}^T\boldsymbol{Q}_m(\boldsymbol{W} + \boldsymbol{H})^T\big[\boldsymbol{Q}_m(\boldsymbol{W} + \boldsymbol{H}) - \boldsymbol{Q}_m(\boldsymbol{W})\big]\boldsymbol{u}^T\right|}_{(1)}$$

$$+ \underbrace{\left|\frac{\lambda^2}{n}\boldsymbol{u}^T\big[\boldsymbol{Q}_m(\boldsymbol{W} + \boldsymbol{H}) - \boldsymbol{Q}_m(\boldsymbol{W})\big]^T\boldsymbol{Q}_m(\boldsymbol{W})\boldsymbol{u}\right|}_{(2)}$$

For (1), we have

$$\left| \frac{\lambda^2}{n} \boldsymbol{u}^T \boldsymbol{Q}_m(\boldsymbol{W} + \boldsymbol{H})^T \Big[ \boldsymbol{Q}_m(\boldsymbol{W} + \boldsymbol{H}) - \boldsymbol{Q}_m(\boldsymbol{W}) \Big] \boldsymbol{u}^T \right|$$

$$= \left| \frac{\lambda^2}{n} \frac{1}{m} \boldsymbol{u}^T \boldsymbol{Q}_m(\boldsymbol{W} + \boldsymbol{H})^T \boldsymbol{Q}_m(\boldsymbol{W} + \boldsymbol{H})(\hat{\boldsymbol{U}}_n - \gamma \hat{\boldsymbol{V}}_n)^T \Big[ \boldsymbol{\Sigma}_{\hat{\mathcal{S}}}(\boldsymbol{W} + \boldsymbol{H})^T \boldsymbol{\Sigma}_{\hat{\mathcal{S}}}(\boldsymbol{W} + \boldsymbol{H}) \right.$$

$$\left. - \boldsymbol{\Sigma}_{\hat{\mathcal{S}}}(\boldsymbol{W})^T \boldsymbol{\Sigma}_{\hat{\mathcal{S}}}(\boldsymbol{W}) \Big] \hat{\boldsymbol{U}}_n \boldsymbol{Q}_m(\boldsymbol{W}) \boldsymbol{u}^T \right|$$

$$= \left| \frac{\lambda^2}{n} \frac{1}{m} \boldsymbol{u}^T \boldsymbol{Q}_m(\boldsymbol{W} + \boldsymbol{H})^T \boldsymbol{Q}_m(\boldsymbol{W} + \boldsymbol{H})(\hat{\boldsymbol{U}}_n - \gamma \hat{\boldsymbol{V}}_n)^T \Big[ \boldsymbol{\Sigma}_{\hat{\mathcal{S}}}(\boldsymbol{W} + \boldsymbol{H})^T \big[ \boldsymbol{\Sigma}_{\hat{\mathcal{S}}}(\boldsymbol{W} + \boldsymbol{H}) - \boldsymbol{\Sigma}_{\hat{\mathcal{S}}}(\boldsymbol{W}) \big] \right.$$

$$\left. + \big[ \boldsymbol{\Sigma}_{\hat{\mathcal{S}}}(\boldsymbol{W} + \boldsymbol{H}) - \boldsymbol{\Sigma}_{\hat{\mathcal{S}}}(\boldsymbol{W}) \big]^T \boldsymbol{\Sigma}_{\hat{\mathcal{S}}}(\boldsymbol{W}) \Big] \hat{\boldsymbol{U}}_n \boldsymbol{Q}_m(\boldsymbol{W}) \boldsymbol{u}^T \right|$$

$$\leq \left| \frac{\lambda^2}{n} \frac{1}{m} \boldsymbol{u}^T \boldsymbol{Q}_m(\boldsymbol{W} + \boldsymbol{H})^T \boldsymbol{Q}_m(\boldsymbol{W} + \boldsymbol{H})(\hat{\boldsymbol{U}}_n - \gamma \hat{\boldsymbol{V}}_n)^T \boldsymbol{\Sigma}_{\hat{\mathcal{S}}}(\boldsymbol{W} + \boldsymbol{H})^T \big[ \boldsymbol{\Sigma}_{\hat{\mathcal{S}}}(\boldsymbol{W} + \boldsymbol{H}) \right.$$

$$\left. - \boldsymbol{\Sigma}_{\hat{\mathcal{S}}}(\boldsymbol{W}) \big] \hat{\boldsymbol{U}}_n \boldsymbol{Q}_m(\boldsymbol{W}) \boldsymbol{u}^T \right|$$

$$+ \left| \frac{\lambda^2}{n} \frac{1}{m} \boldsymbol{u}^T \boldsymbol{Q}_m(\boldsymbol{W} + \boldsymbol{H})^T \boldsymbol{Q}_m(\boldsymbol{W} + \boldsymbol{H})(\hat{\boldsymbol{U}}_n - \gamma \hat{\boldsymbol{V}}_n)^T \big[ \boldsymbol{\Sigma}_{\hat{\mathcal{S}}}(\boldsymbol{W} + \boldsymbol{H}) \right.$$

$$\left. - \boldsymbol{\Sigma}_{\hat{\mathcal{S}}}(\boldsymbol{W}) \big]^T \boldsymbol{\Sigma}_{\hat{\mathcal{S}}}(\boldsymbol{W}) \hat{\boldsymbol{U}}_n \boldsymbol{Q}_m(\boldsymbol{W}) \boldsymbol{u}^T \right|$$

$$\leq \frac{\lambda^2}{n} K^2 K_{\boldsymbol{u}}^2 \left\| \frac{1}{\sqrt{m}} \boldsymbol{Q}_m(\boldsymbol{W} + \boldsymbol{H})(\hat{\boldsymbol{U}}_n - \gamma \hat{\boldsymbol{V}}_n)^T \boldsymbol{\Sigma}_{\hat{\mathcal{S}}}(\boldsymbol{W} + \boldsymbol{H})^T \right\| \left\| \frac{1}{\sqrt{m}} \big[ \boldsymbol{\Sigma}_{\hat{\mathcal{S}}}(\boldsymbol{W} + \boldsymbol{H}) - \boldsymbol{\Sigma}_{\hat{\mathcal{S}}}(\boldsymbol{W}) \big] \right\|$$

$$+ \frac{2\lambda^2}{n} K^2 K_{\boldsymbol{u}}^2 \left\| \frac{1}{\sqrt{m}} \big[ \boldsymbol{\Sigma}_{\hat{\mathcal{S}}}(\boldsymbol{W} + \boldsymbol{H}) - \boldsymbol{\Sigma}_{\hat{\mathcal{S}}}(\boldsymbol{W}) \big] \right\| \left\| \frac{1}{\sqrt{m}} \boldsymbol{\Sigma}_{\hat{\mathcal{S}}}(\boldsymbol{W}) \hat{\boldsymbol{U}}_n \boldsymbol{Q}_m(\boldsymbol{W}) \right\|$$

From Lemma I.4, we know there exists a real $K' > 0$ such that, for all $m$, we have

$$\left\| \frac{1}{\sqrt{m}} \boldsymbol{\Sigma}_{\hat{\mathcal{S}}}(\boldsymbol{W}) \hat{\boldsymbol{U}}_n \boldsymbol{Q}_m(\boldsymbol{W}) \right\| \leq K'$$

and

$$\left\| \frac{1}{\sqrt{m}} \boldsymbol{Q}_m(\boldsymbol{W} + \boldsymbol{H})(\hat{\boldsymbol{U}}_n - \gamma \hat{\boldsymbol{V}}_n)^T \boldsymbol{\Sigma}_{\hat{\mathcal{S}}}(\boldsymbol{W} + \boldsymbol{H})^T \right\| \leq 2K'.$$

From those results, we conclude for (1) that

$$\left| \frac{\lambda^2}{n} \boldsymbol{u}^T \boldsymbol{Q}_m(\boldsymbol{W} + \boldsymbol{H})^T \Big[ \boldsymbol{Q}_m(\boldsymbol{W} + \boldsymbol{H}) - \boldsymbol{Q}_m(\boldsymbol{W}) \Big] \boldsymbol{u}^T \right|$$

$$\leq \frac{4\lambda^2 K^2 K_{\boldsymbol{u}}^2 K'}{n} \left\| \frac{1}{\sqrt{m}} \big[ \boldsymbol{\Sigma}_{\hat{\mathcal{S}}}(\boldsymbol{W} + \boldsymbol{H}) - \boldsymbol{\Sigma}_{\hat{\mathcal{S}}}(\boldsymbol{W}) \big] \right\|$$

$$\leq \frac{4\lambda^2 K^2 K_{\boldsymbol{u}}^2 K'}{n} \left\| \frac{1}{\sqrt{m}} \big[ \boldsymbol{\Sigma}_{\hat{\mathcal{S}}}(\boldsymbol{W} + \boldsymbol{H}) - \boldsymbol{\Sigma}_{\hat{\mathcal{S}}}(\boldsymbol{W}) \big] \right\|_F$$

$$\leq \frac{4\lambda^2 K^2 K_{\boldsymbol{u}}^2 K' K_{\boldsymbol{\sigma}}}{n\sqrt{m}} \| \boldsymbol{H} \boldsymbol{S} \|_F$$

$$= \frac{4\lambda^2 K^2 K_{\boldsymbol{u}}^2 K' K_{\boldsymbol{\sigma}}}{n\sqrt{m}} \sqrt{\text{Tr} \left( \boldsymbol{H} \boldsymbol{S} \boldsymbol{S}^T \boldsymbol{H}^T \right)}$$

$$\leq \frac{4\lambda^2 K^2 K_{\boldsymbol{u}}^2 K' K_{\boldsymbol{\sigma}}}{n\sqrt{m}} \| \boldsymbol{S} \| \, \| \boldsymbol{H} \|_F.$$

The last inequality is obtained because $|\text{Tr}(\boldsymbol{A}\boldsymbol{B})| \leq \|\boldsymbol{B}\| \text{Tr}(\boldsymbol{A})$ for some semi-positive-definite matrix $\boldsymbol{A}$. With a similar reasoning, we can prove for (2) that

$$\left| \frac{\lambda^2}{n} \boldsymbol{u}^T \Big[ \boldsymbol{Q}_m(\boldsymbol{W} + \boldsymbol{H}) - \boldsymbol{Q}_m(\boldsymbol{W}) \Big]^T \boldsymbol{Q}_m(\boldsymbol{W}) \boldsymbol{u} \right| \leq \frac{4\lambda^2 K^2 K_{\boldsymbol{u}}^2 K' K_{\boldsymbol{\sigma}}}{n\sqrt{m}} \| \boldsymbol{S} \| \, \| \boldsymbol{H} \|_F.$$

We thus prove that $f$ is Lipschitz with parameter $\frac{8\lambda^2 K^2 K_u^2 K' K_\sigma}{n\sqrt{m}} \|\boldsymbol{S}\|$, and applying Lemma L.1 gives

$$\Pr\left( \left| \frac{\lambda^2}{n} \boldsymbol{u}^T \boldsymbol{Q}_m(\boldsymbol{W})^T \boldsymbol{Q}_m(\boldsymbol{W}) \boldsymbol{u} - \frac{\lambda^2}{n} \boldsymbol{u}^T \mathbb{E}[\boldsymbol{Q}_m(\boldsymbol{W})^T \boldsymbol{Q}_m(\boldsymbol{W})] \boldsymbol{u} \right| > t \right)$$
$$\leq C e^{-\frac{cn^2 m t^2}{2^6 \lambda^2 K^4 K_u^4 K'^2 K_\sigma^2 K_\varphi^2 \|\boldsymbol{S}\|^2}},$$

for some $C, c > 0$ independent of other parameters. $\qquad\square$

## L  INTERMEDIARY LEMMAS

**Lemma L.1** (Normal Concentration). *((Ledoux, 2001, Corollary 2.6, Propositions 1.3, 1.8) or (Tao, 2012, Theorem 2.1.12)) For $d \in \mathbb{N}$, consider $\mu$ the canonical Gaussian probability on $\mathbb{R}^d$ defined through its density $d\mu(\boldsymbol{w}) = (2\pi)^{-\frac{d}{2}} e^{-\frac{1}{2}\|\boldsymbol{w}\|^2}$ and $f : \mathbb{R}^d \to \mathbb{R}$ a $L_f$-Lipschitz function. Then*

$$\mu\left( \left\{ \left| f - \int f d\mu \right| \geq t \right\} \right) \leq C e^{-c\frac{t^2}{L_f^2}}, \tag{86}$$

*where $C, c > 0$ are independent of $d$ and $L_f$.*

**Lemma L.2** (Resolvent Identity). *For invertible matrices $\boldsymbol{A}, \boldsymbol{B} \in \mathbb{R}^{n \times n}$,*

$$\boldsymbol{A}^{-1} - \boldsymbol{B}^{-1} = \boldsymbol{A}^{-1}(\boldsymbol{B} - \boldsymbol{A})\boldsymbol{B}^{-1}$$

**Lemma L.3** (Sherman–Morrison–Woodbury Matrix Identity). *(Horn & Johnson, 2012, Theorem 0.7.4) Let $\boldsymbol{A} \in \mathbb{R}^{n \times n}$ be a non-singular matrix with a known inverse $\boldsymbol{A}^{-1}$; let $\boldsymbol{M} = \boldsymbol{A} + \boldsymbol{UCV}$, in which $\boldsymbol{U} \in \mathbb{R}^{k \times n}$, $\boldsymbol{V} \in \mathbb{R}^{n \times k}$, and $\boldsymbol{C}^{k \times k}$ is non-singular. If $\boldsymbol{M}$ and $\boldsymbol{C}^{-1} + \boldsymbol{V} \boldsymbol{A}^{-1} \boldsymbol{U}$ are non-singular then*

$$(\boldsymbol{A} + \boldsymbol{UCV})^{-1} = \boldsymbol{A}^{-1} - \boldsymbol{A}^{-1} \boldsymbol{U} \left( \boldsymbol{C}^{-1} + \boldsymbol{V} \boldsymbol{A}^{-1} \boldsymbol{U} \right)^{-1} \boldsymbol{V} \boldsymbol{A}^{-1}, \tag{87}$$

*In particular $(\boldsymbol{A} + \boldsymbol{UV})^{-1} \boldsymbol{U} = \boldsymbol{A}^{-1} \boldsymbol{U} \left( \boldsymbol{I}_n + \boldsymbol{V} \boldsymbol{A}^{-1} \boldsymbol{U} \right)^{-1}$ and $\boldsymbol{V} (\boldsymbol{A} + \boldsymbol{UV})^{-1} = \left( \boldsymbol{I}_n + \boldsymbol{V} \boldsymbol{A}^{-1} \boldsymbol{U} \right)^{-1} \boldsymbol{V} \boldsymbol{A}^{-1}$.*

**Lemma L.4** (Sherman–Morrison Formula). *Let $\boldsymbol{A} \in \mathbb{R}^{n \times n}$ be a non-singular matrix with a known inverse $\boldsymbol{A}^{-1}$; let $\boldsymbol{M} = \boldsymbol{A} + \boldsymbol{uv}^T$, in which $\boldsymbol{u}, \boldsymbol{v} \in \mathbb{R}^n$. If $\boldsymbol{M}$ is non-singular and $1 + \boldsymbol{v}^T \boldsymbol{A}^{-1} \boldsymbol{u} \neq 0$ then*

$$\left( \boldsymbol{A} + \boldsymbol{uv}^T \right)^{-1} = \boldsymbol{A}^{-1} - \frac{\boldsymbol{A}^{-1} \boldsymbol{uv}^T \boldsymbol{A}^{-1}}{1 + \boldsymbol{v}^T \boldsymbol{A}^{-1} \boldsymbol{u}}. \tag{88}$$

*In particular, $\left( \boldsymbol{A} + \boldsymbol{uv}^T \right)^{-1} \boldsymbol{u} = \frac{\boldsymbol{A}^{-1}\boldsymbol{u}}{1 + \boldsymbol{v}^T \boldsymbol{A}^{-1}\boldsymbol{u}}$ and $\boldsymbol{v}^T \left( \boldsymbol{A} + \boldsymbol{uv}^T \right)^{-1} = \frac{\boldsymbol{v}^T \boldsymbol{A}^{-1}}{1 + \boldsymbol{v}^T \boldsymbol{A}^{-1}\boldsymbol{u}}$. This Lemma is an extension of Lemma L.3.*

**Lemma L.5** (Ostrowski's Theorem). *(Horn & Johnson, 2012, Theorem 4.5.9) Let $\boldsymbol{A}, \boldsymbol{S} \in \mathbb{R}^{n \times n}$ with $\boldsymbol{A}$ Hermitian and $\boldsymbol{S}$ nonsingular. Let the eigenvalues of $\boldsymbol{A}$, $\boldsymbol{SAS}^T$, and $\boldsymbol{SS}^T$ be arranged in nondecreasing order. Let $\sigma_1 \geq \ldots \geq \sigma_n > 0$ be the singular values of $\boldsymbol{S}$. For each $k \in [n]$ there is a positive real number $\theta_k \in [\sigma_n^2, \sigma_1^2]$ such that*

$$\nu_k(\boldsymbol{SAS}^T) = \theta_k \nu_k(\boldsymbol{A})$$

**Lemma L.6** (Weinstein–Aronszajn Identity). *For $\boldsymbol{A} \in \mathbb{R}^{m \times n}$, $\boldsymbol{B} \in \mathbb{R}^{n \times m}$ and $\lambda \in \mathbb{R} \setminus \{0\}$,*

$$\det(\boldsymbol{AB} - \lambda \boldsymbol{I}_m) = (-\lambda)^{m-n} \det(\boldsymbol{BA} - \lambda \boldsymbol{I}_n).$$

*It follows that the non-zero eigenvalues of $\boldsymbol{AB}$ and $\boldsymbol{BA}$ are the same.*

**Lemma L.7.** *Let $\boldsymbol{A} \in \mathbb{R}^{n \times n}$ and $\lambda > 0$.*

$$\|(\boldsymbol{A} + \lambda \boldsymbol{I}_n)^{-1}\| \leq \frac{1}{\lambda}$$

*if and only if $\boldsymbol{AA}^T + \lambda(\boldsymbol{A} + \boldsymbol{A}^T)$ is positive definite. In particular, for matrix $\boldsymbol{A} \in \mathbb{R}^{n \times n}$ whose the Hermitian part $H(\boldsymbol{A}) = \frac{\boldsymbol{A} + \boldsymbol{A}^T}{2}$ is semi-positive-definite we have*

$$\|(\boldsymbol{A} + \lambda \boldsymbol{I}_n)^{-1}\| \leq \frac{1}{\lambda}$$

*Proof.*

$$\begin{aligned}
\|(\boldsymbol{A} + \lambda\boldsymbol{I}_n)^{-1}\|^2 &= \nu_{max}\left((\boldsymbol{A} + \lambda\boldsymbol{I}_n)^{-1\,T}(\boldsymbol{A} + \lambda\boldsymbol{I}_n)^{-1}\right) \\
&= \nu_{max}\left(\left[(\boldsymbol{A} + \lambda\boldsymbol{I}_n)\left(\boldsymbol{A}^T + \lambda\boldsymbol{I}_n\right)\right]^{-1}\right) \\
&= \nu_{max}\left(\left(\boldsymbol{A}\boldsymbol{A}^T + \lambda(\boldsymbol{A} + \boldsymbol{A}^T) + \lambda^2\boldsymbol{I}_n\right)^{-1}\right) \\
&= \nu_{min}\left(\left(\boldsymbol{A}\boldsymbol{A}^T + \lambda(\boldsymbol{A} + \boldsymbol{A}^T) + \lambda^2\boldsymbol{I}_n\right)\right)^{-1}
\end{aligned} \tag{89}$$

where $\nu_{max}(\boldsymbol{B})$ and $\nu_{min}(\boldsymbol{B})$ denotes the maximum eigenvalue and minimum eigenvalues of a matrix $\boldsymbol{B}$. Since $\boldsymbol{A}$ is positive-definite the matrix $\boldsymbol{A}\boldsymbol{A}^T + \lambda(\boldsymbol{A} + \boldsymbol{A}^T)$ is semi-positive-definite and has positive nonzeros eigenvalues. Therefore, $\nu_{min}\left(\left(\boldsymbol{A}\boldsymbol{A}^T + \lambda(\boldsymbol{A} + \boldsymbol{A}^T) + \lambda^2\boldsymbol{I}_n\right)\right) > \lambda^2$ and $\|(\boldsymbol{A} + \lambda\boldsymbol{I}_n)^{-1}\| \leq \frac{1}{\lambda}$

$\square$

**Lemma L.8.** *([Yates, 1995](), Theorem 2) If a mapping $f : [0, \infty) \to [0, \infty)$*

- *is monotonically increasing, i.e $x \geq x' \implies f(x) \geq f(x')$,*

- *is scalable, i.e $\forall \alpha > 1, \ \alpha f(x) > f(\alpha x)$,*

- *admits $x_0 \in [0, \infty)$ such that $x_0 \geq f(x_0)$,*

*then $f$ has a unique fixed-point.*

**Lemma L.9.** *Let $\boldsymbol{A} \in \mathbb{R}^{m \times n}$ and $\boldsymbol{B} \in \mathbb{R}^{n \times m}$. If $\boldsymbol{A}\boldsymbol{B} + \lambda\boldsymbol{I}_m$ is invertible, then*

$$\left[\boldsymbol{A}\boldsymbol{B} + \lambda\boldsymbol{I}_m\right]^{-1}\boldsymbol{A} = \boldsymbol{A}\left[\boldsymbol{B}\boldsymbol{A} + \lambda\boldsymbol{I}_n\right]^{-1}.$$

*Proof.* We have

$$\boldsymbol{A}\left[\boldsymbol{B}\boldsymbol{A} + \lambda\boldsymbol{I}_n\right] = \left[\boldsymbol{A}\boldsymbol{B} + \lambda\boldsymbol{I}_m\right]\boldsymbol{A}$$

Since both $\boldsymbol{A}\boldsymbol{B}$ and $\boldsymbol{B}\boldsymbol{A}$ share the same non-zero eigenvalues from Lemma L.6, we deduce $\boldsymbol{B}\boldsymbol{A} + \lambda\boldsymbol{I}_n$ is also invertible. By multiplying the equation above with both the inverse of $\left[\boldsymbol{B}\boldsymbol{A} + \lambda\boldsymbol{I}_n\right]$ and $\left[\boldsymbol{A}\boldsymbol{B} + \lambda\boldsymbol{I}_m\right]$, we get

$$\left[\boldsymbol{A}\boldsymbol{B} + \lambda\boldsymbol{I}_m\right]^{-1}\boldsymbol{A} = \boldsymbol{A}\left[\boldsymbol{B}\boldsymbol{A} + \lambda\boldsymbol{I}_n\right]^{-1}$$

$\square$

