# OpenReview forum: "On Double Descent in Reinforcement Learning with LSTD and Random Features"
_ICLR.cc/2024/Conference — ICLR 2024 poster_

### Official Review · Reviewer_jHhA · 2023-10-31

**Soundness:** 3 good
**Presentation:** 3 good
**Contribution:** 3 good
**Rating:** 6
**Confidence:** 3

**Summary:**

This paper studies a theoretical model (based on random features) of temporal difference learning of the value function in a markov reward process using neural networks with one hidden layer. The key high-level parameters studied are the number of parameters $N$ in the hidden layer, the number of distinct states visited $m$, and the $\ell_2$ regularization parameter $\lambda$. The paper focuses on the regime where $N,m \to \infty$ but the ratio $N/m$ remains constant. Under suitable assumptions, the main results of the paper give numerically solvable equations for computing the asymptotic mean-squared Bellman error (both the true error, and the empirical error observed during training). These results then imply that the double-descent phenomenon occurs in this regime i.e. that the true mean-squared Bellman error decreases when the ratio $N/m < 1$ (the under-parameterized regime), has a large spike at $N/m =1$ (the interpolating regime), and then further decreases when $N/m > 1$ (the over-parameterized regime). Furthermore, the paper shows that increasing the regularization parameter $\lambda$ mitigates this double-descent phenomenon, with sufficiently large values of $\lambda$ eliminating the spike at $N/m =1$ altogether. Empirical results on simple reinforcement learning tasks are in very good agreement with the theoretical predictions.

**Strengths:**

The paper gives strong theoretical grounding for the occurrence of the double-descent phenomenon in temporal difference learning.
Furthermore, the results demonstrate that double-descent can be mitigated by sufficient regularization, which could potentially have practical applications.
Finally, the empirical results in the paper demonstrate that the theoretical predictions are sufficiently accurate (in terms of the rates at which the predicted quantities concentrate) that even in small examples there is good agreement between theory and the empirical measurements.

**Weaknesses:**

The paper gives strong results for the random features model, but it is somewhat unclear how practically relevant these are.
While it is typical to study such models in order to prove high-level statistical properties that may be relevant to practical neural networks, it would be nice to have some discussion of when/whether these results might be relevant.
For example, when proving results in the NTK regime, it is actually true that even deep neural networks of sufficient width will behave as predicted given appropriate random initialization. Of course, the NTK regime requires a width that is unrealistic and thus does not necessarily capture the practical dynamics well. The mean-field regime is designed to (at least somewhat) mitigate this issue with the NTK regime.
It is unclear from the discussion in this paper precisely how the double-asymptotic regime fits-in to this picture with regards to modelling deep neural networks (as opposed to single-hidden layer networks).
Such discussion of the relevance of the double-asymptotic regime to deep-neural networks would also help to clarify the novelty of this work when compared with prior work on value learning in the NTK and mean-field regimes (e.g. the cited 2022 paper by Agazzi and Lu).

**Questions:**

As discussed in the weaknesses above, is there something additional that can be said about how studying the double-asymptotic regime where $N/m$ is constant can give insights into the behavior of deep neural networks? In particular, the NTK and mean-field regimes may allow for theorems that apply to deep neural networks, but perhaps with somewhat unrealistic levels of over-parameterization. Is there any similar statement that can be made in this regime? Or is there some fundamental trade-off where the realistic parameter choices in the double-asymptotic regime preclude results applying beyond a single hidden layer.

---

> ### Author Response · Authors · 2023-11-16
>
> Dear reviewer *jHhA*,
>
> Thank you for your thoughtful review and questions.
>
> ### Weaknesses \& Questions.
> ***
>
> Based on your questions, we extended the discussion of related work in the paper in order to clarify the relationship between the three regimes: NTK, Mean-field, and double asymptotic (ours). All three have in common that they consider an infinite-width single-layer network. Empirical studies for NTK and Mean-field indicate that some of the results carry over to deep networks. In Supervised Learning, the study of double descent was also initiated to explain experimental observations [1, 2]. We expect a similar link between theory and practice for our double asymptotic regime, but this will be the subject of future work.
>
> A major difference between our regime and the others, NTK and Mean-field, is that our theory covers the entire spectrum from under- to overparameterization. In experiments, our theory is closely matched by measurements on systems with just a few dozen parameters. Judging from the literature, it seems that this underparameterization is relatively common in practice in RL.
>
> The work by [3] considers only the case of overparameterization and assumes that the gradient descent has data covering the entire state space. In addition to covering the case of underparameterization, our work considers actual observed data, which allows us to characterize the relationship between the percentage of the state space that has been visited and the learning performance.
>
> ### References
> ***
> [1] Zhang, Chiyuan, et al. "Understanding deep learning (still) requires rethinking generalization." (2021)
>
> [2] Mikhail Belkin, Daniel Hsu, Siyuan Ma, and Soumik Mandal "Reconciling modern machine learning practice and the bias-variance trade-off" (2018)
>
> [3] Agazzi, Andrea, and Jianfeng Lu. "Temporal-difference learning with nonlinear function approximation: lazy training and mean field regimes." (2022)

---

> > ### Comment · Reviewer_jHhA · 2023-11-22
> >
> > I appreciate the clarifications in your response and in the updated submission. I will keep my score and would be happy to see the paper accepted.

---

### Official Review · Reviewer_a6mC · 2023-11-01

**Soundness:** 3 good
**Presentation:** 3 good
**Contribution:** 3 good
**Rating:** 8
**Confidence:** 3

**Summary:**

This paper examines the behavior of the (regularized) LSTD estimator within the random feature model and the on-policy evaluation context. It investigates a doubly asymptotic regime in which both the number of parameters and the number of distinct visited states tend to infinity at the same rate. A double descent phenomenon is observed between the empirical mean squared Bellman error (MSBE) and the population MSBE. The authors derive analytical equations for both the asymptotic empirical MSBE and the asymptotic population MSBE. Based on these derivations, they demonstrate that the double-descent phenomenon diminishes as either the regularization increases or the number of distinct unvisited states decreases. Empirical experiments corroborate the theoretical findings.

**Strengths:**

The setting provides a clean model to study TD methods in the over-parametrized regime, and the results are also fairly clean (at least conceptually, despite some of the linear algebra expressions being messy). The findings clearly demonstrate a theoretical over-parametrized double-descent regime for LSTD in this asymptotic setting, which to my knowledge, is a novel and interesting finding. I have not checked the proofs in great detail but from what I have seen they do check out. The experimental results, despite only looking at simple environments, appropriately confirm the theory.

**Weaknesses:**

One weakness is that the double descent behaviour is said to be related to the behaviour of the correction factor $\delta$ but I do not think that this is demonstrated analytically. What is precisely happening to \delta when N=m, or alternatively what is happening to the difference between the empirical MSBE and the true MSBE? This is one of the main selling points of the paper so it would be good to expand on this.

Other than that, there are a few writing nitpicks, which I outline below.
1. "We use the regularized LSTD algorithm, [...] because it converges to the same solution as other TD algorithms." This is true without regularization, but is it true with regularization?
2. "Theoretical studies of TD algorithms often focus on asymptotic regimes, in either finite or infinite state space, where the number of samples n -> \infty while keeping the number of model parameters N fixed". The authors likely meant that it has not been studied what happens when parameters and samples both converge to infinity, but this sentence also implies that only asymptotic analyses of TD exist (as opposed to finite-sample), which is not correct.
3. Section 3 "Notations" paragraph: what is the "Euclidean norm" of a matrix? Does the authors mean the spectral norm or the Frobenius norm?
4. Items 4 and 5 in the "Contributions" paragraph both repeat that the correction terms vanish when the regularization increases and/or the number of unvisited states go to zero.
5. The notation for feature matrix $\Sigma_S$ is unfortunate since it clashes with typical notation for covariance matrices. Is there a reason not to use the more conventional $\Phi$ notation for this matrix.
6. Section 3 "Linear Temporal-Difference Methods" section: the Mean-Square Projected Bellman Error is undefined.
7. Section 4.1. For the sampling paradigm I do not think the authors literally mean $(s,s') \sim \mu(s) P(s,s')$. This latter expression would be the marginal probability of observing s' only, and is just a number (not a distribution one can sample from).
8. Section 4.1. "We consider the on-policy setting, where $D_train$ is derived from a sample path of the MRP or its stationary distribution $\pi$. These are very different (one is i.i.d.), are both truly considered?
9. Some vector norms are indexed by $\pi$, but many other norms are not indexed (e.g. Equation 8, 9,10). Is this a typo, or are all unindexed norms the Euclidean norm?
10. Relatedly: the closed form for the LSTD estimator is defined without the usual diagonal D_\pi, which implies that the algorithm performs regression on the Euclidean norm rather than $L_2(\pi)$ (one can compare with the more classical weighted version e.g. in Amortila, Jiang, Szepesvári ICML 2023). In this case the LSTD estimator is different from the usual weighted one, can the authors discuss whether this is a meaningful difference and/or whether the results would hold for the weighted estimator?
11. "Under the assumption that all states are visited $\hat{MSBE}(\theta)$ converges to $MSBE(\theta)$ with probability 1, as the number of collected transitions $n \rightarrow \infty$ (Bradtke & Barto, 1996)." I do not think that this is shown in that paper, which instead deals with the projected bellman error.
12. Typo in Assumption 3: should be O(1/\sqrt{m})

**Questions:**

Questions:
1. Why are the two asymptotic quantities the parameter number and the number of unvisited states as opposed to the parameter number and just the sample size?
2. What are the technical difficulties when studying regularized LSTD for policy evaluation as opposed to regularized least squares for the regression case ($\gamma = 0$). The text mentions that there is a lack of positive definiteness in the LSTD matrix -- is this the only difficulty? How is this difficulty overcome?

---

> ### Author Response · Authors · 2023-11-16
>
> Dear reviewer *a6mC*,
>
> We thank you for your detailed and helpful comments.
>
> ### Weaknesses
> ***
>
> A precise analytical characterization of the relationship between the correction factor and double descent is challenging because of the complex nonlinear dependencies. To the best of our knowledge, it is not yet available even in the case of supervised learning (which is covered by our Theorems as the special case $\gamma=0$). We do show analytically in Sect. 5 and Appendix C that the correction factors $\delta, \Delta, \hat \Delta$ vanish as $N/m \to \infty$ (over-parameterization) or as the $l_2$ regularization parameter $\lambda \to \infty$.
>
> To shed more light on the difference between the asymptotic empirical MSBE and the asymptotic true MSBE, we added a reformulation of our theorem in Appendix C that highlights the similarities between the second-order correction terms $\hat \Delta$ and $\Delta$.
>
> Concerning your detailed comments:
>
> 1. Yes, it is also the case! We have added the citation [1].
>
> 2. Thank you for highlighting the lack of clarity on that point. We intended to express that previous studies have not considered the ratio $N/m$, even when $N \to \infty$ and either $n \to \infty$ or $m \to \infty$. We hope our revision clarifies this aspect.
>
> 3. Yes, it was indeed the operator norm.
>
> 4. We reformulated the claim to clarify the different aspects.
>
> 5. We adopted the same notation as [2] for the matrix $\Phi$ and $\Sigma$.
>
> 6. We have included a citation to guide readers to a more comprehensive explanation of the differences between the Mean-Square Error and the Mean-Square Projected Bellman error [3].
>
> 7. Thank you for pointing out this problem. We have updated the notation.
>
> 8. We just need one of this sampling to satisfy Assumption 3. Furthermore, it is easier in the on-policy setting to get Assumption 2.
>
> 9. All unindexed vector norms refer to the Euclidean norm; we have extended the 'Notations' section to clarify this point. We have also added further clarification to the definition of the empirical MSBE.
>
> 10. For regularization, we adopted the definition of regularized LSTD used by [3, 4, 5, 6] where the regularization parameter is applied to the empirical MSBE. Considering a different regularization parameter could alter the solution produced by the algorithm and, consequently, the result of the theorems.
>
> 11. Thank you for pointing this out; we updated the sentence with the correct citation.
> 12. Thank you for pointing out this typo.
>
> ### Questions:
> ***
>
> 1 - In the case of continuous systems, both the sample size $n$ and the number of visited states $m$ are usually equivalent. However, this is not true for discrete systems, where the same states can be sampled multiple times.
> Using $m$, we cover both discrete and continuous cases.
>
> 2 - The main challenge in extending the results from Supervised Learning to RL stems from the absence of positive definiteness in $Q_m(\lambda)$. Without this property, there is no guarantee that the eigenvalues are positive reals, which would yield the existence of the resolvent and the fixed-point $\delta$, bound the operator norm of the resolvent, address some technical aspects in the proofs, and help interpret the correction terms. Fortunately, even though $Q_m(\lambda)$ is not positive-definite, its unique structure still allows us to derive these results, albeit in a more complex manner. For instance, we exploited that the matrix  $H(\hat U_n(\hat U_n - \gamma \hat V_n)^T)$ is positive-definite. We make this matrix appear at different places through reformulations, e.g., applying the Woodbury identity to $Q_m(\lambda)$ modifies its structure, enabling us to leverage the symmetric/skew-symmetric decomposition of the adjusted matrix.
>
> ### References
> ***
>
> [1]  Berthier, Eloïse, Ziad Kobeissi, and Francis Bach. "A Non-asymptotic Analysis of Non-parametric Temporal-Difference Learning."(2022)
>
> [2] Louart, Cosme, Zhenyu Liao, and Romain Couillet. "A random matrix approach to neural networks." (2018)
>
> [3] Dann, Christoph, Gerhard Neumann, and Jan Peters. "Policy evaluation with temporal differences: A survey and comparison." (2014)
>
> [4] Kolter, J. Zico, and Andrew Y. Ng. "Regularization and feature selection in least-squares temporal difference learning." (2009, June)
>
> [5] Hoffman, Matthew W., et al. "Regularized least squares temporal difference learning with nested ℓ2 and ℓ1 penalization." (2011, September)
>
> [6] Chen, Shenglei, Geng Chen, and Ruijun Gu. "An efficient L2-norm regularized least-squares temporal difference learning algorithm."

---

> > ### Author Response · Authors · 2023-11-22
> >
> > Dear reviewer a6mC,
> >
> > Today is the last day of the discussion period. Are you satisfied with our response?  Do you have any other questions that we can help with?

---

> > > ### Comment · Reviewer_a6mC · 2023-11-22
> > > **Thanks for your reply**
> > >
> > > Thanks for your reply and for the improvements/clarifications made to the paper. No further questions or comments at this time.

---

### Official Review · Reviewer_7TMf · 2023-11-01

**Soundness:** 4 excellent
**Presentation:** 4 excellent
**Contribution:** 4 excellent
**Rating:** 10
**Confidence:** 4

**Summary:**

The authors investigate a double-descent phenomenon in RL, where the mean-squared Bellman error (MSBE) initially decreases as the number of learnable parameters $N$ increases, then increases when $N$ equals the number of distinct visited states $m$, and finally decreases again in the overparameterized regime as $N \to \infty$. An analogous effect has been previously identified in supervised learning but not RL. The authors focus their analysis on least-squares temporal difference (LSTD) with random features to simulate a wide 1-layer network without needing to model the complexities of feature extraction. Analytical expressions for the MSBE as a function of the ratio $N / m$ and the $l^2$-regularization coefficient $\lambda$ are derived, which indicate a spike in MSBE near $N / m = 1$ and $\lambda \approx 0$. This behavior is verified empirically by LSTD experiments in small environments, such as Taxi. Notably, it is experimentally shown that the peak at $N / m = 1$ disappears when either $m = |S|$ (the state space is covered) or $\lambda$ is increased.

**Strengths:**

- I really like the idea of studying LSTD with random features to model neural networks. It is a useful abstraction to remove the noise of TD updates and other complexities of online learning like state visitations, changing features, and potential divergence with nonlinear FA.
- The mathematical analysis is excellent and highly significant. The results are clear and the remarks after the theorems provide helpful interpretations for them. The math is correct to the best of my knowledge (although I only checked a small part of the appendix). It is apparent that a lot of thought and effort was put into this paper.
- The experiments clearly demonstrate the hypothesized double-descent phenomenon, and also answer focused questions like to what extent regularization or changing the discount factor can change the double-descent peak. The analysis is good and connects the experimental observations back to the derived terms in the theorems.
- The paper organization is great. The contributions, related work, and preliminaries/system model are explained well and help ease the reader into the main results.

**Weaknesses:**

- I wonder if the observed double-descent behavior may be specifically due to the analysis of MSBE rather than some other learning objective, like the mean-squared value error (MSVE), the error relative to $v_\pi$. For example, it is known that TD methods that optimize MSPBE can converge to a solution far away from the minimum MSBE, which may explain the large peak around $N = m$. However, if we measure against MSVE instead, would we still see this? I imagine that if we have a set of features with some MSVE, and then add more features, we could just set those corresponding new weights to zero to preserve the MSVE. This seems to imply to me that the MSVE would be monotonically non-increasing as $N$ increases, although I am not sure if I am missing something. Perhaps the authors could better justify their choice of MSBE in this regard.
- The paper says that only one random seed (42) was used to generate the training data, so I have to guess that only one seed was used to generate the random features as well. Since the MSBE is highly dependent on these “network” features, it is impossible to tell from just one trial that the double-descent observation is a common phenomenon over the distribution of possible feature initializations. It would be helpful to show the results averaged over a number of different networks, with confidence intervals shown.

**Minor:**
- On the start of page 4: “Deep RL can be cast as a special case of LFA, where the neural network learns both the feature vectors and the parameter vector.” While I see what you mean here, I personally found it confusing to describe deep RL as a special case of linear FA, since deep RL is nonlinear FA. I think this could be reworded for clarity.
- Also page 4: “Linear TD methods are LFA methods that minimize the MSBE” but then later in the paragraph it is correctly noted that TD methods “minimize MSPBE rather than MSBE.” The first part should say something like TD methods *try to* minimize the MSBE, but actually minimize MSPBE.

**Questions:**

1. Can the norm term in $\bar{\text{MSBE}}(\hat{\theta})$ (Theorem 5.3) be simplified to $\|| \bar{r} + (\gamma P - I) \frac{1}{\sqrt{n}} \frac{N}{m} \frac{1}{1+\delta} \Phi_S U_n \bar{Q}_m (\lambda) r \||^2_D$, i.e., factoring out $\gamma P - I$? Or was it left expanded intentionally?
1. Why was recursive LSTD used for the experiments? Since only the final weights are needed, would it be more efficient to just sample data from the behavior policy and then explicitly calculate the TD fixed point like $w = A^{-1} b$?

---

> ### Author Response · Authors · 2023-11-16
>
> Dear reviewer *7TMf*,
>
> We thank you for the thorough review and encouraging comments. We are happy to see that we were able to convey a part of the enjoyment of this research.
>
> ### Weaknesses
> ***
>
> You raise the question of whether double descent also occurs when learning objective functions other than the MSBE. We have considered this problem in detail and added both theoretical and experimental results to the appendix. Additional experiments in Appendix B indicate that the double descent phenomenon is also observed for the Mean-Square Value Error (MSVE). With a proof similar to that for Theorem 3, we derive a deterministic limit for the MSVE, as detailed in Appendix B. We focused on the MSBE for our study because both the MSBE and the MSPBE converge when $N \to \infty$ and since the MSPBE is the true learning objective function used by TD learning algorithms.
>
> Thank you for highlighting the lack of detail regarding the seeds used for generating random features. We have provided additional details in the description of the experimental section in order to clarify this issue. Each curve in the experiments represents the average over 30 training runs. A different seed is used to generate the random features in each training run.
> For the sake of clarity in the presentation, we chose not to include standard deviations in the figure of the experimental section, but they are illustrated in Figure 1.
>
> ### Minor:
> ***
> Thank you for your comments; we have updated the paper accordingly.
>
> ### Questions:
> ***
>
> 1. The factorization is correct, but we have opted for the expanded form in order to explicitly indicate that it represents an MSBE and for visual consistency with the other definitions of MSBE throughout the paper.
>
> 2. We use the Recursive LSTD because the classic LSTD requires more computational resources to process large datasets, particularly for inverting large matrices. Although Recursive LSTD may introduce a greater potential for instability, our experimental results are consistent with the theoretical expectations.

---

> > ### Author Response · Authors · 2023-11-22
> >
> > Dear reviewer 7TMf,
> >
> > Today is the last day of the discussion period. Are you satisfied with our detailed response regarding the MSVE (Annex B)? Since you kindly evaluated soundness, presentation, and contribution of the paper as excellent, would you consider raising the score?

---

> > > ### Comment · Reviewer_7TMf · 2023-11-23
> > >
> > > Thanks to the authors for answering my questions and adding the section about MSVE. It is interesting to see the double-descent phenomenon appear in this new setting as well, both theoretically and empirically. Importantly, these results demonstrate that double descent is not specific to one particular objective but is a more general phenomenon in RL.
> > >
> > > Overall, the paper is excellent. As all of my weaknesses have now been addressed, I must raise my score. Well done!

---

### Official Review · Reviewer_xcHJ · 2023-11-04

**Soundness:** 3 good
**Presentation:** 3 good
**Contribution:** 3 good
**Rating:** 6
**Confidence:** 3

**Summary:**

In this work the authors provide a theoretical analysis of regularized LSTD in a setting where the number of parameters of the value function approximation N and the number of distinct visited states m go to infinity with a fixed ratio N/m. The authors do so by relying on random matrix tools, in particular using features of the form sigma(WA) where W is a random feature matrix (Nxd) applied to any set of d-dimensional states A. The connection to neural networks is via this simplification via random matrices and can be likened to an N-dimensional single layer network, where the simplification is less objectionable due to the asymptotic regime.

In this setting, with some minor assumptions, the authors go on to show that the asymptotic empirical MSBE can be related to the two-norm of the resolvent applied to the reward vector, plus a second order correction term, and that the empirical MSBE approaches the asymptotic almost surely. The authors then show, including empirically for a small number of examples, that it is the second order term that contributes to double-descent behavior. A similar story can be shown for the true MSBE, albeit with a more complicated term and second-order factor.

The authors also show that the empirical variant reduces to a similar approach of Louart et al. for supervised learning when gamma, the discount factor, goes to zero. Additionally as the number of features, equivalent to the width of the network, or the regularization term increases the correction factor diminishes (thus leading to less prominence of the double-descent behavior).

**Strengths:**

The paper provides a clear, concise (as much as possible) presentation of their results. They relate the work to the literature well, particularly how this complements and extends work from the supervised learning literature. Similarly the effect of increasing the regularization strength or width of the "network" is beneficial. The empirical experiments are similarly good in explaining the effect of the theoretical work.

**Weaknesses:**

I don't think the paper has particular weaknesses, however I would ask what the practical implications of this work are? This is less a comment on this work and more a question directed towards the theoretical analsysis of double-descent in general, in that the random-feature assumption while seemingly valid asymptotically departs from how such work might be used in practice. Note that the authors do leave an extension to "deep" neural networks for later work. Similarly the authors discuss policy evaluation as a next step.

Overall I would like to more discussion of how to place this within the "standard/practical" setting, although I do admit that actually attacking that problem is beyond the scope of this already long paper.

**Questions:**

See above.

---

> ### Author Response · Authors · 2023-11-16
>
> Dear reviewer *xcHJ*,
>
> Thank you for the clear and concise assessment of our paper. We try to address your questions regarding theory vs practice below.
>
> ### Weaknesses \& Questions.
> ***
>
> The link between theory and practice in double descent is indeed a very active topic of research in Supervised Learning. Idealized assumptions are necessary to develop the theory, and in practice, these assumptions are violated to varying degrees. Nonetheless, the theoretical results in Supervised Learning have provided insight into the effects of overparameterization and regularization with practical implications, e.g., on the choice of hyperparameters. We expect that further studies will show similar connections between our theory and RL in practice.
>
> In Supervised Learning, the double descent phenomenon was first observed empirically, which then prompted a theoretical investigation
> [1, 2]. The theory is based on the hypothesis of a single-layer network of infinite width and random features. Nonetheless, in practice, double descent has been observed empirically for both shallow and deep networks with large layers and both with and without random features. Furthermore, the theoretical benefits of random features have inspired their use in practice, e.g., to reduce spectral bias [3].
>
> Similarly, our theory on double descent in RL assumes an infinitely large single-layer architecture, but this assumption does not seem to be required in practice since the phenomenon is also observed empirically in systems with less than a hundred parameters.
> Further studies are required to clarify the impact of random features, with sufficient computational resources for a thorough empirical analysis.
>
> ### References
> ***
>
> [1] Zhang, Chiyuan, et al. "Understanding deep learning (still) requires rethinking generalization." (2021)
>
> [2] Belkin, Mikhail, et al. "Reconciling modern machine-learning practice and the classical bias–variance trade-off." (2019)
>
> [3] Tancik, Matthew, et al. "Fourier features let networks learn high frequency functions in low dimensional domains." (2020)

---

> > ### Author Response · Authors · 2023-11-22
> >
> > Dear reviewer xcHJ,
> >
> > Today is the last day of the discussion period. Are you satisfied with our response? Since you did not point out any particular weaknesses, would you consider raising the score?

---

> > > ### Author Response · Authors · 2023-11-23
> > >
> > > Dear reviewer xcHJ,
> > >
> > > This is a gentle reminder that the rebuttal period ends in two hours. We would love to have your feedback to help us improve the paper.

---

### Author Response · Authors · 2023-11-23

Dear reviewers,

We thank you all for the encouraging and constructive feedback, which allowed us to add important clarifications to the paper. Thank you for all the time and effort you put into reviewing our work.

---

### Meta-Review · Area_Chair_KEzm · 2023-12-07

**Metareview:**

This paper examines the behavior of the (regularized) LSTD estimator within the random feature model and the on-policy evaluation context. It investigates a doubly asymptotic regime in which both the number of parameters and the number of distinct visited states tend to infinity at the same rate. A double descent phenomenon is observed between the empirical mean squared Bellman error (MSBE) and the population MSBE. The authors derive analytical equations for both the asymptotic empirical MSBE and the asymptotic population MSBE. Based on these derivations, they demonstrate that the double-descent phenomenon diminishes as either the regularization increases or the number of distinct unvisited states decreases. Empirical experiments corroborate the theoretical findings.
All reviewers agree this paper studies an interesting setting (double descent and RL) and provides a thorough theoretical analysis. The AC agrees and thus recommends acceptance.

**Justification For Why Not Higher Score:**

This paper studies a highly idealized setting, and the analysis techniques are borrowed from existing ones from the supervised learning double descent literature.

**Justification For Why Not Lower Score:**

The setting is interesting and the analysis is quite thorough.

---

### Decision · Program_Chairs · 2024-01-16

Accept (poster)